# Which Algorithms Can Graph Neural Networks Learn?

**Solveig Wittig** [* 1]   **Antonis Vasileiou** [* 1]   **Robert R. Nerem** [* 2]   **Timo Stoll** [1]   **Floris Geerts** [3]   **Yusu Wang** [2]
**Christopher Morris** [1]

## Abstract

In recent years, there has been growing interest in understanding neural architectures' ability to learn to execute discrete algorithms, a line of work often referred to as neural algorithmic reasoning. The goal is to integrate algorithmic reasoning capabilities into larger neural pipelines. Many such architectures are based on (message-passing) graph neural networks (MPNNs), owing to their permutation equivariance and ability to deal with sparsity and variable-sized inputs. However, much existing work is either largely empirical and lacks formal guarantees or it focuses solely on expressivity, leaving open the question of when and how such architectures generalize beyond a finite training set. In this work, we propose a general theoretical framework that characterizes sufficient conditions under which MPNNs can learn an algorithm from a training set of small instances and provably approximate its behavior on inputs of arbitrary size with worst-case guarantees. Our framework applies to a broad class of algorithms, including single-source shortest paths, minimum spanning trees, and general dynamic programming problems, such as the 0-1 knapsack problem. In addition, we establish impossibility results for a wide range of algorithmic tasks, showing that standard MPNNs cannot learn them and derive more expressive MPNN-like architectures that overcome these limitations. Finally, we refine our analysis for the Bellman–Ford algorithm, yielding substantially smaller required training sets and significantly extending the recent work of Nerem et al. (2025) by allowing for a differentiable regularization loss. Empirical results largely support our theoretical findings.

*Equal contribution   [1]RWTH Aachen University, Germany   [2]University of California San Diego, USA   [3]University of Antwerp, Belgium. Correspondence to: Solveig Wittig <solveig.wittig@log.rwth-aachen.de>.

*Proceedings of the $43^{rd}$ International Conference on Machine Learning*, Seoul, South Korea. PMLR 306, 2026. Copyright 2026 by the author(s).

*Graph neural networks* (GNNs), and in particular *message-passing neural networks* (MPNNs), constitute a versatile and expressive class of neural architectures for learning over graph-structured data (Gilmer et al., 2017; Scarselli et al., 2009). Their permutation equivariance and ability to operate on sparse, variable-sized inputs have made them a central tool across a wide range of applications, including drug design (Wong et al., 2023), global medium-range weather forecasting (Lam et al., 2023), and combinatorial optimization (Cappart et al., 2023; Gasse et al., 2019; Scavuzzo et al., 2024; Qian et al., 2024b).

Recently, MPNNs have played a central role in *neural algorithmic reasoning* (NAR), a research direction that seeks to bridge classical algorithm design and neural computation (Cappart et al., 2023; Velickovic & Blundell, 2021; Xu et al., 2020). The goal of NAR is to enable neural networks to learn, execute, and generalize discrete algorithms, thereby seamlessly integrating algorithmic reasoning into end-to-end trainable neural pipelines. Due to their close correspondence with iterative, local graph computation, MPNNs have emerged as a natural architectural backbone for learning graph algorithms such as shortest paths, minimum spanning trees, and dynamic programming procedures (Velickovic et al., 2020). A key motivation for studying such algorithmic primitives is neural combinatorial optimization, where understanding how MPNNs learn and exploit algorithmic structure may help in designing models for more complex optimization problems for which efficient algorithms are unknown or difficult to integrate into differentiable pipelines. In this context, canonical graph algorithms provide a controlled setting for studying learnability and size generalization in neural architectures, rather than replacing classical solvers where these are already available.

Despite substantial empirical progress, theoretical understanding of neural algorithmic reasoning remains limited. Much prior work is either primarily empirical, demonstrating performance on benchmark instances without formal guarantees, or focuses on expressivity questions, characterizing which algorithms can in principle be represented by a given architecture, e.g., Azizian & Lelarge (2021); Chen et al. (2020); He & Vitercik (2025); Loukas (2020); Morris et al. (2019); Xu et al. (2019), with a recent focus on transformer architectures, e.g., de Luca & Fountoulakis

(2024); de Luca et al. (2025); Merrill & Sabharwal (2025); Sanford et al. (2024b;a); Yehudai et al. (2025); Zhou et al. (2024). Some recent works have also begun to study sample complexity and learnability in neural algorithmic reasoning settings, including analyses based on algorithmic alignment (Xu et al., 2019, Theorem 3.6), VC-dimension and covering-number bounds for expressive GNN architectures (Pellizzoni et al., 2025, Section 4) and Franks et al. (2024), and PAC-style norm-based generalization bounds for transformer architectures capable of simulating MPC computation (de Luca et al., 2025, Section 6). However, existing analyses primarily provide in-distribution or PAC-style guarantees and offer limited insight into worst-case or out-of-distribution generalization to larger or structurally different graphs. As a consequence, it remains unclear when learned models provably generalize beyond the training distribution, a property essential for reliable algorithmic deployment.

A notable recent exception is the work of Nerem et al. (2025), which provides theoretical guarantees for learning graph algorithms with MPNNs. However, their analysis is restricted to a single algorithm (i.e., Bellman–Ford) and enforces correctness via a non-differentiable regularization term. These assumptions limit the applicability of the results to broader classes of algorithms and to standard gradient-based learning pipelines.

**Present work** In this work, we develop a general theoretical framework for learning algorithms with MPNNs that addresses these limitations. We characterize sufficient conditions under which an MPNN trained on a small set of instances can provably generalize to inputs of arbitrary size with worst-case guarantees, covering a broad class of graph algorithms while remaining compatible with fully differentiable training objectives. Our framework clarifies which algorithms MPNNs can *learn* from finite data and which they *cannot learn*. Concretely, our contributions are as follows.

1. We introduce a theoretical framework characterizing when standard MPNNs (or more expressive MPNNs) can learn the cost function of graph algorithms, uniformly over graphs of arbitrary size, from finite data by minimizing an empirical loss; see Section 2.

2. Using this framework, we identify conditions under which GNNs can learn standard graph algorithms, including single-source shortest-path, minimum spanning tree, and dynamic programming algorithms.

3. For single-source shortest-path, we propose a differentiable $\ell_1$-regularization term that reduces required training data by balancing empirical risk minimization with regularization to enforce a sparsity pattern; see Section 2.3.

4. We empirically validate that these insights translate

into practice, underscoring the role of training data and the proposed regularization; see Section 5.

*Taken together, our framework provides a precise characterization of which algorithms standard and more expressive MPNNs can learn, enabling a more principled understanding of the capabilities and limitations of GNN-based, data-driven algorithmic design.*

See Appendix A for a detailed discussion on related work.

## 1. Background

In the following sections, we introduce the notation and provide the necessary background.

**Basic notations** Let $\mathbb{N} \coloneqq \{1,2,\dots\}$, $\mathbb{N}_0 \coloneqq \mathbb{N} \cup \{0\}$, $\mathbb{R}^+$ denote the non-negative reals, and $\mathbb{R}_{>0}$ the positive reals. For $n \in \mathbb{N}$, let $[n] \coloneqq \{1,\dots,n\}$ and $[n]_0 \coloneqq \{0,1,\dots,n\}$. We use $\{\!\{\dots\}\!\}$ to denote multisets. For non-empty sets $X,Y$, let $Y^X$ be the set of functions $X \to Y$. For $A \subset X$, let $1_A \colon X \to \{0,1\}$ be the indicator function. For a matrix $M \in \mathbb{R}^{n \times m}$, $M_{i,\cdot}$ and $M_{\cdot,j}$ denote its $i$th row and $j$th column. The symbol $\mathbf{o}$ denotes an all-zero vector of appropriate dimension. Functions are applied to sets, multisets, and matrices element-wise. For $x \in \mathbb{R}^n$, define $\|x\|_2 \coloneqq (\sum_{i=1}^n x_i^2)^{1/2}$ and $\|x\|_\infty \coloneqq \max_{i \in [n]} |x_i|$. For $M \in \mathbb{R}^{n \times m}$, define the Frobenius norm $\|M\|_{\mathrm{F}} \coloneqq (\sum_{i,j} M_{ij}^2)^{1/2}$, the operator norm $\|M\|_2 \coloneqq \sup_{x \neq \mathbf{o}} \|Mx\|_2/\|x\|_2$, and $\|M\|_\infty \coloneqq \max_{i \in [n]} \sum_{j=1}^m |M_{ij}|$.

**Graphs** An (undirected) *graph* $G$ is a pair $(V(G), E(G))$ with finite vertex set $V(G)$ and edge set $E(G) \subseteq \{\{u,v\} \subseteq V(G) \mid u \neq v\}$. The *order* of $G$ is $|V(G)|$; if $|V(G)| = n$ we call $G$ an *$n$-order graph*. A *directed graph* satisfies $E(G) \subseteq V(G)^2$. For an $n$-order graph with $V(G) = [n]$, the *adjacency matrix* $A(G) \in \{0,1\}^{n \times n}$ is defined by $A(G)_{vw} = 1$ iff $\{v,w\} \in E(G)$. The *neighborhood* of $v \in V(G)$ is $N_G(v) \coloneqq \{u \in V(G) \mid \{u,v\} \in E(G)\}$.

An *attributed graph* is a pair $(G, a_G)$ with $a_G \colon V(G) \to \mathbb{R}^d$; the *attribute* of $v$ is $a_G(v)$. An *edge-featured graph* is a pair $(G, w_G)$ with $w_G \colon E(G) \to \mathbb{R}^d$; in the special case $d = 1$ and $w_G(e) \in \mathbb{R}_{>0}$, $G$ is *edge-weighted* and $w_G(e)$ is the *edge weight*. When unambiguous, we write $w(e)$ or $w_e$.

For graphs without edge features, the degree of $u$ is $\deg_G(u) \coloneqq |N_G(u)|$; for edge-weighted graphs, $\deg_G(u) \coloneqq \sum_{v \in N_G(u)} w_G(u,v)$. We omit the subscript when the graph is clear.

For a graph class $\mathcal{G}$ and $k \in \mathbb{N}$, let $V_k(\mathcal{G}) \coloneqq \{(G, \boldsymbol{v}) \mid G \in \mathcal{G}, \ \boldsymbol{v} \in V(G)^k\}$, and set $V_0(\mathcal{G}) \coloneqq \mathcal{G}$.

See Appendix B for additional graph-related notation.

**Invariants** Let $\mathcal{G}$ be a set of graphs, a *graph-level invariant*

*(regarding $\mathcal{G}$)* is a function $h \colon \mathcal{G} \to \mathbb{R}^d$, $d > 0$ such that $h(G) = h(H)$, for $G$ and $H$ being isomorphic. In addition, for $k > 0$, a *k-tuple invariant* is a function $h \colon V_k(\mathcal{G}) \to \mathbb{R}^d$, such that $h(G, \boldsymbol{v}) = h(H, \boldsymbol{w})$ whenever $(G, \boldsymbol{v})$ and $(H, \boldsymbol{w})$ are isomorphic. For $k = 0$, this recovers the notion of a graph invariant.

### 1.1. Metric spaces, covering numbers, and partitions

Here, we define pseudo-metric spaces, continuity assumptions, covering numbers, and partitions, which play an essential role in the following.

**Metric spaces** In the remainder of the paper, "distances" between graphs play an essential role, which we make precise by defining a *pseudo-metric* (on the set of graphs). Let $\mathcal{X}$ be a set equipped with a pseudo-metric $d \colon \mathcal{X} \times \mathcal{X} \to \mathbb{R}^+$, i.e., $d$ is a function satisfying $d(x, x) = 0$ and $d(x, y) = d(y, x)$ for $x, y \in \mathcal{X}$, and $d(x, y) \le d(x, z) + d(z, y)$, for $x, y, z \in \mathcal{X}$. The latter property is called the triangle inequality. The pair $(\mathcal{X}, d)$ is called a *pseudo-metric space*. For $(\mathcal{X}, d)$ to be a *metric space*, $d$ additionally needs to satisfy $d(x, y) = 0 \Rightarrow x = y$, for $x, y \in \mathcal{X}$.[1]

**Lipschitz continuity on metric spaces** Let $(\mathcal{X}, d_{\mathcal{X}})$ and $(\mathcal{Y}, d_{\mathcal{Y}})$ be two pseudo-metric spaces. A function $f \colon \mathcal{X} \to \mathcal{Y}$ is called $c_f$-*Lipschitz continuous*, for $c_f \in \mathbb{R}_{>0}$, if, for $x, x' \in \mathcal{X}$,

$$d_{\mathcal{Y}}(f(x), f(x')) \le c_f \cdot d_{\mathcal{X}}(x, x').$$

**Covering numbers** Let $(\mathcal{X}, d)$ be a pseudo-metric space. Given an $\varepsilon > 0$, an $\varepsilon$-*cover* of $\mathcal{X}$ is a subset $C \subseteq \mathcal{X}$ such that for all elements $x \in \mathcal{X}$ there is an element $y \in C$ such that $d(x, y) \le \varepsilon$. Given $\varepsilon > 0$ and a pseudo-metric $d$ on the set $\mathcal{X}$, we define the *covering number* of $\mathcal{X}$,

$$\mathcal{N}(\mathcal{X}, d, \varepsilon) := \min\{m \mid \exists \text{ an } \varepsilon\text{-cover of } \mathcal{X} \text{ of cardinality } m\},$$

i.e., the smallest number $m$ such that there exists a $\varepsilon$-cover of cardinality $m$ of the set $\mathcal{X}$ with regard to the pseudo-metric $d$.

### 1.2. Message-passing graph neural networks

One particular, well-known class of graph machine learning architectures is MPNNs. MPNNs learn a $d$-dimensional real-valued vector of each vertex in a graph by aggregating information from neighboring vertices. Following Gilmer et al. (2017), let $(G, a_G, w_G)$ be an attributed, edge-weighted graph with initial vertex feature $\boldsymbol{h}_v^{(0)} := a_G(v) \in \mathbb{R}^{d_0}$, $d_0 \in \mathbb{N}$, for $v \in V(G)$. An *L-layer MPNN architecture* consists of a composition of $L$ neural network layers for some

---

[1] Observe that computing a metric on the set of graphs $\mathcal{G}$ up to isomorphism is at least as hard as solving the graph isomorphism problem on $\mathcal{G}$.

$L > 0$. In each *layer*, $t \in \mathbb{N}$, we compute a node feature

$$\boldsymbol{h}_v^{(t)} := \mathsf{UPD}_{\boldsymbol{u}_t}^{(t)} \Big( \boldsymbol{h}_v^{(t-1)}, \mathsf{AGG}_{\boldsymbol{a}_t}^{(t)} \big( \\ \{\!\!\{ (\boldsymbol{h}_v^{(t-1)}, \boldsymbol{h}_u^{(t-1)}, w_G(v, u)) \mid u \in N(v) \}\!\!\} \big) \Big), \quad (1)$$

$d_t \in \mathbb{N}$, for $v \in V(G)$, where $\mathsf{UPD}_{\boldsymbol{u}_t}^{(t)}$ and $\mathsf{AGG}_{\boldsymbol{a}_t}^{(t)}$ are functions, parameterized by $\boldsymbol{u}_t \in \boldsymbol{U}_t$ and $\boldsymbol{a}_t \in \boldsymbol{A}_t$, e.g., neural networks, with $\boldsymbol{U}_t$ and $\boldsymbol{A}_t$ being sets of parameters, e.g., $\mathbb{R}^d$. In the case of graph-level tasks, e.g., graph classification, one also uses a *readout*, where

$$\boldsymbol{h}_G := \mathsf{READOUT}_{\boldsymbol{r}} \Big( \{\!\!\{ \boldsymbol{h}_v^{(L)} \mid v \in V(G) \}\!\!\} \Big) \in \mathbb{R}^d, \quad (2)$$

to compute a single vectorial representation based on learned vertex features after iteration $L$. Again, $\mathsf{READOUT}_{\boldsymbol{r}}$ is a a parameterized function, for $\boldsymbol{r}$ in some parameter set $\boldsymbol{R}$. Throughout the paper, we consider a variety of MPNN architectures; all of them can be viewed as special cases of the general MPNN formulation introduced above. We distinguish between *node-* and *graph-level* MPNNs, i.e., the former compute a feature for each node in a given graph while the latter compute a single feature for the whole graph; see Appendix B.1 in the appendix for details.

See Appendix B for connections to the graph isomorphism problem and for definitions of simple heuristics based on variants of the 1-*dimensional Weisfeiler–Leman algorithm* (namely 1-iWL and $(1,1)$-WL), as well as the separation and approximation abilities of MPNNs.

## 2. What and how *can* GNNs learn

Based on the definition of invariants in Section 1, we can view algorithms as invariant maps from graphs (or their nodes) to scalars or real-valued vectors. Given a hypothesis class, we can compare its distinguishability with that of an algorithm via their induced equivalence relations, closely related to uniform approximation (see Proposition 7 in Appendix B). However, these results do not explain how to choose a function in the hypothesis class that approximates a given algorithm, and, moreover, the notion of approximation is non-uniform, applying only to graphs of fixed size. Consequently, it does not, by itself, imply learnability, particularly for larger instances. In this section, we address these limitations by deriving suitable loss functions and finite datasets; training under these settings yields guarantees on how well MPNNs can learn to approximate algorithms uniformly across graph sizes.

We distinguish between learning algorithms that are graph-level invariants, e.g., the cost of the minimum spanning tree, and $k$-tuple invariants, e.g., the shortest-path distance from a source node to all other nodes in a graph.

## 2.1. Regularization-induced extrapolation

Here, we develop a general theoretical framework showing that, with suitable regularization and carefully chosen datasets, learning systems can extrapolate beyond the training range, provided the input features lie in a compact set with an appropriate topology. In particular, our results apply to extrapolation to arbitrarily large domains. We establish this theory in a general learning setting by proving basic learnability properties of Lipschitz functions on compact sets. In Section 2.2, we specialize the analysis to MPNNs with different architectures, yielding our main extrapolation (size generalization) results.

We begin by defining the notion of a *finite Lipschitz class*: a set of parameterized functions over a bounded domain for which the parameters control the Lipschitz constant.

**Definition 1.** Let $\mathcal{X}$ be a non-empty set. Given a hypothesis class $\mathcal{F}_{\boldsymbol{\Theta}} \coloneqq \{f_{\boldsymbol{\theta}} : \boldsymbol{\theta} \in \boldsymbol{\Theta}\}$ with $f_{\boldsymbol{\theta}} : \mathcal{X} \to \mathbb{R}$ and $\boldsymbol{\Theta}$ being a set of parameters. We say that $\mathcal{F}_{\boldsymbol{\Theta}}$ is a *finite Lipschitz class* if there exists a (pseudo-)metric $d_{\mathcal{X}}$ on $\mathcal{X}$ such that for every $\boldsymbol{\theta} \in \boldsymbol{\Theta}$, $f_{\boldsymbol{\theta}}$ is Lipschitz with minimal Lipschitz constant[2] $M_{\boldsymbol{\theta}} < \infty$, and the covering number $\mathcal{N}(\mathcal{X}, d, \varepsilon)$ is finite for all $\varepsilon > 0$.

In what follows, we assume that whenever the above definition is satisfied, one can compute an upper bound $B_{\boldsymbol{\theta}} \in \mathbb{R}_{>0}$ such that $M_{\boldsymbol{\theta}} \leq B_{\boldsymbol{\theta}}$, for all $\boldsymbol{\theta} \in \boldsymbol{\Theta}$, which we call a *Lipschitz certificate*. A target function $f^*$ is approximable (with respect to $\mathcal{F}_{\boldsymbol{\Theta}}$ with certificate $B_{f^*}$ if for all $\epsilon > 0$, there exists $\boldsymbol{\theta} \in \boldsymbol{\Theta}$ such that $|f_{\boldsymbol{\theta}}(x) - f^*(x)| < \epsilon$ for all $x \in \mathcal{X}$.

Let $N \in \mathbb{N}$ and $\mathcal{F}_{\boldsymbol{\Theta}}$ be a finite Lipschitz class, let $f^* \colon \mathcal{X} \to \mathbb{R}$ be a target function, and let $X \coloneqq \{x_1, \ldots, x_N\} \subseteq \mathcal{X}$. For the dataset $\{(x_i, y_i)\}_{i=1}^N$ with $y_i \coloneqq f^*(x_i)$, the *empirical loss* of a hypothesis $f_{\boldsymbol{\theta}} \in \mathcal{F}_{\boldsymbol{\Theta}}$ is

$$\mathcal{L}_X^{\text{emp}}(f_{\boldsymbol{\theta}}) \coloneqq \frac{1}{N} \sum_{i=1}^N |f_{\boldsymbol{\theta}}(x_i) - y_i|,$$

and its *regularized loss* is

$$\mathcal{L}_X(f_{\boldsymbol{\theta}}) \coloneqq \mathcal{L}_X^{\text{emp}}(f_{\boldsymbol{\theta}}) + \mathcal{L}^{\text{reg}}(f_{\boldsymbol{\theta}}),$$

where $\mathcal{L}^{\text{reg}} \colon \mathcal{F}_{\boldsymbol{\Theta}} \to \mathbb{R}^+$ is a regularization term which, in practice, depends on the certificate $B_{\boldsymbol{\theta}}$ described above. Such regularizers typically depend on the norms of the trainable weights; a concrete example is given below.

**Example of finite Lipschitz class and certificates** A common example a finite Lipschitz class is given by standard feedforward neural networks of the form $\boldsymbol{x} \mapsto \boldsymbol{W}_L \sigma(\boldsymbol{W}_{L-1} \sigma(\cdots \sigma(\boldsymbol{W}_1 \boldsymbol{x}) \cdots))$ with 1-Lipschitz nonlinearities, for which one may take $B_{\boldsymbol{\theta}} = \prod_{\ell=1}^L \|\boldsymbol{W}_\ell\|_2$ with respect to the induced Euclidean metric and assume inputs

---

[2]The infimum is used if the minimum does not exist.

$\boldsymbol{x}$ in a closed Euclidean ball. More generally, certificates for compositions are obtained by multiplying per-layer or operator bounds, and many architectures (including constrained residual or normalized variants) admit similarly computable certificates. Likewise, we assume we have a known upper bound $B_{f^*} \in \mathbb{R}_{>0}$ such that the target $f^*$ is Lipschitz regarding $d_{\mathcal{X}}$ with minimal Lipschitz constant at most $B_{f^*}$.

The following result shows that for a finite Lipschitz class, controlling the Lipschitz certificate of that class, and the Lipschitz continuity of the target class imply that a target function can be learned from finite data; see Figure 1 for an illustration.

**Theorem 2** (Informal). *Let $\mathcal{F}_{\boldsymbol{\Theta}}$ be a finite Lipschitz class on $(\mathcal{X}, d_{\mathcal{X}})$ with certificates $B_{\boldsymbol{\theta}}$, and let $f^*$ be a Lipschitz target that is approximable with certificate $B_{f^*}$. Then, defining the regularization term as*

$$\mathcal{L}^{\text{reg}}(f_{\boldsymbol{\theta}}) = \eta \operatorname{ReLU}(B_{\boldsymbol{\theta}} - B_{f^*}), \quad \text{for some } \eta > 0,$$

*it follows that for any $\varepsilon > 0$, there exist $\varepsilon'(\varepsilon), r(\varepsilon) > 0$ and a dataset $X \subset \mathcal{X}$ with cardinality $X = \mathcal{N}(\mathcal{X}, d_{\mathcal{X}}, r(\varepsilon))$ such that if the regularized loss on $X$ is smaller than $\varepsilon'(\varepsilon)$, then*

$$\sup_{x \in \mathcal{X}} |f_{\boldsymbol{\theta}}(x) - f^*(x)| < \varepsilon.$$

*Proof sketch.* The proof follows a standard cover-plus-Lipschitz-control argument. Start by taking the dataset $X$ to be an $r$-cover of $(\mathcal{X}, d_{\mathcal{X}})$, where $r > 0$ is chosen as a function of the desired error $\varepsilon$. Let $f_{\boldsymbol{\theta}} \in \mathcal{F}_{\boldsymbol{\Theta}}$ satisfy $\mathcal{L}_X(f_{\boldsymbol{\theta}}) < \varepsilon'$. Since the regularized loss is the sum of the empirical loss and the certificate regularizer, both terms are small. For any $x \in \mathcal{X}$, there is $x_i \in X$ such that $d_{\mathcal{X}}(x, x_i) \leq r$. The triangle inequality then gives $|f_{\boldsymbol{\theta}}(x) - f^*(x)| \leq |f_{\boldsymbol{\theta}}(x) - f_{\boldsymbol{\theta}}(x_i)| + |f_{\boldsymbol{\theta}}(x_i) - f^*(x_i)| + |f^*(x_i) - f^*(x)|$. The middle term $|f_{\boldsymbol{\theta}}(x_i) - f^*(x_i)|$ is controlled by the small empirical loss on $X$. The first term $|f_{\boldsymbol{\theta}}(x) - f_{\boldsymbol{\theta}}(x_i)|$ is controlled by the regularizer, since a small regularization term bounds $B_{\boldsymbol{\theta}}$ and therefore the Lipschitzness of $f_{\boldsymbol{\theta}}$. The last term $|f^*(x_i) - f^*(x)|$ is controlled by the Lipschitzness of $f^*$. Since $x$ was arbitrary, taking $r$ and $\varepsilon'$ sufficiently small gives $\sup_{x \in \mathcal{X}} |f_{\boldsymbol{\theta}}(x) - f^*(x)| < \varepsilon$. $\square$

The above result shows that, under suitable assumptions (i.e., that the target algorithm the can be uniformly approximated by the considered hypothesis class) for any prescribed approximation error, one can carefully construct a sufficiently informative training dataset such that minimizing the empirical regularized loss over this dataset guarantees that the learned model approximates the target algorithm up to the desired error level. Note that, in this setting, the regularized loss has infimum zero, since the assumptions allow the training data to be fit arbitrarily well while keeping $B_{\boldsymbol{\theta}} \leq B_{f^*}$.

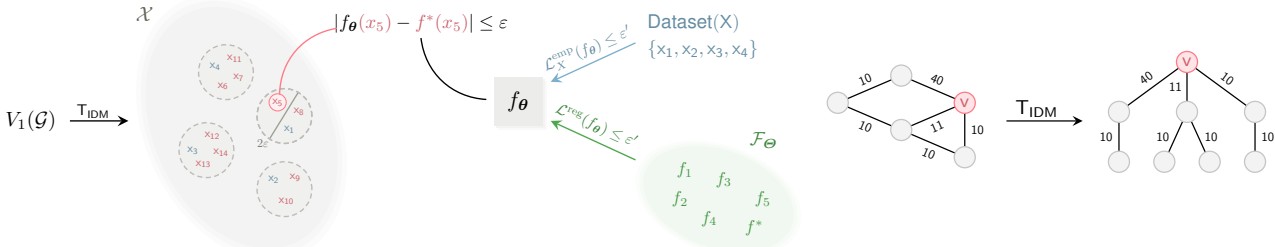

*Figure 1.* An illustration of the learnability result in Theorem 2, applied to MPNNs by first mapping the space $V_1(\mathcal{G})$ to a pseudometric space (via IDMs or computation trees; see Appendix K.1), satisfying Definition 1 and then applying Theorem 2. A computation-tree construction is shown on the right.

**On the $B_{f^*}$ certificate assumption** The key assumption in Theorem 2 is not that the *true* Lipschitz constant stays controlled while approximating the target, but that the *Lipschitz certificate $B_\theta$* does. That is, we can approximate $f^*$ arbitrarily well by functions $f_\theta$ whose computable bounds $B_\theta$ do not blow up (a strengthened notion of approximate realizability). Since $B_\theta$ is directly computable from the parameters (e.g., via operator-norm or other parameter-norm bounds), it can be used in training even when the true Lipschitz constant $M_\theta$ is intractable. This yields a practical regularizer that controls $B_\theta$.

The bound for $B_{f^*}$ is trivial if the target function lies in the hypothesis class, but often $f^* \notin \mathcal{F}_\Theta$ and must be approximated. For standard FNNs, obtaining a nontrivial bound on $B_{f^*}$ is difficult. However, fully expressive FNN alternatives have been proposed for adversarial robustness (Anil et al., 2019); their Lipschitz constants can be bounded via parameter constraints, and the resulting function class is dense in the Lipschitz functions, so they can meet the theorem's conditions. Similarly, 1-Lipschitz residual networks are dense in the set of scalar 1-Lipschitz functions on any compact domain (Murari et al., 2025), with Lipschitz constants bounded through parameter constraints.

A more general theorem in Appendix D addresses cases where functions cannot be approximated with a bounded certificate, including non-Lipschitz targets.

**Connection to distinguishability** If the conclusion of Theorem 2 holds, then a necessary condition is that, whenever $\mathcal{F}_\Theta$ cannot distinguish two inputs, $f^*$ must also assign them the same value. This observation is particularly relevant for hypothesis classes with limited distinguishing power, such as MPNNs. In that case, uniform learnability is restricted to targets $f^*$ whose distinguishability does not exceed that of the model class.

### 2.2. Finite Lipschitzness in MPNNs

Below, we study MPNNs that satisfy the finite Lipschitz learning property from Definition 1, which suffices for the

learnability guarantee in Theorem 2. Throughout, the input space consists of pairs $(G, u)$ with a graph $G$ and node $u \in V(G)$, i.e., $V_1(\mathcal{G})$ for a graph class $\mathcal{G}$.

We equip $V_1(\mathcal{G})$ with a pseudo-metric that controls the Lipschitz behavior of MPNNs. Our analysis covers (i) normalized sum aggregation, (ii) mean aggregation, and (iii) max/min aggregation; see Appendix I for formal definitions. These are special cases of the general MPNN formulation in Equation (1) and capture many classical graph algorithms.

We first establish finite Lipschitzness for normalized sum aggregation and extend it to mean aggregation. While inspired by work on iterated degree measures and computation trees (e.g., Grebik & Rocha (2022); Böker et al. (2023); Rauchwerger et al. (2024)), our approach does not require compactness of the input space: Definition 1 only assumes finite covering numbers, simplifying the construction (see Appendix K).

Crucially, we also show that max/min aggregation—essential for many algorithmic invariants—satisfies finite Lipschitzness. Here, the analysis proceeds on Hausdorff spaces with the Hausdorff distance (see Appendix L). To our knowledge, these are the first explicit Lipschitzness guarantees for max- and min-aggregation MPNNs, despite widespread use of related stability arguments (Levie, 2023; Böker et al., 2023; Rauchwerger et al., 2024; Vasileiou et al., 2024a). These architectures cover a range of common graph algorithms (see Section 2.2.1). Precise assumptions, conditions, and constants appear in Appendix I; we state the main result informally next.

**Theorem 3** (Informal)*. The hypothesis class $\mathcal{F}_\Theta$ induced by MPNNs using normalized sum aggregation, mean aggregation, or max (or min) aggregation satisfies Definition 1.*

See Theorems 103 and 107 for the formal statements and proofs. In the following, we identify graph invariants expressible by these MPNN architectures, and hence learnable from finite data by Theorem 3.

### 2.2.1. ALGORITHMS WITHIN A FINITE LIPSCHITZ CLASS

Here, we outline examples of algorithms that are contained in finite Lipschitz classes, i.e., following Definition 1, there exists a (pseudo-)metric with finite covering number and the algorithm is Lipschitz with respect to this (pseudo-)metric.

**Normalized-sum aggregation** In particular, normalized-sum aggregation is of special interest, as on graphs with a fixed number of nodes it allows MPNNs to represent any graph invariant with distinguishing power equivalent to the 1-WL. However, as we will see in Section 3.2, this learnability property cannot be maintained without restricting attention to graphs of fixed order.

**Truncated PageRank** We now show that a mean-aggregation MPNN learning truncated PageRank satisfies Theorem 2. Let $(G, w_G)$ be an edge-weighted graph, $u \in V(G)$, $\xi \in (0,1)$ a damping factor, and $K \in \mathbb{N}$ a truncation depth. The $K$-*truncated weighted PageRank* value $r_u^{(K)}$ is defined recursively for $t \in [K]$ as

$$r_u^{(t)} := (1 - \xi) + \frac{\xi}{\deg(u)} \sum_{v \in N(u)} w_{uv}\, r_v^{(t-1)}, \quad (3)$$

where $r_u^{(0)} := 1$, $\deg(u)$ denotes the weighted degree of $u$, and $w_{uv} := w_G(u, v)$. This generalizes standard PageRank (Page et al., 1998) to edge-weighted graphs; the unweighted case has $w_{uv} = 1$ for all $(u, v) \in E(G)$.

This algorithm is represented exactly by a $K$-layer mean-aggregation MPNN (Equation (22)). Set $\boldsymbol{h}_u^{(0)} := 1$ and define $\phi_t(x, y) := (1 - \xi) + \xi y$ for $t \in [K]$. Then the MPNN update

$$\boldsymbol{h}_u^{(t)} := \phi_t\!\Big(\boldsymbol{h}_u^{(t-1)}, \ \frac{1}{\deg(u)} \sum_{v \in N(u)} w_{uv}\, \boldsymbol{h}_v^{(t-1)}\Big)$$

matches (3), yielding $\boldsymbol{h}_u^{(t)} = r_u^{(t)}$ for $t \in [K]$. Since PageRank does not use self-loops, $\phi_t$ ignores its first argument. Lipschitz continuity of truncated PageRank follows from Theorem 3, as it lies in the hypothesis class of mean-aggregation MPNNs, which satisfies Definition 1. A key property is that the truncation depth $K$ needed for error $\varepsilon > 0$ depends only on $\varepsilon$ and $\xi$, not on $|V(G)|$. Since PageRank is a contraction with factor $\xi$, we have $|r_u^{(K)} - r_u^{(\infty)}| \leq \xi^K$, where $r_u^{(\infty)}$ is full PageRank. Thus, choosing $K \geq \lceil \log_\xi(\varepsilon) \rceil$ guarantees error at most $\varepsilon$ regardless of graph size. Therefore, by Theorem 2, a finite training set with regularization suffices to train an MPNN to learn truncated PageRank and extrapolate to graphs of arbitrary size. Moreover, since truncation error is size-independent, the trained model approximates full PageRank to the same error across all graphs, despite being trained on finite data.

**Bellman–Ford** The Bellman–Ford algorithm for single-source shortest paths provides another example fitting into the framework. For an edge-weighted graph $(G, w_G)$ with source vertex $r \in V(G)$, the $K$-step Bellman–Ford algorithm computes shortest path distances $x_v^{(K)}$ from the root vertex $r$ to each vertex $v$ via the recurrence $x_r^{(0)} := 0$, $x_v^{(0)} := \beta$ for $v \neq r$, where $\beta$ is a large constant, and $x_v^{(t)} := \min\{x_u^{(t-1)} + w_G(u, v) \mid u \in N(v) \cup \{v\}\}$, for $t \in [K]$. This can be represented exactly by an MPNN with min aggregation (Equation (23)) using $K$ layers, where the update function $\phi_t$ implements the identity and the aggregation function $M_t$ computes $M_t(x_u, w_G(u, v)) := x_u + w_G(u, v)$, for $u \in N(v)$. By Theorem 3, MPNNs with min aggregation satisfy Definition 1, thus it follows from Theorem 2 that a finite training set with regularization can be used to train an MPNN to learn $K$ iterations of Bellman–Ford, enabling learning from a finite number of training examples that will extrapolate to graphs of arbitrary size. In Section 2.3, we adopt an approach specifically designed for this setting to derive a small training set that guarantees extrapolation. More generally, Zhu et al. (2021) showed that a generalized Bellman–Ford algorithm, based on path formulations, captures invariants such as the Katz index, widest path, and most reliable path. These can likewise be represented by MPNNs with mean and max aggregation and thus fall into a finite Lipschitz hypothesis class.

**Dynamic programming** It is well-known that many problems that can be solved by dynamic programming can be cast as a shortest-path problem on a transformed graph (Frieze, 1976). Using the 0-1 knapsack problem as an illustration, for $n \in \mathbb{N}$, given items $i \in [n]$ with integer weights $s_i$ and values $v_i$, and a capacity $S$, consider the directed acyclic graph with vertices $(i, j)$, for $i \in [n]_0$ and $j \in [S]_0$, where $(i, j)$ represents having considered the first $i$ items and accumulated total weight $j$. Now, we add edges $((i-1, j), (i, j))$ of weight 0 ("do not choose item $i$") and, whenever $j + s_i \leq S$, edges $((i-1, j), (i, j+s_i))$ of weight $-v_i$ ("choose item $i$"); finally, connect each vertex $(n, j)$ to a sink $t$ by a zero-weight edge. Then every $s$-$t$ path (with $s := (0,0)$) encodes a feasible subset and has total weight equal to minus its total value, so the shortest-path distance to $t$ equals $-\text{OPT}$. Moreover, the standard knapsack recurrence is exactly the shortest-path relaxation on this graph, i.e., initializing $x_s^{(0)} := 0$ and $x_u^{(0)} := \beta$, for $u \neq s$, one has, for $i \in [n]$ and $j \in [S]_0$,

$$x_{(i,j)}^{(i)} := \min\Big\{ x_{(i-1,j)}^{(i-1)}, \ x_{(i-1, j-s_i)}^{(i-1)} - v_i \Big\},$$

where the second term is omitted when $j < s_i$, and $x_t^{(n)} := \min_{j \leq S} x_{(n,j)}^{(n)}$. As in the Bellman–Ford example, this computation can be represented exactly by a min-aggregation MPNN with $K = n$ layers by taking $\phi_t$ to be the identity and $M_t(x_u, w(u, v)) := x_u + w(u, v)$; thus, by Theorem 3 the hypothesis class satisfies Definition 1, and for families in which the number of stages $n$ is bounded (so $K$ is inde-

pendent of the size of the transformed graph, which grows with $S$), Definition 1 implies learnability from a finite training set with regularization and extrapolation to arbitrarily large capacities. We note that the same reasoning applies to a large class of problems that can be cast as dynamic programs, e.g., the longest increasing subsequence or edit distances between strings.

### 2.3. Improved learning guarantees for single-source shortest path (SSSP) algorithms

Now, the following result shows that we can explicitly construct a constant-size training set and a differentiable regularization term such that a small loss on the training set implies approximating the Bellman–Ford algorithm for arbitrarily large graphs. We consider the standard SSSP setting, in which each instance includes a designated source node (root) explicitly marked in the input. This assumption is important because it provides the model with access to the dynamic program's initial condition. Consider learning $K$-steps of Bellman–Ford using a min-aggregation MPNN with $K$ layers and $m$-layer feed-forward neural networks. We train the MPNN by minimizing a loss function $\mathcal{L}(\boldsymbol{\theta})$ that contains a weighted variant of $\ell_1$-regularization. In particular, $\mathcal{L}^{\mathrm{reg}}(\boldsymbol{\theta})$ is a weighted sum over the $\ell_1$ norms of the weight matrices and bias vectors, with layer-dependent weights.

**Theorem 4** (Informal). *There exists a training set $X$ of size $K + 1$ such that, for appropriate choice of regularization parameter $\eta$, if a $K$-layer min-aggregation MPNN achieves a loss $\mathcal{L}_X(\boldsymbol{\theta})$ within $\varepsilon < 1/2$ of its global minimum, then for any SSSP instance, the MPNN approximates every $K$-step shortest path distance $x^{(K)}$ within additive error $\varepsilon(x^{(K)} + 1)$.*

Although the above statement may appear existential, its proof (in Theorem 16) explicitly constructs the training dataset that guarantees the learnability property. This is possible due to the simple recursive structure underlying the Bellman–Ford algorithm.

The theorem differs from the main theorem of Nerem et al. (2025) in several ways. On the positive side, our result uses a smaller training set, namely $K + 1$ path instances, and replaces their non-differentiable $\ell_0$ penalty with a differentiable $\ell_1$ regularizer with layer-specific weights. This differentiability allows our loss to be optimized directly. On the other hand, our analysis assumes a somewhat more restricted model, i.e, the depth is fixed to exactly $K$ message-passing layers (matching $K$ Bellman–Ford steps), and the aggregation dimension is assumed to be 1. These choices are made to simplify the analysis. Furthermore, in our result, the regularization parameter $\eta$ and the edge weights in the training set both scale exponentially in $K$, which may be prohibitive in some settings. However, in Appendix F.2 we outline ways

to circumvent these limitations.

## 3. What GNNs *cannot* learn

In the previous section, we identified sufficient conditions under which a hypothesis class admits learnability guarantees via regularized empirical risk minimization. In particular, when a class of MPNNs forms a finite Lipschitz class with respect to a suitable (pseudo)metric on the input space, invariant algorithms that can be uniformly approximated within this class are learnable from finite samples. In this section, we turn to the complementary question, i.e., which invariant algorithms cannot be learned by MPNN hypothesis classes? We first show cases of invariants that are not expressible by any MPNN of the general form Equation (1). Then we show that even for invariants that are expressible by MPNNs, Theorem 2 may not be applied since Definition 1 is not satisfied by these types of MPNNs.

### 3.1. Expressivity limitations

This section assumes familiarity with the classical graph problems of *single-source shortest path* (SSSP), and *minimum spanning tree* (MST); see Appendix B.4 for formal definitions. We start by showing that MPNNs are often not expressive enough to learn simple graph algorithms. To that end, we first show that standard MPNNs, see Equation (1), are not expressive enough to determine the costs of the SSSP and MST.

Formally, let $\mathcal{G}$ denote the class of edge-weighted graphs. Given an edge-weighted graph $(G, w_G)$ and a source vertex $s \in V(G)$, we view the costs for the SSSP problem as a 2-tuple invariant $\mathsf{SSSP} \colon V_2(\mathcal{G}) \to \mathbb{R}$ such that $\mathsf{SSSP}(G, (s, v)) \coloneqq \mathsf{cost}_G(P_G(s, v))$ for $v \in V(G)$, where $P_G(s, v)$ denotes a shortest path from $s$ to $v$ in $G$. Similarly, we view the cost of the MST problem as a graph-level invariant $\mathsf{MST} \colon \mathcal{G} \to \mathbb{R}$ such that $\mathsf{MST}(G) \coloneqq \sum_{e \in E(T)} w_G(e)$ for a minimal spanning tree $T$ of $G$.

The following results highlight limitations of MPNN architectures in approximating classical graph algorithmic invariants. In particular, without individualization of nodes, e.g., marking the root node as such, no MPNN architecture can approximate the SSSP cost or the MST cost arbitrarily well. Moreover, this limitation persists for MST even when considering 1-iWL-simulating MPNNs (see Appendix B.3). Formal statements and proofs are deferred to Appendix G.

**Proposition 5** (Informal). *There does not exist an 1-WL-expressive MPNN that can approximate the invariants $\mathsf{SSSP}$ and $\mathsf{MST}$. In contrast, there exist 1-iWL- and (1,1)-WL-expressive MPNN architectures that can approximate the invariants $\mathsf{SSSP}$ and $\mathsf{MST}$, respectively.*

Here, 1-iWL and (1,1)-WL refer to MPNN architectures that

simulate individualized variants of the 1-WL refinement, as introduced in Appendix B.2. These variants support vertex marking and provide a simple mechanism for increasing expressivity beyond standard MPNNs; see Appendix G.1 for details.

### 3.2. Expressible but (possibly) not learnable invariants

Beyond expressivity limitations, there exist invariant algorithms that are representable by MPNNs but for which our sufficient conditions for learnability do not apply. In such cases, the obstruction is not a lack of expressive power, but rather the absence of a metric structure on the input space that yields finite covering numbers, as required by Definition 1.

As a simple example, consider a variant of the MPNN architecture in which the aggregation operator in Equation (1) is replaced by an unnormalized sum. Restricting to graphs without node or edge features (or, equivalently, to graphs with constant node and edge features across all vertices and edges) and to one-layer architectures, the resulting hypothesis class clearly contains the degree invariant $(G, u) \mapsto \deg_G(u)$, since node degrees are computed exactly by summing over neighbors.

However, if we don't restrict our space to graphs with bounded maximum degree, this expressivity already prevents the hypothesis class from being a finite Lipschitz class. Indeed, for any (pseudo-)metric under which the degree map is Lipschitz with a finite constant, the induced metric space necessarily has infinite covering number (for sufficiently small radius) as shown next.

**Lemma 6** (Informal). *Let $\mathcal{K}$ be the family of all complete graphs. Let $d$ be any (pseudo-)metric on $V_1(\mathcal{K})$ such that the degree invariant*

$$\deg : V_1(\mathcal{K}) \to \mathbb{N}$$

*is L-Lipschitz for some $L \in \mathbb{R}_{>0}$. Then, for every $\varepsilon \in (0, \frac{1}{L})$, $\mathcal{N}(V_1(\mathcal{K}), d, \varepsilon) = \infty$. Consequently, no hypothesis class containing the degree invariant can satisfy Definition 1 on any graph space containing $V_1(\mathcal{K})$.*

A similar obstruction applies to any invariant equivalent to the 1-WL, since its first iteration already captures the degree invariant. Thus, any hypothesis class expressive enough to represent all 1-WL invariants inherits the same covering-number pathology on graph spaces with unbounded degrees.

## 4. Limitations and future directions

While our framework provides the first general, provable learning guarantees for a broad class of graph algorithms with MPNNs, it relies on several structural assumptions that limit its scope. In particular, our theory assumes access to carefully constructed training datasets that form suitable covers of the underlying graph space. Although Theorem 2 guarantees the existence of such datasets, in full generality, the result is not constructive, and recovering the corresponding datasets may require computing $\varepsilon$-covers under graph pseudometrics, which can be computationally expensive for the classes considered in Theorem 3. In contrast, for more structured target functions such as Bellman–Ford (Theorem 4), we later obtain a genuinely constructive result that explicitly specifies the required training dataset. More broadly, developing high-probability, sampling-based methods for constructing informative training sets remains an important direction for future work. Moreover, our results are formulated in terms of achieving sufficiently small regularized training loss. Yet we do not guarantee that standard gradient-based optimization methods will reliably converge to parameter settings that generalize.

Hence, *looking forward*, bridging our learning-theoretic analysis with convergence results for gradient descent, therefore, remains an important direction for future work. Finally, our current analysis is restricted to polynomial-time algorithms; extending the framework to approximation algorithms for computationally hard problems and studying whether exploiting the data distribution can yield approximation ratios beyond worst-case guarantees constitutes another promising avenue for future research.

## 5. Experimental study

In the following, we investigate the extent to which our theoretical results translate into practice. Specifically, we answer the following questions.

**Q1** Does gradient descent converge to parameter assignments that allow for size generalization?

**Q2** Do the more expressive MPNN architectures of Section 3 lead to improved predictive performance in practice?

**Q3** Does the differentiable regularization term from Section 2.3 leads to improved generalization errors compared to $p$-norm based regularization term?

We use an MPNN aligned with the theoretical results observed in Section 2.3. Due to the construction of the training set, we conduct the experiments on the SSSP problem outlined in Section 2.3. For this, we generate synthetic training and test datasets based on Erdős–Rényi graphs and path graphs derived from Theorem 4. We then train two- and three-layer MPNNs to predict two or three steps, respectively, of the Bellman-Ford algorithm. See Appendix N for details on dataset construction, experimental settings, and additional results. The source code of all methods and evaluation procedures is available at https://github.com/

`Timo-SH/exact_nar.`

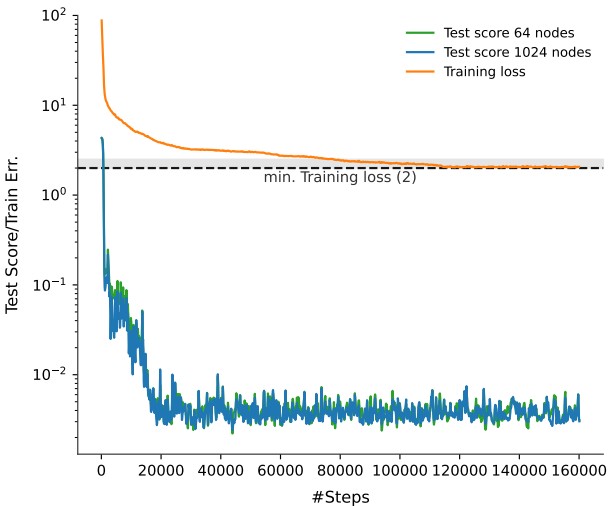

*Figure 2.* Training error and test score (lower is better) for size generalization experiments in **Q1** using test datasets with 64 and 1024 nodes, respectively. Values were smoothed using Gaussian smoothing with $\sigma = 1$. The gray region indicates the loss values for which Theorem 4 guarantees extrapolation.

**Results and discussion** In the following, we address **Q1** to **Q3**.

Regarding **Q1**, Section 5 shows size generalization properties for 64 and 1024 nodes on test graphs. The underlying MPNNs were trained on the same training dataset with a fixed edge weight for the path graphs outlined in Section 2.3. In addition, weights were uniformly sampled for test set edges. To demonstrate size generalization, we provide three test sets with increasing graph diversity. Across all test sets, test error does not increase with node count for each graph. Furthermore, the average node degree, as shown between the degree-bound Er-constdeg dataset and the unbounded ER dataset, does not affect test set performance. For the construction of the datasets, the behavior of the weight parameter, and further results, see Appendix N.

Regarding **Q2**, Table 1 shows the inability of a standard MPNN to generalize to unseen graphs in the test set at all. In addition, compared to the 1-iWL-equivalent MPNN, we observe a significantly higher training loss.

Regarding **Q3**, as seen in Table 2, replacing the proposed differentiable regularization term with a $p$-norm based regularization leads to similar results using $\ell_1$ regularization. While $\ell_1$ and $\ell_2$ norms allow for sufficient training and generalization results, size generalization improves with the proposed regularization term across graph sizes. Furthermore, with $\ell_2$ regularization, training behavior is noticeably less stable and yields slightly worse test-set performance. These results align with Theorem 4, highlighting the application

*Table 1.* Test score results for **Q2** with a standard and 1-iWL-equivalent MPNNs. All results were obtained across three seeds, and test scores are averaged. The number in brackets indicates the number of nodes for the test graphs.

| | Dataset | | |
|---|---|---|---|
| **Task** (Test score ↓) | ER-CONSTDEG | ER | GENERAL |
| 1-WL (64) | $0.9725 \pm 0.0001$ | $0.9725 \pm 0.0001$ | $0.8393 \pm 0.0001$ |
| 1-iWL (64) | $0.0035 \pm 0.0002$ | $0.0034 \pm 0.0004$ | $0.0032 \pm 0.0002$ |
| 1-WL (256) | $0.9896 \pm 0.0006$ | $0.9765 \pm 0.0001$ | $0.8368 \pm 0.0010$ |
| 1-iWL (256) | $0.0033 \pm 0.0002$ | $0.0037 \pm 0.0001$ | $0.0033 \pm 0.0002$ |
| 1-WL (1024) | $0.9948 \pm 0.0005$ | $0.9645 \pm 0.0001$ | $0.8217 \pm 0.0010$ |
| 1-iWL (1024) | $0.0030 \pm 0.0001$ | $0.0038 \pm 0.0002$ | $0.0033 \pm 0.0002$ |

*Table 2.* Comparison of regularization between our $\ell_1$ based method (here named $\ell_{\mathrm{reg}}$), $\ell_1$ and $\ell_2$ regularization terms. Across all experiments, $\eta = 0.1$ holds, and the MPNN from **Q1** was used. Furthermore, the General dataset as outlined in Appendix N is used for all experiments. Results are obtained across three seeds.

| | Nodes | | | | |
|---|---|---|---|---|---|
| **Reg.** (↓) | 64 | 128 | 256 | 512 | 1024 |
| $\ell_1$ | $0.0061 \pm 0.0010$ | $0.0053 \pm 0.0011$ | $0.0054 \pm 0.0010$ | $0.0060 \pm 0.0009$ | $0.0056 \pm 0.0009$ |
| $\ell_2$ | $0.0336 \pm 0.0503$ | $0.0336 \pm 0.0500$ | $0.0343 \pm 0.0510$ | $0.0133 \pm 0.0123$ | $0.0369 \pm 0.0557$ |
| $\ell_{\mathrm{reg}}$ | $0.0032 \pm 0.0002$ | $0.0033 \pm 0.0002$ | $0.0033 \pm 0.0002$ | $0.0033 \pm 0.0003$ | $0.0033 \pm 0.0002$ |

of a differentiable regularization term to the SSSP problem, opposed to a non-differentiable regularization previously required.

## 6. Conclusion

We developed a general theoretical framework for learning graph algorithms with GNNs. Our framework characterizes the conditions under which an MPNN, or more expressive variants, can be trained on a finite set of instances and provably generalize to inputs of arbitrary size by minimizing a supervised, regularized loss. By connecting algorithmic learning to notions of metric structure, covering numbers, and regularization-induced extrapolation, we move beyond purely expressivity-based analyses and provide learning-theoretic guarantees for neural algorithmic reasoning on graphs. Building on this framework, we identified broad classes of graph algorithms, ranging from shortest paths to dynamic programming problems, that suitably expressive GNNs can learn, and we also identified fundamental limitations of standard MPNNs. For the single-source shortest-path problem, we further showed how a differentiable $p$-norm-based regularization significantly reduces the size of the required training set. *In summary, our results provide a precise characterization of which algorithms GNNs can learn from finite data and which they cannot, thereby enabling a more principled understanding of data-driven algorithmic design and its potential for reliable generalization beyond the training regime.*

## Acknowledgement

SW and AV are supported by the German Research Foundation (DFG) within Research Training Group 2236/2 (UnRAVeL). SW is supported by the DFG Heinz Maier-Leibnitz award. TS and CM are partially funded by a DFG Emmy Noether grant (468502433) and RWTH Junior Principal Investigator Fellowship under Germany's Excellence Strategy. RN and YW are supported by US National Science Foundation (NSF) grant CCF-2112665

## Impact statement

This paper presents work whose goal is to advance the field of machine learning. There are many potential societal consequences of our work, none of which we feel must be specifically highlighted here.

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

## A. Related work

In the following, we discuss related work.

**MPNNs** MPNNs (Gilmer et al., 2017; Scarselli et al., 2009) emerged as the most prominent graph machine learning architecture. Notable instances of this architecture include, e.g., Duvenaud et al. (2015); Hamilton et al. (2017); Kipf & Welling (2017) and Veličković et al. (2018), which can be subsumed under the message-passing framework introduced in Gilmer et al. (2017). In parallel, approaches based on spectral information were introduced in, e.g., Bruna et al. (2014); Defferrard et al. (2016); Gama et al. (2019); Kipf & Welling (2017); Levie et al. (2019), and Monti et al. (2017)—all of which descend from early work in Baskin et al. (1997); Goller & Küchler (1996); Kireev (1995); Merkwirth & Lengauer (2005); Micheli & Sestito (2005); Micheli (2009); Scarselli et al. (2009), and Sperduti & Starita (1997).

**Expressivity of MPNNs** The *expressivity* of an MPNN is the architecture's ability to express or approximate different functions over a set of graphs. High expressivity means the neural network can represent many functions over this domain. In the literature, the expressivity of MPNNs is modeled mathematically using two main approaches: separation power compared to graph isomorphism test (Morris, 2021), and universal approximation theorems (Azizian & Lelarge, 2021; Geerts & Reutter, 2022). Works following the first approach study if an MPNN, by choosing appropriate weights, can distinguish the same pairs of non-isomorphic graphs with a given graph isomorphism test. The most commonly used graph isomorphism test for analyzing the expressive power of MPNNs is the 1-*dimensional Weisfeiler–Leman algorithm* 1-WL, a well-studied heuristic for the graph isomorphism problem, and its more expressive variants. Here, an MPNN distinguishes two non-isomorphic graphs if it can compute distinct vector representations for them. Specifically, Morris et al. (2019) and Xu et al. (2019) showed that the 1-WL limits the expressive power of any possible MPNN architecture in distinguishing non-isomorphic graphs. In turn, these results have been generalized to the $k$-dimensional Weisfeiler–Leman algorithm, e.g., Azizian & Lelarge (2021); Geerts (2020); Maron et al. (2019); Morris et al. (2019; 2020b; 2022) and (ordered) subgraph GNNs Bevilacqua et al. (2022); Cotta et al. (2021); Li et al. (2020); Qian et al. (2022); Zhang et al. (2023). Works following the second approach, which operates over a set of graphs, can be approximated arbitrarily closely by an MPNN (Azizian & Lelarge, 2021; Böker et al., 2023; Chen et al., 2019; Geerts & Reutter, 2022; Maehara & NT, 2019).

**Generalization abilities of MPNNs** Early work by Scarselli et al. (2018), building on classical learning theory (Karpinski & Macintyre, 1997; Vapnik, 1995), bounded the Vapnik–Chervonenkis theory dimension of MPNNs with piecewise polynomial activations on fixed graphs by $\mathcal{O}(P^2 n \log n)$, where $P$ is the number of parameters and $n$ the graph's order; see also Hammer (2001). However, their MPNN model differs from modern architectures (Gilmer et al., 2017; D'Inverno et al., 2025). Garg et al. (2020) bounded the empirical Rademacher complexity of a simple sum-aggregation MPNN in terms of graph degree, depth, Lipschitz constants, and parameter norms, assuming weight sharing. This line was extended to $E(n)$-equivariant MPNNs by Karczewski et al. (2024) and refined via PAC-Bayesian analyses by Liao et al. (2021); Ju et al. (2023); see also Lee et al. (2024) for knowledge graphs. Morris et al. (2023a) connected MPNNs' expressivity and generalization via the Vapnik–Chervonenkis theory, showing that VC dimension depends on the number of 1-WL equivalence classes, logarithmically on the number of colors, and polynomially on the number of parameters. Their discrete pseudo-metric assumption was extended by Pellizzoni et al. (2024), who studied node-individualized MPNNs using covering numbers, though without explicit metric bounds. Related refinements include VC lower bounds for restricted MPNNs (Daniëls & Geerts, 2024), margin-based analyses (Franks et al., 2024; Li et al., 2024; Chuang et al., 2021), and more expressive MPNNs (Franks et al., 2024; Maskey et al., 2025). Several works analyze generalization under structural assumptions. Maskey et al. (2022; 2024); Wang et al. (2025) considered random graph models, while Levie (2023); Rauchwerger et al. (2024); Vasileiou et al. (2024a) derived bounds using covering numbers. Transductive generalization was studied via algorithmic stability (Verma & Zhang, 2019) and Rademacher complexity under stochastic block models (Esser et al., 2021; Tang & Liu, 2023). For semi-supervised node classification, Baranwal et al. (2021) analyzed MPNNs on mixtures of Gaussians over stochastic block models. Importantly, the above work analyzed MPNNs' generalization ability in the classical uniform convergence regime. In contrast, the present work examines the generalization of a single function to larger graphs than those seen during training. Yehudai et al. (2021) derived negative generalization results for larger graphs than those seen in the training set, while Levin et al. (2025) derived necessary conditions of generalization to larger graphs. See Vasileiou et al. (2024b) for a survey on generalization analyses of MPNNs and related architectures.

**Empirical work on NAR on graphs** A large body of empirical work studies NAR for graphs problems (Velickovic & Blundell, 2021; Cappart et al., 2023). Early empirical studies demonstrated that neural architectures can imitate algorithmic execution when trained on intermediate computation traces (Velickovic et al., 2020). MPNNs have emerged as a particularly effective backbone for NAR on graph-structured problems due to their close correspondence with local, iterative graph

computations (Cappart et al., 2023). Empirical results show that MPNNs can learn graph algorithms such as single-source shortest paths, breadth-first search, minimum spanning trees, as well as dynamic programming-style problems such as knapsack (Pándy et al., 2022; Velickovic et al., 2020; Yonetani et al., 2021; Požgaj et al., 2025). Subsequent work investigated architectural refinements and training strategies to improve stability and generalization, e.g., (Grötschla et al., 2022; Ibarz et al., 2022; Jain et al., 2023; Jürß et al., 2023; Numeroso et al., 2023; Rodionov & Prokhorenkova, 2025; Xhonneux et al., 2021). Surveys (Cappart et al., 2023) and benchmark studies (Velickovic et al., 2022) further systematized these empirical findings and highlighted both the potential and the limitations of machine-learning-enhanced approaches to algorithmic reasoning and combinatorial optimization.

**Theoretical work on algorithmic reasoning on graphs** There is a substantial body of work studying the expressive capabilities of message-passing neural networks (MPNNs) and related architectures for representing graph algorithms. For example, Loukas (2020) studied the depth and width requirements of MPNNs for solving problems such as minimum vertex cover, leveraging results from distributed computing. Xu et al. (2019); Morris et al. (2019) showed that MPNNs are inherently limited by the 1-WL test in their ability to distinguish non-isomorphic graphs. Qian et al. (2024a) demonstrated that MPNNs can express each step of the primal–dual interior-point method for solving linear optimization problems, while Yau et al. (2024) investigated their ability to represent approximation algorithms for hard combinatorial problems such as maximum cut and minimum vertex cover; see also Sato et al. (2019). Dudzik & Velickovic (2022) used category-theoretic tools to establish a connection between MPNNs and dynamic programming. Hertrich & Skutella (2023); Hertrich & Sering (2025) studied the size requirements of recurrent neural networks for solving knapsack and maximum-flow problems. More recently, Rosenbluth & Grohe (2025) devised a general framework for understanding the ability of recurrent MPNNs to simulate algorithms on arbitrarily large instances, and He & Vitercik (2025) showed that MPNNs can simulate classical primal–dual approximation schemes. Recently, several works have begun to study the ability of transformer architectures to simulate (graph) algorithms, e.g., de Luca & Fountoulakis (2024); de Luca et al. (2025); Merrill & Sabharwal (2025); Sanford et al. (2024b;a); Yehudai et al. (2025); Zhou et al. (2024). Overall, these works primarily focus on expressivity, largely ignoring questions related to learning and optimization.

Xu et al. (2020) addressed this gap by proposing PAC-style sample-complexity bounds for learning common (graph) algorithms using MPNNs, demonstrating that architectural alignment with the target algorithm can improve sample efficiency. Nerem et al. (2025) demonstrated that MPNNs trained on small datasets, equipped with a regularization term, and optimized to a sufficiently small loss can execute the Bellman–Ford algorithm on arbitrarily large graphs. However, their analysis crucially relies on a non-differentiable regularization term.

# B. Extended background

Here, give additional background.

**Graphs** An *(undirected) graph* $G$ is a pair $(V(G), E(G))$ with *finite* sets of *vertices* $V(G)$ and *edges* $E(G) \subseteq \{\{u,v\} \subseteq V(G) \mid u \neq v\}$. *vertices* or *nodes* $V(G)$ and *edges* $E(G) \subseteq \{\{u,v\} \subseteq V(G) \mid u \neq v\}$. The *order* of a graph $G$ is its number $|V(G)|$ of vertices. We call $G$ an *n-order graph* if $G$ has order $n$. In a *directed graph*, we define $E(G) \subseteq V(G)^2$, where each edge $(u,v)$ has a direction from $u$ to $v$. Given a directed graph $G$ and vertices $u, v \in V(G)$, we say that $v$ is a *child* of $u$ if $(u,v) \in E(G)$. For a graph $G$ and an edge $e \in E(G)$, we denote by $G \setminus e$ the *graph induced by removing* the edge $e$ from $G$. For an $n$-order graph $G$, assuming $V(G) = [n]$, we denote its *adjacency matrix* by $\boldsymbol{A}(G) \in \{0,1\}^{n \times n}$, where $\boldsymbol{A}(G)_{vw} = 1$ if, and only if, $\{v,w\} \in E(G)$. The *neighborhood* of a vertex $v \in V(G)$ is denoted by $N_G(v) \coloneqq \{u \in V(G) \mid \{v,u\} \in E(G)\}$, where we usually omit the subscript for ease of notation.

An *attributed graph* is a pair $(G, a_G)$ with a graph $G = (V(G), E(G))$ and a (vertex-)attribute function $a_G \colon V(G) \to \mathbb{R}^{1 \times d}$, for $d > 0$. The *attribute* or *feature* of $v \in V(G)$ is $a_G(v)$. Similarly, we consider graphs equipped with edge features. An *edge-featured graph* is a pair $(G, w_G)$, where $G = (V(G), E(G))$ is a graph and $w_G \colon E(G) \to \mathbb{R}^{1 \times d}$ assigns a (possibly vector-valued) feature to each edge. For an edge $e \in E(G)$, the vector $w_G(e)$ is referred to as the *edge feature* of $e$. The special case $p = 1$ with $w_G(e) \in \mathbb{R}^+$ for all $e \in E(G)$ corresponds to an *edge-weighted graph*, in which case $w_G(e)$ is called the *(edge) weight* of $e$. When the underlying graph is clear from the context, we simply write $w(e)$ or $w_e$ to denote the edge feature of $e \in E(G)$.

For a graph $G$ without edge features, the *degree* of a node $u \in V(G)$ is defined as $\deg_G(u) \coloneqq |N_G(u)|$. For an edge-weighted graph $(G, w_G)$, the *weighted degree* of $u \in V(G)$ is given by $\deg_G(u) \coloneqq \sum_{v \in N_G(u)} w_G(u, v)$. When the underlying graph is clear from the context, we omit the subscript $G$ and simply write $\text{degree}(u)$.

Let $G$ be graph, a path $P$ on $G$ of *length* $k$ is a sequence of vertices $(v_0, v_1, v_2, \ldots, v_k)$ such that for $i \in [k]$, it holds that $(v_{i-1}, v_i) \in E(G)$. We denote the set of paths between vertices $v, w \in V(G)$ by $\mathcal{P}_G(v, w)$. A graph is *connected* if $\mathcal{P}(v, w) \neq \emptyset$, for all $v, w \in V(G)$. A graph $G$ is a *tree* if it is connected, but $G \setminus e$ is disconnected for any $e \in E(G)$. A tree or a disjoint collection of trees is known as a forest.

A *rooted tree* $(G, r)$ is a tree where a specific vertex $r$ is marked as the *root*. For a rooted (undirected) tree, we can define an implicit direction on all edges as pointing away from the root; thus, when we refer to the *children* of a vertex $u$ in a rooted tree, we implicitly consider this directed structure. For $S \subseteq V(G)$, the graph $G[S] := (S, E_S)$ is the *subgraph induced by* $S$, where $E_S := \{(u, v) \in E(G) \mid u, v \in S\}$. A *(vertex-)labeled graph* is a pair $(G, \ell_G)$ with a graph $G = (V(G), E(G))$ and a (vertex-)label function $\ell_G \colon V(G) \to \Sigma$, where $\Sigma$ is an arbitrary countable label set. For a vertex $v \in V(G)$, $\ell_G(v)$ denotes its *label*.

Two graphs $G$ and $H$ are *isomorphic* if there exists a bijection $\varphi \colon V(G) \to V(H)$ that preserves adjacency, i.e., $(u, v) \in E(G)$ if and only if $(\varphi(u), \varphi(v)) \in E(H)$. The bijection $\varphi$ is called an isomorphism. In the case of attributed graphs, we additionally require $a_G(v) = a_H(\varphi(v))$ for all $v \in V(G)$, and similarly for edge- or feature-labeled graphs. More generally, for $k \geq 1$, and $(G, \boldsymbol{v}), (H, \boldsymbol{w}) \in V_k(\mathcal{G})$ for some graph space $\mathcal{G}$, we say that $(G, \boldsymbol{v})$, and $(H, \boldsymbol{w})$ are isomorphic if there exists an isomorphism $\varphi \colon V(G) \to V(H)$ with $\varphi(\boldsymbol{v}) = \boldsymbol{w}$ (applied componentwise). Given two graphs $G$ and $H$ with disjoint vertex sets, we denote their disjoint union by $G \mathbin{\dot\cup} H$.

## B.1. Node-level and graph-level MPNN classes

Since Equation (1) and Equation (2) are parametrized functions, we can define function classes of MPNNs that operate at the node and graph levels. Let $\mathcal{G}_n$ be a set of $n$-order graphs, and let $L > 0$, $d > 0$. Furthermore, let $\mathcal{S}_L := (\mathsf{UPD}^{(1)}, \mathsf{AGG}^{(1)}, \ldots, \mathsf{UPD}^{(L)}, \mathsf{AGG}^{(L)})$ be a sequence of parameterized functions following Equation (1) and $\mathcal{P}_L := (\boldsymbol{U}_1, \boldsymbol{A}_1, \ldots, \boldsymbol{U}_L, \boldsymbol{A}_L)$ be a corresponding sets of parameters. We then define

$$\mathsf{MPNN}^{\mathcal{P}_L}_{(\mathcal{S}_L, d, n)}(\mathcal{G}_n) := \left\{ h \colon V_1(\mathcal{G}_n) \to \mathbb{R}^{n \times d} \;\middle|\; h(G)_v = \boldsymbol{h}_v^{(t)}, G \in \mathcal{X}, \text{ where } \boldsymbol{u}_t \in \boldsymbol{U}_t, \boldsymbol{a}_t \in \boldsymbol{A}_t \right\}.$$

We call such a set of functions a *node-level MPNN class*. Similarly, let $\mathcal{T}_L := (\mathsf{UPD}^{(1)}, \mathsf{AGG}^{(1)}, \ldots, \mathsf{UPD}^{(L)}, \mathsf{AGG}^{(L)}, \mathsf{READOUT})$ be a sequence of parameterized functions following Equations (1) and (2) and $\mathcal{Q}_L := (\boldsymbol{U}_1, \boldsymbol{A}_1, \ldots, \boldsymbol{U}_L, \boldsymbol{A}_L, \boldsymbol{R})$ be a corresponding set of parameters. We then define

$$\mathsf{MPNN}^{\mathcal{Q}_L}_{(\mathcal{T}_L, d)}(\mathcal{G}_n) := \left\{ h \colon \mathcal{G}_n \to \mathbb{R} \;\middle|\; h(G) := \boldsymbol{h}_G, G \in \mathcal{X}, \text{ where } \boldsymbol{u}_t \in \boldsymbol{U}_t, \boldsymbol{a}_t \in \boldsymbol{A}_t, \text{ and } \boldsymbol{r} \in \boldsymbol{R} \right\}.$$

We call such a set of functions a *graph-level MPNN class*. We call a concrete choice of parameters, e.g., $((\boldsymbol{u}_t, \boldsymbol{a}_t)_{t \in [L]}, \boldsymbol{r})$ of an graph-level MPNN architecture *parametrization*.

## B.2. The $1$-dimensional Weisfeiler–Leman algorithm and variants

Here, we introduce the 1-dimensional Weisfeiler–Leman algorithm and some variants.

**The $1$-dimensional Weisfeiler–Leman algorithm** The $1$-*dimensional Weisfeiler–Leman algorithm* (1-$\mathsf{WL}$) or *color refinement* is a well-studied heuristic for the graph isomorphism problem, originally proposed by Weisfeiler & Leman (1968).[3] Intuitively, the algorithm determines if two graphs are non-isomorphic by iteratively coloring or labeling vertices. Given an initial coloring or labeling of the vertices of both graphs, e.g., their degree or application-specific information, in each iteration, two vertices with the same label get different labels if the number of identically labeled neighbors is unequal. These labels induce a vertex partition, and the algorithm terminates when, after some iterations, it does not refine the current partition, i.e., when a *stable coloring* or *stable partition* is obtained. Then, if the number of vertices with a specific label differs between the two graphs, we can conclude that the graphs are not isomorphic. It is easy to see that the algorithm cannot distinguish all non-isomorphic graphs (Cai et al., 1992). However, it is a powerful heuristic that can successfully decide isomorphism for a broad class of graphs (Arvind et al., 2015; Babai & Kucera, 1979).

In the following, we formally describe a variant of the 1-$\mathsf{WL}$ that also considers edge weights. Formally, let $(G, \ell_G)$ be a labeled graph and let $w_G \colon E(G) \to \mathbb{R}$ be an edge-weight function for $G$. In each iteration, $t > 0$, the 1-$\mathsf{WL}$ computes a

---

[3]Strictly speaking, the 1-$\mathsf{WL}$ and color refinement are two different algorithms. That is, the 1-$\mathsf{WL}$ considers neighbors and non-neighbors to update the coloring, resulting in a slightly higher expressive power when distinguishing vertices in a given graph; see Grohe (2021) for details. Following the conventions in the machine learning literature, we treat both algorithms as equivalent.

*vertex coloring* $C_t^1 \colon V(G) \to \mathbb{N}$, depending on the coloring of the neighbors and the weights of the incident edges. That is, in iteration $t > 0$, we set

$$C_t^1(v) \coloneqq \mathsf{RELABEL}\Big(\big(C_{t-1}^1(v), \{\!\{(C_{t-1}^1(u), w_G(u,v)) \mid u \in N(v)\}\!\}\big)\Big),$$

for vertex $v \in V(G)$, where RELABEL injectively maps the above pair to a unique natural number, which has not been used in previous iterations. In iteration 0, the coloring $C_0^1 \coloneqq \ell_G$ is used.[4] To test whether two graphs $G$ and $H$ are non-isomorphic, we run the above algorithm in "parallel" on both graphs. If the two graphs have a different number of vertices colored $c \in \mathbb{N}$ at some iteration, the 1-WL *distinguishes* the graphs as non-isomorphic. Moreover, if the number of colors between two iterations, $t$ and $(t+1)$, does not change, i.e., the cardinalities of the images of $C_t^1$ and $C_{i+t}^1$ are equal, or, equivalently,

$$C_t^1(v) = C_t^1(w) \iff C_{t+1}^1(v) = C_{t+1}^1(w),$$

for all vertices $v, w \in V(G \dot\cup H)$, then the algorithm terminates. For such $t$, we define the *stable coloring* $C_\infty^1(v) = C_t^1(v)$, for $v \in V(G \dot\cup H)$. The stable coloring is reached after at most $\max\{|V(G)|, |V(H)|\}$ iterations (Grohe, 2017).

It is straightforward to show that the 1-WL has limited expressivity in distinguishing pairs of non-isomorphic graphs. Hence, in the following, we derive two more expressive variants that allow us to characterize the needed expressivity to capture well-known graph algorithms.

**The 1-dimensional Weisfeiler–Leman algorithm on individualized graphs** We consider the 1-WL on *individualized graphs*, i.e., graphs equipped with a distinguished vertex. Intuitively, given a root vertex $r$, we *individualize* $r$ by assigning it a unique initial label and then run the standard 1-WL refinement. Formally, let $(G, w_G)$ be an edge-weighted graph with uniform vertex labels $\ell_G$, and let $r \in V(G)$. We define initial labels by setting $\ell_G(r) \coloneqq [*]$, where $[*]$ is a fresh label not used for any other vertex. Then, for each iteration $t > 0$, the algorithm computes a coloring $C_t^{1,r} \colon V(G) \to \mathbb{N}$ by the usual 1-WL update rule, i.e., $C_t^{1,r}$ is obtained from $C_{t-1}^{1,r}$ by aggregating the multiset of neighbor colors together with the incident edge weights, exactly as in Section 1. Equivalently, $C_0^{1,r}$ is the coloring induced by $\ell_G$ with $r$ labeled $[*]$.

For two edge-weighted graphs $(G, w_G)$ and $(H, w_H)$ with individualized vertices $v \in V(G)$ and $w \in V(H)$, we say that 1-WL *distinguishes* the (individualized) graphs $(G, v)$ and $(H, w)$ if, when running the above refinement in parallel on $(G, v)$ and $(H, w)$, the resulting color multisets differ at some iteration (analogously to the usual 1-WL notion of distinction). We write 1-iWL when 1-WL is used on individualized graphs.

**The 1.1-dimensional Weisfeiler–Leman algorithm** The 1.1-*dimensional Weisfeiler–Leman algorithm* (1,1)-WL (Rattan & Seppelt, 2023; Qian et al., 2024b) can be seen as an extension of the 1-iWL, which tries every possible placement for the unique label $[*]$ and runs the 1-iWL in parallel on the disjoint union of these individualized graphs. Formally, the (1,1)-WL does *not distinguish* a pair of graphs $(G, H)$ if there exists a bijection $\pi \colon V(G) \to V(H)$ such that, for $v \in V(G)$, running the 1-iWL in "parallel" on $G$, with $v$ individualized, and $H$, with $\pi(v)$ individualized, does not distinguish between the two graphs.

Observe that we can define 1-tuple or graph-level invariants based on these 1-WL variants.

## B.3. Separation and approximation abilities of MPNNs

Morris et al. (2019) and Xu et al. (2019) established that the graph-distinguishing power of any MPNN architecture is upper bounded by the 1-WL test. Moreover, for MPNNs with sum aggregation, Morris et al. (2019) showed that, on any finite set of graphs, suitable parameter choices yield expressivity matching that of 1-WL; see Grohe (2021) and Morris et al. (2023b) for further discussion. Analogous statements can be lifted to the 1-iWL and (1,1)-WL settings.

To formalize distinguishability, following Azizian & Lelarge (2021) we express the ability of a function class to distinguish graphs via an induced equivalence relation. Let $\mathcal{G}$ be a set of graphs, and let $\mathcal{F}$ be a class of functions $f \colon \mathcal{G} \to \mathbb{D}$ for some domain $\mathbb{D}$. We define the equivalence relation $\rho_\mathcal{G}(\mathcal{F})$ on $\mathcal{G}$ by

$$(G, H) \in \rho_\mathcal{G}(\mathcal{F}) \iff f(G) = f(H) \text{ for all } f \in \mathcal{F}.$$

In case $(G, H) \in \rho_\mathcal{G}(\mathcal{F})$ we say that $\mathcal{F}$ cannot distinguish $G$ and $H$. If $\mathcal{F} = \{f\}$ is a singleton, we write $\rho(f)$ instead of $\rho(\{f\})$.

---

[4]Here, we implicitly assume an injective function from $\Sigma$ to $\mathbb{N}$.

We extend the definition to $k$-tuple invariants as follows. Let $V_k(\mathcal{G})$ denote the set of pairs $(G, \boldsymbol{v})$ with $G \in \mathcal{G}$ and $\boldsymbol{v} \in V(G)^k$. Furthermore, we let $V_0(\mathcal{G}) = \mathcal{G}$. For a given $k \in \mathbb{N}$ and for a class $\mathcal{F}$ of functions $f \colon V_k(\mathcal{G}) \to \mathbb{D}$, for some domain $\mathbb{D}$, define the equivalence relation $\rho_{V_k(\mathcal{G})}(\mathcal{F})$ on $V_k(\mathcal{G})$ by

$$(G, \boldsymbol{v}, H, \boldsymbol{w}) \in \rho_{V_k(\mathcal{G})}(\mathcal{F}) \iff f(G, \boldsymbol{v}) = f(H, \boldsymbol{w}) \text{ for all } f \in \mathcal{F}$$

We can now rephrase the various notions of distinguishability from Appendix B.2, as follows. Let $\mathcal{G}$ be a set of graphs.

$$\rho_{V_1(\mathcal{G})}(\text{1-WL}) = \left\{ (G, v, H, w) \in (V_1(\mathcal{G}))^2 \mid C_\infty^1(v) = C_\infty^1(w) \right\},$$
$$\rho_{\mathcal{G}}(\text{1-WL}) = \left\{ (G, H) \in \mathcal{G}^2 \mid \exists \pi : V(G) \to V(H), \right.$$
$$\left. \forall v \in V(G) : (G, v, H, \pi(v)) \in \rho_{V_1(\mathcal{G})}(\text{1-WL}) \right\},$$
$$\rho_{V_2(\mathcal{G})}(\text{1-iWL}) = \left\{ (G, (r, v), H, (s, w)) \in (V_2(\mathcal{G}))^2 \mid C_\infty^{1.5, r}(v) = C_\infty^{1.5, s}(w) \right\},$$
$$\rho_{V_1(\mathcal{G})}(\text{1-iWL}) = \left\{ (G, r, H, s) \in (V_1(\mathcal{G}))^2 \mid \exists \pi : V(G) \to V(H), \; \forall v \in V(G), \right.$$
$$\left. (G, (r, v), H, (s, \pi(v))) \in \rho_{V_2(\mathcal{G})}(\text{1-iWL}) \right\},$$
$$\rho_{\mathcal{G}}((1,1)\text{-WL}) = \left\{ (G, H) \in \mathcal{G}^2 \mid \exists \pi : V(G) \to V(H), \; \forall v \in V(G), \; (G, v, H, \pi(v)) \in \rho_{V_1(\mathcal{G})}(\text{1-iWL}) \right\},$$

where $\pi : V(G) \to V(H)$ denotes a bijection. We remark that for 1-iWL, we interpret an element $(G, v) \in V_1(\mathcal{G})$ as graphs $G \in \mathcal{G}$ in which $v$ is individualized.

Given any $\text{alg} \in \{\text{1-WL}, \text{1-iWL}, (1,1)\text{-WL}\}$ and class of functions $\mathcal{F} : V_k(\mathcal{G}) \to \mathbb{D}$ for some $k \in \mathbb{N}$ and domain $\mathbb{D}$, we say that $\mathcal{F}$ is *alg-simulating* if and only if

$$\rho_{V_k(\mathcal{G})}(\mathcal{F}) = \rho_{V_k(\mathcal{G})}(\text{alg}).$$

That is, the distinguishing power of $\mathcal{F}$ is precisely that of alg. With this notation, the expressiveness result from Morris et al. (2019) can be stated as

$$\rho_{\mathcal{G}}(\text{MPNN}) = \rho_{\mathcal{G}}(\text{1-WL}) \quad \text{and} \quad \rho_{V_1(\mathcal{G})}(\text{MPNN}) = \rho_{V_1(\mathcal{G})}(\text{1-WL}),$$

where we abuse notation and let MPNN denote both the class of all graph-level MPNNs (for $\rho_{\mathcal{G}}(\text{MPNN})$) and the class of all node-level MPNNs (for $\rho_{V_1(\mathcal{G})}(\text{MPNN})$), see also Section 1.2. As noted above, one can similarly verify the existence of MPNN variants that are 1-iWL-simulating (at rooted/tuple-level) and $(1,1)$-WL-simulating (at graph-level).

Finally, Azizian & Lelarge (2021) and Geerts & Reutter (2022) showed that, under mild regularity assumptions, separation entails approximation. We remark that these approximations require fixing the order of the underlying graphs and restricting features to a compact domain.

**Proposition 7.** *Let $\mathcal{G}$ be the set of attributed graphs with $n$ vertices and node/edge attributes taking values in a compact set of $\mathbb{R}^d$. Let $\text{alg} \in \{1\text{-WL}, 1\text{-iWL}, (1,1)\text{-WL}\}$, and let $g : V_k(\mathcal{G}) \to \mathbb{R}$ be a $k$-tuple invariant such that $\rho_{V_k(\mathcal{G})}(\text{alg}) \subseteq \rho_{V_k(\mathcal{G})}(g)$. Assuming standard technical conditions on a class of MPNNs that simulates $\rho_{V_k(\mathcal{G})}(\text{alg})$, for every $\epsilon > 0$ there exists an MPNN $m$ such that*

$$\sup_{(G, \boldsymbol{v}) \in V_k(\mathcal{G})} |g(G, \boldsymbol{v}) - m(G, \boldsymbol{v})| < \epsilon.$$

In other words, whenever an invariant alg has enough information to distinguish everything that matters for the function $g$ of interest, an alg-simulating class of MPNNs can approximate $g$ arbitrarily well. We refer to Appendix H.1 for details.

**Feedforward neural networks** Let $J \in \mathbb{N}$ and $(d_0, \ldots, d_J) \in \mathbb{N}^{J+1}$. Given weights $\mathcal{W} = (\boldsymbol{W}^1, \ldots, \boldsymbol{W}^J) \in \prod_{i=1}^{J} \mathbb{R}^{d_i \times d_{i-1}} =: \boldsymbol{\Theta}_W$, biases $\mathcal{B} = (\boldsymbol{b}^1, \ldots, \boldsymbol{b}^J) \in \prod_{i=1}^{J} \mathbb{R}^{d_i} =: \boldsymbol{\Theta}_b$, and $j \in [J]_0$ define the *feed-forward neural network* (FNN) with parameters $(\mathcal{W}, \mathcal{B})$ up to layer $j$, as the map $\text{FNN}_j^{(J)}(\mathcal{W}, \mathcal{B}) \colon \mathbb{R}^{d_0} \to \mathbb{R}^{d_j}$ such that

$$\text{FNN}_j^{(J)}(\mathcal{W}, \mathcal{B})(\boldsymbol{x}) := \sigma\left( \boldsymbol{W}^{(j)} \cdots \sigma\left( \boldsymbol{W}^{(2)} \sigma\left( \boldsymbol{W}^{(1)} \boldsymbol{x} + \boldsymbol{b}^{(1)} \right) + \boldsymbol{b}^{(2)} \right) \cdots + \boldsymbol{b}^{(j)} \right) \in \mathbb{R}^{d_j},$$

for $\boldsymbol{x} \in \mathbb{R}^{d_0}$. Here, the function $\sigma \colon \mathbb{R} \to \mathbb{R}$ is an *activation function*, applied component-wisely, e.g., a *rectified linear unit* (ReLU), where $\sigma(x) := \max(0, x)$. Further we will write $\boldsymbol{\theta} := (\mathcal{W}, \mathcal{B}) \in \boldsymbol{\Theta} := \boldsymbol{\Theta}_W \times \boldsymbol{\Theta}_b$ to denote the whole parameter set. In case $j = J$, we denote the $J$-layer *feed-forward neural network* by

$$\text{FNN}^{(J)}(\boldsymbol{\theta})(\boldsymbol{x}) := \text{FNN}_J^{(J)}(\boldsymbol{\theta})(\boldsymbol{x}).$$

## B.4. Considered graph problems

In the following, we formally introduce the studied graph problems, namely, the *single-source shortest path* (SSSP) problem, the *shortest path-finding* (SPF) problem, the *minimum spanning tree* (MST) problem, and the *knapsack problem*.

*Solving the SSSP problem*, given an edge-weighted graph $(G, w_G)$ and a *source vertex* $s \in V(G)$, amounts to finding the *shortest path* from the *source vertex* $s$ to all other vertices in the graph $G$. That is, for a vertex $v \in V(G)$, we aim to find a path

$$P_G^*(s, v) \coloneqq \arg \min_{P \in \mathcal{P}_G(s,v)} \sum_{e \in P} w_G(e).$$

The *cost* $\mathsf{cost}_G(P)$ of a (shortest) path $P$ is $\sum_{e \in P} w_G(e)$. Given an edge-weighted graph $(G, w_G)$, *determining the cost of the SSSP problem* amounts to determining the cost of a shortest path $P_G^*(s, v)$ from the source vertex $s$ to $v$, for all $v \in V(G)$.

Similarly, *solving the SPF problem*, given an edge-weighted graph $(G, w_G)$ a source vertex $s \in V(G)$, a *target vertex* $t \in V(G)$, amounts to finding the shortest path from the source vertex $s$ to the target vertex $t$. That is, we aim to find a path

$$P_G^*(s, t) \coloneqq \arg \min_{P \in \mathcal{P}_G(s,t)} \sum_{e \in P} w_G(e).$$

Given an edge-weighted graph $(G, w_G)$ *determining the cost of the SPF problem* amounts to determining the cost of a shortest path $P_G^*(s, t)$ from the source vertex $s$ to the target vertex $t$.

*Solving the MST problem*, given an edge-weighted graph $(G, w_G)$, amounts to finding a tree over all vertices in the graph $G$ with minimum overall edge weight, the *minimum spanning tree*. That is, we aim to find a tree

$$T_G^* \coloneqq \arg \min_{\substack{V(T)=V(G) \\ T \text{ is a tree.}}} \sum_{e \in E(T)} w_G(e).$$

The *cost* $\mathsf{cost}(T)$ of a (minimum) spanning tree $T$ is $\sum_{e \in T(E)} w_G(e)$. Given an edge-weighted graph $(G, w_G)$, *determining the cost of the MST* amounts to determining the cost of a minimum spanning tree.

Finally, we consider the *knapsack problem*. Given a finite set of items $I = \{1, \ldots, n\}$, each item $i \in I$ is associated with a *value* $v_i \in \mathbb{R}_{>0}$ and a *weight* $w_i \in \mathbb{R}_{>0}$. Given a *capacity* $C \in \mathbb{R}_{>0}$, the knapsack problem consists of selecting a subset of items whose total weight does not exceed $C$ and whose total value is maximized. Formally, we aim to find a subset

$$S^* \coloneqq \arg \max_{\substack{S \subseteq I \\ \sum_{i \in S} w_i \leq C}} \sum_{i \in S} v_i.$$

The *cost* (or value) of a solution $S$ is given by $\sum_{i \in S} v_i$. Given $(\{(v_i, w_i)\}_{i \in I}, C)$, *determining the cost of the knapsack problem* amounts to determining the maximum achievable total value under the capacity constraint. While the knapsack problem is not a graph problem in its standard formulation, it admits a classical reduction to a shortest-path problem on a suitably constructed directed graph; see Section 2.2.1.

## C. Proof of Theorem 2

In this appendix, we prove Theorem 2. We begin with a lemma showing that, for finite Lipschitz classes equipped with certificates, one can control the deviation from a given Lipschitz-continuous target function.

**Lemma 8.** *Consider a hypothesis class $\mathcal{F}_\Theta \subset \mathbb{R}^{\mathcal{X}}$ and a target function $f^* \in \mathbb{R}^{\mathcal{X}}$. Assume $f^*$ is Lipschitz continuous with respect to $d_{\mathcal{X}}$ with Lipschitz constant $B_{f^*}$. Let $r > 0$, $\varepsilon > 0$ and assume $\mathcal{F}_\Theta$ is a finite Lipschitz class with certificates $\{B_\theta\}_{\theta \in \Theta}$ and (pseudo-)metric $d_{\mathcal{X}}$. Consider the regularized loss $\mathcal{L}_X(f_\theta)$ with regularization term $\mathcal{L}^{\mathrm{reg}}(f_\theta)$. Let $X \coloneqq \{x_i\}_{i=1}^n$ be an $r$-cover of $\mathcal{X}$ of minimum cardinality, i.e., $n = \mathcal{N}(\mathcal{X}, d_{\mathcal{X}}, r)$. If the regularized loss satisfies $\mathcal{L}_X(f_\theta) \leq \varepsilon$, then*

$$\|f_\theta - f^*\|_\infty \leq (B_\theta + B_{f^*})r + \mathcal{N}(\mathcal{X}, d_{\mathcal{X}}, r)\,\varepsilon.$$

*Proof.* Since $\mathcal{L}_X(f_\theta) \leq \varepsilon$ and $\mathcal{L}_X^{\mathrm{emp}}(f_\theta) \geq 0$, we have $\mathcal{L}^{\mathrm{reg}}(f_\theta) \leq \mathcal{L}_X(f_\theta) \leq \varepsilon$.

Since $X$ is an $r$-cover of $\mathcal{X}$, for any $x \in \mathcal{X}$ there exists $x_i \in X$ with $d_{\mathcal{X}}(x, x_i) \leq r$. By the triangle inequality and Lipschitz continuity:

$$
\begin{aligned}
|f_{\boldsymbol{\theta}}(x) - f^*(x)| &\leq |f_{\boldsymbol{\theta}}(x) - f_{\boldsymbol{\theta}}(x_i)| + |f_{\boldsymbol{\theta}}(x_i) - f^*(x_i)| + |f^*(x_i) - f^*(x)| \\
&\leq B_{\boldsymbol{\theta}} \cdot d_{\mathcal{X}}(x, x_i) + |f_{\boldsymbol{\theta}}(x_i) - f^*(x_i)| + B_{f^*} \cdot d_{\mathcal{X}}(x, x_i) \\
&\leq (B_{\boldsymbol{\theta}} + B_{f^*}) \cdot r + |f_{\boldsymbol{\theta}}(x_i) - f^*(x_i)|.
\end{aligned}
$$

By definition of $y_i$, we have

$$
\frac{1}{N} \sum_{i=1}^{N} |f_{\boldsymbol{\theta}}(x_i) - f^*(x_i)| = \mathcal{L}_X^{\text{emp}}(f_{\boldsymbol{\theta}}) \leq \mathcal{L}_X(f_{\boldsymbol{\theta}}) \leq \varepsilon.
$$

Thus,

$$
\max_{1 \leq j \leq N} |f_{\boldsymbol{\theta}}(x_j) - f^*(x_j)| \leq \sum_{j=1}^{N} |f_{\boldsymbol{\theta}}(x_j) - f^*(x_j)| \leq N\varepsilon.
$$

Therefore, for the $x_i$ with $d_{\mathcal{X}}(x, x_i) \leq r$, we have $|f_{\boldsymbol{\theta}}(x_i) - f^*(x_i)| \leq N\varepsilon$, and

$$
|f_{\boldsymbol{\theta}}(x) - f^*(x)| \leq (B_{\boldsymbol{\theta}} + B_{f^*})r + N\varepsilon,
$$

for all $x \in \mathcal{X}$. $\qquad\square$

We are now ready to prove the main theorem. For completeness, we restate it below in a formal form.

**Theorem 9** (Theorem 2 in the main text)**.** *Let $\mathcal{F}_{\Theta}$ be a finite Lipschitz class through the (pseudo)metric $d_{\mathcal{X}}$, and let $f^*$ be a target function. Assume there is a constant $B_{f^*} \in \mathbb{R}^+$ such that $f^*$ is Lipschitz continuous regarding $d_{\mathcal{X}}$ with Lipschitz constant $B_{f^*}$. Moreover, suppose that, for every $\delta > 0$, there exists $\boldsymbol{\theta}$ satisfying $\sup_{x \in \mathcal{X}} |f_{\boldsymbol{\theta}}(x) - f^*(x)| < \delta$ and the certificate $B_{\boldsymbol{\theta}} \leq B_{f^*}$.*

*For $X := \{x_1, \ldots, x_n\} \subset \mathcal{X}$ and given dataset $\{(x_i, f^*(x_i))\}_{i=1}^n$, and $\eta > 0$, consider the regularized loss $\mathcal{L}_X(f_{\boldsymbol{\theta}})$ with $\mathcal{L}_X^{\text{reg}}(f_{\boldsymbol{\theta}}) = \eta \mathrm{ReLU}(B_{\boldsymbol{\theta}} - B_{f^*})$, which satisfies $\inf_{\boldsymbol{\theta} \in \Theta}\{\mathcal{L}_X(f_{\boldsymbol{\theta}})\} = 0$. Then, for $\varepsilon \in (0,1)$, there exists $r > 0$ and $\varepsilon' > 0$ such that if we take a dataset $X$ that is an $r$-cover of minimum cardinality with $|X| = K_r = \mathcal{N}(\mathcal{X}, d_{\mathcal{X}}, r)$, then*

$$
\mathcal{L}_X(f_{\boldsymbol{\theta}}) < \varepsilon',
$$

*implies*

$$
|f_{\boldsymbol{\theta}}(x) - f^*(x)| < \varepsilon, \quad \text{for all } x \in \mathcal{X}.
$$

*In particular, the above conclusion holds for $r = \frac{\varepsilon}{6(1 + B_{f^*})}$ and any $\varepsilon' < \min\left\{ \frac{\varepsilon}{3\mathcal{N}(\mathcal{X}, d, r)}, \frac{\varepsilon\eta}{6(1 + B_{f^*})} \right\}$; in this case, the required number of samples is $K_r = \mathcal{N}(\mathcal{X}, d_{\mathcal{X}}, r)$.*

*Proof.* We recall that we assume that, for every $\delta > 0$, there exists $\boldsymbol{\theta}$ satisfying $\sup_{x \in \mathcal{X}} |f_{\boldsymbol{\theta}}(x) - f^*(x)| < \delta$ and the certificate $B_{\boldsymbol{\theta}} \leq B_{f^*}$. This implies that there exists a sequence $f_{\boldsymbol{\theta}_k}$ such that $\|f_{\boldsymbol{\theta}_k} - f^*\|_{\infty} \to 0$ and $B_{\boldsymbol{\theta}_k} \leq B_{f^*}$.

Note

$$
\mathcal{L}^{\text{reg}}(f_{\boldsymbol{\theta}}) = \eta \mathrm{ReLU}(B_{\boldsymbol{\theta}} - B_{f^*}) \leq B,
$$

implies

$$
B_{\boldsymbol{\theta}} \leq \frac{1}{\eta} B + B_{f^*}.
$$

Choose $r = \frac{\varepsilon}{6(1 + B_{f^*})}$, let $K_r = \mathcal{N}(\mathcal{X}, d_{\mathcal{X}}, r)$, and choose $\varepsilon' < \min\left\{ \frac{\varepsilon}{3K_r}, \frac{\varepsilon\eta}{6(1 + B_{f^*})} \right\}$. By the assumption on $\mathcal{F}_{\boldsymbol{\theta}}$, there exists an $r$-cover $X$ with $|X| = K_r$.

Let $L := \inf_{\boldsymbol{\theta} \in \Theta} \{\mathcal{L}_X(f_{\boldsymbol{\theta}})\}$. We claim that $L = 0$. Indeed, consider the sequence $\{f_k\}$ and $\{B_{\boldsymbol{\theta}_k}\}$ mentioned earlier. Then $B_{\boldsymbol{\theta}_k} \leq B_{f^*}$ and thus $\mathcal{L}^{\text{reg}}(f_{\boldsymbol{\theta}_k}) = 0$. Further $\mathcal{L}_X^{\text{emp}}(f_{\boldsymbol{\theta}_k}) \leq \|f^* - f_{\boldsymbol{\theta}_k}\|_\infty \to 0$. Therefore

$$L = \inf_{\boldsymbol{\theta} \in \Theta} \{\mathcal{L}_X(f_{\boldsymbol{\theta}})\} \leq \lim_{k \to \infty} \mathcal{L}_X(f_{\boldsymbol{\theta}_k}) = 0.$$

Now we apply Lemma 8 to the set $X = \{x_i\}_{i=1}^{K_r}$, which is an $r$-cover by construction. We consider the dataset $\{(x_i, f^*(x_i))\}_{i=1}^{K_r}$. By assumption, $\mathcal{L}_X(f_{\boldsymbol{\theta}}) < \varepsilon'$. Hence, we have $\eta \text{ReLU}(B_{\boldsymbol{\theta}} - B_{f^*}) < \varepsilon'$, which implies $B_{\boldsymbol{\theta}} < B_{f^*} + \frac{\varepsilon'}{\eta} \leq B_{f^*} + \frac{\varepsilon}{6(1 + B_{f^*})}$.

Thus by Lemma 8,

$$\|f_{\boldsymbol{\theta}} - f^*\|_\infty \leq (B_{\boldsymbol{\theta}} + B_{f^*})r + K_r \varepsilon'$$
$$\leq \left(B_{f^*} + \frac{\varepsilon}{6(1 + B_{f^*})} + B_{f^*}\right) \cdot \frac{\varepsilon}{6(1 + B_{f^*})} + K_r \cdot \frac{\varepsilon}{3K_r}$$
$$= \left(2B_{f^*} + \frac{\varepsilon}{6(1 + B_{f^*})}\right) \cdot \frac{\varepsilon}{6(1 + B_{f^*})} + \frac{\varepsilon}{3}$$
$$= \frac{2B_{f^*}\varepsilon}{6(1 + B_{f^*})} + \frac{\varepsilon^2}{36(1 + B_{f^*})^2} + \frac{\varepsilon}{3}.$$

We now assume $\varepsilon < 1$ so that $\frac{\varepsilon^2}{36(1 + B_{f^*})^2} \leq \frac{\varepsilon}{36}$, and observe that $\frac{B_{f^*}}{1 + B_{f^*}} < 1$. From these inequalities, it follows that

$$\|f_{\boldsymbol{\theta}} - f^*\|_\infty < \frac{\varepsilon}{3} \cdot \frac{B_{f^*}}{1 + B_{f^*}} + \frac{\varepsilon}{3} + \frac{\varepsilon}{36}$$
$$< \frac{\varepsilon}{3} + \frac{\varepsilon}{3} + \frac{\varepsilon}{36}$$
$$= \frac{2\varepsilon}{3} + \frac{\varepsilon}{36}$$
$$= \frac{24\varepsilon}{36} + \frac{\varepsilon}{36} = \frac{25\varepsilon}{36} < \varepsilon.$$

The required number of samples is $K_r = \mathcal{N}(\mathcal{X}, d_{\mathcal{X}}, r)$ where $r = \frac{\varepsilon}{6(1 + B_{f^*})}$. $\qquad \square$

## D. Learning non-realizable functions

This appendix records a variant of Theorem 2 that applies to a larger class of target functions, particularly functions that cannot be approximated with a bounded Lipschitz constant. The approximation properties of the target are summarized by a *certificate profile*, which captures how large a certificate budget is required to approximate the target to a given accuracy.

**Certificate profile.** Let $(\mathcal{X}, d_{\mathcal{X}})$ be a pseudo-metric space and let $\mathcal{F}_\Theta = \{f_{\boldsymbol{\theta}} : \mathcal{X} \to \mathbb{R} \mid \boldsymbol{\theta} \in \Theta\}$ be a hypothesis class. Assume that for each $\boldsymbol{\theta} \in \Theta$ we can compute a certificate $B_{\boldsymbol{\theta}} \in \mathbb{R}_+$ such that $f_{\boldsymbol{\theta}}$ is $B_{\boldsymbol{\theta}}$-Lipschitz with respect to $d_{\mathcal{X}}$. For a target function $f^* : \mathcal{X} \to \mathbb{R}$ and $\varepsilon > 0$, define

$$\widetilde{B}_{f^*}(\varepsilon) := \inf\left\{B \geq 0 \ \Big| \ \exists \boldsymbol{\theta} \in \Theta : \|f_{\boldsymbol{\theta}} - f^*\|_\infty \leq \varepsilon \text{ and } B_{\boldsymbol{\theta}} \leq B\right\}. \tag{4}$$

Note that $\widetilde{B}_{f^*}(\varepsilon)$ may be finite even if $f^* \notin \mathcal{F}_\Theta$.

**Lemma 10.** *Let $f^* : \mathcal{X} \to \mathbb{R}$ be a target and fix $\varepsilon \in (0,1)$, $\eta > 0$, and $\varepsilon' > 0$. Set*

$$r := \frac{\varepsilon}{4\big(1 + 2\widetilde{B}_{f^*}(\varepsilon/4) + \varepsilon'/\eta\big)}. \tag{5}$$

*Let $X = \{x_1, \ldots, x_K\} \subset \mathcal{X}$ be a minimum-cardinality $r$-cover (so $K = \mathcal{N}(\mathcal{X}, d_{\mathcal{X}}, r)$) and set $y_i := f^*(x_i)$. For $f_{\boldsymbol{\theta}} \in \mathcal{F}_\Theta$, define*

$$\mathcal{L}_X(f_{\boldsymbol{\theta}}) := \frac{1}{K} \sum_{i=1}^{K} |f_{\boldsymbol{\theta}}(x_i) - y_i| + \eta \, \text{ReLU}(B_{\boldsymbol{\theta}} - \widetilde{B}_{f^*}(\varepsilon/4)). \tag{6}$$

*If $\mathcal{L}_X(f_{\boldsymbol{\theta}}) \leq \varepsilon'$ and*

$$\varepsilon' \;\leq\; \frac{\varepsilon}{4K}, \tag{7}$$

*then $\|f_{\boldsymbol{\theta}} - f^*\|_\infty \leq \varepsilon$.*

*In particular, taking $B = \widetilde{B}_{f^*}(\varepsilon/4)$ yields the sample size*

$$K(\varepsilon, \eta, \varepsilon') \;=\; \mathcal{N}\!\left(\mathcal{X}, d_{\mathcal{X}}, \frac{\varepsilon}{4\big(1 + 2(\widetilde{B}_{f^*}(\varepsilon/4) + \varepsilon'/\eta)\big)}\right).$$

*Proof.* By the definition of $\widetilde{B}_{f^*}(\varepsilon/4)$ there exists $\boldsymbol{\theta}^* \in \Theta$ such that

$$\|f_{\boldsymbol{\theta}^*} - f^*\|_\infty \leq \varepsilon/4 \qquad \text{and} \qquad B_{\boldsymbol{\theta}^*} \leq B. \tag{8}$$

If $\mathcal{L}_X(f_{\boldsymbol{\theta}}) \leq \varepsilon'$, then the regularizer term in (6) satisfies $\eta \, \mathrm{ReLU}(B_{\boldsymbol{\theta}} - \widetilde{B}_{f^*}(\varepsilon/4)) \leq \varepsilon'$, hence $\mathrm{ReLU}(B_{\boldsymbol{\theta}} - \widetilde{B}_{f^*}(\varepsilon/4)) \leq \varepsilon'/\eta$ and therefore

$$B_{\boldsymbol{\theta}} \leq \widetilde{B}_{f^*}(\varepsilon/4) + \varepsilon'/\eta. \tag{9}$$

Since $y_i = f^*(x_i)$, we have

$$\frac{1}{K} \sum_{i=1}^{K} |f_{\boldsymbol{\theta}}(x_i) - f^*(x_i)| \leq \mathcal{L}_X(f_{\boldsymbol{\theta}}) \leq \varepsilon'.$$

Hence,

$$\max_{1 \leq i \leq K} |f_{\boldsymbol{\theta}}(x_i) - f^*(x_i)| \leq \sum_{i=1}^{K} |f_{\boldsymbol{\theta}}(x_i) - f^*(x_i)| \leq K \varepsilon'. \tag{10}$$

Fix $x \in \mathcal{X}$ and choose $x_i \in X$ with $d_{\mathcal{X}}(x, x_i) \leq r$. Using the triangle inequality,

$$|f_{\boldsymbol{\theta}}(x) - f^*(x)| \leq |f_{\boldsymbol{\theta}}(x) - f_{\boldsymbol{\theta}^*}(x)| + |f_{\boldsymbol{\theta}^*}(x) - f^*(x)|.$$

The second term is at most $\varepsilon/4$ by (8). For the first term, Lipschitz continuity of $f_{\boldsymbol{\theta}}$ and $f_{\boldsymbol{\theta}^*}$ yields

$$\begin{aligned}
|f_{\boldsymbol{\theta}}(x) - f_{\boldsymbol{\theta}^*}(x)| &\leq |f_{\boldsymbol{\theta}}(x) - f_{\boldsymbol{\theta}}(x_i)| + |f_{\boldsymbol{\theta}}(x_i) - f_{\boldsymbol{\theta}^*}(x_i)| + |f_{\boldsymbol{\theta}^*}(x_i) - f_{\boldsymbol{\theta}^*}(x)| \\
&\leq (B_{\boldsymbol{\theta}} + B_{\boldsymbol{\theta}^*}) \, d(x, x_i) + |f_{\boldsymbol{\theta}}(x_i) - f_{\boldsymbol{\theta}^*}(x_i)| \\
&\leq (B_{\boldsymbol{\theta}} + B_{\boldsymbol{\theta}^*}) \, r + |f_{\boldsymbol{\theta}}(x_i) - f^*(x_i)| + |f^*(x_i) - f_{\boldsymbol{\theta}^*}(x_i)|.
\end{aligned}$$

By (9) and (8), $B_{\boldsymbol{\theta}} + B_{\boldsymbol{\theta}^*} \leq (B + \varepsilon'/\eta) + B$, and by (10) and (8),

$$|f_{\boldsymbol{\theta}}(x_i) - f^*(x_i)| \leq K\varepsilon', \qquad |f^*(x_i) - f_{\boldsymbol{\theta}^*}(x_i)| \leq \varepsilon/4.$$

Combining these bounds gives

$$|f_{\boldsymbol{\theta}}(x) - f^*(x)| \leq (B_{\boldsymbol{\theta}} + B_{\boldsymbol{\theta}^*}) \, r + K\varepsilon' + \varepsilon/2. \tag{11}$$

By the choice of $r$ in (5),

$$(B_{\boldsymbol{\theta}} + B_{\boldsymbol{\theta}^*}) \, r \leq \frac{(B_{\boldsymbol{\theta}} + B_{\boldsymbol{\theta}^*})\varepsilon}{4(1 + 2\widetilde{B}_{f^*}(\varepsilon/4) + \varepsilon'/\eta)} \leq \varepsilon/4,$$

and by (7), $K\varepsilon' \leq \varepsilon/4$. Substituting into (11) yields $|f_{\boldsymbol{\theta}}(x) - f^*(x)| \leq \varepsilon$. Since $x \in \mathcal{X}$ was arbitrary, $\|f_{\boldsymbol{\theta}} - f^*\|_\infty \leq \varepsilon$. □

**Remark** The dependence on $\eta$ and the tolerance $\varepsilon'$ appears through the term $\varepsilon'/\eta$ in (5): weaker regularization (smaller $\eta$) permits larger certificates $B_{\boldsymbol{\theta}} \leq \widetilde{B}_{f^*}(\varepsilon/4) + \varepsilon'/\eta$, which shrinks the admissible cover radius and increases the sample size $\mathcal{N}(\mathcal{X}, d_{\mathcal{X}}, r)$. Conversely, stronger regularization tightens certificate control but may increase the achievable regularized loss when an accurate approximation requires larger budgets (as captured by $\widetilde{B}_{f^*}(\cdot)$).

# E. Improved learning guarantees for SSSPs algorithms

While the above sections derive conditions under which a finite training dataset exists, and the algorithm can be learned, here we derive a concrete, small training dataset for learning the Bellman–Ford algorithm for the SSSP problem. Our analysis significantly extends the analysis of Nerem et al. (2025) by deriving a simpler training dataset and, unlike the former work, a differentiable regularization term.

In the following, we formalize the Bellman–Ford update, which corresponds exactly to the update performed by the Bellman–Ford algorithm at each iteration on all nodes of a graph. Our objective is to learn the $K$-fold application of this update.

**Definition 11** (Bellman–Ford instance, update, and distance)**.** An attributed, edge-weighted graph $G$ is called a *Bellman–Ford instance* (BF instance) if both its node labels $a_G(v) \geq 0$, for all $v \in V(G)$, and its edge weights $w_G(e) \geq 0$, for all $e \in E(G)$, are non-negative real-valued functions. For notational convenience, we additionally assume that

$$N_G(v) = N_G(v) \cup \{v\}, \qquad w_G(v,v) = 0, \qquad \forall v \in V(G).$$

We denote by $\mathcal{G}_{\mathrm{BF}}$ the set of all BF instances. We further define the map $\Gamma \colon \mathcal{G}_{\mathrm{BF}} \to \mathcal{G}_{\mathrm{BF}}$ to be the operator that maps a BF instance $G$ to the BF instance $\Gamma(G)$ with the same vertices, edges, and edge weights as $G$, but whose node labels are updated according to

$$a_{\Gamma(G)}(v) \;\coloneqq\; \min\{a_G(u) + w_G(u,v) \colon u \in N_G(v)\}, \qquad \forall v \in V(G).$$

We refer to $\Gamma$ as the *Bellman–Ford update*. Note that $\Gamma(G) \in \mathcal{G}_{\mathrm{BF}}$ for all $G \in \mathcal{G}_{\mathrm{BF}}$.

Given $G \in \mathcal{G}_{\mathrm{BF}}$, we define the *$t$-step Bellman–Ford distance of a node $v$* as

$$x_v^{(t)} \;\coloneqq\; a_{\Gamma^t(G)}(v),$$

where $\Gamma^t$ denotes the $t$-fold composition of $\Gamma$.

Consequently, the learning task studied in this work is to approximate the node-level mapping induced by $\Gamma^K$ using a message-passing neural network.

## E.1. Employed MPNN architecture

Consider $G \in \mathcal{G}_{BF}$. Let $K \in \mathbb{N}$ denote the number of iterations of the Bellman-Ford Update we wish to learn. We want to employ MPNNs as defined in Section 1.2 with minimum aggregation and $m$-layer ReLU FNNs, where $m \in \mathbb{N}$, as update and aggregation functions. We initialize the node representations by setting

$$\boldsymbol{h}_v^{(0)} \coloneqq a_G(v) \in \mathbb{R}_{\geq 0}, \quad \text{for } v \in V(G),$$

More precisely, we define the update and aggregation functions through

$$\boldsymbol{h}_v^{(t)} \coloneqq \mathsf{UPD}_{\boldsymbol{u}_t}^{(t)}\Big(\boldsymbol{h}_v^{(t-1)}, \mathsf{AGG}_{\boldsymbol{a}_t}^{(t)}\big\{\!\!\big\{\big(\boldsymbol{h}_v^{(t-1)}, \boldsymbol{h}_u^{(t-1)}, w_G(v,u)\big) \mid u \in N(v)\big\}\!\!\big\}\Big)$$

$$\coloneqq \mathsf{FNN}^{(m)}(\boldsymbol{\theta}_{\mathsf{UPD}}^{(t)})\left(\min_{u \in N(v)} \mathsf{FNN}^{(m)}(\boldsymbol{\theta}_{\mathsf{AGG}}^{(t)})\left(\begin{pmatrix}\boldsymbol{h}_u^{(t-1)} \\ w_G(v,u)\end{pmatrix}\right)\right),$$

where $t \in [K]$, $v \in V(G)$, and

$$\boldsymbol{\theta}_{\mathsf{UPD}}^{(t)} \coloneqq (\boldsymbol{W}^{(\mathsf{UPD},t,1)}, \boldsymbol{b}^{(\mathsf{UPD},t,1)}, \ldots, \boldsymbol{W}^{(\mathsf{UPD},t,m)}, \boldsymbol{b}^{(\mathsf{UPD},t,m)}),$$
$$\boldsymbol{\theta}_{\mathsf{AGG}}^{(t)} \coloneqq (\boldsymbol{W}^{(\mathsf{AGG},t,1)}, \boldsymbol{b}^{(\mathsf{AGG},t,1)}, \ldots, \boldsymbol{W}^{(\mathsf{AGG},t,m)}, \boldsymbol{b}^{(\mathsf{AGG},t,m)}),$$

are parameter sets, each consisting of $m$ weight matrices and bias vectors of appropriately chosen dimension, which we will specify below. Finally, we denote the output of the MPNN after $K$ message-passing steps for vertex $v \in V(G)$ by $\boldsymbol{h}_v^{(K)}$.

**Global Indexing and hidden dimensions** Note that in the above, each weight matrix and bias vector is indexed depending on whether they belong to an update or aggregation FNN, the index $t \in [K]$ of the aggregation layer, and the layer index within the FNN. We now introduce an alternative global indexing scheme, in which each weight matrix and bias vector is labeled by its global position within the MPNN, from innermost to outermost. More precisely, let $J := 2mK$ be the number of total layers. Then define a bijection

$$\phi \colon [J]_0 \to \{(0)\} \cup \{(f,t,l) \mid t \in [K], l \in [m], f \in \{\mathsf{AGG}, \mathsf{UPD}\}\}$$

such that

$$(\phi(0), \phi(1), \phi(2), \dots, \phi(J)) := (0, (\mathsf{AGG}, 1, 1), \dots, (\mathsf{AGG}, 1, m), (\mathsf{UPD}, 1, 1), \dots,$$
$$(\mathsf{UPD}, 1, m), (\mathsf{AGG}, 2, 1), \dots, (\mathsf{UPD}, K, m)).$$

If the context allows it, we will abuse notation and write $j = \phi(j)$. With both indices at hand, we now want to specify the dimensions of the hidden features and, by extension, those of the parameters. Let $d_{\phi(j)} = d_j \in \mathbb{N}$ denote the dimension of the feature vector of the $j$-th global layer. Further let $d_{\phi(0)} = d_0 = 1$ denote the input dimension. We then require

$$\boldsymbol{b}^j \in \mathbb{R}^{d_j}, \quad \boldsymbol{W}^j \in \begin{cases} \mathbb{R}^{d_j \times (d_{j-1}+1)} & \phi(j) \in \{(\mathsf{AGG}, t, 1) \mid t \in [K]\} \\ \mathbb{R}^{d_j \times d_{j-1}} & \text{otherwise,} \end{cases}$$

where for each edge inserting layer, i.e., a layer of the type $(\mathsf{AGG}, t, 1)$, $t \in [K]$, the input dimension is given by the sum of the feature dimension of the previous layer and the dimension of edge weights (which is equal to 1). For our purposes, we will choose $d_0 = d_J = 1$. Further, we will choose all hidden dimensions to be equal to some fixed $d \in \mathbb{R}$, except for $j$ such that $\phi(j) \in \{(\mathsf{AGG}, t, m) \mid t \in [K]\}$ where we let $d_j = 1$.

In the following proofs, we will primarily use the global indexing. To identify *aggregation layers*, we define, for each $k \in [K]$, an index $a_k \in [J]$ such that $\phi(a_k) = (\mathsf{AGG}, k, m)$, i.e., $a_k$ is the last layer of the $k$-th aggregation FNN. Similarly, to identify *edge-inserting layers*, we define, for each $k \in [K]$, an index $j_k \in [J]$ such that $\phi(j_k) = (\mathsf{AGG}, k, 1)$, i.e., $j_k$ is the first layer of the $k$-th aggregation FNN. For convenience, we additionally set $j_0 := 0$.

Further, we want to split the weight matrices of the edge-inserting layers into components that act on hidden node features and on the inserted edge weights. To this end, we define

$$\tilde{\boldsymbol{W}}^{j_k} := \begin{pmatrix} \boldsymbol{W}^{j_k}_{-,1} & \cdots & \boldsymbol{W}^{j_k}_{-,d_{(j_k-1)}} \end{pmatrix} \in \mathbb{R}^{d_j \times d_{j-1}}, \quad \boldsymbol{C}^k := \begin{pmatrix} \boldsymbol{W}^{j_k}_{-,d_{(j_k-1)}+1} \end{pmatrix} \in \mathbb{R}^{d_j \times 1}$$

such that

$$\boldsymbol{W}^{j_k} \begin{pmatrix} \boldsymbol{h}^{(k-1)}_u \\ w_G(v,u) \end{pmatrix} = \tilde{\boldsymbol{W}}^{j_k} \boldsymbol{h}^{(k-1)}_u + \boldsymbol{C}^k w_G(v,u).$$

Finally to avoid unnecessary case distinctions we redefine $\boldsymbol{W}^{j_k} := \tilde{\boldsymbol{W}}^{j_k}$.

**Parameter sets** We define $\mathcal{B} := (\boldsymbol{b}^{(1)}, \dots, \boldsymbol{b}^{(J)}) \in \prod_{j \in [J]} \mathbb{R}^{d_j} =: \boldsymbol{\Theta}_b$ as the vector of biases, $\mathcal{C} := (\boldsymbol{C}^{(1)}, \dots, \boldsymbol{C}^{(K)}) \in \prod_{k \in [K]} \mathbb{R}^{d_{j_k} \times 1} =: \boldsymbol{\Theta}_C$ as the vector of edge inserting weight matrices and $\mathcal{W} := (\boldsymbol{W}^{(1)}, \dots, \boldsymbol{W}^{(J)}) \in \prod_{j \in [J]} \mathbb{R}^{d_j \times d_{j-1}} =: \boldsymbol{\Theta}_W$ as the vector of weight matrices acting on node features. Further we define $\boldsymbol{\theta} := (\mathcal{W}, \mathcal{C}, \mathcal{B})$ to denote the collection of all parameters and $\boldsymbol{\Theta}_{\mathrm{BF}} := \boldsymbol{\Theta}_W \times \boldsymbol{\Theta}_C \times \boldsymbol{\Theta}_b$ to denote the parameter space such that $\boldsymbol{\theta} \in \boldsymbol{\Theta}_{\mathrm{BF}}$.

### E.2. Loss function

Following Appendix B.3, let

$$X \subset \mathcal{X} := V_1(\mathcal{G}_{\mathrm{BF}}) = \{(G,v) : G \in \mathcal{G}_{\mathrm{BF}}, v \in V(G)\}$$

denote the training set used to learn a node-level invariant, and let $N := |X|$ be its cardinality.

For notational convenience, we identify each node $v \in V(G)$ with the pair $(G, v)$, and write

$$\mathcal{X} = \{v : G \in \mathcal{G}_{\mathrm{BF}}, v \in V(G)\}$$

to simplify expressions in the remainder of the section.

We define the loss function acting on parameter configurations $\boldsymbol{\theta} \in \boldsymbol{\Theta}_{\mathrm{BF}}$ as the sum of the MAE loss and some weighted $\ell_1$-norm as regularization. More precisely, we let

$$\mathcal{L}(\boldsymbol{\theta}) := \mathcal{L}^{\mathrm{emp}}(\boldsymbol{\theta}) + \eta \mathcal{L}^{\mathrm{reg}}(\boldsymbol{\theta}),$$

for some fixed $\eta > 0$, where

$$\mathcal{L}^{\mathrm{emp}}(\boldsymbol{\theta}) := \frac{1}{N} \sum_{v \in X} \left| \boldsymbol{h}_v^{(K)} - x_v^{(K)} \right|$$

where $\boldsymbol{h}_v^{(K)}$ denotes the final output of the MPNN and $x_v^{(K)}$ is the targeted $K$-step BF-distance, and

$$\mathcal{L}^{\mathrm{reg}}(\boldsymbol{\theta}) := \sum_{k=0}^{K} \sum_{j > j_k}^{J} \|\boldsymbol{W}^j\|_1 + \sum_{k \in [K]} \|\boldsymbol{C}^k\|_1 + \sum_{j=1}^{J} \|\boldsymbol{b}^j\|_1.$$

*Remark* 12. Using layer-wise indexing, one can show that the regularization rewrites as

$$\mathcal{L}^{\mathrm{reg}}(\theta) = \sum_{k \in [K]} \Big( \sum_{l=2}^{m} k \|\boldsymbol{W}^{(\mathsf{AGG},k,l)}\|_1 + \sum_{i \in [d_{(\mathsf{UPD},k-1,m)}]} k \|\boldsymbol{W}_{-,i}^{(\mathsf{AGG},k,m)}\|_1 + \|\boldsymbol{W}_{-,d_{(\mathsf{UPD},k-1,m)}+1}^{(\mathsf{AGG},k,m)}\|_1 \Big)$$
$$+ \sum_{k \in [K]} \sum_{l \in [m]} \big( (k+1) \|\boldsymbol{W}^{(\mathsf{UPD},k,l)}\|_1 + \|\boldsymbol{b}^{(\mathsf{AGG},k,l)}\|_1 + \|\boldsymbol{b}^{(\mathsf{UPD},k,l)}\|_1 \big),$$

i.e., all weight matrices acting on feature vectors of the $k$-th aggregation layer are weighted with $k$, all weight matrices acting on edge components and all bias vectors are weighted with 1, and all weight matrices of the $k$-th update layer are weighted with $k+1$. Intuitively, the weight matrices are weighted depending on their importance for the network, which will become clearer during the proof.

### E.3. Construction of the training set for learning Bellman–Ford algorithm

In the following, we outline the construction of the training set for training an MPNN to execute the $K$-fold Bellman–Ford update.

**Definition 13** (Path-Graphs). Let $K > 0$, let $\boldsymbol{w} \in \mathbb{R}_{\geq 0}^{K+1}$, and $\beta > 0$, we now define node-attributed, edge-weighted paths where the edge-weights are given by the entries of a real-valued vector. That is, let $P(\boldsymbol{w}) := P_\beta(\boldsymbol{w}) \in \mathcal{G}_{BF}$ be the path graph such that

$$V(P(\boldsymbol{w})) := \{v_0^{\boldsymbol{w}}, v_1^{\boldsymbol{w}}, \ldots, v_K^{\boldsymbol{w}}\} := \{v_0, v_1, \ldots, v_K\}$$

with labels $a_{P(\boldsymbol{w})}(v_0) = w_0$, $a_{P(\boldsymbol{w})}(v_i) = \beta$ for all $i \in [K]$ and

$$E(P(\boldsymbol{w})) := \{(v_{i-1}, v_i) \mid i \in [K]\},$$

where the edge-weight function $w_{P(\boldsymbol{w})} \colon E(P(\boldsymbol{w})) \to \mathbb{R}_{\geq 0}$ is defined as $(v_{i-1}, v_i) \mapsto w_i$, for $i \in [K]$.

*Remark* 14. Consider a graph $P_\beta(\boldsymbol{w})$ such that $\beta > \|\boldsymbol{w}\|_1$. Then one can easily verify that the $K$-step BF-distance of $v_K^{\boldsymbol{w}}$ in $P_\beta(\boldsymbol{w})$ is given by

$$x_{v_{\boldsymbol{w}}^K}^{(K)} = \|\boldsymbol{w}\|_1.$$

**Definition 15** (Bellman–Ford path training set). Let $x \in \mathbb{R}_{\geq 0}$, and define

$$S_{x,K} := \{x \boldsymbol{e}_k^{(K+1)} \mid k \in [K]_0\},$$

where $\{\boldsymbol{e}_\ell^{K+1}\}_{\ell=0}^K$ denotes the canonical unit-length basis of $\mathbb{R}^{K+1}$. That is, $S_{x,K}$ contains scaled versions of the canonical unit-length basis vectors.

We now use these scaled vectors to define the edge weights in our training set, which consists of paths. Let $K > 0$, $\beta = \beta(x, N) \geq 2(N + x + 1)$ and let $P_\beta(\boldsymbol{w})$ denote an edge-weighted path as in Definition 13. We then define the *Bellman–Ford path training set*, parameterized by $x \in \mathbb{R}_{\geq 0}$, as

$$T_{S_{x,K}} := \{v_K^{\boldsymbol{w}} \mid \boldsymbol{w} \in S_{x,K}, v_K^{\boldsymbol{w}} \in P_\beta(\boldsymbol{w})\}.$$

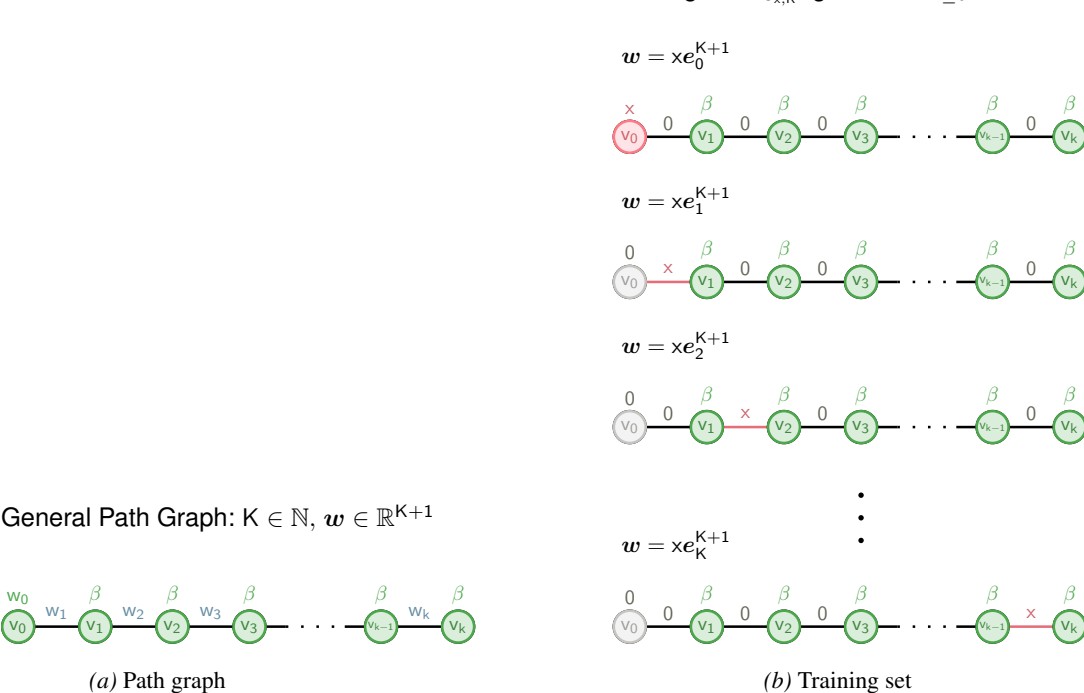

*Figure 3.* Bellman–Ford training graphs. Dots indicate omitted intermediate nodes, and edge weights are shown on the edges. **(a)** General path graph associated with $\boldsymbol{w} \in \mathbb{R}^{K+1}$ as in Definition 13. The initial node is labeled $a(v_0) = w_0$, while all other nodes have label $\beta \gg 0$. The path has length $K$ with edge weights $w_1, \ldots, w_K$. **(b)** Bellman–Ford training set for arbitrary $K$ as defined in Definition 15, consisting of $K + 1$ path graphs corresponding to the scaled unit vectors $x\boldsymbol{e}_0^{K+1}, \ldots, x\boldsymbol{e}_K^{K+1}$. Each path contains $K + 1$ vertices $v_0, \ldots, v_K$. The root node satisfies $a(v_0) = x$ if $k = 0$ and $a(v_0) = 0$ otherwise, while all other nodes have label $\beta$. Exactly one edge per path has weight $x$, and all remaining edges have weight 0.

### E.4. Theorem

The following provides a quantitative approximation guarantee for the learned Bellman–Ford dynamics under near-optimal training loss.

**Theorem 16.** *Let $K \in \mathbb{N}$, assume we want to learn $K$-steps of Bellman–Ford, i.e., the function $\Gamma^K$, using the min-aggregation MPNN architecture with $K$ layers and $m$-layer FNN as update and aggregation functions as described above in Appendix E.1. In addition, assume that the regularization parameter $\eta$ satisfies $\eta \geq 2K \exp(m(K^2 + 3K))$, and for the edge weight scaler $x$ of the training set $T_{S_{x,K}}$ it holds $x \geq 4mKN\eta$, where $N$ is the cardinality of the training set $X$. Then if the Bellman-Ford path training set is contained in the training set, i.e., $T_{S_{x,K}} \subset X$ and the loss $\mathcal{L}(\theta)$ is within $\varepsilon \leq \frac{1}{2}$ of its global minimum, the following holds, then, for any Bellman–Ford instance $G \in \mathcal{G}_{BF}$ and any $v \in V(G)$,*

$$\left| \boldsymbol{h}_v^{(K)} - x_v^{(K)} \right| \leq \varepsilon(x_v^{(K)} + 1),$$

*where $\boldsymbol{h}_v^{(K)}$ is the feature representation output by the MPNN and $x_v^{(K)}$ is the targeted $K$-step BF-distance of the vertex $v$.*

This theorem differs from the main theorem of Nerem et al. (2025) in several ways. On the positive side, our result uses a smaller training set, namely $K + 1$ path instances from $T_{S_{x,K}}$, and replaces their non-differentiable $\ell_0$ penalty with a differentiable $\ell_1$ regularizer that has layer-specific weights $w(k)$. This differentiability allows our loss to be optimized directly. On the other hand, our analysis assumes a more restricted model: the depth is fixed to exactly $K$ message-passing layers (matching $K$ Bellman–Ford steps), and the aggregation dimension is 1. These choices are made to simplify the analysis. Furthermore, unlike Nerem et al. (2025), the regularization parameter $\eta$ and the training weight scale $x$ both scale exponentially in $K$, which may be prohibitive in some settings.

# F. Proof of Theorem 16

We begin with a proof outline that highlights the main lemmas and ideas underlying the argument.

## F.1. Proof outline

**Appendix F.3: Properties of FNNs.**   The aim of this section is to develop algebraic properties of feedforward neural networks (FNNs) that are needed to handle the ReLU nonlinearity and to investigate the dependence of the network output on its weights and biases. Later, we will see that the features produced by the MPNN can be upper bounded by the output of a single FNN of depth $J$ (the total number of global layers of the MPNN), which is where the results derived in this section will be applied.

The main results include linearity of FNNs in the regime where all parameters are non-negative, as well as monotonicity with respect to both the network parameters and the input. The key ingredient is that, for non-negative inputs, the ReLU activation acts as the identity.

In addition, we prove the following corollary.

**Corollary 17.** *Let $\mathcal{W} \in \boldsymbol{\Theta}_W$ and $\mathcal{B}, \mathcal{B}_C \in \boldsymbol{\Theta}_b$. Then for any $\boldsymbol{y} \in \mathbb{R}_{\geq 0}^{d_0}$ it holds*

$$\left| \mathsf{FNN}^{(J)}(\mathcal{W}, \mathcal{B} + \mathcal{B}_C)(\boldsymbol{y}) - \mathsf{FNN}^{(J)}(\mathcal{W}^+, \mathcal{B}_C^+)(\boldsymbol{y}) \right| \; \leq \; G(\mathcal{W}, \mathcal{B}_C)(\boldsymbol{y}) + \mathsf{FNN}^{(J)}(|\mathcal{W}|, |\mathcal{B}|)(0).$$

Here, $\mathcal{B}$ should be interpreted as the collection of biases of the MPNN, while $\mathcal{B}_C$ has the same structure and represents the contribution of edge features along a path to a node $v$ in the training set. Thus, the LHS side can be viewed as measuring the difference between the output of the MPNN with parameters $(\mathcal{W}, \mathcal{C}, \mathcal{B})$ (cf. Appendix E.1) and the output of the MPNN with parameters $(\mathcal{W}^+, \mathcal{C}^+, \mathbf{o})$, that is, where all weights are replaced by their positive parts and all biases are set to zero.

The latter parameter configuration is significantly easier to analyze, since the effect of the ReLU nonlinearity and the biases vanish. Consequently, this result allows us to replace the features of the MPNN with parameters $(\mathcal{W}, \mathcal{C}, \mathcal{B})$ by the output of the MPNN with parameters $(\mathcal{W}^+, \mathcal{C}^+, \mathbf{o})$ up to some error controlled by the RHS.

**Appendix F.4: FNNs along computation trees.**   In this section, we define the set of computation trees $\mathcal{T}_G^K(v)$, which encodes the essential information needed to describe all possible choices available during aggregation. We then define the FNN along a computation tree $t \in \mathcal{T}_G^K(v)$, denoted by $H^{(k)}(\boldsymbol{\theta})(t)$. Intuitively, this quantity represents the feature that the MPNN would output if the aggregation followed the computation tree $t$.

This framework allows us to define the set of minimum-aggregation trees $\mathcal{T}_{G,\boldsymbol{\theta}}^k(v)$, i.e., the computation trees selected by the min-aggregation. In particular, the MPNN feature can be expressed as an FNN evaluated along a minimum-aggregation tree.

**Lemma 18.** *Let $G \in \mathcal{G}_{BF}$, $v \in V(G)$, and $\boldsymbol{\theta} \in \boldsymbol{\Theta}_{BF}$. Then for any $k \in [K]_0$ and any $\tau_v^k \in \mathcal{T}_{G,\boldsymbol{\theta}}^k(v)$,*

$$\boldsymbol{h}_v^{(k)} = H^{(k)}(\boldsymbol{\theta})(\tau_v^k).$$

The main purpose of introducing FNNs along computation trees is to enable comparisons between the MPNN output and the output obtained from alternative computation trees. In particular, this allows us to upper-bound the MPNN features by FNNs evaluated along trees that are not chosen by the MPNN itself.

This idea is formalized in the next lemma, which shows that although the MPNN performs a layer-wise, "greedy" minimization, the resulting computation tree $\tau_v^k \in \mathcal{T}_{G,\boldsymbol{\theta}}^K(v)$ minimizes the feature globally, up to the effect of negative weights.

**Lemma 19.** *Let $G \in \mathcal{G}_{BF}$, $v \in V(G)$, and $\boldsymbol{\theta} \in \boldsymbol{\Theta}_{BF}$. Then for any $k \in [K]_0$, $\tau_v^k \in \mathcal{T}_{G,\boldsymbol{\theta}}^k(v)$, and $t_v^k \in \mathcal{T}_G^k(v)$ it holds*

$$H^{(k)}(\boldsymbol{\theta})(\tau_v^k) \; \leq \; H^{(k)}(\boldsymbol{\theta}^+)(t_v^k)$$

*element-wise.*

This is the point where the use of min-aggregation becomes crucial.

**Appendix F.5: Walk-lifted FNNs.** In this section, we define the Walk-lifted FNN $H_{\mathrm{wl}}(\boldsymbol{\theta})(\boldsymbol{z})$, where $\boldsymbol{z} \in \mathbb{R}_{\geq 0}^{K+1}$, which can be represented as an FNN with suitably chosen biases. As a consequence, it inherits all properties derived in Appendix F.3. In particular, Corollary 17 translates directly into the following corollary for Walk-lifted FNNs.

**Corollary 20.** *For all $\boldsymbol{z} \in \mathbb{R}_{\geq 0}^{K+1}$ and parameters $(\mathcal{W}, \mathcal{C}, \mathcal{B}) \in \boldsymbol{\Theta}_{BF}$, it holds*

$$\left| H_{\mathrm{wl}}(\mathcal{W}, \mathcal{C}, \mathcal{B})(\boldsymbol{z}) - H_{\mathrm{wl}}(\mathcal{W}^+, \mathcal{C}^+, 0)(\boldsymbol{z}) \right| \ \leq \ G(\mathcal{W}, \mathcal{B}(\boldsymbol{z}, \mathcal{C}))(\boldsymbol{z}) + H_{\mathrm{wl}}(|\mathcal{W}|, |\mathcal{C}|, |\mathcal{B}|)(0).$$

Here, $\mathcal{B}(\boldsymbol{z}, \mathcal{C})$ represents the effective bias induced by the edge features encoded in $\boldsymbol{z}$ and the parameters $\mathcal{C}$. As before, the right-hand side compares the Walk-lifted FNN with arbitrary parameters to the corresponding object with non-negative weights and vanishing biases.

We further show that the Walk-lifted FNN coincides with the FNN along a computation tree whenever the computation tree is essentially a walk.

**Lemma 21.** *Let $G \in \mathcal{G}_{BF}$ and $(\mathcal{W}, \mathcal{C}, \mathcal{B}) = \boldsymbol{\theta} \in \boldsymbol{\Theta}_{BF}$. Then for any $v \in V(G)$ and any $p \in \mathcal{P}_G^K(v)$ it holds*

$$H_{\mathrm{wl}}(\boldsymbol{\theta})(\boldsymbol{z}^p) = H^{(k)}(\boldsymbol{\theta})(t^K(p)).$$

Here, $p$ should be interpreted as a walk in the graph $G$, $t^K(p)$ as its corresponding computation tree representation, and $\boldsymbol{z}^p$ as a vector encoding the edge weights along $p$.

Combining the three lemmas above, we obtain the following result, which allows us to replace the MPNN features with the walk-lifted FNN.

**Lemma 22.** *Let $\boldsymbol{\theta} \in \boldsymbol{\Theta}_{BF}$ and $G \in \mathcal{G}_{BF}$. Then for any $v \in V(G)$ and any $p \in \mathcal{P}_G^K(v)$ it holds*

$$\boldsymbol{h}_v^{(K)} \ \leq \ H_{\mathrm{wl}}(\boldsymbol{\theta}^+)(\boldsymbol{z}^p),$$

*and*

$$\boldsymbol{h}_v^{(K)} = H_{\mathrm{wl}}(\boldsymbol{\theta})(\boldsymbol{z}^p), \quad \text{if } t^K(p) \in \mathcal{T}_{G, \boldsymbol{\theta}}^K(v).$$

In the subsequent sections, this lemma will be the key ingredient for replacing the MPNN features with an object that is significantly easier to control and analyze.

**Appendix F.6: Small empirical loss and structure of parameters.** The goal of this section is to show that if the loss is close to its global minimum and if the parameters $\eta$ and $x$ are chosen sufficiently large, then the network parameters must exhibit a highly constrained structure. More precisely, we prove that

- all biases are close to zero, i.e. $\mathcal{B} \approx 0$,
- all weights are either positive or close to zero, i.e. $\mathcal{W}^- \approx 0$,
- the factors multiplying edge weights are approximately equal to 1 or larger,
- the empirical loss is small, i.e. $\mathcal{L}^{\mathrm{emp}} \leq 2\varepsilon$.

The key idea underlying this analysis is that the parameters are subject to two competing forces:

- **Regularization**, which pushes all parameters toward zero, and
- **Empirical loss**, which pushes those parameters necessary to fit the targets toward 1.

We begin by introducing a constant $L \in \mathbb{R}_{>0}$ and show that $\eta L$ is an upper bound on the global minimum of the loss, i.e.,

$$\min_{\boldsymbol{\theta} \in \boldsymbol{\Theta}_{\mathrm{BF}}} \mathcal{L}(\boldsymbol{\theta}) \leq \eta L.$$

In fact, $\eta L$ is the global minimum, which we establish implicitly at the end of the section.

Next, we derive coarse *a priori* bounds on the effect that non-zero biases and negative weights can have on the MPNN output, uniformly over all seen and unseen inputs.

**Corollary 23.** *Let $\boldsymbol{\theta} \in \boldsymbol{\Theta}_{BF}$ and $0 \leq \varepsilon \leq \eta L$. Assume that $\mathcal{L}(\boldsymbol{\theta})$ lies within $\varepsilon$ of its global minimum. Then*

$$H_{\mathrm{wl}}(|\mathcal{W}|, |\mathcal{C}|, |\mathcal{B}|)(0) \leq \exp(L) \sum_{j=1}^{J} \|\boldsymbol{b}^j\|_1,$$

$$G(\mathcal{W}, \mathcal{B}(\boldsymbol{z}, \mathcal{C}))(\boldsymbol{z}) \leq \exp(L) \Big( \sum_{l=1}^{J} \|(\boldsymbol{W}^l)^-\|_1 + \sum_{k=1}^{K} \|(\boldsymbol{C}^k)^-\|_1 \Big) \|\boldsymbol{z}\|_1, \quad \boldsymbol{z} \in \mathbb{R}_{\geq 0}^{K+1}.$$

Here, $H_{\mathrm{wl}}(|\mathcal{W}|, |\mathcal{C}|, |\mathcal{B}|)(0) = \sum_{l \in [J]} \prod_{j>l} |\boldsymbol{W}^j| \, |\boldsymbol{b}^l|$ represents the maximal deviation that the biases $\mathcal{B} = (b^1, \dots, b^J)$ can induce in the MPNN output. Similarly, $G(\mathcal{W}, \mathcal{B}(\boldsymbol{z}, \mathcal{C}))(\boldsymbol{z})$ bounds the maximal contribution of negative weights.

Controlling both quantities is crucial, since their sum appears on the right-hand side of Corollary 20 and thus governs the error incurred when replacing the original parameters $(\mathcal{W}, \mathcal{C}, \mathcal{B})$ by the simplified configuration $(\mathcal{W}^+, \mathcal{C}^+, \mathbf{o})$. The exponential dependence on $L$ arises from bounding products of the form $\prod_{j>l} |\boldsymbol{W}^j|$ solely in terms of $\sum_{j>l} \|\boldsymbol{W}^j\|_1 \leq \mathcal{L}(\boldsymbol{\theta}) \leq 2\eta L$.

We then introduce a reduced set of transformed parameters together with a modified loss function. The key idea is that this new loss depends on fewer variables while still capturing the essential behavior of the original optimization problem.

**Definition 24** (Modified loss function). Given $\boldsymbol{\theta} = (\mathcal{W}, \mathcal{C}, \mathcal{B}) \in \boldsymbol{\Theta}_{\mathrm{BF}}$, we define the transformed parameters

$$\tilde{\boldsymbol{\theta}} := (\gamma_0, \dots, \gamma_K, B, w^-) \in \mathbb{R}_{\geq 0}^{K+3}.$$

To avoid case distinctions, we introduce $C^0 := 1$. We then set

$$\gamma_k := \prod_{j=j_k+1}^{J} (\boldsymbol{W}^j)^+ (\boldsymbol{C}^k)^+, \quad k \in \{0, \dots, K\},$$

$$B := \|\mathcal{B}\|_1,$$

$$w^- := \sum_{l=1}^{J} \|(\boldsymbol{W}^l)^-\|_1 + \sum_{k=1}^{K} \|(\boldsymbol{C}^k)^-\|_1.$$

We define the modified loss

$$\tilde{\mathcal{L}}(\tilde{\boldsymbol{\theta}}) := \tilde{\mathcal{L}}^{\mathrm{emp}}(\tilde{\boldsymbol{\theta}}) + \eta \tilde{\mathcal{L}}^{\mathrm{reg}}(\tilde{\boldsymbol{\theta}}),$$

where

$$\tilde{\mathcal{L}}^{\mathrm{emp}}(\tilde{\boldsymbol{\theta}}) := \frac{1}{N} \sum_{k=0}^{K} \sigma\big((1 - \gamma_k)x - \exp(L)B\big),$$

$$\tilde{\mathcal{L}}^{\mathrm{reg}}(\tilde{\boldsymbol{\theta}}) := B + \sum_{k=0}^{K} l_k \gamma_k^{1/l_k} + w^-.$$

Up to negative parameters, $\gamma_k$ represents the factor by which the $k$-th edge weight along an aggregation path is multiplied, while $B$ and $w^-$ capture the total magnitude of biases and negative weights, respectively. Our objective is therefore to show that $B \approx 0$, $w^- \approx 0$, and $\gamma_k \approx 1$.

To justify working with $\tilde{\mathcal{L}}$ instead of $\mathcal{L}$, we show that near-optimality of the original loss implies near-optimality of the modified loss. Since the regularization terms are chosen such that both losses attain the same global minimum $\eta L$, it suffices to prove that $\tilde{\mathcal{L}}(\tilde{\boldsymbol{\theta}}) \leq \mathcal{L}(\boldsymbol{\theta})$.

In fact, we prove the following.

**Lemma 25.** *Let $\boldsymbol{\theta} = (\mathcal{W}, \mathcal{C}, \mathcal{B}) \in \boldsymbol{\Theta}_{BF}$ and let $\tilde{\boldsymbol{\theta}}$ denote its transformed parameter vector. Assume $T_{S_{x,K}} \subset X$ and that $\mathcal{L}(\boldsymbol{\theta})$ lies within $\varepsilon \leq \eta L$ of its global minimum. Then*

$$\mathcal{L}^{\mathrm{emp}}(\boldsymbol{\theta}) \geq \tilde{\mathcal{L}}^{\mathrm{emp}}(\tilde{\boldsymbol{\theta}}), \qquad \mathcal{L}^{\mathrm{reg}}(\boldsymbol{\theta}) \geq \tilde{\mathcal{L}}^{\mathrm{reg}}(\tilde{\boldsymbol{\theta}}).$$

To prove the empirical-loss inequality $\mathcal{L}^{\mathrm{emp}}(\boldsymbol{\theta}) \geq \tilde{\mathcal{L}}^{\mathrm{emp}}(\tilde{\boldsymbol{\theta}})$, we upper bound the MPNN features $\boldsymbol{h}_v^{(K)}$ using Lemma 22. and the *a priori* bound from Corollary 23, to obtain

$$\boldsymbol{h}_v^{(K)} \leq \sum_{k=0}^{K} \gamma_k z_k + \exp(L)B,$$

where $\boldsymbol{z} \in \mathbb{R}^{K+1}$ denotes the edge weights along a shortest path to $v$. By the choice of training samples $T_{S_{x,K}}$, we have $\sum_{k=0}^{K} \gamma_k z_k = \gamma_k x$ for some $k \in [K]$. Consequently,

$$\left| x_v^{(K)} - \boldsymbol{h}_v^{(K)} \right| \geq \sigma\left( x - \gamma_k x - \exp(L)B \right).$$

Summing over all samples in $T_{S_{x,K}}$ yields $\mathcal{L}^{\mathrm{emp}}(\boldsymbol{\theta}) \geq \tilde{\mathcal{L}}^{\mathrm{emp}}(\tilde{\boldsymbol{\theta}})$.

To establish $\mathcal{L}^{\mathrm{reg}}(\boldsymbol{\theta}) \geq \tilde{\mathcal{L}}^{\mathrm{reg}}(\tilde{\boldsymbol{\theta}})$, we exploit the specific structure of the regularizer. For each $k \in [K]_0$ and each weight matrix appearing in

$$\gamma_k = \prod_{j=j_k+1}^{J} (\boldsymbol{W}^j)^+ (\boldsymbol{C}^k)^+,$$

there exists exactly one corresponding normalized term in the regularization. A variational argument, combined with a separation of biases and negative weights from the remaining terms, yields the claimed bound.

Based on this lemma, we now characterize parameter configurations whose loss is close to the global minimum.

**Lemma 26** (Parameter characterization near the global minimum). *Let $\boldsymbol{\theta} \in \boldsymbol{\Theta}_{BF}$. Assume $T_{S_{x,K}} \subset X$ and that $\mathcal{L}(\boldsymbol{\theta})$ lies within $0 \leq \varepsilon \leq \eta L$ of its global minimum. Further assume*

$$\eta \geq 2K \exp(L) \qquad and \qquad x \geq 2N\eta J.$$

*Then the following estimates hold:*

$$H_{\mathrm{wl}}(|\mathcal{W}|, |\mathcal{C}|, |\mathcal{B}|)(0) \leq \varepsilon, \tag{12}$$

$$\gamma_k \geq 1 - \frac{\varepsilon}{\eta J}, \qquad k \in [K]_0, \tag{13}$$

*and, for any $\boldsymbol{z} \in \mathbb{R}_{\geq 0}^{K+1}$,*

$$\left| H_{\mathrm{wl}}(\mathcal{W}, \mathcal{C}, \mathcal{B})(\boldsymbol{z}) - H_{\mathrm{wl}}(\mathcal{W}^+, \mathcal{C}^+, 0)(\boldsymbol{z}) \right| \leq \left( \tfrac{1}{2}\|\boldsymbol{z}\|_1 + 1 \right)\varepsilon. \tag{14}$$

*Moreover,*

$$\mathcal{L}^{\mathrm{emp}}(\boldsymbol{\theta}) \leq 2\varepsilon.$$

The estimate (12) implies that the effect of the biases on the MPNN output is negligible, i.e. $\mathcal{B} \approx 0$. The bound (13) shows that the factors multiplying the edge weights are close to one (or larger). Inequality (14) allows us to replace MPNN features along a path by the simpler expression $H_{\mathrm{wl}}(\mathcal{W}^+, \mathcal{C}^+, 0)(\boldsymbol{z})$, and the final inequality ensures that the empirical loss is small.

The key idea in the proof is that the previous lemma allows us to analyze the modified loss $\tilde{\mathcal{L}}$ instead of $\mathcal{L}$. The modified loss removes couplings between the factors $\gamma_k$ and eliminates ReLU nonlinearities between layers, thereby isolating the two competing forces acting on the parameters. The regularization drives $B$, the negative weights $w^-$, and the factors $\gamma_k, k \in [K]_0$, toward zero, while the empirical loss pushes $B$ and the $\gamma_k$'s away from zero. This implies $w^- \approx 0$. Since $\exp(L) < \eta$ by assumption, the regularization force on $B$ dominates, yielding $B \approx 0$. Finally, for sufficiently large $x$, the loss is minimized when $\gamma_k \approx 1$, which implies that the regularization loss is approximately $L$ and hence that the empirical loss is small.

**Appendix F.7: Upper and Lower Bound**    In this section, we additionally assume that the hidden dimension before each minimum aggregation is equal to 1. Intuitively, this restriction ensures that the minimum aggregation cannot decrease the hidden features on graphs that contain more branching than simple path graphs, which provide strictly fewer aggregation options. As a result, path graphs represent the worst case for the minimum operator.

Formally, this assumption implies that the MPNN features can be expressed via a Walk-lifted FNN along any computation path. This allows us to apply the parameter characterization from Lemma 26 and deduce the following lower bound.

**Corollary 27** (Lower bound). *Let $\boldsymbol{\theta} \in \boldsymbol{\Theta}_{BF}$, $\eta \geq 2K \exp(L)$, and $x \geq 2N\eta J$. Assume $T_{S_{x,K}} \subset X$ and that $\mathcal{L}(\boldsymbol{\theta})$ lies within $0 \leq \varepsilon \leq \eta L$ of its global minimum. Then for any $G \in \mathcal{G}_{BF}$, $v \in V(G)$, and $t \in \mathcal{T}_{\theta}^J(v)$,*

$$\boldsymbol{h}_v^{(K)} \geq (1-\varepsilon)\|\boldsymbol{z}^t\|_1 - \varepsilon.$$

*In particular,*

$$\boldsymbol{h}_v^{(K)} \geq (1-\varepsilon)x_v^{(K)} - \varepsilon,$$

*where $x_v^{(K)}$ denotes the Bellman–Ford distance.*

The first inequality shows that the MPNN features can be bounded from below by approximately the length of any min-aggregation computation path. Since, by definition, the Bellman–Ford distance $x_v^{(K)}$ is the minimal length among all paths ending at $v \in V(G)$, this immediately implies the claimed lower bound.

We next establish the corresponding upper bound. Here, an additional difficulty arises: although the modified loss is minimized at $\gamma_k = 1$, it does not provide a tight *a priori* upper bound on $\gamma_k$, since the additional regularization cost incurred by choosing $\gamma_k > 1$ is relatively small. We therefore first show that excessively large values of $\gamma_k$ necessarily lead to overly large MPNN features and hence to an empirical loss that is too large. This observation allows us to control $\gamma_k$ from above and, in combination with Lemma 22, to derive the desired upper bound.

**Theorem 28** (Upper bound). *Let $\boldsymbol{\theta} \in \boldsymbol{\Theta}_{BF}$, $\eta \geq 2K \exp(L)$, and $x \geq 2N\eta J$. Assume $T_{S_{x,K}} \subset X$ and that $\mathcal{L}(\boldsymbol{\theta})$ lies within $0 < \varepsilon < \frac{1}{2}$ of its global minimum. Then, for any $G \in \mathcal{G}_{BF}$ and any $v \in V(G)$,*

$$\boldsymbol{h}_v^{(K)} \leq (1+\varepsilon)x_v^{(K)} + \varepsilon.$$

Combining the lower and upper bounds yields the claimed theorem.

### F.2. Outlook

In this subsection, we outline several directions in which the results of this the paper can be strengthened and extended.

**Removing the exponential dependence on $\eta$ without biases.** We first observe that if all biases are set to zero, the proofs reveal that the exponential lower bound on the regularization parameter $\eta$ is no longer necessary. In this setting, the arguments can be repeated with only minor modifications, yielding the following result.

**Theorem 29.** *Let $K \in \mathbb{N}$ and suppose we aim to learn $K$ steps of the Bellman–Ford algorithm, i.e. the function $\Gamma^K$, using a min-aggregation MPNN with $K$ layers and $m$-layer feedforward neural networks as update and aggregation functions, as described in Appendix E.1, but without any biases. Assume that the Bellman–Ford path training set is contained in the training set, i.e. $T_{S_{x,K}} \subset X$, and that the edge-weight scaling satisfies $x \geq 4mKN\eta$, where $N := |X|$ is the cardinality of the training set.*

*Then there exists a constant $C > 0$ such that if $\mathcal{L}(\theta)$ lies within $0 \leq \varepsilon < \min\{\eta 2mK(K+3), \frac{1}{2}\}$ of its global minimum, then for any Bellman–Ford instance $G \in \mathcal{G}_{BF}$ and any $v \in V(G)$,*

$$\left|\boldsymbol{h}_v^{(K)} - x_v^{(K)}\right| \leq \varepsilon C x_v^{(K)},$$

*where $\boldsymbol{h}_v^{(K)}$ denotes the MPNN feature representation and $x_v^{(K)}$ the target $K$-step Bellman–Ford distance.*

**Removing the exponential dependence with biases.** The exponential lower bound on $\eta$ can also be removed without eliminating biases by including the Bellman–Ford path training set twice with different edge-weight scalings, for example, by assuming that $T_{S_{x,K}}, T_{S_{2x,K}} \subset X$. By comparing feature values obtained from identical path graphs with different scalings, we can eliminate the bias terms in a first step and show that $\mathcal{L}^{\mathrm{emp}}(\boldsymbol{\theta}) \leq 2\varepsilon$ and $B \leq \varepsilon/\eta$ without any restriction on $\eta$. In a second step, bounds analogous to those in Lemma 26 can be derived. The proofs of the upper and lower bounds then remain unchanged.

**Extension to $\ell_1$-regularization.** We further aim to generalize our analysis to standard $\ell_1$-regularization. From a theoretical perspective, we conjecture that size generalization results as in Theorem 16 continue to hold in this setting, though potentially with weaker error bounds and increased optimization difficulty.

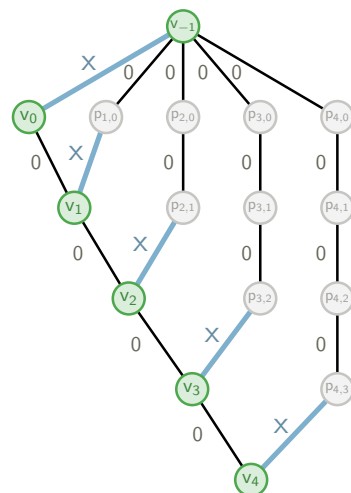

*Figure 4.* Edge case graph from Definition 30 for learning the $K$-fold Bellman–Ford update with higher aggregation dimension. The root node is $v_{-1}$. Shown here is the instance for $K = 4$. At each node $v_i$, $i \in [K]$, the network can choose between multiple paths of equal total weight $x$.

**Removing the one-dimensional aggregation assumption.** Finally, we seek to remove the assumption that the hidden dimension before each aggregation equals one, i.e. to extend the theory to $d_{a_k} > 1$. This assumption is only used in Appendix F.7. For higher aggregation dimensions, the minimum operator may exploit additional degrees of freedom and produce feature values that are significantly smaller than the true shortest-path distance, thereby breaking generalization. We conjecture that this issue can be resolved by augmenting the training set with a carefully designed edge case graph that forces the MPNN to confront the most extreme configurations of shortest paths.

**Definition 30** (Edge case graph). Let $x > 0$ and $K \in \mathbb{N}$. We define the edge case graph $G(x, K) = (V, E)$ as follows.

First, add a path with nodes $v_{-1}, v_0, \ldots, v_K \in V$ and edges $(v_{i-1}, v_i) \in E$ for $i \in [K]_0$, with edge weights $w(v_{-1}, v_0) = x$ and $w(v_{i-1}, v_i) = 0$ for all $i \in [K]$.

Next, for each $i \in [K]$, add a path of length $i + 1$ from $v_{-1}$ to $v_i$ whose edge weights are zero except for the final edge, which has weight $x$. More precisely, for each $i \in [K]$ we introduce nodes $p_{i,j} \in V$ for $j \in [i - 1]_0$ and edges

$$\{v_{-1}, p_{i,0}\}, \{p_{i,0}, p_{i,1}\}, \ldots, \{p_{i,i-2}, p_{i,i-1}\}, \{p_{i,i-1}, v_i\} \in E,$$

with weights

$$w(\{v_{-1}, p_{i,0}\}) = w(\{p_{i,0}, p_{i,1}\}) = \cdots = w(\{p_{i,i-2}, p_{i,i-1}\}) = 0, \qquad w(\{p_{i,i-1}, v_i\}) = x.$$

Finally, we assign node features corresponding to the one-step Bellman–Ford distances with root node $v_{-1}$. Specifically, we set $a(v_{-1}) = a(p_{i,0}) = 0$ for all $i \in [K]$, $a(v_0) = x$, and $a(v) = \beta$ for all remaining nodes $v \in V \backslash \{v_{-1}, p_{1,0}, \ldots, p_{K,0}, v_0\}$, where $\beta > 0$ is chosen sufficiently large.

We conjecture that augmenting the training set with the edge case graph $G(x, K)$ suffices to restore size generalization guarantees with a constant aggregation dimension, and that, in this regime, the performance gap between the tailored loss used in this work and the standard $\ell_1$-loss vanishes.

### F.3. Properties of FNNs

In the following, let $J \in \mathbb{N}$ and $(d_0, \ldots, d_J) \in \mathbb{N}^{J+1}$ be fixed.

*Remark* 31. By definition of the feedforward neural network (cf. Section 1.2), for any parameters $\boldsymbol{\theta} = (\mathcal{W}, \mathcal{B}) \in \boldsymbol{\Theta}$ and any input $\boldsymbol{x} \in \mathbb{R}^{d_0}$, we have

$$\mathsf{FNN}_0^J(\boldsymbol{\theta})(\boldsymbol{x}) = \boldsymbol{x},$$

and for all $j \geq 1$,

$$\mathsf{FNN}_j^J(\boldsymbol{\theta})(\boldsymbol{x}) = \sigma\big(\boldsymbol{W}^{(j)}\,\mathsf{FNN}_{j-1}^J(\boldsymbol{\theta})(\boldsymbol{x}) + \boldsymbol{b}^{(j)}\big).$$

We will repeatedly use this recursive representation in inductive arguments.

*Remark* 32. We regard $\mathcal{W} \in \boldsymbol{\Theta}_W$, $\mathcal{B} \in \boldsymbol{\Theta}_b$, and $\boldsymbol{\theta} \in \boldsymbol{\Theta}$ as vectors in their respective parameter spaces. Accordingly, element-wise operations on $\mathcal{W}$, $\mathcal{B}$, or $\boldsymbol{\theta}$ are defined componentwise on the matrices $\boldsymbol{W}^{(j)}$ and vectors $\boldsymbol{b}^{(j)}$, $j \in [J]$. For example,

$$\mathcal{B}^+ := \big((\boldsymbol{b}^{(1)})^+, \dots, (\boldsymbol{b}^{(J)})^+\big),$$

and operations such as $\mathcal{B} + \mathcal{B}'$ for $\mathcal{B}, \mathcal{B}' \in \boldsymbol{\Theta}_b$ are well defined.

**Lemma 33** (Linearity). *Let $\boldsymbol{\theta} := (\mathcal{W}, \mathcal{B}) \in \boldsymbol{\Theta}$ and assume that all parameters of $\boldsymbol{\theta}$ are (element-wise) non-negative. Then for all $j \in [J]_0$ and $\boldsymbol{y} \in \mathbb{R}_{\geq 0}^{d_0}$ it holds that*

$$\mathsf{FNN}_j^{(J)}(\mathcal{W}, \mathcal{B})(\boldsymbol{y}) = \sum_{l=1}^{j} \Big( \prod_{s=l+1}^{j} \boldsymbol{W}^s \Big) \boldsymbol{b}^l + \Big( \prod_{s=1}^{j} \boldsymbol{W}^s \Big)(\boldsymbol{y}).$$

**Lemma 34.** *Let $j \in [J]_0$, $\boldsymbol{y}_1, \boldsymbol{y}_2 \in \mathbb{R}_{\geq 0}^{d_0}$, and $\boldsymbol{\theta}_1 := (\mathcal{W}_1, \mathcal{B}_1), \boldsymbol{\theta}_2 := (\mathcal{W}_2, \mathcal{B}_2) \in \boldsymbol{\Theta}$. Assume that $\boldsymbol{y}_1 \leq \boldsymbol{y}_2$, $\boldsymbol{\theta}_1 \leq \boldsymbol{\theta}_2$ and $\mathcal{W}_2 \geq 0$ element-wise. Then it holds that*

$$\mathsf{FNN}_j^{(J)}(\mathcal{W}_1, \mathcal{B}_1)(\boldsymbol{y}_1) \;\leq\; \mathsf{FNN}_j^{(J)}(\mathcal{W}_2, \mathcal{B}_2)(\boldsymbol{y}_2),$$

*element-wise.*

*Proof.* Let $\mathcal{W}_i = (\boldsymbol{W}_i^1, \dots, \boldsymbol{W}_i^J)$ and $\mathcal{B}_i = (\boldsymbol{b}_i^1, \dots, \boldsymbol{b}_i^J)$ for $i = 1, 2$. We argue by induction on $j \in [J]_0$.

**Base case ($j = 0$).** It holds

$$\mathsf{FNN}_j^{(J)}(\mathcal{W}_1, \mathcal{B}_1)(\boldsymbol{y}_1) = \boldsymbol{y}_1 \leq \boldsymbol{y}_2 = \mathsf{FNN}_j^{(J)}(\mathcal{W}_2, \mathcal{B}_2)(\boldsymbol{y}_2).$$

**Induction step.** Assume the statement holds for $j - 1$, then we obtain

$$
\begin{aligned}
\mathsf{FNN}_j^{(J)}(\mathcal{W}_1, \mathcal{B}_1)(\boldsymbol{y}_1) &= \sigma\big(\boldsymbol{W}_1^j\,\mathsf{FNN}_{j-1}^{(J)}(\mathcal{W}_1, \mathcal{B}_1)(\boldsymbol{y}_1) + b_1^j\big) && \text{(since } \mathsf{FNN}_{j-1}^{(J)}(\mathcal{W}_1, \mathcal{B}_1)(\boldsymbol{y}_1) \geq 0) \\
&\leq \sigma\big(\boldsymbol{W}_2^j\,\mathsf{FNN}_{j-1}^{(J)}(\mathcal{W}_1, \mathcal{B}_1)(\boldsymbol{y}_1) + b_2^j\big) && \text{(since } \boldsymbol{W}_2^j \geq \boldsymbol{W}_1^j, b_2^j \geq b_1^j) \\
&\leq \sigma\big(\boldsymbol{W}_2^j\,\mathsf{FNN}_{j-1}^{(J)}(\mathcal{W}_2, \mathcal{B}_2)(\boldsymbol{y}_2) + b_2^j\big) && \text{(induction hypothesis and } \boldsymbol{W}_2^j \geq 0) \\
&= \mathsf{FNN}_j^{(J)}(\mathcal{W}_2, \mathcal{B}_2)(\boldsymbol{y}_2).
\end{aligned}
$$

Thus, the inequality holds for $j$, completing the proof. $\qquad\square$

**Lemma 35** (Lipschitz–type inequality). *Let $j \in [J]_0$, $\boldsymbol{y}_1, \boldsymbol{y}_2 \in \mathbb{R}^{d_0}$, $\mathcal{W} \in \boldsymbol{\Theta}_W$, and $\mathcal{B}_1, \mathcal{B}_2 \in \boldsymbol{\Theta}_b$. Then, the following holds*

$$\big|\mathsf{FNN}_j^{(J)}(\mathcal{W}, \mathcal{B}_1)(\boldsymbol{y}_1) - \mathsf{FNN}_j^{(J)}(\mathcal{W}, \mathcal{B}_2)(\boldsymbol{y}_2)\big| \;\leq\; \mathsf{FNN}_j^{(J)}(|\mathcal{W}|, |\mathcal{B}_1 - \mathcal{B}_2|)(|\boldsymbol{y}_1 - \boldsymbol{y}_2|),$$

*element-wise.*

Recall that $|\mathcal{W}|$ denotes the vector or collection of biases where the absolute value is applied point-wise to all parameters, i.e. $|\mathcal{W}| := (|\boldsymbol{W}^1|, \dots, |\boldsymbol{W}^J|)$ where $(|\boldsymbol{W}|^j)_{l,k} = \big|(\boldsymbol{W}^j)_{l,k}\big|$ for all $j \in [J]$ and indexes $l, k$.

*Proof.* As before, we let $\mathcal{B}_i = (\boldsymbol{b}_i^1, \dots, \boldsymbol{b}_i^J)$ for $i = 1, 2$. We proceed by induction on $j \in [J]_0$.

**Base case ($j = 0$).** It holds

$$\big|\mathsf{FNN}_0^{(J)}(\mathcal{W}, \mathcal{B}_1)(\boldsymbol{y}_1) - \mathsf{FNN}_0^{(J)}(\mathcal{W}, \mathcal{B}_2)(\boldsymbol{y}_2)\big| = |\boldsymbol{y}_1 - \boldsymbol{y}_2| = \mathsf{FNN}_0^{(J)}(|\mathcal{W}|, |\mathcal{B}_1 - \mathcal{B}_2|)(|\boldsymbol{y}_1 - \boldsymbol{y}_2|).$$

**Induction step.** Assume the statement holds for $j - 1$.

Since ReLU satisfies $|\sigma(a) - \sigma(a')| \leq |a - a'|$, we obtain

$$\left| \mathsf{FNN}_j^{(J)}(\mathcal{W}, \mathcal{B}_1)(\boldsymbol{y}_1) - \mathsf{FNN}_j^{(J)}(\mathcal{W}, \mathcal{B}_2)(\boldsymbol{y}_2) \right|$$
$$\leq |\boldsymbol{W}^j| \left| \mathsf{FNN}_{j-1}^{(J)}(\mathcal{W}, \mathcal{B}_1)(\boldsymbol{y}_1) - \mathsf{FNN}_{j-1}^{(J)}(\mathcal{W}, \mathcal{B}_2)(\boldsymbol{y}_2) \right| + |b_1^j - b_2^j|.$$

By the induction hypothesis,

$$\left| \mathsf{FNN}_{j-1}^{(J)}(\mathcal{W}, \mathcal{B}_1)(\boldsymbol{y}_1) - \mathsf{FNN}_{j-1}^{(J)}(\mathcal{W}, \mathcal{B}_2)(\boldsymbol{y}_2) \right| \leq \mathsf{FNN}_{j-1}^{(J)}(|\mathcal{W}|, |\mathcal{B}_1 - \mathcal{B}_2|)(|\boldsymbol{y}_1 - \boldsymbol{y}_2|).$$

Substituting gives

$$\left| \mathsf{FNN}_j^{(J)}(\mathcal{W}, \mathcal{B}_1)(\boldsymbol{y}_1) - \mathsf{FNN}_j^{(J)}(\mathcal{W}, \mathcal{B}_2)(\boldsymbol{y}_2) \right| \leq |\boldsymbol{W}^j| \, \mathsf{FNN}_{j-1}^{(J)}(|\mathcal{W}|, |\mathcal{B}_1 - \mathcal{B}_2|)(|\boldsymbol{y}_1 - \boldsymbol{y}_2|) + |b_1^j - b_2^j|$$
$$= \mathsf{FNN}_j^{(J)}(|\mathcal{W}|, |\mathcal{B}_1 - \mathcal{B}_2|)(|\boldsymbol{y}_1 - \boldsymbol{y}_2|).$$

This completes the induction. $\qquad\square$

Next, we study how negative parameters affect the outputs of FNNs.

**Definition 36.** Let $\mathcal{W} \in \boldsymbol{\Theta}_W$, and $\mathcal{B} \in \boldsymbol{\Theta}_b$. For $l \in [J]$ define

$$\mathcal{B}^l := (\boldsymbol{b}^1, \boldsymbol{b}^2, \ldots, \boldsymbol{b}^{l-1}, \mathbf{o}, \ldots) \in \boldsymbol{\Theta}_b$$

and

$$\mathcal{W}^l := ((\boldsymbol{W}^1)^+, \ldots, (\boldsymbol{W}^{l-1})^+, (\boldsymbol{W}^l)^-, (\boldsymbol{W}^{l+1})^+, \ldots, (\boldsymbol{W}^J)^+) \in \boldsymbol{\Theta}_W$$

**Definition 37.** Let $\mathcal{W} \in \boldsymbol{\Theta}_W$, and $\mathcal{B}_1, \mathcal{B}_2, \mathcal{B} \in \boldsymbol{\Theta}_b$. For $j \in [J]_0$, define

$$G_j(\mathcal{W}, \mathcal{B}_1, \mathcal{B}_2)(\boldsymbol{y}) := \sum_{l=1}^{j} \mathsf{FNN}_j^{(J)}(\mathcal{W}^l, \mathcal{B}_1^l)(\boldsymbol{y}) + \mathsf{FNN}_j^{(J)}(\mathcal{W}^+, \mathcal{B}_2)(\mathbf{0}),$$

and

$$G(\mathcal{W}, \mathcal{B})(\boldsymbol{y}) := G_J(\mathcal{W}, \mathcal{B}^+, \mathcal{B}^-)(\boldsymbol{y}).$$

*Remark* 38 (Intuition for $G$). We see that in the case of $G(\mathcal{W}, \mathcal{B})(\boldsymbol{y})$, all parameters of all FNNs that are being summed over in the above definition are positive. Thus we can apply Lemma 33 to see that, for any $\boldsymbol{y} \in \mathbb{R}_{\geq 0}^{d_0}$,

$$G(\mathcal{W}, \mathcal{B})(\boldsymbol{y}) = \sum_{l \in [J]_0} \sum_{l'=0}^{l-1} (\boldsymbol{W}^J)^+ \ldots (\boldsymbol{W}^{l+1})^+ (\boldsymbol{W}^l)^- (\boldsymbol{W}^{l-1})^+ \ldots (\boldsymbol{W}^{l'+1})^+ (\boldsymbol{b}^{l'})^+$$

$$\tag{15}$$

$$+ \sum_{l' \in [J]_0} \left( \prod_{s=l'+1}^{J} (\boldsymbol{W}^s)^+ \right) (\boldsymbol{b}^{l'})^-,$$

where we let $\boldsymbol{b}^0 := \boldsymbol{y}$ for notational convenience.

Note that the right-hand side can be interpreted as follows: it corresponds to evaluating $\mathsf{FNN}_j^J(\mathcal{W}, \mathcal{B})(\boldsymbol{y})$ after replacing all weight matrices and bias vectors by their positive parts, except for exactly one of them, which is replaced by its negative part, and then summing over all such choices.

For the estimates that follow, we aim to bound the difference between the output of the MPNN with parameters $(\mathcal{W}, \mathcal{C}, \mathcal{B})$ (cf. Appendix E.1) and that of the MPNN with parameters $(\mathcal{W}^+, \mathcal{C}^+, \mathbf{o})$, i.e., where all weights are replaced by their positive parts and all biases are set to zero.

The next lemma provides a first step toward such a bound using the function $G$. It has the flavor of an inclusion–exclusion lower bound and formalizes the idea of swapping a single weight matrix or bias vector with its negative part.

*Remark* 39 (Upper bound on $G$). When deriving error bounds on the loss and parameters in Appendix F.6 we will need to bound $G(\mathcal{W}, \mathcal{B})(\boldsymbol{y})$. We will do this through

$$\|G(\mathcal{W}, \mathcal{B})(\boldsymbol{b}^0)\|_1 \leq \sum_{l \in [J]_0} \sum_{l'=0}^{l-1} \Big( \prod_{\substack{s=l'+1 \\ s \neq l}}^{J} \|(\boldsymbol{W}^s)^+\|_1 \Big) \|(\boldsymbol{W}^l)^+\|_1 \|(\boldsymbol{b}^{l'})^+\|_1 + \sum_{l' \in [J]_0} \Big( \prod_{s=l'+1}^{J} \|(\boldsymbol{W}^s)^+\|_1 \Big) \|(\boldsymbol{b}^{l'})^-\|_1,$$

where $\|\boldsymbol{W}^j\|_1 = \sum_{k \in [d_j]} \sum_{l \in [d_{j-1}]} \big| \boldsymbol{W}_{k,l}^j \big|$ denotes the 1-norm of matrices, which is not to be understood as a point-wise application in contrast to the absolute value $|\cdot|$ and the positive part $(\cdot)^+$. Note that this is a direct consequence of Equation (15).

**Lemma 40.** *Let* $\mathcal{W} \in \boldsymbol{\Theta}_W$, *and* $\mathcal{B}_1, \mathcal{B}_2, \mathcal{B} \in \boldsymbol{\Theta}_b$ *such that* $\mathcal{B}_1 - \mathcal{B}_2 = \mathcal{B}$, *and* $\mathcal{B}_1, \mathcal{B}_2 \geq 0$. *Then for any* $\boldsymbol{y} \in \mathbb{R}_{\geq 0}^{d_0}$ *it holds*

$$\mathsf{FNN}_j^{(J)}(\mathcal{W}, \mathcal{B})(\boldsymbol{y}) \geq \mathsf{FNN}_j^{(J)}(\mathcal{W}^+, \mathcal{B}_1)(\boldsymbol{y}) - G_j(\mathcal{W}, \mathcal{B}_1, \mathcal{B}_2)(\boldsymbol{y}).$$

*Proof.* We argue by induction over $j$.

**Base case** $j = 0$**.** Since $G_0(\mathcal{W}, \mathcal{B}_1, \mathcal{B}_2)(\boldsymbol{y}) = 0$,

$$\mathsf{FNN}_0^{(J)}(\mathcal{W}, \mathcal{B})(\boldsymbol{y}) = \boldsymbol{y} = \mathsf{FNN}_0^{(J)}(\mathcal{W}^+, \mathcal{B}_1)(\boldsymbol{y}) - G_0(\mathcal{W}, \mathcal{B}_1, \mathcal{B}_2)(\boldsymbol{y}).$$

**Inductive step.** In the following, we omit the argument $(\boldsymbol{y})$ whenever it is clear from the context. Let $j > 0$ and assume the claim holds for $j - 1$. Using $\boldsymbol{W}^j = (\boldsymbol{W}^j)^+ - (\boldsymbol{W}^j)^-$, the induction hypothesis and Lemma 34

$$\begin{aligned}
\mathsf{FNN}_j^{(J)}(\mathcal{W}, \mathcal{B}) &= \sigma\big(\boldsymbol{W}^j \mathsf{FNN}_{j-1}^{(J)}(\mathcal{W}, \mathcal{B}) - \boldsymbol{b}^j\big) \\
&= \sigma\big((\boldsymbol{W}^j)^+ \mathsf{FNN}_{j-1}^{(J)}(\mathcal{W}, \mathcal{B}) - (\boldsymbol{W}^j)^- \mathsf{FNN}_{j-1}^{(J)}(\mathcal{W}, \mathcal{B}) + \boldsymbol{b}_1^j - \boldsymbol{b}_2^j\big) \\
&\geq (\boldsymbol{W}^j)^+ \big(\mathsf{FNN}_{j-1}^{(J)}(\mathcal{W}^+, \mathcal{B}_1) - G_{j-1}(\mathcal{W}, \mathcal{B}_1, \mathcal{B}_2)\big) - (\boldsymbol{W}^j)^- \mathsf{FNN}_{j-1}^{(J)}(\mathcal{W}^+, \mathcal{B}) + \boldsymbol{b}_1^j - \boldsymbol{b}_2^j \\
&= \big((\boldsymbol{W}^j)^+ \mathsf{FNN}_{j-1}^{(J)}(\mathcal{W}^+, \mathcal{B}_1) + \boldsymbol{b}_1^j\big) - \big((\boldsymbol{W}^j)^+ G_{j-1}(\mathcal{W}, \mathcal{B}_1, \mathcal{B}_2) + (\boldsymbol{W}^j)^- \mathsf{FNN}_{j-1}^{(J)}(\mathcal{W}^+, \mathcal{B}) + \boldsymbol{b}_2^j\big).
\end{aligned}$$

Since $\mathsf{FNN}_{j-1}^{(J)}(\mathcal{W}^+, \mathcal{B})$ is only dependent on the first $j-1$ bias vectors we note that $\mathsf{FNN}_{j-1}^{(J)}(\mathcal{W}^+, \mathcal{B}) = \mathsf{FNN}_{j-1}^{(J)}(\mathcal{W}^+, \mathcal{B}^j)$ by definition of $\mathcal{B}^j$ (cf. Definition 36). By using the definition of $G$ and that $\mathcal{B}_2 \geq 0$ we obtain that

$$\begin{aligned}
&(\boldsymbol{W}^j)^+ G_{j-1}(\mathcal{W}, \mathcal{B}_1, \mathcal{B}_2) + (\boldsymbol{W}^j)^- \mathsf{FNN}_{j-1}^{(J)}(\mathcal{W}^+, \mathcal{B}) + \boldsymbol{b}_2^j \\
&= (\boldsymbol{W}^j)^+ \sum_{l=i+1}^{j-1} \mathsf{FNN}_{j-1}^{(J)}(\mathcal{W}^l, \mathcal{B}_1^l) + (\boldsymbol{W}^j)^- \mathsf{FNN}_{j-1}^{(J)}(\mathcal{W}^+, \mathcal{B}_1^j) + (\boldsymbol{W}^j)^+ \mathsf{FNN}_{j-1}^{(J)}(\mathcal{W}^+, \mathcal{B}_2)(0) + \boldsymbol{b}_2^j \\
&= \sum_{l=i+1}^{j-1} \sigma\big((\boldsymbol{W}^j)^+ \mathsf{FNN}_{j-1}^{(J)}(\mathcal{W}^l, \mathcal{B}_1^l)\big) + \sigma\big((\boldsymbol{W}^j)^- \mathsf{FNN}_{j-1}^{(J)}(\mathcal{W}^+, \mathcal{B}_1^j)\big) + \sigma\big((\boldsymbol{W}^j)^+ \mathsf{FNN}_{j-1}^{(J)}(\mathcal{W}^+, \mathcal{B}_2)(0) + \boldsymbol{b}_2^j\big) \\
&= \sum_{l=i+1}^{j} \mathsf{FNN}_j^{(J)}(\mathcal{W}^l, \mathcal{B}_1^l) + \mathsf{FNN}_j^{(J)}(\mathcal{W}^+, \mathcal{B}_2)(0) = G_j(\mathcal{W}, \mathcal{B}_1, \mathcal{B}_2)
\end{aligned}$$

Inserting this into the above equation gives

$$\begin{aligned}
\mathsf{FNN}_j^{(J)}(\mathcal{W}, \mathcal{B}) &\geq \big((\boldsymbol{W}^j)^+ \mathsf{FNN}_{j-1}^{(J)}(\mathcal{W}^+, \mathcal{B}_1)(\boldsymbol{y}) + \boldsymbol{b}_1^j\big) - G_j(\mathcal{W}, \mathcal{B}_1, \mathcal{B}_2) \\
&= \mathsf{FNN}_j^{(J)}(\mathcal{W}^+, \mathcal{B}_1) - G_j(\mathcal{W}, \mathcal{B}_1, \mathcal{B}_2)
\end{aligned}$$

and the result follows. $\qquad\square$

The next lemma finally establishes the bound we will later use to bound the difference between the output of the MPNN with parameters $(\mathcal{W}, \mathcal{C}, \mathcal{B})$ and that of the MPNN with parameters $(\mathcal{W}^+, \mathcal{C}^+, \mathbf{o})$.

**Corollary 41.** *Let* $\mathcal{W} \in \boldsymbol{\Theta}_W$, *and* $\mathcal{B}, \mathcal{B}_C \in \boldsymbol{\Theta}_b$. *Then for any* $\boldsymbol{y} \in \mathbb{R}^{d_0}_{\geq 0}$ *it holds*

$$\left| \mathsf{FNN}^{(J)}(\mathcal{W}, \mathcal{B} + \mathcal{B}_C)(\boldsymbol{y}) - \mathsf{FNN}^{(J)}(\mathcal{W}^+, \mathcal{B}_C^+)(\boldsymbol{y}) \right| \leq G(\mathcal{W}, \mathcal{B}_C)(\boldsymbol{y}) + \mathsf{FNN}^{(J)}(|\mathcal{W}|, |\mathcal{B}|)(0).$$

*Proof.* By Lemma 34 and Lemma 33, we obtain the upper bound:

$$\mathsf{FNN}^{(J)}(\mathcal{W}, \mathcal{B} + \mathcal{B}_C)(\boldsymbol{y}) \leq \mathsf{FNN}^{(J)}(\mathcal{W}^+, \mathcal{B}^+ + \mathcal{B}_C^+)(\boldsymbol{y})$$

$$\leq \mathsf{FNN}^{(J)}(\mathcal{W}^+, \mathcal{B}^+)(0) + \mathsf{FNN}^{(J)}(\mathcal{W}^+, \mathcal{B}_C^+)(\boldsymbol{y}) \leq \mathsf{FNN}^{(J)}(|\mathcal{W}|, |\mathcal{B}|)(0) + \mathsf{FNN}^{(J)}(\mathcal{W}^+, \mathcal{B}_C^+)(\boldsymbol{y}).$$

We also obtain a lower bound by applying Lemma 35:

$$\mathsf{FNN}^{(J)}(\mathcal{W}, \mathcal{B} + \mathcal{B}_C)(\boldsymbol{y}) \geq \mathsf{FNN}^{(J)}(\mathcal{W}, \mathcal{B}_C)(\boldsymbol{y}) - \mathsf{FNN}^{(J)}(|\mathcal{W}|, |\mathcal{B}|)(0).$$

Furthermore, by letting $\mathcal{B}_1 = \mathcal{B}_C^+$ and $\mathcal{B}_2 = \mathcal{B}_C^-$ in Lemma 40 we arrive at

$$\mathsf{FNN}^{(J)}(\mathcal{W}, \mathcal{B} + \mathcal{B}_C)(\boldsymbol{y}) \geq \mathsf{FNN}^{(J)}(\mathcal{W}, \mathcal{B}_C)(\boldsymbol{y}) - \mathsf{FNN}^{(J)}(|\mathcal{W}|, |\mathcal{B}|)(0)$$

$$\geq \mathsf{FNN}^{(J)}(\mathcal{W}^+, \mathcal{B}_C^+)(\boldsymbol{y}) - G(\mathcal{W}, \mathcal{B}_C)(\boldsymbol{y}) - \mathsf{FNN}^{(J)}(|\mathcal{W}|, |\mathcal{B}|)(0)$$

Combining the upper and lower bounds yields the claimed estimate. □

### F.4. FNNs along computation trees

To avoid analyzing the layers of the MPNN one by one, we aim to express the entire MPNN in terms of an object that can be defined similarly to the FNN in the previous section. Unfortunately, in general, no choice of bias and weight vectors in $\mathsf{FNN}^{(J)}(\mathcal{W}, \mathcal{B})(\boldsymbol{y})$ can capture the full expressivity of the aggregation operation. To address this, we introduce *FNNs along computation paths* in this section. For an appropriate choice of the underlying tree, these networks coincide with the MPNN output.

Recall that each layer of the MPNN is given by

$$\boldsymbol{h}_v^{(t)} := \mathsf{FNN}^{(m)}(\boldsymbol{\theta}_{\mathsf{UPD}}^{(t)}) \left( \min_{u \in N(v)} \mathsf{FNN}^{(m)}(\boldsymbol{\theta}_{\mathsf{AGG}}^{(t)}) \left( \begin{smallmatrix} \boldsymbol{h}_u^{(t-1)} \\ w_G(v,u) \end{smallmatrix} \right) \right).$$

For ease of notation, we write

$$f_{\boldsymbol{\theta}}^{\mathsf{UPD},k} := \mathsf{FNN}^{(m)}(\boldsymbol{\theta}_{\mathsf{UPD}}^{(k)}), \qquad f_{\boldsymbol{\theta}}^{\mathsf{AGG},k} := \mathsf{FNN}^{(m)}(\boldsymbol{\theta}_{\mathsf{AGG}}^{(k)}),$$

and use this shorthand throughout. Further, recall that we index global layers by $j \in [J]$, where $\{j_k : k \in [K]\}$ denote the edge-inserting layers and $\{a_k : k \in [K]\}$ the aggregation layers. The output dimension of the $j$-th global layer is denoted by $d_j \in \mathbb{N}$. Parameter configurations are written as $\boldsymbol{\theta} = (\mathcal{W}, \mathcal{C}, \mathcal{B}) \in \boldsymbol{\Theta}_{\mathsf{BF}}$ (see Appendix E.1).

**Definition 42** (Rooted tree with indexed children). Let $\mathcal{T}$ denote the set of node-attributed, edge-weighted rooted trees, and let $G$ be an edge-weighted graph. Given $v \in V(G)$, neighbors $u_1, \ldots, u_d \in N_G(v)$, and trees $t_1, \ldots, t_d \in \mathcal{T}$, we define

$$T_v(t_1, \ldots, t_d) = T \in \mathcal{T}$$

to be the rooted tree with root $r$ labeled $v$, and children $1, \ldots, d$, where for each $i \in [d]$ the subtree rooted at $i$ is $t_i$ and edge-weights

$$w_T(r, i) := w_G(v, u_i).$$

For $t \in \mathcal{T}$, we denote the label of its root by $\mathrm{root}(t)$.

**Definition 43** (Set of computation trees). Let $G \in \mathcal{G}$ denote a graph. We define $\mathcal{T}_G^k(v)$, the *set of computation trees of $G$ rooted in $v \in V(G)$ of length $k \in [K]_0$* recursively as follows. For $k = 0$, let $\mathcal{T}_G^0(v)$ be the set that contains the tree $t_v^0$ given by a single vertex $r$ with the node-label $a_{t_v^0}(r) = a_G(v)$. For $k > 0$, we define

$$\mathcal{T}_G^k(v) := \{T_v(t_1, \ldots, t_{d_{a_k}}) : u_i \in N_G(v), t_i \in \mathcal{T}_G^{k-1}(u_i), i \in [d_{a_k}]\},$$

where $d_{a_k}$ is the output dimension of the $k$-th aggregation.

**Definition 44** (FNN along computation tree)**.** Let $G \in \mathcal{G}_{\text{BF}}$, and $\boldsymbol{\theta} \in \Theta_{\text{BF}}$. We define the *FNN of $G$ and parameters $\boldsymbol{\theta}$ along* $t_v^k \in \mathcal{T}_G^k(v)$ recursively as follows. For $k = 0$ and $t_v^0 \in \mathcal{T}_G^0(v)$ with $\text{root}(t_v^0) = r$ let

$$H^{(0)}(\boldsymbol{\theta})(t_v^0) := a_{t_v^0}(r) = a_G(v).$$

Further for $k > 0$ and $t_v^k = T_v(t_1, \ldots, t_{d_{a_k}}) \in \mathcal{T}_G^k(v)$ with $\text{root}(t_v^k) = r$ define

$$H^{(k)}(\boldsymbol{\theta})(t_v^k) := f_{\boldsymbol{\theta}}^{\text{UPD},k} \left( \left( f_{\boldsymbol{\theta}}^{\text{AGG},k} \begin{pmatrix} H^{(k-1)}(\boldsymbol{\theta})(t_i) \\ w_{t_v^k}(i,r) \end{pmatrix} \right)_{i \in [d_{a_k}]} \right)$$

$$= f_{\boldsymbol{\theta}}^{\text{UPD},k} \left( \left( f_{\boldsymbol{\theta}}^{\text{AGG},k} \begin{pmatrix} H^{(k-1)}(\boldsymbol{\theta})(t_i) \\ w_G(v, \text{root}(t_i)) \end{pmatrix} \right)_{i \in [d_{a_k}]} \right).$$

**Definition 45** (Min-aggregation computation trees)**.** Let $G \in \mathcal{G}_{\text{BF}}$, and $\boldsymbol{\theta} \in \Theta_{\text{BF}}$. We define $\mathcal{T}_{G,\boldsymbol{\theta}}^k(v)$, the *set of min-aggregation computation trees of $G$ and a network $\boldsymbol{\theta}$ rooted in $v \in V(G)$ of length $k \in [K]_0$* recursively as follows. For $k = 0$ let $\mathcal{T}_{G,\boldsymbol{\theta}}^0(v) := \mathcal{T}_G^0(v)$ be the set that contains the tree given by the single vertex-labeled $a_G(v)$.

For $k > 0$ we define

$$\mathcal{T}_{G,\boldsymbol{\theta}}^k(v) := \left\{ T_v(\tau_1, \ldots, \tau_{d_{a_k}}) \; \middle| \; \forall i \in [d_{a_k}] : \; \tau_i \in \arg \min_{\tau \in \mathcal{S}_{\boldsymbol{\theta}}^{k-1}(v)} \left( f_{\boldsymbol{\theta}}^{\text{AGG},k} \begin{pmatrix} H^{(k-1)}(\boldsymbol{\theta})(\tau) \\ w_G(v, \text{root}(\tau)) \end{pmatrix} \right)_i \right\},$$

where

$$\mathcal{S}_{\boldsymbol{\theta}}^{k-1}(v) := \bigcup_{u \in N_G(v)} \mathcal{T}_{G,\boldsymbol{\theta}}^{k-1}(u).$$

The next lemma establishes that the FNN along a min-aggregation tree rooted in $v \in V(G)$ is equal to the feature of $v$.

**Lemma 46.** *Let $G \in \mathcal{G}_{\text{BF}}$, $v \in V(G)$ and $\boldsymbol{\theta} \in \Theta_{\text{BF}}$. Then for any $k \in [K]_0$ and $\tau_v^k \in \mathcal{T}_{G,\boldsymbol{\theta}}^k(v)$*

$$\boldsymbol{h}_v^{(k)} = H^{(k)}(\boldsymbol{\theta})(\tau_v^k)$$

*Proof.* Fix $G \in \mathcal{G}_{\text{BF}}$, and $\boldsymbol{\theta} \in \Theta_{\text{BF}}$. We prove the claim by induction on $k \in [K]_0$.

**Base case $k = 0$.** For any $\tau_v^0 \in \mathcal{T}_{G,\boldsymbol{\theta}}^0(v)$,

$$\boldsymbol{h}_v^{(0)} = a_G(v) = H^{(0)}(\boldsymbol{\theta})(\tau_v^0).$$

**Induction step.** Fix $v \in V(G)$ and $\tau_v^k \in \mathcal{T}_{G,\boldsymbol{\theta}}^k(v)$ such that $\tau_v^k = T_v(\tau_1, \ldots, \tau_{d_{a_k}})$ where $\tau_i \in \mathcal{S}_{\boldsymbol{\theta}}^{k-1}(v)$ for $i \in [d_{a_k}]$. Assume the claim holds for $k-1$. Then for every $\tau \in \mathcal{S}_{\boldsymbol{\theta}}^{k-1}(v)$ with root labeled $u \in N_G(v)$ it holds $\boldsymbol{h}_u^{(k-1)} = H^{(k-1)}(\boldsymbol{\theta})(\tau)$. Thus for every $i \in [d_{a_k}]$

$$\min_{\tau \in \mathcal{S}_{\boldsymbol{\theta}}^{k-1}(v)} \left( f_{\boldsymbol{\theta}}^{\text{AGG},k} \begin{pmatrix} H^{(k-1)}(\boldsymbol{\theta})(\tau_i) \\ w_G(v, \text{root}(\tau_i)) \end{pmatrix} \right)_i = \min_{u \in N(v)} \left( f_{\boldsymbol{\theta}}^{\text{AGG},k} \begin{pmatrix} \boldsymbol{h}_u^{(k-1)} \\ w_G(v, u) \end{pmatrix} \right)_i$$

and hence

$$H^{(k)}(\boldsymbol{\theta})(\tau_v^k) = f_{\boldsymbol{\theta}}^{\text{UPD},k} \left( \left( f_{\boldsymbol{\theta}}^{\text{AGG},k} \begin{pmatrix} H^{(k-1)}(\boldsymbol{\theta})(\tau_i) \\ w_G(v, \text{root}(\tau_i)) \end{pmatrix} \right)_{i \in [d_{a_k}]} \right)$$

$$= f_{\boldsymbol{\theta}}^{\text{UPD},k} \left( \min_{\tau \in \mathcal{S}_{\boldsymbol{\theta}}^{k-1}(v)} f_{\boldsymbol{\theta}}^{\text{AGG},k} \begin{pmatrix} H^{(k-1)}(\boldsymbol{\theta})(\tau) \\ w_G(v, \text{root}(\tau)) \end{pmatrix} \right)$$

$$= f_{\boldsymbol{\theta}}^{\text{UPD},k} \left( \min_{u \in N(v)} f_{\boldsymbol{\theta}}^{\text{AGG},k} \begin{pmatrix} \boldsymbol{h}_u^{(k-1)} \\ w_G(v, u) \end{pmatrix} \right) = \boldsymbol{h}_v^{(k)},$$

where the minimum is taken element-wise. $\qquad\square$

The next lemma shows that even though the MPNN minimizes features somewhat "greedily" layer by layer, the chosen computation tree of the MPNN $\tau_v^J \in \mathcal{T}_{G,\boldsymbol{\theta}}^J(v)$ minimizes the feature "globally" up to negative weights.

**Lemma 47.** *Let $G \in \mathcal{G}_{BF}$, $v \in V(G)$ and $\boldsymbol{\theta} \in \boldsymbol{\Theta}_{BF}$. Then for any $k \in [K]_0$, $\tau_v^k \in \mathcal{T}_{G,\boldsymbol{\theta}}^k(v)$ and $t_v^k \in \mathcal{T}_G^k(v)$ it holds*

$$H^{(k)}(\boldsymbol{\theta})(\tau_v^k) \;\leq\; H^{(k)}(\boldsymbol{\theta}^+)(t_v^k)$$

*element-wise.*

*Proof.* We prove the statement by induction on $k \in [K]_0$.

**Base case ($j = 0$).** Let $v \in V(G)$, $\tau_v^0 \in \mathcal{T}_{G,\boldsymbol{\theta}}^0(v)$ and $t_v^0 \in \mathcal{T}_G^0(v)$, then

$$H^{(0)}(\boldsymbol{\theta})(\tau_v^0) = a_G(v) = H^{(0)}(\boldsymbol{\theta}^+, G)(t_v^0).$$

**Inductive step.** Assume the hypothesis holds for $k-1$. Let $u \in V(G)$, $\tau \in \mathcal{T}_{G,\boldsymbol{\theta}}^{k-1}(u)$ and $t \in \mathcal{T}_G^{k-1}(u)$. Applying the monotonicity result Lemma 34 and the induction hypothesis gives

$$f_{\boldsymbol{\theta}}^{\mathsf{AGG},k}\begin{pmatrix} H^{(k-1)}(\boldsymbol{\theta})(\tau) \\ w_G(v, \mathrm{root}(\tau)) \end{pmatrix} \leq f_{\boldsymbol{\theta}^+}^{\mathsf{AGG},k}\begin{pmatrix} H^{(k-1)}(\boldsymbol{\theta}^+, G)(t) \\ w_G(v, \mathrm{root}(t))). \end{pmatrix}$$

Now fix $v \in V(G)$ and define $S^{k-1}(v) := \cup_{u \in N(v)} \mathcal{T}_G^{k-1}(u)$. Then the above implies

$$\min_{\tau \in \mathcal{S}_{\boldsymbol{\theta}}^{k-1}(v)} f_{\boldsymbol{\theta}}^{\mathsf{AGG},k}\begin{pmatrix} H^{(k-1)}(\boldsymbol{\theta})(\tau) \\ w_G(v, \mathrm{root}(\tau)) \end{pmatrix} = \min_{u \in N(v)} \min_{\tau \in \mathcal{T}_{G,\boldsymbol{\theta}}^{k-1}(u)} f_{\boldsymbol{\theta}}^{\mathsf{AGG},k}\begin{pmatrix} H^{(k-1)}(\boldsymbol{\theta})(\tau) \\ w_G(v, \mathrm{root}(\tau)) \end{pmatrix}$$

$$\leq \min_{u \in N(v)} \min_{t \in \mathcal{T}_G^{k-1}(u)} f_{\boldsymbol{\theta}^+}^{\mathsf{AGG},k}\begin{pmatrix} H^{(k-1)}(\boldsymbol{\theta}^+, G)(t) \\ w_G(v, \mathrm{root}(t))). \end{pmatrix} = \min_{t \in \mathcal{S}^{k-1}(v)} f_{\boldsymbol{\theta}^+}^{\mathsf{AGG},k}\begin{pmatrix} H^{(k-1)}(\boldsymbol{\theta}^+, G)(t) \\ w_G(v, \mathrm{root}(t))) \end{pmatrix}.$$

Further fix $\tau_v^k \in \mathcal{T}_{G,\boldsymbol{\theta}}^k(v)$ and $t_v^k \in \mathcal{T}_G^k(v)$. W.l.o.g. assume that $t_v^k = T_v(t_1, \ldots, t_{d_{a_k}})$ and $\tau_v^k = T_v(\tau_1, \ldots, \tau_{d_{a_k}})$ where $t_i \in S^{k-1}(v)$ and $\tau_i \in \mathcal{S}_{\boldsymbol{\theta}}^{k-1}(v)$ for all $i \in [d_{a_k}]$. Applying the monotonicity result 34 on $f_{\boldsymbol{\theta}}^{\mathsf{UPD},k}$ together with the above gives

$$H^{(k)}(\boldsymbol{\theta})(\tau_v^k) = f_{\boldsymbol{\theta}}^{\mathsf{UPD},k}\left( \left( f_{\boldsymbol{\theta}}^{\mathsf{AGG},k}\begin{pmatrix} H^{(k-1)}(\boldsymbol{\theta})(\tau_i) \\ w_G(v,\mathrm{root}(\tau_i)) \end{pmatrix} \right)_{i \in [d_{a_k}]} \right)$$

$$= f_{\boldsymbol{\theta}}^{\mathsf{UPD},k}\left( \min_{\tau \in \mathcal{S}_{\boldsymbol{\theta}}^{k-1}(v)} f_{\boldsymbol{\theta}}^{\mathsf{AGG},k}\begin{pmatrix} H^{(k-1)}(\boldsymbol{\theta})(\tau) \\ w_G(v,\mathrm{root}(\tau)) \end{pmatrix} \right)$$

$$\leq f_{\boldsymbol{\theta}^+}^{\mathsf{UPD},k}\left( \min_{t \in \mathcal{S}^{k-1}(v)} f_{\boldsymbol{\theta}^+}^{\mathsf{AGG},k}\begin{pmatrix} H^{(k-1)}(\boldsymbol{\theta}^+,G)(t) \\ w_G(v,\mathrm{root}(t)) \end{pmatrix} \right)$$

$$\leq f_{\boldsymbol{\theta}^+}^{\mathsf{UPD},k}\left( \left( f_{\boldsymbol{\theta}^+}^{\mathsf{AGG},k}\begin{pmatrix} H^{(k-1)}(\boldsymbol{\theta}^+,G)(t_i) \\ w_G(v,\mathrm{root}(t_i)) \end{pmatrix} \right)_{i \in [d_{a_k}]} \right) = H^{(k)}(\boldsymbol{\theta}^+)(t_v^k)$$

which concludes the induction step. $\qquad\square$

### F.5. Application to Walk-lifted FNNs

In this section, we introduce a third and final parametrized FNN–the Walk-lifted FNN—so that we can easily access all relevant parameters. We will see that the Walk-lifted FNN inherits all the properties of the simple FNN, as shown in Appendix F.3. Moreover, we show that when the aggregation indeed follows a single path, the MPNN can be expressed by the Walk-lifted FNN.

**Definition 48** (Set of walks). Let $G \in \mathcal{G}_{BF}$ denote a graph and $v_K \in V(G)$. We define

$$\mathcal{P}_G^K(v_K) := \{(v_0, \ldots, v_K) \in V(G)^{K+1} \colon (v_{k-1}, v_k) \in E(G) \text{ for all } k \in [K]\}$$

as the *set of walks* of length $K \in \mathbb{N}$ in $G$ and ending in $v_K$. Let $p := (v_0, \ldots, v_K) \in \mathcal{P}_G^K(v_K)$. We define the *weight vector of p* as the sequence of weights

$$z^p := (a_G(v_0), w_G(v_0, v_1), \ldots, w_G(v_{K-1}, v_K)) \in \mathbb{R}_{\geq 0}^{K+1}.$$

Recalling that by definition, BF-graphs have non-negative real-valued node and edge features $a_G, w_G \geq 0$. Further, we define the *tree $t^k(p)$ of p of length $k \in [K]_0$* recursively as follows. For $k = 0$, let $t^0(p) \in \mathcal{T}_G^0(v_0)$ denote the tree with a single vertex with node label $a_G(v_0)$. For $k > 0$. we define

$$t^k(p) := T_{v_k}(t^{k-1}(p), \ldots, t^{k-1}(p)) \in \mathcal{T}_G^k(v_k)$$

by attaching $d_{a_k}$ copies of the tree $t^{k-1}(p)$ underneath a root node with label $v_k$ (where $d_{a_k}$ denotes the output dimension of the $k$-th aggregation layer) .

*Remark* 49 (BF-distance). Let $G \in \mathcal{G}_{\mathrm{BF}}$ and $v \in V(G)$. Then the $K$-step BF-distance (see Definition 11) is given by

$$x_v^{(K)} = \min\{\|z^p\|_1 : p \in \mathcal{P}_G^K(v)\}.$$

*Remark* 50 (Path vector of path graphs). Let $w \in \mathbb{R}^{K+1}$ and recall that the path graph $P(w)$ was defined by $V(P(w)) := \{v_0, v_1, \ldots, v_K\}$ with node-labels given by $a_G(v_0) = w_0$ and $a_G(v_i) = \beta$ for all $i \in [K]$ and $E(P(w)) := \{(v_{i-1}, v_i) \mid i \in [K]\}$, where the edge-weight function $w_{P(w)} \colon E(P(w)) \to \mathbb{R}^+$ is defined as $(v_{i-1}, v_i) \mapsto w_i$, for $i \in [K]_0$. If we let $p = (v_0, \ldots, v_K) \in \mathcal{P}_G^K(v_K)$. Then the above implies that

$$z^p = (a_G(v_0), w_G(v_0, v_1), \ldots, w_G(v_{K-1}, v_K)) = w.$$

Hence for any training sample $v_K \in V(P_\beta(w))$, where $w \in S_{x,K}$ it holds for the path $p$ as above that

$$z^p = w$$

which we will make use of later on.

*Remark* 51 (1D aggregation). If $d_{a_k} = 1$ for all $k \in [K]$. then any tree in $\mathcal{T}_G^K(v)$ is in fact a walk, and we can identify

$$\mathcal{T}_G^K(v) \equiv \mathcal{P}_G^K(v).$$

In this case we write $z^t$ for $t \in \mathcal{T}_G^K(v)$, meaning $z^p$ where $p \in \mathcal{P}_G^K(v)$ is such that $t^K(p) = t$.

Next, we define the Walk-lifted FNN, which is a simple FNN as in Appendix F.3, but with a very specifically chosen bias vector. Intuitively, we choose the bias vector so that each edge weight along a given walk is added to the corresponding edge-insertion layer. In general, the Walk-lifted FNN does not capture the same expressivity as an FNN along a tree, and thus is not powerful enough to express any possible feature of the MPNN. In case any minimum aggregation tree is essentially a walk, we will see that the Walk-lifted FNN is in fact equal to the feature of the MPNN.

**Definition 52** (Bias of weight vector and Walk-lifted FNN). Let $\mathcal{C} \in \Theta_C$ and $z = (z_0, \ldots, z_K) \in \mathbb{R}_{\geq 0}^{K+1}$ we define the *bias of a weight vector* as

$$\mathcal{B}(z, \mathcal{C}) := (b^1(z, \mathcal{C}), \ldots, b^J(z, \mathcal{C}))$$

where

$$b^j(z, \mathcal{C}) := \begin{cases} C^k z_k & j = j_k, k \in [K] \\ \mathbf{0} & \text{else} \end{cases} \in \mathbb{R}^{d_j}, \quad j \in [J].$$

Further let $\mathcal{W} \in \Theta_W$ and $\mathcal{B} \in \Theta_b$. We define the *Walk-lifted FNN* as

$$H_{\mathrm{wl}}(\mathcal{W}, \mathcal{C}, \mathcal{B})(z) := \mathsf{FNN}^{(J)}(\mathcal{W}, \mathcal{B} + \mathcal{B}(z, \mathcal{C}))(z_0), \quad z \in \mathbb{R}_{\geq 0}^{K+1}.$$

The next lemma establishes the connection between the FNN of a computation tree and the Walk-lifted FNN in case the tree is essentially a walk.

**Lemma 53.** *Let $G \in \mathcal{G}_{BF}$ and $(\mathcal{W}, \mathcal{C}, \mathcal{B}) = \theta \in \Theta_{BF}$. Then for any $v \in V(G)$ and $p \in \mathcal{P}_G^K(v)$ it holds*

$$H_{\mathrm{wl}}(\theta)(z^p) = H^{(k)}(\theta)(t^K(p)).$$

*Proof.* We extend the above definition to $k \in [K]_0$ as follows, using the equivalent global indexing

$$H_{\mathrm{wl}}^{(k)}(\mathcal{W}, \mathcal{C}, \mathcal{B})(\boldsymbol{z}) := \mathrm{FNN}_{(\mathrm{UPD}, k, m)}^{(J)}(\mathcal{W}, \mathcal{B} + \mathcal{B}(\boldsymbol{z}, \mathcal{C}))(\boldsymbol{z}_0), \quad \boldsymbol{z} \in \mathbb{R}_{\geq 0}^{K+1}.$$

We now prove

$$H_{\mathrm{wl}}^{(k)}(\boldsymbol{\theta})(\boldsymbol{z}^p) = H^{(k)}(\boldsymbol{\theta})(t^k(p))$$

by induction on $k \in [K]_0$. Once proven, this concludes the lemma.

**Base case ($j = 0$).** It follows from the above definitions and the fact that $t^0(p) \in \mathcal{T}_G^0(v_0)$

$$H_{\mathrm{wl}}^{(0)}(\boldsymbol{\theta})(\boldsymbol{z}^p) = \mathrm{FNN}_0^{(J)}(\mathcal{W}, \mathcal{B} + \mathcal{B}(\boldsymbol{z}, \mathcal{C}))(\boldsymbol{z}_0^t) = \boldsymbol{z}_0^t = a_G(v_0) = H^{(0)}(\boldsymbol{\theta})(t^0(p)).$$

**Induction Step.** Assuming the claim holds for $k - 1$, we find

$$
\begin{aligned}
H^{(k)}(\boldsymbol{\theta})(t^k(p)) &= f_{\boldsymbol{\theta}}^{\mathrm{UPD},k} \left( f_{\boldsymbol{\theta}}^{\mathrm{AGG},k} \left( \begin{smallmatrix} H^{(k-1)}(\boldsymbol{\theta})(t^{k-1}(p)) \\ w_G(v_k, \mathrm{root}(t^{k-1}(p))) \end{smallmatrix} \right) \right) \\
&= f_{\boldsymbol{\theta}}^{\mathrm{UPD},k} \left( f_{\boldsymbol{\theta}}^{\mathrm{AGG},k} \left( \begin{smallmatrix} H_{\mathrm{wl}}^{(k-1)}(\boldsymbol{\theta})(\boldsymbol{z}^p) \\ \boldsymbol{z}_k^p \end{smallmatrix} \right) \right) \\
&= \sigma(\boldsymbol{W}^{(\mathrm{UPD},k,m)} \cdots \sigma(\boldsymbol{W}^{(\mathrm{AGG},k,1)} H_{\mathrm{wl}}^{(k-1)}(\boldsymbol{\theta})(\boldsymbol{z}^p) + \boldsymbol{C}^k \boldsymbol{z}_k^p + \boldsymbol{b}^{(\mathrm{AGG},k,1)}) \cdots + \boldsymbol{b}^{(\mathrm{UPD},k,m)}) \\
&= H_{\mathrm{wl}}^{(k)}(\boldsymbol{\theta})(\boldsymbol{z}^p)
\end{aligned}
$$

which proves the induction step. $\qquad\square$

The following lemmas are all immediate consequences of the above and the properties derived in the last two sections. Combined, they will be the base of all estimations of features in the two following sections.

**Lemma 54** (Linearity). *Let $\boldsymbol{\theta} = (\mathcal{W}, \mathcal{C}, \mathcal{B}) \in \Theta_{BF}$ and assume that all parameters of $\boldsymbol{\theta}$ are (element-wise) non-negative. Then for all $\boldsymbol{z} \in \mathbb{R}_{\geq 0}^{K+1}$,*

$$H_{\mathrm{wl}}(\mathcal{W}, \mathcal{C}, \mathcal{B})(\boldsymbol{z}) = \sum_{l=1}^{J} \Big( \prod_{s=l+1}^{J} \boldsymbol{W}^s \Big) \boldsymbol{b}^l + \Big( \prod_{s=1}^{J} \boldsymbol{W}^s \Big) \boldsymbol{z}_0 + \sum_{k \in [K]} \Big( \prod_{s=j_k+1}^{J} \boldsymbol{W}^s \Big) \boldsymbol{C}^k \boldsymbol{z}_k.$$

*Moreover, the Walk-lifted FNN is linear in $\boldsymbol{z}$, i.e.,*

$$H_{\mathrm{wl}}(\mathcal{W}, \mathcal{C}, \mathcal{B})(\boldsymbol{z}) = \sum_{k=0}^{K} \boldsymbol{z}_k \, H_{\mathrm{wl}}(\mathcal{W}, \mathcal{C}, 0)(\boldsymbol{e}_k^{K+1}) + H_{\mathrm{wl}}(\mathcal{W}, \mathcal{C}, \mathcal{B})(0)$$

*where $\{\boldsymbol{e}_l^{K+1}\}_{l=0}^{K}$ denotes the canonical unit-length basis of $\mathbb{R}^{K+1}$.*

*Proof.* Combining Lemma 33 and Definition 52 of $\mathcal{B}(\boldsymbol{z}, \mathcal{C})$. $\qquad\square$

**Lemma 55.** *Let $\boldsymbol{z}_1, \boldsymbol{z}_2 \in \mathbb{R}_{\geq 0}^{K+1}$, and $\boldsymbol{\theta}_1 = (\mathcal{W}_1, \mathcal{C}_1, \mathcal{B}_1), \boldsymbol{\theta}_2 = (\mathcal{W}_2, \mathcal{C}_2, \mathcal{B}_2) \in \Theta_{BF}$. Assume that $\boldsymbol{z}_1 \leq \boldsymbol{z}_2$, $\boldsymbol{\theta}_1 \leq \boldsymbol{\theta}_2$ and $\mathcal{W}_2, \mathcal{C}_2 \geq 0$, element-wise. Then*

$$H_{\mathrm{wl}}(\boldsymbol{\theta}_1)(\boldsymbol{z}_1) \leq H_{\mathrm{wl}}(\boldsymbol{\theta}_2)(\boldsymbol{z}_2),$$

*element-wise.*

*Proof.* Note that by assumption $\mathcal{B}_1 + \mathcal{B}(\boldsymbol{z}_2, \mathcal{C}_1) \leq \mathcal{B}_2 + \mathcal{B}(\boldsymbol{z}_2, \mathcal{C}_2)$. Thus, the lemma follows by Lemma 34. $\qquad\square$

**Corollary 56.** *For all $\boldsymbol{z} \in \mathbb{R}_{\geq 0}^{K+1}$ and parameter $(\mathcal{W}, \mathcal{C}, \mathcal{B}) \in \Theta_{BF}$,*

$$\left| H_{\mathrm{wl}}(\mathcal{W}, \mathcal{C}, \mathcal{B})(\boldsymbol{z}) - H_{\mathrm{wl}}(\mathcal{W}^+, \mathcal{C}^+, 0)(\boldsymbol{z}) \right| \leq G(\mathcal{W}, \mathcal{B}(\boldsymbol{z}, \mathcal{C}))(\boldsymbol{z}) + H_{\mathrm{wl}}(|\mathcal{W}|, |\mathcal{C}|, |\mathcal{B}|)(0).$$

*Proof.* The statement follows by choosing $\mathcal{B}_C = \mathcal{B}(\mathbf{z}, \mathcal{C})$ and $\mathcal{B} = \mathcal{B}$ in Corollary 41. $\qquad\square$

**Lemma 57.** *Let $\boldsymbol{\theta} \in \boldsymbol{\Theta}_{BF}$, and $G \in \mathcal{G}_{BF}$. Then for any $v \in V(G)$ and $p \in \mathcal{P}_G^K(v)$ it holds*

$$\boldsymbol{h}^{(K)}(v) \le H_{\mathrm{wl}}(\boldsymbol{\theta}^+)(\boldsymbol{z}^p).$$

*and*

$$\boldsymbol{h}^{(K)}(v) = H_{\mathrm{wl}}(\boldsymbol{\theta})(\boldsymbol{z}^p).$$

*if $t^K(p) \in \mathcal{T}_{G,\boldsymbol{\theta}}^K(v)$*

*Proof.* By Lemma 46, $\boldsymbol{h}_v^{(K)} = H^{(k)}(\boldsymbol{\theta})(\tau_v^K)$ for any $\tau_v^K \in \mathcal{T}_{G,\boldsymbol{\theta}}^K(v)$. Thus if $t^K(p) \in \mathcal{T}_{G,\boldsymbol{\theta}}^K(v)$ it must be

$$\boldsymbol{h}_v^{(K)} = H^{(k)}(\boldsymbol{\theta})(t^K(p)) = H_{\mathrm{wl}}(\boldsymbol{\theta})(\boldsymbol{z}^p)$$

by Lemma 53. Similarly, choosing $t_v^K = t^K(p) \in \mathcal{T}_G^K(v)$ in Lemma 47 and applying Lemma 53 gives

$$\boldsymbol{h}_v^{(K)} = H^{(k)}(\boldsymbol{\theta})(\tau_v^K) \le H^{(k)}(\boldsymbol{\theta}^+)(t^K(p)) = H_{\mathrm{wl}}(\boldsymbol{\theta}^+)(\boldsymbol{z}^p).$$

$\qquad\square$

In the preceding sections, we introduced and analyzed three objects: FNNs, FNNs along computation trees, and Walk-lifted FNNs.

We first introduced FNNs as a simple setting in which we could derive fundamental structural properties. These properties were then applied in the subsequent sections to the aggregation and update functions of the MPNN, as well as to Walk-lifted FNNs. Next, FNNs along computation trees allowed us to express MPNN features in closed form, enabling us to compare the MPNN output with that obtained when aggregation follows a computation tree different from the min-aggregation tree.

The Walk-lifted FNN is the main object we will work with from now on. As shown above, it inherits the relevant properties established for simple FNNs while explicitly revealing its dependence on the network parameters and on the edge weights along a given path. This explicit structure makes it particularly convenient to analyze. Moreover, as established in the previous lemma, the Walk-lifted FNN can be used to express or bound the features of the MPNN.

We will use these results as follows.

In the next section, we derive bounds on the network parameters under the assumption that the loss is close to its global minimum. To this end, we apply the previous lemma to obtain a lower bound on the empirical loss by removing the ReLU nonlinearity and replacing the MPNN feature $\boldsymbol{h}_v^K$ with the Walk-lifted FNN $H_{\mathrm{wl}}(\boldsymbol{\theta}^+)(\boldsymbol{z}^p)$.

Finally, in the last section, we show that for the path training samples $T_{S_{x,K}}$, the min-aggregation computation tree must follow the path and is therefore essentially a walk. In this case as well, the MPNN feature can be replaced by the Walk-lifted FNN, allowing us to work entirely within this framework.

### F.6. Small empirical loss and structure of parameters

For the remainder of this section, we will consider the MPNN and its parameters as defined in Appendix E.

**Definition 58.** For each $k \in [K]_0$, we define

$$l_k := \big|\{\, j \in [J] \colon j > j_k \,\}\big| + \mathbf{1}_{k \ne 0} = J - j_k + \mathbf{1}_{k \ne 0}$$

where $j_k$ is the $k$-th edge-inserting layer (c.f. Appendix E.1). Further let

$$L := \sum_{k=0}^K l_k.$$

*Remark* 59. We will see that $l_k$ is the number of weight matrices that the $k$-th edge weight is multiplied by on a given walk. This will be crucial for lower-bounding the regularization term. Further explicit computation gives $L = mK(K + 3)$.

**Definition 60** (Bellman-Ford implementing parameters). We define the *Bellman-Ford (BF) parameter configuration* $\psi \in \boldsymbol{\Theta}_{\mathrm{BF}}$ as follows. For each layer $j \in [J]$, let the bias vector be given by

$$\boldsymbol{b}_\psi^j := \mathbf{o}^{\,d_j},$$

and define the weight matrix $\boldsymbol{W}_\psi^j \in \mathbb{R}^{d_j \times d_{j-1}}$ by

$$\boldsymbol{W}_\psi^j := \begin{pmatrix} 1 & 0 & \cdots & 0 \\ 0 & 0 & \cdots & 0 \\ \vdots & \vdots & \ddots & \vdots \\ 0 & 0 & \cdots & 0 \end{pmatrix},$$

i.e., the matrix whose only nonzero entry is a single "1" in the top-left corner. Further for each $k \in [K]$, let

$$\boldsymbol{C}_\psi^k := (1,0,\dots,0)^\top \in \mathbb{R}^{d_{j_k} \times 1},$$

the vector with a single nonzero entry in the first coordinate. We call the resulting parameter tuple

$$\psi \;=\; \big((\boldsymbol{W}_\psi^j)_{j=1}^J,\; (\boldsymbol{C}_\psi^k)_{k=1}^K,\; (\boldsymbol{b}_\psi^j)_{j=1}^J\big) \in \boldsymbol{\Theta}_{\mathrm{BF}}$$

the *BF parameter configuration*.

**Lemma 61.** *For $L$ defined as above, it holds that*

$$\min_{\boldsymbol{\theta} \in \boldsymbol{\Theta}_{BF}} \mathcal{L}(\boldsymbol{\theta}) \;\leq\; \eta\, L.$$

*Proof.* Let $\psi$ denote the BF configuration from Definition 60. First note that the BF parameters exactly implement the Bellman-Ford algorithm and hence $\mathcal{L}_{\mathrm{emp}}(\psi) = 0$. Thus by definition of $\mathcal{L}$, and using $\boldsymbol{b}_\psi^j = \mathbf{o}^{\,d_j}$, we have

$$\mathcal{L}(\psi) = \eta \mathcal{L}(\psi)_{\mathrm{reg}} = \eta \left( \sum_{k=0}^K \sum_{j > j_k} \|\boldsymbol{W}_\psi^j\|_1 \;+\; \sum_{k \in [K]} \|\boldsymbol{C}_\psi^k\|_1 \right).$$

Since each $\boldsymbol{W}_\psi^j$, $j \in [J]$ and each $\boldsymbol{C}_\psi^k$, $k \in [K]$ contains exactly one nonzero entry,

$$\|\boldsymbol{W}_\psi^j\|_1 = 1, \qquad \|\boldsymbol{C}_\psi^k\|_1 = 1.$$

Hence

$$\mathcal{L}(\psi) = \eta \left( \sum_{k=0}^K \big(J - j_k + \mathbf{1}_{k \neq 0}\big) \right) = \eta \sum_{k=0}^K l_k = \eta L.$$

Since $\psi \in \boldsymbol{\Theta}_{\mathrm{BF}}$, we obtain

$$\min_{\boldsymbol{\theta} \in \boldsymbol{\Theta}_{\mathrm{BF}}} \mathcal{L}(\boldsymbol{\theta}) \;\leq\; \mathcal{L}(\psi) \;=\; \eta L.$$

$\square$

**Lemma 62.** *Let $C \in \mathbb{R}$, $T$ denote some index set and $t := |T|$. Then the solution to*

$$\min_{\boldsymbol{v} \in \mathbb{R}^t} \quad \sum_{l \in T} |v_l|$$

$$s.t. \quad \prod_{l \in T} v_l = C$$

*is given by $v_l = C^{\frac{1}{t}}$, for all $l \in T$.*

*Proof.* Because of the symmetry of the absolute value, it suffices to consider $v \in \mathbb{R}^t_{\geq 0}$. We then see from the inequality of arithmetic and geometric means that

$$\frac{1}{t}\sum_{l \in T}|v_l| \geq \Big(\prod_{l \in T}v_l\Big)^{\frac{1}{t}} = C^{\frac{1}{t}}.$$

Thus $\frac{1}{t}\sum_{l \in T}|v_l|$ is minimal in case of equality, which is exactly the case if $v_l = C^{\frac{1}{t}}$. $\qquad\square$

**Lemma 63.** *Let $M \in \mathbb{R}_{\geq 0}$. Then $\left(\frac{M}{x}\right)^x \leq \exp\left(\frac{M}{e}\right)$, for all $x \in \mathbb{R}_{>0}$.*

*Proof.* We want to bound the maximum of the function $f\colon \mathbb{R}_{>0} \to \mathbb{R}_{>0}$, $f(x) := \left(\frac{M}{x}\right)^x$. To this end, we compute its extreme points as

$$0 = f'(x) = \exp\left((\ln(M) - \ln(x))x\right)' = \left(-\tfrac{1}{x}x + \ln(M) - \ln(x)\right)\exp\left((\ln(M) - \ln(x))x\right)$$
$$= (-1 + \ln(M) - \ln(x))f(x).$$

Since $f(x) \neq 0$, we conclude that the extreme point is given by

$$-1 + \ln(M) = \ln(x) \implies x = \exp(-1 + \ln(M)) = \tfrac{M}{e}.$$

Further it holds

$$f(\tfrac{M}{e}) = \exp\left(\tfrac{M}{e}\right)$$
$$\lim_{x \to \infty}f(x) \leq \lim_{x \to \infty}\tfrac{1}{2}^x = 0 \leq \exp\left(\tfrac{M}{e}\right)$$
$$\lim_{x \to 0}f(x) = \lim_{x \to \infty}f(\tfrac{1}{x}) = \lim_{x \to \infty}\exp\left(\tfrac{\ln(Mx)}{x}\right) = 1 \leq \exp\left(\tfrac{M}{e}\right)$$

and thus $\left(\frac{M}{x}\right)^x \leq \exp\left(\frac{M}{e}\right)$, for all $x \in \mathbb{R}_{>0}$. $\qquad\square$

To prove Theorem 16, we begin by deriving basic bounds on the effect that nonzero biases and negative entries in the weight matrices can have on the output of the MPNN.

**Lemma 64.** *Let $\boldsymbol{\theta} \in \boldsymbol{\Theta}_{BF}$ and $0 \leq \varepsilon \leq \eta L$. Assume that $\mathcal{L}(\boldsymbol{\theta})$ is within $\varepsilon$ of its global minimum. Then the following holds.*

1. *For any $T \subset [J]$,*

$$\prod_{j \in T}\|\boldsymbol{W}^j\|_1 \leq \exp(L).$$

2. *For any $k \in [K]$ and $T \subset \{j_k + 1, \ldots, J\}$,*

$$\prod_{j \in T}\|\boldsymbol{W}^j\|_1\|\boldsymbol{C}^k\|_1 \leq \exp(L).$$

*Proof.* We prove the first statement; the second is analogous.

Let $t := |T|$ and $C := \prod_{j \in T}\|\boldsymbol{W}^j\|_1$. By Lemma 62,

$$tC^{1/t} \leq \sum_{j \in T}\|\boldsymbol{W}^j\|_1.$$

Applying Lemma 61 gives

$$\eta\sum_{j \in T}\|\boldsymbol{W}^j\|_1 \leq \mathcal{L}(\boldsymbol{\theta}) \leq \eta L + \varepsilon \leq 2\eta L,$$

and therefore

$$C^{1/t} \leq \frac{2L}{t} \quad \Rightarrow \quad C \leq \left(\frac{2L}{t}\right)^t.$$

Finally, by Lemma 63, $C \leq \left(\frac{2L}{t}\right)^t \leq \exp(2L/e) \leq \exp(L)$, which proves the claim. $\qquad\square$

**Corollary 65.** *Let $\boldsymbol{\theta} \in \boldsymbol{\Theta}_{BF}$ and $0 \leq \varepsilon \leq \eta L$. Assume that $\mathcal{L}(\boldsymbol{\theta})$ is within $\varepsilon$ of its global minimum. Then,*

$$H_{\mathrm{wl}}(|\mathcal{W}|, |\mathcal{C}|, |\mathcal{B}|)(0) \leq \exp(L) \sum_{j=1}^{J} \|\boldsymbol{b}^j\|_1,$$

$$G(\mathcal{W}, \mathcal{B}(\boldsymbol{z}, \mathcal{C}))(z_0) \leq \exp(L) \Big( \sum_{l=1}^{J} \|(\boldsymbol{W}^l)^-\|_1 + \sum_{k=1}^{K} \|(\boldsymbol{C}^k)^-\|_1 \Big) \|\boldsymbol{z}\|_1, \quad \boldsymbol{z} \in \mathbb{R}_{\geq 0}^{K+1}.$$

*Proof.* First note that since $d_J = 1$, it holds that $H_{\mathrm{wl}}(|\mathcal{W}|, |\mathcal{C}|, |\mathcal{B}|)(0)$ and $G(\mathcal{W}, \mathcal{B}(\boldsymbol{z}, \mathcal{C}))(z_0)$ belong to $\mathbb{R}_{\geq 0}$. Using Lemmas 54 and 64, we obtain

$$H_{\mathrm{wl}}(|\mathcal{W}|, |\mathcal{C}|, |\mathcal{B}|)(0) \leq \sum_{j=1}^{J} \prod_{l=j+1}^{J} \|\boldsymbol{W}^l\|_1 \|\boldsymbol{b}^j\|_1 \leq \Big( \max_{j \in [J]} \prod_{l=j+1}^{J} \|\boldsymbol{W}^l\|_1 \Big) \sum_{j=1}^{J} \|\boldsymbol{b}^j\|_1 \leq \exp(L) \sum_{j=1}^{J} \|\boldsymbol{b}^j\|_1.$$

Recall that by Remark 39, it holds for general biases $\tilde{\mathcal{B}}$ that

$$\|G(\mathcal{W}, \tilde{\mathcal{B}})(\tilde{b}^0)\|_1$$
$$\leq \sum_{l \in [J]_0} \sum_{l'=0}^{l-1} \Big( \prod_{\substack{s=l'+1 \\ s \neq l}}^{J} \|(\boldsymbol{W}^s)^+\|_1 \Big) \|(\boldsymbol{W}^l)^-\|_1 \|(\tilde{b}^{l'})^+\|_1 + \sum_{l' \in [J]_0} \Big( \prod_{s=l'+1}^{J} \|(\boldsymbol{W}^s)^+\|_1 \Big) \|(\tilde{b}^{l'})^-\|_1$$

For our purposes we choose $\tilde{\mathcal{B}} = \mathcal{B}(\boldsymbol{z}, \mathcal{C})$ and $\tilde{b}^0 = z_0$. Further we let $C^0 = 1$ for ease of notation. Noting that $b^{l'}(\boldsymbol{z}, \mathcal{C}) = \sum_{k \in [K]_0} \delta_{l', j_k} \boldsymbol{C}^k z_k$ for $l' \in [J]$ we find

$$\|G(\mathcal{W}, \mathcal{B}(\boldsymbol{z}, \mathcal{C}))(z_0)\|_1$$

$$\leq \sum_{l \in [J]_0} \sum_{l'=0}^{l-1} \Big( \prod_{\substack{s=l'+1 \\ s \neq l}}^{J} \|(\boldsymbol{W}^s)^+\|_1 \Big) \|(\boldsymbol{W}^l)^-\|_1 \sum_{k \in [K]_0} \delta_{l', j_k} \|(\boldsymbol{C}^k)^+\|_1 z_k$$

$$+ \sum_{l' \in [J]_0} \Big( \prod_{s=l'+1}^{J} \|(\boldsymbol{W}^s)^+\|_1 \Big) \sum_{k \in [K]_0} \delta_{l', j_k} \|(\boldsymbol{C}^k)^-\|_1 z_k$$

$$\overset{64}{\leq} \Big( \sum_{l \in [J]_0} \sum_{l'=0}^{l-1} \|(\boldsymbol{W}^l)^-\|_1 \sum_{k \in [K]_0} \delta_{l', j_k} z_k + \sum_{l' \in [J]_0} \sum_{k \in [K]_0} \delta_{l', j_k} \|(\boldsymbol{C}^k)^-\|_1 z_k \Big) \exp(L)$$

$$= \Big( \sum_{l \in [J]_0} \|(\boldsymbol{W}^l)^-\|_1 \sum_{k \in [K]_0} \Big( \sum_{l'=0}^{l-1} \delta_{l', j_k} \Big) z_k + \sum_{k \in [K]_0} \Big( \sum_{l' \in [J]_0} \delta_{l', j_k} \Big) \|(\boldsymbol{C}^k)^-\|_1 z_k \Big) \exp(L)$$

$$\leq \Big( \sum_{l \in [J]_0} \|(\boldsymbol{W}^l)^-\|_1 \sum_{k \in [K]_0} z_k + \sum_{k \in [K]_0} \|(\boldsymbol{C}^k)^-\|_1 z_k \Big) \exp(L)$$

$$\leq \Big( \sum_{l \in [J]_0} \|(\boldsymbol{W}^l)^-\|_1 \|\boldsymbol{z}\|_1 + \sum_{k \in [K]_0} \|(\boldsymbol{C}^k)^-\|_1 \|\boldsymbol{z}\|_1 \Big) \exp(L)$$

$$\leq \exp(L) \Big( \sum_{l \in [J]_0} \|(\boldsymbol{W}^l)^-\|_1 + \sum_{k \in [K]_0} \|(\boldsymbol{C}^k)^-\|_1 \Big) \|\boldsymbol{z}\|_1$$

applying lemma Lemma 64 with appropriate choice for $T$ in each considered summand. $\square$

To simplify the analysis of the loss function and highlight the main forces acting on the parameters, we introduce a modified loss function.

**Definition 66** (Modified Loss Function). We begin by defining a transformation of the neural network parameters. Given $\boldsymbol{\theta} = (\mathcal{W}, \mathcal{C}, \mathcal{B}) \in \boldsymbol{\Theta}_{\mathrm{BF}}$, we define the transformed parameters

$$\tilde{\boldsymbol{\theta}} := (\gamma_0, \ldots, \gamma_K, B, w^-) \in \mathbb{R}_{\geq 0}^{K+3}$$

as follows. To avoid a case distinction, we introduce $C^0 := 1$ as a fixed parameter. Then we set

$$\gamma_k := \prod_{j=j_k+1}^{J} (\boldsymbol{W}^j)^+ (\boldsymbol{C}^k)^+ \in \mathbb{R}_{\geq 0}, \quad k \in \{0, \ldots, K\},$$

$$B := \|\mathcal{B}\|_1 \in \mathbb{R}_{\geq 0},$$

$$w^- := \sum_{l=1}^{J} \|(\boldsymbol{W}^l)^-\|_1 + \sum_{k=1}^{K} \|(\boldsymbol{C}^k)^-\|_1 \in \mathbb{R}_{\geq 0}.$$

Next, we define a new loss function $\tilde{\mathcal{L}}$ on the transformed parameters. As before, we separate the loss into an empirical term and a regularization term:

$$\tilde{\mathcal{L}}(\tilde{\boldsymbol{\theta}}) := \tilde{\mathcal{L}}^{\mathrm{emp}}(\tilde{\boldsymbol{\theta}}) + \eta \tilde{\mathcal{L}}^{\mathrm{reg}}(\tilde{\boldsymbol{\theta}}),$$

where

$$\tilde{\mathcal{L}}^{\mathrm{emp}}(\tilde{\boldsymbol{\theta}}) := \frac{1}{N} \sum_{k=0}^{K} \sigma\big((1 - \gamma_k)x - \exp(L)B\big),$$

$$\tilde{\mathcal{L}}^{\mathrm{reg}}(\tilde{\boldsymbol{\theta}}) := B + \sum_{k=0}^{K} l_k \gamma_k^{\frac{1}{l_k}} + w^-,$$

where $x \in \mathbb{R}_{\geq 0}$ is the scale of the weights in the training set $T_{S_{x,K}}$.

*Remark* 67. Applying Lemma 54 and the definition of $\gamma_k$, $k \in [K]_0$ we see that

$$H_{\mathrm{wl}}(\mathcal{W}^+, \mathcal{C}^+, 0)(\boldsymbol{z}) = \sum_{k=0}^{K} \Big( \prod_{s=j_k+1}^{J} (\boldsymbol{W}^s)^+ \Big)(\boldsymbol{C}^k)^+ z_k = \sum_{k=0}^{K} \gamma_k z_k$$

for any $\boldsymbol{z} \in \mathbb{R}^{K+1}$. Thus, up to biases and negative weights, $\gamma_k$ is the factor that the $k$-th edge weight $z_k$ is multiplied with, in case the aggregation follows the path $p$ such that $\boldsymbol{z}^p = \boldsymbol{z}$.

The next lemma establishes the key connection between the original and modified loss functions. It shows that the modified loss lower-bounds the contributions of the original parameters, allowing us to analyze $\tilde{\mathcal{L}}$ in place of $\mathcal{L}$.

**Lemma 68.** *Let* $\boldsymbol{\theta} = (\mathcal{W}, \mathcal{C}, \mathcal{B}) \in \boldsymbol{\Theta}_{\mathrm{BF}}$ *and let* $\tilde{\boldsymbol{\theta}}$ *be its transformed parameter vector. Further, assume* $T_{S_{x,K}} \subset X$ *and that* $\mathcal{L}(\boldsymbol{\theta})$ *is within* $\varepsilon \leq \eta L$ *of its global minimum. Then*

$$\mathcal{L}^{\mathrm{emp}}(\boldsymbol{\theta}) \geq \tilde{\mathcal{L}}^{\mathrm{emp}}(\tilde{\boldsymbol{\theta}}), \quad \text{and} \quad \mathcal{L}^{\mathrm{reg}}(\boldsymbol{\theta}) \geq \tilde{\mathcal{L}}^{\mathrm{reg}}(\tilde{\boldsymbol{\theta}}).$$

*Proof.* We first prove $\mathcal{L}^{\mathrm{emp}}(\boldsymbol{\theta}) \geq \tilde{\mathcal{L}}^{\mathrm{emp}}(\tilde{\boldsymbol{\theta}})$. Fix $\boldsymbol{w} \in S_{x,K}$ and consider the corresponding path graph used for training $G := P(\boldsymbol{w})$ with nodes $v_0^{\boldsymbol{w}}, \ldots, v_K^{\boldsymbol{w}}$. Let $p_{\boldsymbol{w}} = (v_0^{\boldsymbol{w}}, \ldots, v_K^{\boldsymbol{w}}) \in P_G^K(v_K)$ be the walk that starts in $v_0^{\boldsymbol{w}}$ and ends in $v_K^{\boldsymbol{w}}$. Then it holds $\boldsymbol{z}^{p_{\boldsymbol{w}}} = \boldsymbol{w}$ (see Remark 50). Thus, Lemma 57 implies

$$\boldsymbol{h}_{v_K^{\boldsymbol{w}}}^{(K)} \leq H_{\mathrm{wl}}(\boldsymbol{\theta}^+)(\boldsymbol{z}^{p_{\boldsymbol{w}}}) = H_{\mathrm{wl}}(\mathcal{W}^+, \mathcal{C}^+, \mathcal{B}^+)(\boldsymbol{w}).$$

Further since the last node $v_K^{\boldsymbol{w}}$ of the path graph $P(\boldsymbol{w})$ is part of the training set, e.g. $\{v_K^{\boldsymbol{w}} : \boldsymbol{w} \in S_{x,K}\} = T_{S_{x,K}} \subset X$ and for the corresponding target it holds $x_{v_K^{\boldsymbol{w}}}^{(K)} = \|\boldsymbol{w}\|_1$ (c.f. Remark 14) we have

$$\mathcal{L}^{\mathrm{emp}}(\boldsymbol{\theta}) = \frac{1}{N} \sum_{v \in X} \big| \boldsymbol{h}_v^{(K)} - x_v^{(K)} \big| \geq \frac{1}{N} \sum_{\boldsymbol{w} \in S_{x,K}} \big| \boldsymbol{h}_{v_K^{\boldsymbol{w}}}^{(K)} - \|\boldsymbol{w}\|_1 \big|.$$

Combining the above, using Lemma 54, Lemma 55, Corollary 65 and Remark 67, we obtain

$$
\begin{aligned}
N\,\mathcal{L}^{\mathrm{emp}}(\boldsymbol{\theta}) &\geq \sum_{\boldsymbol{w}\in S_{x,K}} |\|\boldsymbol{w}\|_1 - h_{v_K^w}^{(K)}| \geq \sum_{\boldsymbol{w}\in S_{x,K}} \sigma\big(\|\boldsymbol{w}\|_1 - H_{\mathrm{wl}}(\mathcal{W}^+,\mathcal{C}^+,\mathcal{B}^+)(\boldsymbol{w})\big) \\
&\geq \sum_{\boldsymbol{w}\in S_{x,K}} \sigma\big(\|\boldsymbol{w}\|_1 - H_{\mathrm{wl}}(\mathcal{W}^+,\mathcal{C}^+,0)(\boldsymbol{w}) - H_{\mathrm{wl}}(|\mathcal{W}|,|\mathcal{C}|,|\mathcal{B}|)(0)\big) \\
&\geq \sum_{\boldsymbol{w}\in S_{x,K}} \sigma\big(\|\boldsymbol{w}\|_1 - H_{\mathrm{wl}}(\mathcal{W}^+,\mathcal{C}^+,0)(\boldsymbol{w}) - \exp(L)B\big) \\
&= \sum_{\boldsymbol{w}\in S_{x,K}} \sigma\left(\sum_{k=0}^{K}(1-\gamma_k)w_k - \exp(L)B\right) \\
&\geq \sum_{k=0}^{K} \sigma((1-\gamma_k)x - \exp(L)B) = N\,\tilde{\mathcal{L}}^{\mathrm{emp}}(\tilde{\boldsymbol{\theta}}),
\end{aligned}
$$

where in the last inequality we used that $S_{x,K} = \{x\boldsymbol{e}_k^{K+1} : k \in [K]_0\}$.

Next we prove $\mathcal{L}^{\mathrm{reg}}(\boldsymbol{\theta}) \geq \tilde{\mathcal{L}}^{\mathrm{reg}}(\tilde{\boldsymbol{\theta}})$. Observe that,

$$
\gamma_0 \leq \prod_{j>j_0}^{J} \|(\boldsymbol{W}^j)^+\|_1, \quad \gamma_k \leq \prod_{j>j_k}^{J} \|(\boldsymbol{W}^j)^+\|_1\,\|(\boldsymbol{C}^k)^+\|_1, \quad k \in [K].
$$

Thus for each $k \in [K]_0$, by Lemma 62 and the definition of $l_k$ (see Definition 58)

$$
l_k\,\gamma_k^{1/l_k} \leq \sum_{j>j_k}^{J} \|\boldsymbol{W}^j\|_1 + \mathbf{1}_{k\neq0}\|\boldsymbol{C}^k\|_1.
$$

Therefore,

$$
\begin{aligned}
\mathcal{L}^{\mathrm{reg}}(\boldsymbol{\theta}) &= \sum_{k=0}^{K}\left(\sum_{j>j_k}\|\boldsymbol{W}^j\|_1 + \mathbf{1}_{k\neq0}\|\boldsymbol{C}^k\|_1\right) + B \\
&\geq \sum_{k=0}^{K}\left(\sum_{j>j_k}\|(\boldsymbol{W}^j)^+\|_1 + \mathbf{1}_{k\neq0}\|(\boldsymbol{C}^k)^+\|_1\right) + B + \sum_{j=1}^{J}\|(\boldsymbol{W}^j)^-\|_1 + \sum_{k=1}^{K}\|(\boldsymbol{C}^k)^-\|_1 \\
&\geq \sum_{k=0}^{K} l_k\gamma_k^{1/l_k} + B + w^- = \tilde{\mathcal{L}}^{\mathrm{reg}}(\tilde{\boldsymbol{\theta}}).
\end{aligned}
$$

$\square$

Based on the previous lemma, which allows us to analyze the modified loss $\tilde{\mathcal{L}}$ in place of the original loss $\mathcal{L}$, the next result identifies the key structural properties that any parameter set must satisfy when it lies close to the global minimum. There are essentially two main advantages to working with $\tilde{\mathcal{L}}$.

First, the feature contribution is replaced by a simpler upper bound that no longer depends on the ReLU's intricate behavior. An immediate consequence is that non-zero biases or negative entries in the weight matrices can only increase the modified loss, rather than interact in more complicated ways as in the original formulation. Secondly, the parameters appearing in different summands of the modified empirical loss become independent of one another. In particular, increasing $\gamma_{k_1}$ for some $k_1$ does not decrease the term $\sigma\big((1-\gamma_{k_2})x - \exp(L)B\big)$ associated with any $k_2 \neq k_1$. These simplifications eliminate several sources of coupling and nonlinearity present in the original loss, thereby making the subsequent analysis considerably more tractable.

**Lemma 69** (Parameter characterization near the global minimum). *Let $\boldsymbol{\theta} \in \boldsymbol{\Theta}_{BF}$. Assume $T_{S_{x,K}} \subset X$ and that $\mathcal{L}(\boldsymbol{\theta})$ lies within $0 \leq \varepsilon \leq \eta L$ of its global minimum. Further assume*

$$\eta \geq 2K \exp(L) \qquad and \qquad x \geq 2N\eta J.$$

*Then the following estimates hold:*

$$H_{\mathrm{wl}}(|\mathcal{W}|, |\mathcal{C}|, |\mathcal{B}|)(0) \leq \varepsilon, \tag{16}$$

$$\gamma_k \geq 1 - \frac{\varepsilon}{\eta J}, \qquad k \in [K]_0, \tag{17}$$

*and, for any $\boldsymbol{z} \in \mathbb{R}_{\geq 0}^{K+1}$,*

$$|H_{\mathrm{wl}}(\mathcal{W}, \mathcal{C}, \mathcal{B})(\boldsymbol{z}) - H_{\mathrm{wl}}(\mathcal{W}^+, \mathcal{C}^+, 0)(\boldsymbol{z})| \leq \left(\tfrac{1}{2}\|\boldsymbol{z}\|_1 + 1\right)\varepsilon. \tag{18}$$

*Moreover,*

$$\mathcal{L}^{\mathrm{emp}}(\theta) \leq 2\varepsilon.$$

Note that both $\eta$ and $x$ must be chosen sufficiently large. A large value of $\eta$ ensures that the influence of the biases on the empirical loss is small compared to their contribution to the regularization term, so that any non-zero bias is strongly penalized overall. Likewise, taking $x$ much larger than $\eta$ guarantees that the empirical loss dominates the regularizer, so that deviations in the feature values have a significantly stronger impact on the total loss than variations in the regularization cost.

*Proof.* Starting from the definition of $\tilde{\mathcal{L}}$, we compute

$$
\begin{aligned}
\tilde{\mathcal{L}}(\tilde{\boldsymbol{\theta}}) - \eta L - \eta w^- &= \frac{1}{N}\sum_{k=0}^{K}\sigma((1-\gamma_k)x - \exp(L)B) + \eta B + \eta\sum_{k'=0}^{K}l_{k'}(\gamma_{k'}^{1/l_{k'}} - 1) \\
&\geq \frac{1}{N}\sum_{k:\,\gamma_k\leq 1}\left((1-\gamma_k)x - \exp(L)B\right) + \eta B + \eta\sum_{k':\,\gamma_{k'}\leq 1}l_{k'}(\gamma_{k'}^{1/l_{k'}} - 1) \\
&\geq \frac{1}{N}\sum_{k:\,\gamma_k\leq 1}\left((1-\gamma_k)x + \eta l_k(\gamma_k - 1)\right) + \left(\eta - K\exp(L)\right)B \\
&\geq \left(\tfrac{x}{N} - \eta J\right)\sum_{k:\,\gamma_k\leq 1}(1-\gamma_k) + \tfrac{1}{2}\eta B \\
&\geq \eta J\sum_{k:\,\gamma_k\leq 1}(1-\gamma_k) + \tfrac{1}{2}\eta B,
\end{aligned}
$$

where we used that $\gamma_{k'}^{1/l_{k'}} \geq \gamma_{k'}$ if $\gamma_{k'} \leq 1$ and $l_k \leq J$.

Using Lemma 68 and Lemma 61 we obtain

$$\eta J\sum_{k:\,\gamma_k\leq 1}(1-\gamma_k) + \tfrac{1}{2}\eta B + \eta w^- \leq \tilde{\mathcal{L}}(\tilde{\boldsymbol{\theta}}) - \eta L \leq \mathcal{L}(\boldsymbol{\theta}) - \eta L \leq \varepsilon. \tag{19}$$

With (19) at hand, we can now derive the estimates as claimed.

**Proof of** (16) **and** (18)  From (19),

$$\tfrac{1}{2}\eta B \leq \varepsilon, \qquad \eta w^- \leq \varepsilon.$$

Using Corollary 65 and $\eta \geq 2\exp(L)$ by assumption on $\eta$, this yields

$$H_{\mathrm{wl}}(|\mathcal{W}|, |\mathcal{C}|, |\mathcal{B}|)(0) \leq \exp(L)B \leq \exp(L)\frac{2\varepsilon}{\eta} \leq \exp(L)\frac{2\varepsilon}{2\exp(L)} = \varepsilon,$$

$$G(\mathcal{W}, \mathcal{B}(\boldsymbol{z}, \mathcal{C}))(z_0) \leq \exp(L)w^-\|\boldsymbol{z}\|_1 \leq \exp(L)\frac{\varepsilon}{\eta}\|\boldsymbol{z}\|_1 \leq \tfrac{1}{2}\varepsilon\|\boldsymbol{z}\|_1.$$

Thus (16) holds, and by Corollary 56,

$$|H_{\mathrm{wl}}(\mathcal{W}^+, \mathcal{C}^+, 0)(\boldsymbol{z}) - H_{\mathrm{wl}}(\mathcal{W}, \mathcal{C}, \mathcal{B})(\boldsymbol{z})|$$
$$\leq G(\mathcal{W}, \mathcal{B}(\boldsymbol{z}, \mathcal{C}))(z_0) + H_{\mathrm{wl}}(|\mathcal{W}|, |\mathcal{C}|, |\mathcal{B}|)(0) \leq \left(\tfrac{1}{2}\|\boldsymbol{z}\|_1 + 1\right)\varepsilon,$$

proving (18).

**Lower bound on $\gamma_k$**  From (19), for any $k \in [K]_0$

$$\eta J(1 - \gamma_k) \leq \varepsilon \quad \Rightarrow \quad \gamma_k \geq 1 - \frac{\varepsilon}{\eta J},$$

which gives (17).

**Estimate for the empirical loss**  From (19),

$$\sum_{k \,:\, \gamma_k \leq 1} (1 - \gamma_k) \;\leq\; \frac{\varepsilon}{\eta J}$$

and hence,

$$\tilde{\mathcal{L}}^{\mathrm{reg}}(\tilde{\boldsymbol{\theta}}) \geq \sum_{k=0}^{K} l_k \, \gamma_k^{1/l_k} = L - \sum_{k=0}^{K} l_k \left(1 - \gamma_k^{1/l_k}\right) \geq L - J \sum_{k \,:\, \gamma_k \leq 1} (1 - \gamma_k) \geq L - \tfrac{\varepsilon}{\eta}.$$

Therefore, using Lemma 68 and Lemma 61

$$\mathcal{L}^{\mathrm{emp}}(\boldsymbol{\theta}) = \mathcal{L}(\boldsymbol{\theta}) - \eta \mathcal{L}^{\mathrm{reg}}(\boldsymbol{\theta}) \leq \mathcal{L}(\boldsymbol{\theta}) - \eta \tilde{\mathcal{L}}^{\mathrm{reg}}(\tilde{\boldsymbol{\theta}}) \leq \varepsilon + \eta L - \left(\eta L - \varepsilon\right) \;\leq\; 2\varepsilon$$

which completes the proof. $\qquad\square$

### F.7. Upper and Lower bound

In this section, we assume that the dimension of aggregation is 1, i.e., $d_{a_k} = 1$ for all $k \in [K]$. Then any computation tree is in fact a path, and thus, $\boldsymbol{z}^t$ is well-defined for any $t \in \mathcal{T}_G^J(v)$, where $G \in \mathcal{G}_{\mathrm{BF}}$ and $v \in V(G)$ (c.f. Remark 51).

**Corollary 70** (Lower bound). *Let $\boldsymbol{\theta} \in \boldsymbol{\Theta}_{BF}$, $\eta \geq 2K \exp(L)$ and $x \geq 2N\eta J$. Assume $T_{S_{x,K}} \subset X$ and that $\mathcal{L}(\boldsymbol{\theta})$ lies within $0 \leq \varepsilon \leq \eta L$ of its global minimum. Then for any $G \in \mathcal{G}_{BF}$, $v \in V(G)$, and $t \in \mathcal{T}_{\boldsymbol{\theta}}^J(v)$*

$$\boldsymbol{h}_v^{(K)} \;\geq\; (1 - \varepsilon)\, \|\boldsymbol{z}^t\|_1 \;-\; \varepsilon.$$

*In particular,*

$$\boldsymbol{h}_v^{(K)} \;\geq\; (1 - \varepsilon)\, x_v^{(K)} \;-\; \varepsilon$$

*where $x_v^{(K)}$ denotes the Bellman–Ford distance.*

*Proof.* Let $G \in \mathcal{G}_{\mathrm{BF}}$. Fix $v \in V(G)$ and $\tau \in \mathcal{T}_{G,\boldsymbol{\theta}}^K(v)$. Applying Lemma 57, and (18) from Lemma 69 and Remark 67 gives

$$\boldsymbol{h}_v^{(K)} = H_{\mathrm{wl}}(\mathcal{W}, \mathcal{C}, \mathcal{B})(\boldsymbol{z}^\tau) \geq H_{\mathrm{wl}}(\mathcal{W}^+, \mathcal{C}^+, 0)(\boldsymbol{z}^\tau) - \left(\tfrac{1}{2}\|\boldsymbol{z}^\tau\|_1 + 1\right)\varepsilon$$
$$= \sum_{k=0}^{K} \gamma_k \, \boldsymbol{z}_k^\tau - \left(\tfrac{1}{2}\|\boldsymbol{z}^\tau\|_1 + 1\right)\varepsilon.$$

Using $\gamma_k \geq 1 - \tfrac{\varepsilon}{\eta J}$ (by (17) from Lemma 69), $\eta \geq 2$ and $J \geq 1$, we obtain

$$\sum_{k=0}^{K} \gamma_k \, \boldsymbol{z}_k^\tau \;\geq\; \left(1 - \frac{\varepsilon}{\eta J}\right) \|\boldsymbol{z}^\tau\|_1 \;\geq\; \left(1 - \tfrac{\varepsilon}{2}\right) \|\boldsymbol{z}^\tau\|_1.$$

Combining this with the previous estimate gives

$$\boldsymbol{h}_v^{(K)} \ \geq\ (1-\varepsilon)\,\|\boldsymbol{z}^\tau\| - \varepsilon.$$

Finally, note that by definition of the BF-distance $x_v^{(K)}$ it holds $\|\boldsymbol{z}^t\|_1 \geq x_v^{(K)}$ for any $t \in \mathcal{T}_G^K(v) \equiv \mathcal{P}_G^K(v)$ (see Remark 49). Therefore,

$$\boldsymbol{h}_v^{(K)} \ \geq\ (1-\varepsilon)\,x_v^{(K)} - \varepsilon,$$

which completes the proof. $\qquad\square$

The next lemma helps to derive tighter upper bounds on the features.

**Lemma 71.** *Let $\boldsymbol{\theta} \in \boldsymbol{\Theta}_{BF}$, $\eta \geq 2K \exp(L)$ and $x \geq 2N\eta J$. Assume $T_{S_{x,K}} \subset X$ and that $\mathcal{L}(\boldsymbol{\theta})$ lies within $0 < \varepsilon < 1/2 \ (\leq \eta L)$ of its global minimum. Then for any $\boldsymbol{z} \in \mathbb{R}_{\geq 0}^{K+1}$,*

$$\big|\, H_{\mathrm{wl}}(\mathcal{W}^+,\mathcal{C}^+,0)(\boldsymbol{z}) - \|\boldsymbol{z}\|_1 \,\big| \ \leq\ \varepsilon\,\|\boldsymbol{z}\|_1.$$

*Proof.* Let $\boldsymbol{w} \in S_{x,K}$ and $\tau \in \mathcal{T}_{G,\boldsymbol{\theta}}^J(v_K^{\boldsymbol{w}})$ where $v_K^{\boldsymbol{w}}$ denotes the last node of the graph $G := P_\beta(\boldsymbol{w})$ from the training set. Further let $p_{\mathrm{BF}} := (v_0^{\boldsymbol{w}},\ldots,v_K^{\boldsymbol{w}}) \in \mathcal{P}_G^K(v_K^{\boldsymbol{w}})$ and $p_{\mathrm{AGG}} \in \mathcal{P}_G^K(v_K^{\boldsymbol{w}})$ denote the path of $\tau$, i.e. $t^K(p_{\mathrm{AGG}}) = \tau$.

Assume for contradiction that the aggregation does not follow $p_{\mathrm{BF}}$, i.e. $p_{\mathrm{BF}} \neq p_{\mathrm{AGG}}$. Then the first node on $p_{\mathrm{AGG}}$ is not $v_0$ and hence $\boldsymbol{z}_0^{p_{\mathrm{AGG}}} = a_G(v_0) = \beta$ (c.f. Remark 50). Since $\varepsilon \leq \frac{1}{2}$, and by Lemma 69 and Corollary 70, we have

$$
\begin{aligned}
N \geq 2N\varepsilon \geq N\mathcal{L}^{\mathrm{emp}}(\theta) &\geq |\boldsymbol{h}_{v_K^{\boldsymbol{w}}}^{(K)} - \|\boldsymbol{w}\|_1| \geq \boldsymbol{h}_{v_K^{\boldsymbol{w}}}^{(K)} - \|\boldsymbol{w}\|_1 \\
&\geq (1-\varepsilon)\|\boldsymbol{z}^\tau\|_1 - \varepsilon - \|\boldsymbol{w}\|_1 \\
&\geq (1-\varepsilon)\beta - \varepsilon - x \geq \tfrac{1}{2}(\beta - 1) - x \geq N + \tfrac{1}{2}
\end{aligned}
$$

where in the last inequality we use $\beta \geq 2(N + x + 1)$, which is a contradiction. Hence, it must be $p_{\mathrm{BF}} = p_{\mathrm{AGG}}$ and thus $\boldsymbol{w} = \boldsymbol{z}^{p_{\mathrm{BF}}} = \boldsymbol{z}^\tau$ (c.f. Remark 50). Therefore by Lemma 57

$$\boldsymbol{h}_{v^{\boldsymbol{w}}}^{(K)} = H_{\mathrm{wl}}(\mathcal{W},\mathcal{C},\mathcal{B})(\boldsymbol{z}^\tau) = H_{\mathrm{wl}}(\mathcal{W},\mathcal{C},\mathcal{B})(\boldsymbol{w}).$$

Applying Lemma 69, we obtain

$$
\begin{aligned}
2N\varepsilon \geq N\mathcal{L}^{\mathrm{emp}}(\theta) &\geq |\boldsymbol{h}_{v_K^{\boldsymbol{w}}}^{(K)} - \|\boldsymbol{w}\|_1| = |H_{\mathrm{wl}}(\mathcal{W},\mathcal{C},\mathcal{B})(\boldsymbol{w}) - \|\boldsymbol{w}\|_1| \\
&\geq |H_{\mathrm{wl}}(\mathcal{W}^+,\mathcal{C}^+,0)(\boldsymbol{w}) - \|\boldsymbol{w}\|_1| - |H_{\mathrm{wl}}(\mathcal{W},\mathcal{C},\mathcal{B})(\boldsymbol{w}) - H_{\mathrm{wl}}(\mathcal{W}^+,\mathcal{C}^+,0)(\boldsymbol{w})| \\
&\geq |H_{\mathrm{wl}}(\mathcal{W}^+,\mathcal{C}^+,0)(\boldsymbol{w}) - \|\boldsymbol{w}\|_1| - \big(\tfrac{1}{2}\|\boldsymbol{w}\|_1 + 1\big)\varepsilon.
\end{aligned}
$$

For $\boldsymbol{w} = x\boldsymbol{e}_k^{K+1} \in S_{x,K}$, this gives

$$|H_{\mathrm{wl}}(\mathcal{W}^+,\mathcal{C}^+,0)(x\boldsymbol{e}_k^{K+1}) - x| \ = \ |H_{\mathrm{wl}}(\mathcal{W}^+,\mathcal{C}^+,0)(\boldsymbol{w}) - \|\boldsymbol{w}\|_1| \ \leq \ (2N + \tfrac{1}{2}x + 1)\varepsilon \ \leq \ \varepsilon x,$$

where the last inequality follows from our assumptions on $x$ and $\eta$:

$$\frac{2N+1}{\tfrac{1}{2}x} \leq 2\frac{2N+1}{2N\eta} \leq \frac{2N+1}{2NK\exp(L)} \leq \frac{2N+1}{4N} \leq 1.$$

Finally, by the linearity of the Walk-lifted FNN for positive parameter we get for any $\boldsymbol{z} \in \mathbb{R}_{\geq 0}^{K+1}$:

$$|H_{\mathrm{wl}}(\mathcal{W}^+,\mathcal{C}^+,0)(\boldsymbol{z}) - \|\boldsymbol{z}\|_1| \leq \sum_{k\in[K]_0} \frac{z_k}{x}|H_{\mathrm{wl}}(\mathcal{W}^+,\mathcal{C}^+,0)(x\boldsymbol{e}_k^{K+1}) - x| \leq \sum_{k\in[K]_0} \frac{z_k}{x}\varepsilon x \leq \varepsilon\|\boldsymbol{z}\|_1.$$

$\qquad\square$

**Theorem 72** (Upper bound). *Let $\boldsymbol{\theta} \in \boldsymbol{\Theta}_{BF}$, $\eta \geq 2K \exp(L)$ and $x \geq 2N\eta J$. Assume $T_{S_{x,K}} \subset X$ and that $\mathcal{L}(\boldsymbol{\theta})$ lies within $0 < \varepsilon < \frac{1}{2}$ of its global minimum.*

*Then, for any $G \in \mathcal{G}_{BF}$ and any $v \in V(G)$,*

$$\boldsymbol{h}_v^{(K)} \leq (1+\varepsilon)x_v^{(K)} + \varepsilon.$$

*Proof.* Fix $G \in \mathcal{G}_{\mathrm{BF}}$ and let $p \in \mathcal{P}_G^K(v)$. Using Lemma 57, Lemma 54, Lemma 71 and (16) from Lemma 69

$$\boldsymbol{h}_v^{(K)} \leq H_{\mathrm{wl}}(\mathcal{W}^+, \mathcal{C}^+, \mathcal{B}^+)(\boldsymbol{z}^p) \leq H_{\mathrm{wl}}(\mathcal{W}^+, \mathcal{C}^+, 0)(\boldsymbol{z}^p) + H_{\mathrm{wl}}(\mathcal{W}^+, \mathcal{C}^+, \mathcal{B}^+)(0) \leq (1+\varepsilon)\|\boldsymbol{z}^p\| + \varepsilon$$

and thus by definition of the BF-distance (see Remark 49)

$$\boldsymbol{h}_v^{(K)} \leq \min_{p \in \mathcal{P}_G^K(v)} (1+\varepsilon)\|\boldsymbol{z}^p\| + \varepsilon = (1+\varepsilon)x_v^K + \varepsilon.$$

$\square$

# G. What MPNN cannot learn

This appendix provides additional details for Section 3. In particular, Appendix G.1 contains further details related to Section 3.1, including a formal statement and proof of Proposition 5, while Appendix G.2 contains the proof of Lemma 6 from Section 3.2.

## G.1. Expressivity limitations

We begin by formally stating and proving the negative result of Proposition 5, namely that standard MPNN architectures cannot approximate the SSSP and MST invariants.

**Proposition 73** (Proposition 5 (negative result) in the main text). *The following holds.*

1. *For $n \geq 6$, there exists an edge-weighted graph $G$ of order $n$ and vertices $s, t_1, t_2 \in V(G)$ such that, for the class of node-level $\mathsf{MPNN}_{(\mathcal{S}_L, d, n)}^{\mathcal{P}_L}(\{G\})$, for any number of layers $L \geq 0$, $d > 0$, set of parameters $\mathcal{P}_L$, and sequence of parameterized functions $\mathcal{S}_L$, it holds that for all $m \in \mathsf{MPNN}_{(\mathcal{S}_L, d, n)}^{\mathcal{P}_L}(\{G\})$,*

$$|\mathsf{SSSP}(G, (s, t_1)) - \mathsf{SSSP}(G, (s, t_2))| \geq 1 \quad but \quad m(t_1) = m(t_2).$$

2. *For $n \geq 6$, there exist edge-weighted graphs $G, H$ of order $n$ such that, for the class of graph-level $\mathsf{MPNN}_{(\mathcal{T}_L, d)}^{\mathcal{Q}_L}(\{G, H\})$, for any number of layers $L \geq 0$, $d > 0$, set of parameters $\mathcal{Q}_L$, and sequence of parameterized functions $\mathcal{T}_L$, it holds that for all $m \in \mathsf{MPNN}_{(\mathcal{T}_L, d)}^{\mathcal{Q}_L}(\{G, H\})$,*

$$|\mathsf{MST}(G) - \mathsf{MST}(H)| \geq 1 \quad but \quad m(G) = m(H).$$

3. *For $n \geq 14$, there exist edge-weighted graphs $G, H$ of order $n$ such that, for the class of graph-level $1$-iWL-simulating $\mathsf{MPNN}_{(\mathcal{T}_L, d)}^{\mathcal{Q}_L}(\{G, H\})$, for any number of layers $L \geq 0$, $d > 0$, set of parameters $\mathcal{Q}_L$, and sequence of parameterized functions $\mathcal{T}_L$, it holds that for all $m \in \mathsf{MPNN}_{(\mathcal{T}_L, d)}^{\mathcal{Q}_L}(\{G, H\})$,*

$$|\mathsf{MST}(G) - \mathsf{MST}(H)| \geq 1 \quad but \quad m(G) = m(H).$$

*Proof.* For (1) it suffices to consider the edge-weighted graph $(G, w_G)$ with $V(G) := [6]$ and edge set $E(G) := \{(1,2), (1,3), (2,3), (3,4), (4,5), (4,6), (5,6)\}$, with edge weights $w_G(1,2) := 3$, $w_G(1,3) := 1$, $w_G(2,3) := 1$, $w_G(3,4) := 5$, $w_G(4,5) := 1$, $w_G(4,6) := 1$, and $w_G(5,6) := 3$. Hence, the graph $G$ consists of two triangles connected by an edge of high weight. Let $s = 1$, $t_1 = 2$, and $t_2 = 5$. Then, the shortest path from $s$ to $t_1$ has cost 2 via vertices 1,3,2, whereas the shortest path from $s$ to $t_2$ has cost 7 via vertices 1,3,4,5. Hence $|\mathsf{SSSP}(G, (s, t_1)) - \mathsf{SSSP}(G, (s, t_2))| = 5 \geq 1$. For (2) we consider an additional edge-weighted graph $(H, w_H)$, also of order six with $V(H) := [6]$. The graphs $G$ and $H$ have different costs for minimal spanning trees, yet they are indistinguishable by 1-WL. The edge set of $H$ is $E(H) := \{(1,2), (2,4), (4,6), (6,5), (5,3), (3,1), (3,4)\}$, with edge weights $w_H(1,2) := 3$, $w_H(2,4) := 1$, $w_H(4,6) := 1$,

$w_H(6,5) := 3$, $w_H(5,3) := 1$, $w_H(3,1) := 1$, and $w_H(3,4) := 5$. Hence, the graph $H$ consists of a 6-cycle with a high-weight chord. Observe that $\mathsf{MST}(G) = 9$. However, in $H$, because we do not have to include the heavy chord in an MST, we get $\mathsf{MST}(H) = 7$.

We now observe that 1-$\mathsf{WL}$, taking edge weights into account, cannot distinguish the graphs $G$ and $H$, and it also cannot distinguish $t_1$ and $t_2$ in $G$. Indeed, in view of the characterization of 1-$\mathsf{WL}$-distinguishability in terms of unrollings (see Lemma 80), one can verify that $\mathsf{unr}(G, t_1, L) = \mathsf{unr}(G, t_2, L)$ for all $L$. Similarly, there is a bijection $\pi \colon V(G) \to V(H)$ such that for all $v \in V(G)$, $\mathsf{unr}(G, v, L) = \mathsf{unr}(H, \pi(v), L)$ for all $L$. Hence, by Morris et al. (2019, Theorem 1), no node-level MPNN can separate $t_1$ from $t_2$ on $G$, and no graph-level MPNN can separate $G$ from $H$, implying (1) and (2).

For (3) we exhibit two connected graphs $G$ and $H$ such that 1-$\mathsf{iWL}$ does not distinguish them (regarding a suitable choice of roots), yet $\mathsf{MST}(G) \neq \mathsf{MST}(H)$. In both graphs, the vertex set is $V(G) = V(H) := [14]$, where 7 and 8 are two bridge endpoints and both graphs contain the edge $(7,8)$. The edge sets are

$$E(G) := \{(1,2), (1,3), (2,3)\} \cup \{(4,5), (4,6), (5,6)\} \cup \{(3,4)\}$$
$$\cup \{(9,10), (9,11), (10,11)\} \cup \{(12,13), (12,14), (13,14)\} \cup \{(11,12)\}$$
$$\cup \{(7,i) \mid i \in [6]\} \cup \{(8,j) \mid j \in \{9,10,11,12,13,14\}\} \cup \{(7,8)\},$$

and

$$E(H) := \{(1,2), (2,4), (4,6), (6,5), (5,3), (3,1), (3,4)\}$$
$$\cup \{(9,10), (10,12), (12,14), (14,13), (13,11), (11,9), (11,12)\}$$
$$\cup \{(7,i) \mid i \in [6]\} \cup \{(8,j) \mid j \in \{9,10,11,12,13,14\}\} \cup \{(7,8)\}.$$

The weights are defined by $w_G(3,4) = w_G(11,12) := 5$, $w_G(7,i) := 10$ for $i \in [6]$, $w_G(8,j) := 10$ for $j \in \{9, \ldots, 14\}$, and $w_G(e) := 1$ for all other $e \in E(G)$ (in particular $w_G(7,8) = 1$); analogously, $w_H(3,4) = w_H(11,12) := 5$, $w_H(7,i) := 10$ for $i \in [6]$, $w_H(8,j) := 10$ for $j \in \{9, \ldots, 14\}$, and $w_H(e) := 1$ for all other $e \in E(H)$ (again $w_H(7,8) = 1$). It is readily verified that $\mathsf{MST}(G) = 39$ and $\mathsf{MST}(H) = 31$.

Let $v$ and $w$ be vertex 1 in $G$ and $H$, respectively. Then $G$ and $H$ are 1-$\mathsf{iWL}$-indistinguishable regarding $v$ and $w$. Indeed, this follows from the corresponding characterization in terms of unrollings (see Lemma 80) and the existence of a bijection $\pi \colon V(G) \to V(H)$ such that for all $v' \in V(G)$, $\mathsf{unr}(G, v', L, v) = \mathsf{unr}(H, \pi(v'), L, w)$. Hence, by the same argument as Morris et al. (2019, Theorem 1), no 1-$\mathsf{iWL}$-simulating graph-level MPNN can separate $G$ from $H$, which implies (3).  □

Now, the following result shows that 1-$\mathsf{iWL}$- and $(1,1)$-$\mathsf{WL}$-simulating MPNNs can arbitrarily well approximate the costs of SSSP and MST, respectively. We remark that these approximation results require fixing the order of graphs.

**Proposition 74.** *Let $n > 0$, let $C \subseteq \mathbb{R}$ be compact, and let $\mathcal{G}_{n,C}$ be a set of edge-weighted $n$-order graphs with edge weights from $C$. Then the following holds.*

1. *For $n \geq 1$ and $\varepsilon > 0$, there exists a class of 1-$\mathsf{iWL}$-simulating node-level MPNNs $\mathcal{F}_\varepsilon$ and an $f \in \mathcal{F}_\varepsilon$ such that*

$$\sup_{G \in \mathcal{G}_{n,C}, \, s,t \in V(G)} \big| f(G, (s,t)) - \mathsf{SSSP}(G, (s,t)) \big| < \varepsilon.$$

2. *For $n \geq 1$ and $\varepsilon > 0$, there exists a class of $(1,1)$-$\mathsf{WL}$-simulating graph-level MPNNs $\mathcal{F}_\varepsilon$ and an $f \in \mathcal{F}_\varepsilon$ such that*

$$\sup_{G \in \mathcal{G}_{n,C}} \big| f(G) - \mathsf{MST}(G) \big| < \varepsilon.$$

*Proof.* For simplicity, we assume that all graphs are connected. We rely on Proposition 7 and show below, in Lemma 75, that $\rho_2(\text{1-}\mathsf{iWL}) \subseteq \rho_2(\mathsf{SSSP})$, and in Lemma 76 that $\rho((1,1)\text{-}\mathsf{WL}) \subseteq \rho(\mathsf{MST})$. The approximation statements then follow from Proposition 7 and from the fact that we consider simulating MPNNs.  □

The following results state that 1-$\mathsf{iWL}$ determines shortest-path distances and that $(1,1)$-$\mathsf{WL}$ determines the cost of an MST.

**Lemma 75** (Proposition 5 (positve result regarding SSSP) in the main text)**.** *Let $(G, w_G)$ and $(H, w_H)$ be two connected edge-weighted graphs, and let $s, v \in V(G)$ and $t, w \in V(H)$. If $C^{1,s}_\infty(v) = C^{1,t}_\infty(w)$, then $\mathsf{SSSP}(G, (s, v)) = \mathsf{SSSP}(H, (t, w))$.*

*Proof.* We argue by contradiction. Assume that $\mathsf{SSSP}(G, (s, v)) \neq \mathsf{SSSP}(H, (t, w))$. Then, for $L$ large enough, the rooted unrollings satisfy $\mathrm{unr}(G, v, L, s) \neq \mathrm{unr}(H, w, L, t)$. By the characterization of 1-iWL in terms of rooted unrollings (see Lemma 80), this implies $C^{1,s}_\infty(v) \neq C^{1,t}_\infty(w)$, contradicting the assumption. We refer to Appendix H.2 for details. $\quad\square$

**Lemma 76** (Proposition 5 (positve result regarding MST) in the main text)**.** *Let $(G, w_G)$ and $(H, w_H)$ be two connected edge-weighted graphs that are $(1,1)$-WL-indistinguishable. Then $\mathsf{MST}(G) = \mathsf{MST}(H)$.*

*Proof.* We reduce the computation of the cost of a minimal spanning tree to counting the number of connected components of weight-pruned subgraphs. Since $(1,1)$-WL determines the number of connected components (Rattan & Seppelt, 2023), the claim follows. We refer to Appendix H.2 for details. $\quad\square$

### G.2. Proof of Lemma 6

This appendix contains the formal statement and proof of Lemma 6.

**Lemma 77** (Lemma 6 in the main text)**.** *Let $\mathcal{K}$ be the family of all complete graphs, i.e.,*

$$\mathcal{K} := \{G \mid V(G) = [n], \ E(G) = \{\{i, j\} \mid i, j \in [n], \ i \neq j\}, \ \textit{for some } n \in \mathbb{N}\}.$$

*Let $d$ be any (pseudo-)metric on $V_1(\mathcal{K})$ such that the degree invariant*

$$\deg : V_1(\mathcal{K}) \to \mathbb{N}$$

*is L-Lipschitz for some $L < \infty$. Then, for every $\varepsilon \in (0, \frac{1}{L})$, $\mathcal{N}(V_1(\mathcal{K}), d, \varepsilon) = \infty$. Consequently, no hypothesis class containing the degree invariant can satisfy Definition 1 on any graph space containing $V_1(\mathcal{K})$.*

*Proof.* Fix $\varepsilon \in (0, 1/L)$ and set $q := \lceil L\varepsilon \rceil + 1$. For each $k \in \mathbb{N}$, let $K_{1+kq} \in \mathcal{K}$ denote the complete graph on $1 + kq$ vertices, choose an arbitrary $u_k \in V(K_{1+kq})$, and define $x_k := (K_{1+kq}, u_k) \in V_1(\mathcal{K})$. Then $\deg(x_k) = kq$, and for $k \neq \ell$,

$$d(x_k, x_\ell) \ \geq \ \frac{|\deg(x_k) - \deg(x_\ell)|}{L} \geq \frac{q}{L} \ > \ \varepsilon.$$

Thus $\{x_k\}_{k \in \mathbb{N}}$ is an infinite $\varepsilon$-separated subset of $(V_1(\mathcal{K}), d)$ (i.e., $d(x_k, x_\ell) > \varepsilon$ for all $k \neq \ell$), implying $\mathcal{N}(V_1(\mathcal{K}), d, \varepsilon) = \infty$. $\quad\square$

## H. Expressivity limitations

This appendix provides additional technical background for Appendix G.1, including the formal proofs of Lemma 75 and Lemma 76.

### H.1. Separation and approximation

We briefly recall known connections between *discrete separation power* and *continuous approximation power*, following Azizian & Lelarge (2021) and Geerts & Reutter (2022).

Let $C \subseteq \mathbb{R}$ be compact. For $n \in \mathbb{N}$, let $\mathcal{G}_{n,C}$ denote the set of edge-weighted $n$-order graphs with edge weights in $C$. One can represent elements in $\mathcal{G}_{n,C}$ by their weighted adjacency matrices in $C^{n \times n}$. Then, equipped with the product topology, $\mathcal{G}_{n,C}$ is compact.

Let $k \in \mathbb{N}$ and let $\mathcal{F}$ be a class of continuous functions on $V_k(\mathcal{G}_{n,C})$ of the form $f : V_k(\mathcal{G}_{n,C}) \to \mathbb{R}^{\ell_f}$ for some $\ell_f \in \mathbb{N}$ that may depend on $f$. We write $\overline{\mathcal{F}}$ for the closure of $\mathcal{F}$ using the sup norm. That is, a function $h : V_k(\mathcal{G}_{n,C}) \to \mathbb{R}^{\ell_h}$ is in $\overline{\mathcal{F}}$ if there exists a sequence $(f_i)_{i \geq 1} \subseteq \mathcal{F}$ with $f_i : V_k(\mathcal{G}_{n,C}) \to \mathbb{R}^{\ell_h}$ such that

$$\|f_i - h\|_\infty := \sup_{(G, \boldsymbol{v}) \in V_k(\mathcal{G}_{n,C})} \|f_i(G, \boldsymbol{v}) - h(G, \boldsymbol{v})\|_2 \to 0.$$

We assume $\mathcal{F}$ satisfies the following natural assumptions:

(i) *Concatenation-closed:* if $f_1 : V_k(\mathcal{G}_{n,C}) \to \mathbb{R}^d$ and $f_2 : V_k(\mathcal{G}_{n,C}) \to \mathbb{R}^p$ are in $\mathcal{F}$, then $G \mapsto (f_1(G), f_2(G)) \in \mathbb{R}^{d+p}$ is in $\mathcal{F}$.

(ii) *Function/FNN-closed for $\ell$:* if $f : V_k(\mathcal{G}_{n,C}) \to \mathbb{R}^p$ is in $\mathcal{F}$ and $g : \mathbb{R}^p \to \mathbb{R}^\ell$ is continuous or is an FNN, then $g \circ f \in \mathcal{F}$.

For such $\mathcal{F}$ we denote by $\mathcal{F}_\ell$ the subset of functions in $\mathcal{F}$ of the form $V_k(\mathcal{G}_{n,C}) \to \mathbb{R}^\ell$, i.e., with output dimension fixed to $\ell$.

Based on a generalized Stone–Weierstrass theorem (Timofte, 2005), Geerts & Reutter (2022, Theorem 6.1) and Azizian & Lelarge (2021, Lemma 32) combined prove the following characterization.

**Theorem 78.** *Let $C \subseteq \mathbb{R}$ be compact, and let $n, m, k, \ell \in \mathbb{N}$. Let $\mathcal{F}$ be a class of functions on $V_k(\mathcal{G}_{n,C})$ that is concatenation-closed and function/FNN-closed for $\ell$. Then*

$$\overline{\mathcal{F}_\ell} = \big\{ f : V_k(\mathcal{G}_{n,C}) \to \mathbb{R}^\ell \ \big| \ \rho_{V_k(\mathcal{G}_{n,c})}(\mathcal{F}) \subseteq \rho_{V_k(\mathcal{G}_{n,C})}(f) \big\}.$$

Let us fix $L \in \mathbb{N}$ and consider

$$\mathcal{F} := \bigcup_p \mathsf{MPNN}^{\mathcal{Q}_L}_{(\mathcal{T}_L, p)}(\mathcal{G}_{n,C}),$$

i.e., the class of all $L$-layer graph-level MPNNs as defined in Section 1.2, with $\mathcal{T}_L$ implemented by FNNs. Fix $\ell \in \mathbb{N}$. As the class $\mathcal{F}$ of MPNNs is easily verified to be concatenation-closed and FNN-closed for $\ell$, Theorem 78 applies.

**Corollary 79.** *Let $g : \mathcal{G}_{n,C} \to \mathbb{R}^\ell$ be a continuous graph invariant such that $g$ cannot separate more graphs than the class $\mathcal{F}$ of $L$-layer graph-level MPNNs. Then, for any $\epsilon > 0$, there is an $L$-layer graph-level MPNN $f$ in $\mathsf{MPNN}^{\mathcal{Q}_L}_{(\mathcal{T}_L, p)}(\mathcal{G}_{n,\ell})$ satisfying*

$$\sup_{G \in \mathcal{G}_{n,C}} \|g(G) - f(G)\|_2 \leq \epsilon.$$

One can obtain similar statements for invariants $g : V_1(\mathcal{G}_{n,C}) \to \mathbb{R}^\ell$ and the class $\mathcal{F}$ of *node-level* $L$-layer MPNNs, $g : V_2(\mathcal{G}_{n,c}) \to \mathbb{R}^\ell$ and the class $\mathcal{F}$ of node-level $L$-layer MPNNs where each $(G, (r, v)) \in V_2(\mathcal{G}_{n,C})$ as a node $v$ in $G$ in which $r$ is individualized.

In particular, given any $\mathsf{alg} \in \{1\text{-}\mathsf{WL}, 1\text{-}\mathsf{iWL}, (1,1)\text{-}\mathsf{WL}\}$ and class of MPNNs that are $\mathsf{alg}$-simulating, the Corollary implies that this class of MPNNs can arbitrarily approximate any invariant $g$ that is upper bounded in expressive power by $\mathsf{alg}$, resulting in Proposition 7 in the main paper.

In the following, we will use the above result to shed some light on the abilities of MPNNs to approximate or not be able to approximate invariants corresponding to well-known graph problems.

## H.2. Additional details for Section 3.1

We first recall the unrolling-tree characterization of 1-$\mathsf{WL}$ for vertex-labeled, edge-weighted graphs (Morris et al., 2020a) and then adapt it to characterize 1-$\mathsf{iWL}$ as well.

Given a connected vertex-labeled and edge-weighted graph $(G, \ell_G, w_G)$, we define the *unrolling tree* of depth $L \in \mathbb{N}_0$ rooted at a vertex $u \in V(G)$, denoted $\mathsf{unr}(G, u, L)$, inductively as follows.

1. For $L = 0$, $\mathsf{unr}(G, u, 0)$ is the single-vertex tree whose root is labeled $\ell_G(u)$.

2. For $L > 0$, $\mathsf{unr}(G, u, L)$ has a root labeled $\ell_G(u)$ and, for each neighbor $v \in N(u)$, it has a child subtree isomorphic to $\mathsf{unr}(G, v, L-1)$, connected to the root by an edge of weight $w_G(u, v)$.

We now extend this notion to incorporate the distinguished root vertex used in 1-$\mathsf{iWL}$. Given a connected vertex-labeled and edge-weighted graph $(G, \ell_G, w_G)$ and a fixed vertex $r \in V(G)$, we mark $r$ by setting $\ell_G(r) := [*]$. The *unrolling tree regarding $r$* of depth $L \in \mathbb{N}_0$ rooted at $u \in V(G)$, denoted $\mathsf{unr}(G, u, L, r)$, is defined inductively as follows.

1. For $L = 0$, $\mathsf{unr}(G, u, 0, r)$ is the single-vertex tree whose root is labeled $\ell_G(u)$.

2. For $L > 0$, $\mathsf{unr}(G, u, L, r)$ has a root labeled $\ell_G(u)$ and, for each neighbor $v \in N(u)$, it has a child subtree isomorphic to $\mathsf{unr}(G, v, L-1, r)$, connected to the root by an edge of weight $w_G(u, v)$.

The following lemma is immediate.

**Lemma 80** (Follows from Morris et al. (2020a, Lemma 12)). *The following characterizations hold.*

1. *For $L \in \mathbb{N}_0$, given a connected vertex-labeled, edge-weighted graph $(G, \ell_G, w_G)$ and vertices $u, v \in V(G)$, the following are equivalent.*

   - *The vertices $u$ and $v$ have the same color after $L$ iterations of 1-WL.*
   - *The unrolling trees $\mathrm{unr}(G, u, L)$ and $\mathrm{unr}(G, v, L)$ are isomorphic (as rooted, labeled, edge-weighted trees).*

2. *For $L \in \mathbb{N}_0$, given a connected vertex-labeled, edge-weighted graph $(G, \ell_G, w_G)$, a fixed vertex $r \in V(G)$ with $\ell_G(r) = [*]$, and vertices $u, v \in V(G)$, the following are equivalent.*

   - *The vertices $u$ and $v$ have the same color after $L$ iterations of 1-iWL, for individualized $r$.*
   - *The unrolling trees $\mathrm{unr}(G, u, L, r)$ and $\mathrm{unr}(G, v, L, r)$ are isomorphic (as rooted, labeled, edge-weighted trees).*

### H.2.1. PROOF OF LEMMA 75

Let $(G, w_G)$ and $(H, w_H)$ be two connected edge-weighted graphs, and let $s, v \in V(G)$ and $t, w \in V(H)$. We need to show that if $C_\infty^{1.5,s}(v) = C_\infty^{1.5,t}(w)$, then $\mathsf{SSSP}(G, (s, v)) = \mathsf{SSSP}(H, (t, w))$.

Assume for contradiction that $\mathsf{SSSP}(G, (s, v)) \neq \mathsf{SSSP}(H, (t, w))$. Fix $L$ large enough so that the (unique) marked vertices $s$ in $\mathrm{unr}(G, v, L, s)$ and $t$ in $\mathrm{unr}(H, w, L, t)$ appear within depth $L$ along all shortest root-to-marked paths. In $\mathrm{unr}(G, v, L, s)$, the cost of the shortest path from the root $v$ to the unique vertex labeled $[*]$ equals $\mathsf{SSSP}(G, (s, v))$, and analogously, in $\mathrm{unr}(H, w, L, t)$ the cost of the shortest path from the root $w$ to the unique vertex labeled $[*]$ equals $\mathsf{SSSP}(H, (t, w))$. If $\mathsf{SSSP}(G, (s, v)) \neq \mathsf{SSSP}(H, (t, w))$, then the two rooted, vertex-labeled, edge-weighted trees $\mathrm{unr}(G, v, L, s)$ and $\mathrm{unr}(H, w, L, t)$ cannot be isomorphic, contradicting the unrolling-tree characterization of 1-iWL (see Lemma 80) together with the assumption $C_\infty^{1,s}(v) = C_\infty^{1,t}(w)$.

### H.2.2. PROOF OF LEMMA 76

Let $(G, w_G)$ and $(H, w_H)$ be two connected edge-weighted graphs that are $(1,1)$-WL-indistinguishable. We need to show that $\mathsf{MST}(G) = \mathsf{MST}(H)$.

We prove this by showing that the cost of a minimal spanning tree is determined by the number of connected components in threshold subgraphs that only hold edges with weights below a certain threshold. Lemma 75 then follows from the fact that $(1,1)$-WL determines the number of connected components. We detail the argument at the end of this subsection.

For a graph $X$ we write $\mathrm{cc}(X)$ for its set of connected components and $\#\mathrm{cc}(X)$ for the number of connected components. Let $(G, w_G)$ be an (undirected) edge-weighted graph and let $W(G) := \{w_G(e) \mid e \in E(G)\}$ be its set of edge weights. Let the distinct weights be

$$w_1 < w_2 < \cdots < w_m,$$

and set $w_{m+1} := +\infty$. For any $w \in \mathbb{R}_{>0}$ we define the (unweighted) *threshold subgraph*

$$G_{<w} := \big(V(G), \{e \in E(G) \mid w_G(e) < w\}\big).$$

For each $j \in [m+1]$ define $\kappa_j := \#\mathrm{cc}(G_{<w_j})$. Note that $\kappa_{m+1} = \#\mathrm{cc}(G)$.

Let $\mathsf{MSF}(G)$ be the minimum spanning forest cost, i.e., the total weight of a minimum spanning forest of $G$. If $G$ is connected, then we denote $\mathsf{MSF}(G)$ by $\mathsf{MST}(G)$.

**Lemma 81.** *For any edge-weighted graph $(G, w_G)$ with distinct weights $w_1 < \cdots < w_m$,*

$$\mathsf{MSF}(G) = \sum_{j=1}^{m} (\kappa_j - \kappa_{j+1})\, w_j.$$

*Proof.* For $j \in [m]$, set $G_j := G_{<w_{j+1}}$. Then $E(G_j) = \{e \in E(G) \mid w_G(e) \leq w_j\}$ and $\#\mathrm{cc}(G_j) = \kappa_{j+1}$.

We prove by induction on $j$ that

$$\mathsf{MSF}(G_j) = \sum_{i=1}^{j} (\kappa_i - \kappa_{i+1})\, w_i. \tag{20}$$

Taking $j = m$ yields the claim since $G_m = G$.

*Base case $j = 1$.* We have $G_1 = G_{<w_2}$, hence every edge in $G_1$ has weight $w_1$. Any spanning forest of $G_1$ has exactly $|V(G)| - \kappa_2$ edges, so

$$\mathsf{MSF}(G_1) = (|V(G)| - \kappa_2)\, w_1 = (\kappa_1 - \kappa_2)\, w_1,$$

since $G_{<w_1}$ has no edges and thus $\kappa_1 = \#\mathrm{cc}(G_{<w_1}) = |V(G)|$.

*Induction step.* Assume (20) holds for $j-1$. Let $H := G_{j-1} = G_{<w_j}$, so $\#\mathrm{cc}(H) = \kappa_j$ and all edges in $H$ have weight $< w_j$.

*Upper bound.* Let $F$ be a minimum spanning forest of $H$, so $\mathsf{MSF}(H) = \sum_{e \in E(F)} w_G(e)$ and $F$ has $\kappa_j$ components. Since $G_j$ has $\kappa_{j+1}$ connected components, there exists a set $S \subseteq E(G_j)$ of exactly $\kappa_j - \kappa_{j+1}$ edges of weight $w_j$ that connect components of $F$ without creating cycles. Then $F \cup S$ is a spanning forest of $G_j$, and therefore

$$\mathsf{MSF}(G_j) \le \sum_{e \in E(F \cup S)} w_G(e) = \mathsf{MSF}(H) + (\kappa_j - \kappa_{j+1})\, w_j.$$

*Lower bound.* Let $T$ be any spanning forest of $G_j$. Removing all edges of weight $w_j$ from $T$ leaves a forest $T_{<w_j} \subseteq E(H)$, so $T_{<w_j}$ has at least $\kappa_j$ components. Each edge of weight $w_j$ in $T$ can reduce the number of components by at most 1, hence $T$ must contain at least $\kappa_j - \kappa_{j+1}$ edges of weight $w_j$. Thus

$$\sum_{e \in E(T)} w_G(e) \ge \sum_{e \in E(T_{<w_j})} w_G(e) + (\kappa_j - \kappa_{j+1})\, w_j \ge \mathsf{MSF}(H) + (\kappa_j - \kappa_{j+1})\, w_j.$$

Taking the minimum over all such $T$ yields

$$\mathsf{MSF}(G_j) \ge \mathsf{MSF}(H) + (\kappa_j - \kappa_{j+1})\, w_j.$$

Combining the bounds gives $\mathsf{MSF}(G_j) = \mathsf{MSF}(H) + (\kappa_j - \kappa_{j+1})\, w_j$, and substituting the induction hypothesis for $\mathsf{MSF}(H) = \mathsf{MSF}(G_{j-1})$ proves (20). $\qquad\square$

**Claim 82.** *Let $(G, w_G)$ and $(H, w_H)$ be edge-weighted graphs that are $(1,1)$-WL indistinguishable. Then $W(G) = W(H)$. Assume that $W(G) = \{w_1, \ldots, w_m\}$ with $w_1 < w_2 < \cdots < w_m$. Then for every $j \in [m]$,*

$$\#\mathrm{cc}(G_{<w_j}) = \#\mathrm{cc}(H_{<w_j}) \qquad \text{and} \qquad \#\mathrm{cc}(G) = \#\mathrm{cc}(H).$$

*Proof.* By $(1,1)$-WL-indistinguishability there exists a bijection $\pi\colon V(G) \to V(H)$ such that for all $v \in V(G)$, $(G, v, H, \pi(v))$ are 1-iWL-indistinguishable regarding $v$ and $\pi(v)$. In particular, $\mathrm{unr}(G, v, 1, v)$ and $\mathrm{unr}(H, \pi(v), 1, \pi(v))$ are isomorphic, for all $v \in V(G)$. These unrolling trees contain all edge weights in the respective graphs. Hence, $W(G) = W(H)$.

Fix $j \in [m]$. Consider the unweighted graphs $G_{<w_j}$ and $H_{<w_j}$. It is readily verified by unrolling tree characterization that $G_{<w_j}$ and $H_{<w_j}$ are also $(1,1)$-WL-equivalent. Hence, by the spectral characterization of $(1,1)$-WL via equitable matrix maps, the Laplacian spectra of $G_{<w_j}$ and $H_{<w_j}$ coincide (Rattan & Seppelt, 2023). The multiplicity of eigenvalue 0 of the Laplacian equals the number of connected components, hence $\#\mathrm{cc}(G_{<w_j}) = \#\mathrm{cc}(H_{<w_j})$. The same argument yields $\#\mathrm{cc}(G) = \#\mathrm{cc}(H)$. $\qquad\square$

We are now finally ready to formally prove Lemma 75. Indeed, if $(G, w_G)$ and $(H, w_H)$ are $(1,1)$-WL-indistinguishable, then by the previous claim, $\#\mathrm{cc}(G_{<w_j}) = \#\mathrm{cc}(H_{<w_j})$ and $\#\mathrm{cc}(G) = \#\mathrm{cc}(H)$. Applying Lemma 81 the suffices to conclude $\mathsf{MST}(G) = \mathsf{MST}(H)$, as desired.

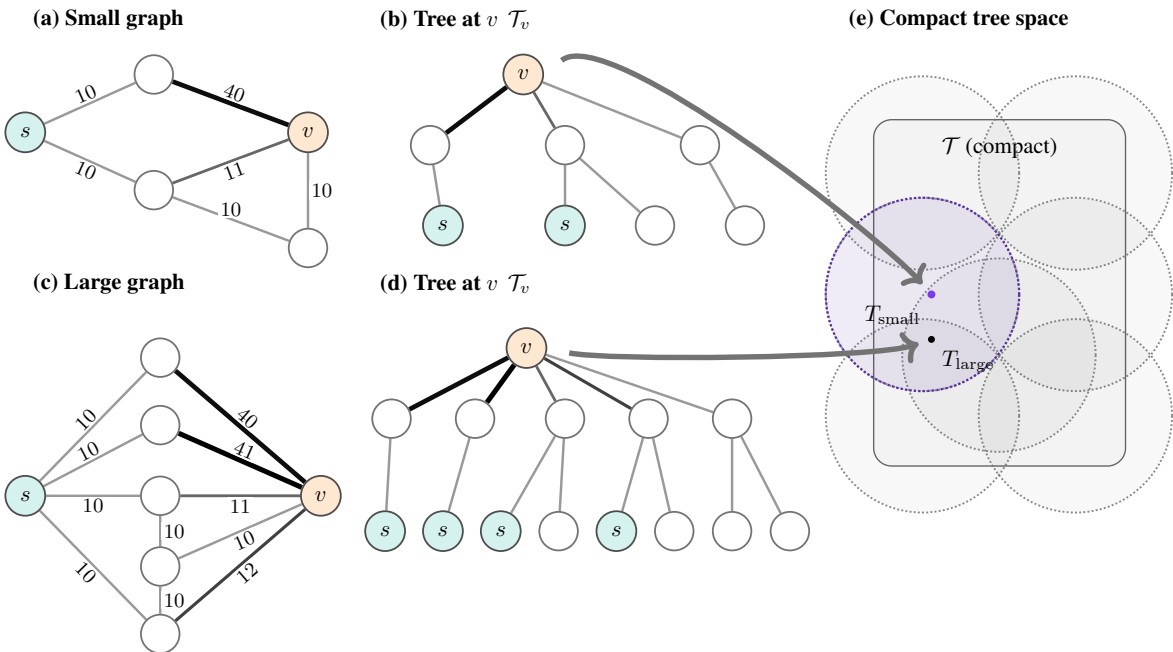

**(a) Small graph**

**(b) Tree at $v$ $\mathcal{T}_v$**

**(e) Compact tree space**

**(c) Large graph**

**(d) Tree at $v$ $\mathcal{T}_v$**

*Figure 5.* A compact space of computation trees enables algorithmic generalization. Under an appropriate metric on $\mathcal{T}$, the computation trees $T_{\text{small}}$ and $T_{\text{large}}$ are close, so regularization-induced Lipschitz continuity of the model implies that good performance on the training instance $T_{\text{small}}$ transfers to good performance on the nearby instance $T_{\text{large}}$. Occurrences of $v$ in the computation tree other than the root are omitted for simplicity.

## I. MPNN classes satisfying finite Lipschitzness

Below, we describe hypothesis classes for graph learning models—which capture many well-known graph algorithms—that satisfy the finite Lipschitz learning property introduced in Definition 1 and, consequently, Theorem 2. Throughout, the input space $\mathcal{X}$ consists of pairs of graphs and nodes belonging to these graphs. The parameter space $\Theta$ is a subset of $\mathbb{R}^P$ for some $P \in \mathbb{N}$, which will be specified later. We equip the input space with a suitable pseudo-metric such that the conditions of Definition 1 are satisfied. More precisely, we first establish this property for message-passing neural networks (MPNNs) with normalized-sum aggregation. This result largely follows existing work (see, e.g., (Rauchwerger et al., 2024; Grebik & Rocha, 2022; Böker et al., 2023)) based on iterated degree measure spaces, a continuous counterpart of computation trees (see Appendix M), endowed with the Kantorovich distance (see Appendix J), but with important simplifications. In particular, in our setting, it is not necessary to show compactness of the input space, since Definition 1 requires only a weaker condition, namely finiteness of the covering number. We then extend the result to MPNNs with mean aggregation.

Finally, using similar tools but applied to different topological spaces—namely, Hausdorff spaces rather than measure spaces, equipped with the Hausdorff distance—we show that MPNNs with max and min aggregations satisfy the finite Lipschitz learning property. These architectures encompass many commonly used graph algorithms, including shortest-path algorithms, minimum spanning tree problems, and related tasks. Below, we present the precise architectures and state the main result.

Overall, we get the following results.

**Assumptions** Let $\mathcal{G}_{r,p_0}$ denote the space of undirected attributed and edge-weighted graphs $(G, a_G, w_G)$, without isolated nodes,[5] with node features $a_G(u) \in B_{r,p_0} := \{x \in \mathbb{R}^{p_0} \mid \|x\|_2 \leq r\}$ for all $u \in V(G)$, where $p_0 \in \mathbb{N}$ and $r > 0$, and edge-weights $w_G(e) \in E \subseteq \mathbb{R}_{>0}$ for some compact set $E$, for all $e \in E(G)$. Also, let $V_1(\mathcal{G}_{r,p_0})$ denote the space consisting of pairs of graphs in $\mathcal{G}_{r,p_0}$ and their nodes (i.e., $V_1(\mathcal{G}_{r,p_0}) = \{(G, u) \mid G \in \mathcal{G}_{r,p_0}, u \in V(G)\}$).

Let $L \in \mathbb{N}$ and let $p_t \in \mathbb{N}$. Let $\phi_1 \colon B_{r,p_0} \times B_{r,p_0} \to \mathbb{R}^{p_1}$, and

$$\phi_t \colon \mathbb{R}^{p_{t-1}} \times \mathbb{R}^{p_{t-1}} \to \mathbb{R}^{p_t} \quad (t > 1), \qquad M_t \colon \mathbb{R}^{p_{t-1}} \times \mathbb{R} \to \mathbb{R}^{p_{t-1}} \quad (t \geq 1).$$

---

[5] We consider non-isolated nodes to avoid division by zero for mean aggregation and empty sets for max aggregation in Appendix L.1.

We assume that for each $t \in [L]$ there exist constants $C_{\phi,1}^{(t)}, C_{\phi,2}^{(t)}, C_{M,1}^{(t)}, C_{M,2}^{(t)} > 0$ such that

$$\|\phi_t(x,y) - \phi_t(x',y')\|_2 \leq C_{\phi,1}^{(t)} \|x - x'\|_2 + C_{\phi,2}^{(t)} \|y - y'\|_2, \quad \forall x, y, x', y' \in \mathbb{R}^{p_{t-1}}.$$

$$\|M_t(x,y) - M_t(x',y')\|_2 \leq C_{M,1}^{(t)} \|x - x'\|_2 + C_{M,2}^{(t)} \|y - y'\|_2, \quad \forall (x,y), (x',y') \in \mathbb{R}^{p_{t-1}} \times \mathbb{R}.$$

Moreover, we assume bounded offsets, i.e., $\|\phi_t(0,0)\|_2 \leq B_t^{(0)}$, for some $B_t > 0$. Finally, for graph level predictions let $\psi \colon \mathbb{R}^{p_L} \to \mathbb{R}^d$ be a Lipschitz function with respect to the $\|\cdot\|_2$ norm.

Based on these assumptions on $\{\phi_t\}_{t \in [L]}$ and $\psi$, we define three specific MPNN architectures, each of which is an instance of the general MPNN framework introduced in Equation (1).

**Normalized sum aggregation** Let $(G, u) \in V_1(\mathcal{G}_{r,p_0})$, define $h_u^{(0)} = a_G(u) \in B_{r,p_0}$ and recursively

$$h_u^{(t)} = \phi_t\left( h_u^{(t-1)}, \frac{1}{|V(G)|} \sum_{v \in N(u)} w_{uv} h_v^{(t-1)} \right), \quad t \in [L]. \tag{21}$$

For graph-level tasks, the final $d$-dimensional representation is given by

$$h_G = \psi\left( \frac{1}{|V(G)|} \sum_{u \in V(G)} h_u^{(L)} \right).$$

A particular instance of such a hypothesis class is given by

$$\mathcal{F}_{\Theta} := \left\{ f \colon V_1(\mathcal{G}_{r,p_0}) \to \mathbb{R}^{p_L} \,\middle|\, f(G,u) = h_u^{(L)}, \; \phi_t(x,y) = \sigma\left( W_1^{(t)} x + W_2^{(t)} y \right), \right.$$
$$\left. W_1^{(t)}, W_2^{(t)} \in \mathbb{R}^{p_t \times p_{t-1}}, \; t \in [L] \right\},$$

where $\sigma$ denotes the element-wise ReLU activation, or more generally any Lipschitz function (e.g., Leaky ReLU). For this class, the Lipschitz constants are $C_{\phi,1}^{(t)} = \|W_1^{(t)}\|_2$ and $C_{\phi,2}^{(t)} = \|W_2^{(t)}\|_2$, while the offset bounds satisfy $B_t^{(0)} = 0$. The parameter space $\Theta$ consists of all matrices $\{W_1^{(t)}, W_2^{(t)}\}_{t \in [L]}$. Although $\phi_t$ is defined here as a single-layer feed-forward network, the same construction extends to arbitrary-depth feed-forward networks with ReLU (or any other 1-Lipschitz) activations; in that case, $C_{\phi,1}^{(t)}$ and $C_{\phi,2}^{(t)}$ are given by the products of the spectral norms of the corresponding weight matrices.

**Mean aggregation** Let $(G, u) \in V_1(\mathcal{G}_{r,p_0})$, define $\overline{h}_u^{(0)} = a_G(u) \in B_{r,p_0}$ and recursively:

$$\overline{h}_u^{(t)} = \phi_t\left( \overline{h}_u^{(t-1)}, \frac{1}{\deg(u)} \sum_{v \in N(u)} w_{uv} \overline{h}_v^{(t-1)} \right), \quad t \in [L]. \tag{22}$$

For graph-level tasks, we again define

$$\overline{h}_G = \psi\left( \frac{1}{|V(G)|} \sum_{u \in V(G)} \overline{h}_u^{(L)} \right).$$

A special case of such a hypothesis class can be derived similarly to normalized sum aggregation MPNNs by replacing $\phi_t$, with FNNs.

**Max-min aggregation** Let $(G, u) \in V_1(\mathcal{G}_{r,p_0})$, define $\hat{h}_u^{(0)} := x_u$ and, recursively:

$$\hat{h}_u^{(t)} = \phi_t\left( \hat{h}_u^{(t-1)}, \max_{v \in N(u)} M_t\left( \hat{h}_v^{(t-1)}, w_{uv} \right) \right), \tag{23}$$

where The maximum in Equation (23) is taken coordinatewise in $\mathbb{R}^{p_{t-1}}$. Note that similarly we can define MPNNs based on min aggregation by replacing the max operator with min.

A special case of such a hypothesis class can be derived similarly to normalized sum aggregation MPNNs by replacing $\phi_t$ and $M_t$ with FNNs.

The following result shows that all the above-defined families of MPNNs satisfied the finite Lipschitz property from Definition 1.

**Theorem 83** (Theorem 3 in the main text). *The hypothesis class $\mathcal{F}_\Theta$ induced by MPNNs using any one of the following aggregation schemes, normalized sum aggregation (Equation (21)), mean aggregation (Equation (22)), or max (or min) aggregation (Equation (23)), satisfies Definition 1.*

*Proof.* see Appendix K, Appendix L. □

**Remark** The above result extends directly to graph-level representations. Indeed, if the readout function $\psi$ is Lipschitz, then the composition of the node-level MPNN with $\psi$ remains Lipschitz with respect to the induced graph-level pseudo-metric. Consequently, the finite Lipschitz learning property continues to hold for graph-level prediction tasks.

**Extensions to mixed aggregation architectures** The exclusivity of the aggregation choices in Theorem 3 reflects a technical limitation of the proof strategy, i.e., the pseudometrics introduced in Appendices K and L are tailored to individual aggregation mechanisms and do not suffice to establish the result simultaneously for all of them, nor for architectures that combine multiple aggregation operators.

Nevertheless, the proof techniques developed in Appendices K and L extend beyond the specific aggregation schemes considered there. By equipping the relevant input spaces with product topologies endowed with summed metrics, one can naturally combine different aggregation mechanisms, such as mean or normalized sum, together with max (or min), within a single GNN architecture. More generally, the arguments can be unified by considering a generic aggregation operator and defining suitable pseudometrics on the associated iterated neighborhood spaces, such as the Kantorovich–Rubinstein distance on spaces of iterated degree measures or the Hausdorff distance on spaces of iterated neighborhood sets. With these choices, the input space remains compact, and both aggregation and update maps remain Lipschitz with respect to the resulting metrics.

## J. Topological, measure-theoretic, and geometric background

This appendix collects all background material required for the proof of Theorem 3. The central objective is to construct suitable metric spaces for node and neighborhood representations induced by message passing neural networks, and to show that these spaces are compact. Compactness will later imply the finiteness of covering numbers, which is the key condition in Definition 1.

The appendix is organized as follows. Section J.1 recalls basic notions from topology and metric geometry. Section J.2 reviews measure-theoretic preliminaries and weak* convergence. Section J.3 introduces iterated degree measures and the Kantorovich–Rubinshtein metric, which underpin the analysis of sum and mean aggregation. Section J.4 develops the hyperspace and Hausdorff-metric framework needed for max (and min) aggregation.

### J.1. Topological background

We begin by recalling standard definitions from topology, emphasizing compactness and product constructions, which will be repeatedly used in later proofs.

**Definition 84** (Topological space). A topological space is a pair $(\mathcal{X}, \tau)$, where $\mathcal{X}$ is a set and $\tau$ is a collection of subsets of $\mathcal{X}$ containing the empty set and $\mathcal{X}$, closed under arbitrary unions and finite intersections. The elements of $\tau$ are called open sets.

Let $(\mathcal{X}, \tau_\mathcal{X})$ and $(\mathcal{Y}, \tau_\mathcal{Y})$ be topological spaces. A map $f \colon \mathcal{X} \to \mathcal{Y}$ is said to be *continuous* if for every open set $U \subseteq \mathcal{Y}$, the preimage $f^{-1}(U)$ is an open subset of $\mathcal{X}$.

Let $\tau_1$ and $\tau_2$ be two topologies on the same set $\mathcal{X}$. We say that $\tau_1$ is *coarser* than $\tau_2$ if $\tau_1 \subseteq \tau_2$.

**Definition 85** (Product topology). Let $\{(\mathcal{X}_i, \tau_i)\}_{i \in I}$ be a family of topological spaces. The product topology on $\prod_{i \in I} \mathcal{X}_i$ is the coarsest topology for which all coordinate projections are continuous.

**Definition 86** (Compact space). A topological space $(\mathcal{X}, \tau)$ is compact if every open cover of $\mathcal{X}$ admits a finite subcover.

**Theorem 87** (Tychonoff). *The product of compact topological spaces is compact with respect to the product topology.*

**Definition 88** (Metric space and induced topology). Let $X$ be a set. A function $d\colon X \times X \to [0, +\infty)$ is a *metric* on $X$ if, for all $x, y, z \in X$,

$$d(x,y) = 0 \iff x = y, \quad d(x,y) = d(y,x), \quad d(x,z) \leq d(x,y) + d(y,z).$$

The pair $(X, d)$ is called a metric space. The metric $d$ induces a topology on $X$ whose open sets are those $U \subset X$ such that for every $x \in U$ there exists $\varepsilon > 0$ with $B_d(x, \varepsilon) \subset U$.

**Definition 89** (Metrizable topology). A topological space $(\mathcal{X}, \tau)$ is said to be *metrizable* if there exists a metric $d$ on $\mathcal{X}$ such that the topology induced by $d$ coincides with $\tau$.

## J.2. Measure-theoretic background

We next recall basic measure-theoretic notions that will be used to model neighborhood aggregation by averaging or summation.

Let $\mathcal{X}$ be a set. A collection $\mathcal{A} \subseteq 2^{\mathcal{X}}$ is a *$\sigma$-algebra* if $\mathcal{X} \in \mathcal{A}$, $\mathcal{A}$ is closed under complements, and under countable unions.

**Definition 90** (Measure). Let $(\mathcal{X}, \mathcal{A})$ be a measurable space. A measure on $(\mathcal{X}, \mathcal{A})$ is a map $\mu\colon \mathcal{A} \to [0, \infty]$ such that $\mu(\emptyset) = 0$ and

$$\mu\left( \bigcup_{i \in \mathbb{N}} A_i \right) = \sum_{i \in \mathbb{N}} \mu(A_i)$$

for every countable collection of pairwise disjoint sets.

Given a collection $\mathcal{S} \subseteq 2^{\mathcal{X}}$, we denote by $\sigma(\mathcal{S})$ the smallest $\sigma$-algebra containing $\mathcal{S}$. Given a topological space $(\mathcal{X}, \tau)$, we denote by $\mathcal{B}(\mathcal{X}) = \sigma(\tau)$ the Borel $\sigma$-algebra on $\mathcal{X}$, by $\mathcal{M}_{\leq 1}(\mathcal{X})$ the space of finite Borel measures with total mass at most $1$, and by $\mathcal{P}(\mathcal{X})$ the space of finite Borel measures with total mass exactly $1$ (i.e., probability measures).

**Definition 91** (Weak* topology). Let $(\mathcal{X}, \tau)$ be a topological space. The weak* topology on $\mathcal{M}_{\leq 1}(\mathcal{X})$ (or similarly on $\mathcal{P}(\mathcal{X})$) is the coarsest topology such that, for every bounded continuous function $f\colon \mathcal{X} \to \mathbb{R}$, the map

$$\mu \mapsto \int_{\mathcal{X}} f \, d\mu$$

is continuous.

**Theorem 92** ((Kechris, 1995), Theorem 17.22). *If $(\mathcal{X}, \tau)$ is a compact metrizable space, then the space $\mathcal{M}_{\leq 1}(\mathcal{X})$ (similarly $\mathcal{P}(\mathcal{X})$), endowed with the weak\* topology, are also compact and metrizable.*

## J.3. Iterated degree measures and Kantorovich–Rubinshtein metric

This subsection develops the measure-valued representation spaces used for normalized-sum and mean aggregation. We recall that $B_{r,p_0} := \{x \in \mathbb{R}^{p_0} \mid \|x\|_2 \leq r\}$.

**Definition 93** (Iterated degree measures). Let $L \in \mathbb{N}$. We define a sequence of spaces recursively $\{M_\ell\}_{\ell=0}^{L}$ and $\{H_\ell\}_{\ell=0}^{L}$ as follows.

- Set
$$M_0 := B_{r,p_0}, \qquad H_0 := M_0.$$

- For each $\ell \geq 0$, define
$$M_{\ell+1} := \mathcal{M}_{\leq 1}(H_\ell), \qquad H_{\ell+1} := \prod_{j=0}^{\ell+1} M_j.$$

Note that, an element of $M_\ell$ is a (sub-)probability measure on the space $H_{\ell-1}$, whereas an element of $H_\ell$ is a tuple

$$(h_0, h_1, \ldots, h_\ell), \qquad h_j \in M_j \text{ for each } j \leq \ell.$$

In particular, for $\ell \geq 1$, the space $H_\ell$ collects the iterated degree measures up to level $\ell$, while $M_{\ell+1}$ consists of measures supported on such tuples.

**Definition 94** (Kantorovich–Rubinshtein distance). Let $(\mathcal{X}, d)$ be a metric space and let $\mu, \nu \in \mathcal{M}_{\leq 1}(\mathcal{X})$. Define

$$\mathrm{Lip}_1(\mathcal{X}, \mathbb{R}) := \{f \colon \mathcal{X} \to \mathbb{R} \mid |f(x) - f(x')| \leq d(x, x') \; \forall x, x' \in \mathcal{X}\}.$$

The Kantorovich–Rubinshtein distance is

$$\mathbf{K}(\mu, \nu) := \sup_{\substack{f \in \mathrm{Lip}_1(\mathcal{X}, \mathbb{R}) \\ \|f\|_\infty \leq 1}} \left| \int_{\mathcal{X}} f \, d\mu - \int_{\mathcal{X}} f \, d\nu \right|.$$

*Remark* 95. When $\mu$ and $\nu$ are probability measures on $\mathcal{X}$, the Kantorovich–Rubinshtein distance coincides with the 1-Wasserstein distance and admits an equivalent formulation as an optimal transportation problem (see (Chuang & Jegelka, 2022)). The restriction to bounded test functions in the supremum is essential for well-definedness when considering finite (sub-probability) measures. Indeed, without imposing a bound on $f$, every constant function $f \equiv c$ is 1-Lipschitz and if $\mu(\mathcal{X}) \neq \nu(\mathcal{X})$, then $\int_{\mathcal{X}} f \, d\mu - \int_{\mathcal{X}} f \, d\nu = c(\mu(\mathcal{X}) - \nu(\mathcal{X}))$, which can be made arbitrarily large in absolute value by letting $|c| \to \infty$. Consequently, the supremum would be infinite.

**Kantorovich–Rubinshtein recursive metric** We define a metric $d_{\mathbf{K},t}$ on $H_L$, for $t \in \mathbb{N}$ recursively as follows. For $x, y \in H_0$, recall $H_0 = M_0 = B_{r,p_0}$, and set $d_0(x, y) = \|x - y\|_2$. For $t \geq 1$, let $x = (\eta, \mu)$ and $x' = (\eta', \mu')$ be elements of $H_t$, where $\eta, \eta' \in H_{t-1}$ and $\mu, \mu' \in M_t$. Define

$$d_t(x, x') := d_{t-1}(\eta, \eta') + \mathbf{K}(\mu, \mu').$$

**Theorem 96.** *For every $L \in \mathbb{N}$, $(H_L, d_L)$ is a compact metric space.*

*Proof.* We prove the theorem by induction on $t$. We begin by stating four claims that will be used throughout the proof.

**Claim 1.** If $(\mathcal{X}, d)$ is a compact metric space, then $\mathcal{M}_{\leq 1}(\mathcal{X})$ is compact with respect to the weak* topology. This follows from Theorem 92.

**Claim 2.** If $(\mathcal{X}, d)$ is a compact metric space, then the Kantorovich–Rubinstein metric $\mathbf{K}$ defined in Definition 94 metrizes the weak* topology on $\mathcal{M}_{\leq 1}(\mathcal{X})$. This follows from Bogachev (2007)[Theorem 8.3.2].

**Claim 3.** If $A$ and $B$ are compact topological spaces, then $A \times B$ is compact with respect to the product topology. This is a direct consequence of Tychonoff's theorem (Theorem 87).

**Claim 4.** Let $(A, d_A)$ and $(B, d_B)$ be metric spaces, and define the sum metric $d$ on $A \times B$ by

$$d\big((a, b), (a', b')\big) := d_A(a, a') + d_B(b, b').$$

Then $d$ metrizes the product topology induced by $d_A$ and $d_B$. Indeed, for any $\varepsilon > 0$,

$$B_{d_A}\big(a, \varepsilon/2\big) \times B_{d_B}\big(b, \varepsilon/2\big) \subseteq B_d\big((a, b), \varepsilon\big) \subseteq B_{d_A}(a, \varepsilon) \times B_{d_B}(b, \varepsilon),$$

where for a metric space $(\mathcal{X}, d)$ and $\varepsilon > 0$ we write

$$B_d(x, \varepsilon) = \{y \in \mathcal{X} \mid d(x, y) \leq \varepsilon\}.$$

We now proceed with the proof. For the base case $t = 0$, we have $H_0 = B_{r,p_0}$ with $d_{\mathbf{K},0}(x, y) = \|x - y\|_2$. Since $B_{r,p_0}$ is closed and bounded in $\mathbb{R}^{p_0}$, it is compact. Hence $(H_0, d_{\mathbf{K},0})$ is compact.

For the induction step, assume that $(H_{t-1}, d_{\mathbf{K},t-1})$ is a compact metric space for some $t \geq 1$. We show that $(H_t, d_{\mathbf{K},t})$ is compact. By the induction hypothesis and Claim 1, $\mathcal{M}_{\leq 1}(H_{t-1}) = M_t$ is compact in the weak* topology. By Claim 2, the metric $\mathbf{K}$ metrizes this topology, so $(M_t, \mathbf{K})$ is a compact metric space. Since $H_t = H_{t-1} \times M_t$ and both $H_{t-1}$ and $M_t$ are compact, Claim 3 implies that $H_t$ is compact with respect to the product topology. Finally, by Claim 4, the topology induced by $d_{\mathbf{K},t}$ coincides with the product topology induced by $d_{\mathbf{K},t-1}$ on $H_{t-1}$ and $\mathbf{K}$ on $M_t$. Therefore, $(H_t, d_{\mathbf{K},t})$ is compact, completing the induction. $\square$

## J.4. Hyperspaces and Hausdorff metrics for max aggregation

We now turn to max (and min) aggregation, where neighborhoods are encoded as compact sets rather than measures.

**Hyperspace and Hausdorff metric**    Let $(\mathcal{X}, d)$ be a metric space and denote by

$$\mathrm{Haus}(\mathcal{X}) := \{A \subseteq \mathcal{X} \mid A \neq \emptyset, \ A \text{ is compact}\}.$$

the hyperspace of nonempty compact subsets of $\mathcal{X}$. The Hausdorff metric on $\mathrm{Haus}(\mathcal{X})$ induced by $d$ is denoted by $\mathbf{H}_d$ and is defined as

$$\mathbf{H}_d(A, B) = \max\left\{ \sup_{a \in A} \inf_{b \in B} d(a, b), \sup_{b \in B} \inf_{a \in A} d(a, b)\right\}.$$

The following result is a corollary of Blaschke Selection Theorem (Schneider, 2013)[Thm. 1.8.4].

**Theorem 97.** *Let $(\mathcal{X}, d)$ be compact. Then $(\mathrm{Haus}(\mathcal{X}), \mathbf{H}_d)$ is compact.*

**Iterated neighborhood set spaces**    Let $E \subset \mathbb{R}^e$ be compact. Define recursively

- $S_0^{\max} = B_{r, p_0}, \quad H_0^{\max} = S_0^{\max}$.
- $Y_t^{\max} = H_t^{\max} \times E, \quad S_{t+1}^{\max} = \mathrm{Haus}(Y_t^{\max}), \quad H_{t+1}^{\max} = H_t^{\max} \times S_{t+1}^{\max}$.

**Hausdorff recursive metric**    We define a metric $d_{\mathbf{H},t}$ on $H_t^{\max}$ recursively. For $x, y \in H_0^{\max}$, set

$$d_{\mathbf{H},0}(x, y) := \|x - y\|_2.$$

Equip $Y_{t-1}^{\max} = H_{t-1}^{\max} \times E$ with the product metric

$$d_{Y,t-1}\big((\eta, e), (\eta', e')\big) := d_{\mathbf{H},t-1}(\eta, \eta') + \|e - e'\|_2.$$

Let $\mathbf{H}_{t-1}$ denote the Hausdorff metric on $\mathrm{Haus}(Y_{t-1}^{\max})$ induced by $d_{Y,t-1}$. For $t \geq 1$ and $x = (\eta, A)$, $x' = (\eta', A')$ in $H_t^{\max} = H_{t-1}^{\max} \times S_t^{\max}$, define

$$d_{\mathbf{H},t}(x, x') := d_{\mathbf{H},t-1}(\eta, \eta') + \mathbf{H}_{t-1}(A, A').$$

**Theorem 98** (Compactness of $(H_t^{max}, d_{\mathbf{H},t})$). *For every $t \in \mathbb{N}$, the space $(H_t^{max}, d_{\mathbf{H},t})$ is a compact metric space.*

*Proof.* We prove the theorem by induction on $t$. For $t = 0$, by definition, $H_0^{\max} = B_{r, p_0}$ and $d_{\mathbf{H},0}(x, y) = \|x - y\|_2$, hence $(H_0^{\max}, d_{\mathbf{H},0})$ is compact.

Assume that $(H_{t-1}^{\max}, d_{\mathbf{H},t-1})$ is compact for some $t \geq 1$. We show that $(H_t^{\max}, d_{\mathbf{H},t})$ is compact. By the induction hypothesis and the compactness of the edge-feature space $E$, the product space $Y_{t-1}^{\max} = H_{t-1}^{\max} \times E$ is compact with respect to the sum metric $d_{Y,t-1}$ (see Claim 4 in the proof of Theorem 96). By Theorem 97, the hyperspace $\mathrm{Haus}(Y_{t-1}^{\max})$ endowed with the Hausdorff metric $\mathbf{H}_{t-1}$ is compact. Hence, the neighborhood set space $S_t^{\max}$ is compact. Since $H_t^{\max} = H_{t-1}^{\max} \times S_t^{\max}$ is a finite product of compact spaces, it is compact in the product topology (Theorem 87). Finally, by construction, the metric

$$d_{\mathbf{H},t}\big((\eta, A), (\eta', A')\big) = d_{\mathbf{H},t-1}(\eta, \eta') + \mathbf{H}_{t-1}(A, A')$$

metrizes the product topology (again see Claim 4 in Theorem 96). Therefore $(H_t^{\max}, d_{\mathbf{H},t})$ is compact. $\square$

## K. Proof of Theorem 3 for sum/mean-aggregation

This section proves Theorem 3 for the normalized-sum and mean aggregation schemes. The overall strategy is to construct, for each $(G, u) \in \mathcal{G}_{r, p_0} \otimes \mathcal{V}$, a canonical *iterated degree measure* (IDM) representation in the compact metric space $(H_L, d_{\mathbf{K},L})$ from Appendix J. This allows us to endow $\mathcal{G}_{r, p_0} \otimes \mathcal{V}$ with a pseudo-metric induced by $d_{\mathbf{K},L}$ and to deduce finiteness of covering numbers from compactness. We then show that the MPNN maps are Lipschitz with respect to these pseudo-metrics, completing the verification of Definition 1 for the two aggregation schemes.

**Roadmap**   We proceed in three steps:

1. In Appendix K.1, we define the induced IDM maps and the corresponding pseudo-metrics on $V_1(\mathcal{G}_{r,p_0})$.

2. In Appendix K.2, we prove Lipschitz continuity of the normalized-sum and mean MPNNs with respect to the induced pseudo-metrics (via the equivalent IDM formulations).

3. In Appendix K.3, we state the main conclusion (Theorem 5 in the main paper) and explain how it follows immediately from the Lipschitz bounds and compactness/covering arguments.

**Normalized-sum and mean-aggregation MPNNs.**   We recall the normalized-sum and mean-aggregation schemes defined in Equation (21) and Equation (22). Fix $L \in \mathbb{N}$. For $(G, u) \in V_1(\mathcal{G}_{r,p_0})$, set $h_u^{(0)} = \overline{h}_u^{(0)} := x_u$ and, for $t \in [L]$, define

$$h_u^{(t)} = \phi_t\left(h_u^{(t-1)}, \frac{1}{|V(G)|} \sum_{v \in N(u)} w_{uv} h_v^{(t-1)}\right)$$

$$\overline{h}_u^{(t)} = \phi_t\left(\overline{h}_u^{(t-1)}, \frac{1}{\deg(u)} \sum_{v \in N(u)} w_{uv} \overline{h}_v^{(t-1)}\right)$$

where $\deg(u) := \sum_{v \in N(u)} w_{uv}$. For each $t \in [L]$, we assume that the update map $\phi_t : \mathbb{R}^{p_{t-1}} \times \mathbb{R}^{p_{t-1}} \to \mathbb{R}^{p_t}$ is Lipschitz, i.e., there exist constants $C_{\phi,1}^{(t)}, C_{\phi,2}^{(t)} > 0$ such that

$$\|\phi_t(x, y) - \phi_t(x', y')\|_2 \leq C_{\phi,1}^{(t)}\|x - x'\|_2 + C_{\phi,2}^{(t)}\|y - y'\|_2, \qquad \forall x, x', y, y' \in \mathbb{R}^{p_{t-1}}.$$

## K.1. Induced IDMs and pseudo-metrics

In this subsection, we construct mappings from the space $V_1(\mathcal{G}_{r,p_0})$ to the space of iterated degree measures (IDMs) $H_L$. This allows us to endow $V_1(\mathcal{G}_{r,p_0})$ with a pseudo-metric defined as the recursive Kantorovich–Rubinshtein metric on $H_L$, evaluated on the induced elements. Using the compactness result proved in Appendix J (in particular, Theorem 96), we will later deduce finiteness of the covering number of $V_1(\mathcal{G}_{r,p_0})$ with respect to this pseudo-metric. Moreover, we aim to construct these mappings in such a way that message passing neural networks (MPNNs) are Lipschitz continuous with respect to the induced pseudo-metric. For this reason, we introduce different mappings from $V_1(\mathcal{G}_{r,p_0})$ to $H_L$, depending on the aggregation scheme under consideration (normalized-sum or mean). We therefore define below two different induced IDMs, one for each aggregation scheme. For this section, without loss of generality, we assume that for all $G \in \mathcal{G}_{r,p_0}$, the edge weights $w_{uv} \in E(0,1]$ for all $\{u, v\} \in E(G)$ (i.e., $E \subset (0,1]$ compact).

**Normalized-sum induced IDMs**   Let $(G, u) \in V_1(\mathcal{G}_{r,p_0})$. We define the normalized-sum induced IDM of order $t$, denoted by $\eta_u^{(t)}$, recursively as follows. Set

$$\eta_u^{(0)} = x_u,$$

where $x_u$ denotes the initial node features of the node $u \in V(G)$. For $t \geq 0$, we define

$$\eta_u^{(t+1)} = \left(\eta_u^{(t)}, \mu_u^{(t+1)}\right) \in H_t \times M_{t+1} = H_{t+1},$$

where

$$\mu_u^{(t+1)} = \frac{1}{|V(G)|} \sum_{v \in N(u)} w_{uv} \delta_{\eta_v^{(t)}}.$$

Here, $\delta_\eta$ denotes the Dirac measure concentrated at the point $\eta \in H_t$, that is, for any measurable set $A \subset H_t$,

$$\delta_\eta(A) = \begin{cases} 1, & \text{if } \eta \in A, \\ 0, & \text{otherwise.} \end{cases}$$

**Mean induced IDMs** Let $(G, u) \in V_1(\mathcal{G}_{r,p_0})$. We define the mean induced IDM of order $t$, denoted by $\overline{\eta}_u^{(t)}$, recursively as follows. Let

$$\deg(u) = \sum_{v \in N(u)} w_{uv} > 0.$$

The positivity of $\deg(u)$ holds since, by definition, graphs in $\mathcal{G}_{r,p_0}$ do not contain isolated nodes. We set

$$\overline{\eta}_u^{(0)} = a_G(u).$$

For $t \geq 0$, we define

$$\overline{\eta}_u^{(t+1)} = \left(\overline{\eta}_u^{(t)}, \pi_u^{(t+1)}\right) \in H_t \times M_{t+1} = H_{t+1},$$

where

$$\pi_u^{(t+1)} = \frac{1}{\deg(u)} \sum_{v \in N(u)} w_{uv}\, \delta_{\overline{\eta}_v^{(t)}}.$$

**Induced pseudo metric on $V_1(\mathcal{G}_{r,p_0})$** Since $d_{\mathbf{K},L}$ is a well-defined metric on $H_L$, the following pseudo-metrics on $V_1(\mathcal{G}_{r,p_0})$ are well defined for $(G, u), (G', u') \in V_1(\mathcal{G}_{r,p_0})$:

(i) the normalized-sum pseudo-metric

$$d_{\mathrm{sum},L}\big((G, u), (G', u')\big) := d_{\mathbf{K},L}\big(\eta_u^{(L)}, \eta_{u'}^{(L)}\big),$$

(ii) the mean-aggregation pseudo-metric

$$d_{\mathrm{mean},L}\big((G, u), (G', u')\big) := d_{\mathbf{K},L}\big(\overline{\eta}_u^{(L)}, \overline{\eta}_{u'}^{(L)}\big).$$

Below, we derive equivalent expressions of the MPNNs defined in Equation (21) and Equation (22) through their induced IDMs. These formulations will be used later to prove the Lipschitz property with respect to the corresponding pseudo-metrics in Propositions 101 and 102.

**Lemma 99.** *For $(G, u) \in V_1(\mathcal{G}_{r,p_0})$, let $h_u^{(t)}$, and $\overline{h}_u^{(t)}$ be as defined in Equation (21), and Equation (22), respectively. Let $\eta_u^{(t)}$, and $\overline{\eta}_u^{(t)}$, denote the corresponding induced IDMs defined above. Then, for all $t \in \mathbb{N}$, the following identities hold:*

*(i)* $h_u^{(t)} = g_{sum}^{(t)}(\eta_u^{(t)}) := \phi_t\left(g_{sum}^{(t-1)}\big(\eta_u^{(t-1)}\big),\ \int g_{sum}^{(t-1)}(z)\, d\mu_u^{(t)}(z)\right),$

*(ii)* $\overline{h}_u^{(t)} = g_{mean}^{(t)}(\overline{\eta}_u^{(t)}) := \phi_t\left(g_{mean}^{(t-1)}\big(\overline{\eta}_u^{(t-1)}\big),\ \int g_{mean}^{(t-1)}(z)\, d\pi_u^{(t)}(z)\right).$

*where $h_u^{(0)} = g_{sum}^{(0)}(\eta_u^{(0)}) = g_{mean}^{(0)}(\eta_u^{(0)}) := \eta_u^{(0)} = a_G(u)$ for all $(G, u) \in V_1(\mathcal{G}_{r,p_0})$.*

*Proof.* The proof proceeds by induction on $t$. By direct application of the definition of IDMs, we obtain

$$\int g^{(t)}(z)\, d\mu^{(t+1)}(z) = \frac{1}{|V(G)|} \sum_{v \in N(u)} w_{uv} g^{(t)}\Big(\eta_v^{(t)}\Big).$$

The claim then follows by applying the induction hypothesis. □

The following lemma reduces differences of vector-valued integrals to scalar-valued ones, which allows us to apply the Kantorovich–Rubinshtein distance later on.

**Lemma 100.** *Let $(\mathcal{X}, \mathcal{A})$ be a measurable space, let $\mu, \mu'$ be finite measures on $(\mathcal{X}, \mathcal{A})$, and let $G \colon \mathcal{X} \to \mathbb{R}^m$ be integrable with respect to both $\mu$ and $\mu'$. Then*

$$\left\| \int_{\mathcal{X}} G\, d\mu - \int_{\mathcal{X}} G\, d\mu' \right\|_2 = \sup_{\|a\|_2 \leq 1} \left| \int_{\mathcal{X}} \langle a, G(x) \rangle\, d\mu(x) - \int_{\mathcal{X}} \langle a, G(x) \rangle\, d\mu'(x) \right|.$$

*Proof.* Set

$$v := \int_{\mathcal{X}} G \, d\mu - \int_{\mathcal{X}} G \, d\mu' \in \mathbb{R}^m.$$

By the dual characterization of the Euclidean norm,

$$\|v\|_2 = \sup_{\|a\|_2 \leq 1} \langle a, v \rangle = \sup_{\|a\|_2 \leq 1} |\langle a, v \rangle|.$$

Using linearity of the integral and the inner product, for each $a \in \mathbb{R}^m$ we have

$$\langle a, v \rangle = \left\langle a, \int_{\mathcal{X}} G \, d\mu - \int_{\mathcal{X}} G \, d\mu' \right\rangle = \int_{\mathcal{X}} \langle a, G(x) \rangle \, d\mu(x) - \int_{\mathcal{X}} \langle a, G(x) \rangle \, d\mu'(x).$$

Substituting into the previous yields the claimed identity. $\qquad\square$

### K.2. Lipschitzness of sum/mean MPNNs

We now establish Lipschitz continuity of the normalized-sum and mean MPNN maps with respect to the induced pseudo-metrics. This is the key analytic step that connects the recursive representation space geometry to the stability of message passing.

**Proposition 101** (Lipschitzness of normalized-sum MPNNs)**.** *Fix $L \in \mathbb{N}$ and consider the normalized-sum MPNN in Equation* (21)*. Let $g_{sum}^{(t)} \colon H_t \to \mathbb{R}^{p_t}$ be defined as in Lemma 99, i.e., $h_u^{(t)} = g_{sum}^{(t)}(\eta_u^{(t)})$ for all $(G, u) \in V_1(\mathcal{G}_{r,p_0})$. Then for each $t \in [L]$, the map $g^{(t)}$ is Lipschitz with respect to $d_{\mathbf{K},t}$. In particular, the function $f_{sum} \colon V_1(\mathcal{G}_{r,p_0}) \to \mathbb{R}^{p_L}$ defined by $f_{sum}(G, u) := h_u^{(L)}$ is Lipschitz with respect to $d_{sum,L}$, i.e., there exists $C_{sum,L} > 0$ such that*

$$\|f_{sum}(G, u) - f_{sum}(G', u')\|_2 \leq C_{sum,L} \, d_{sum,L}\big((G, u), (G', u')\big), \quad \forall (G, u), (G', u') \in V_1(\mathcal{G}_{r,p_0}).$$

*Proof.* We prove by induction on $t$ that there exist constants $C_t, b_t > 0$ such that, for all $x, x' \in H_t$,

$$\|g_{sum}^{(t)}(x) - g_{sum}^{(t)}(x')\|_2 \leq C_t \, d_{\mathbf{K},t}(x, x') \quad \text{and} \quad \sup_{x \in H_t} \|g^{(t)}(x)\|_2 \leq B_t.$$

We have $H_0 = B_{r,p_0}$, $g_{sum}^{(0)}(\eta) = \eta$, and $d_{\mathbf{K},0}(\eta, \eta') = \|\eta - \eta'\|_2$. Hence $C_0 = 1$ and $B_0 = r$.

Assume the claim holds for $t-1$, with constants $C_{t-1}$ and $M_{t-1}$. Let $x = (\eta, \mu)$ and $x' = (\eta', \mu')$ in $H_t$, where $\eta, \eta' \in H_{t-1}$ and $\mu, \mu' \in M_t$. By Lemma 99,

$$g_{sum}^{(t)}(x) = \phi_t\left(g_{sum}^{(t-1)}(\eta), \int g_{sum}^{(t-1)}(z) \, d\mu(z)\right), \qquad g^{(t)}(x') = \phi_t\left(g_{sum}^{(t-1)}(\eta'), \int g_{sum}^{(t-1)}(z) \, d\mu'(z)\right).$$

Using the assumed Lipschitz property of $\phi_t$, we obtain

$$\|g_{sum}^{(t)}(x) - g_{sum}^{(t)}(x')\|_2 \leq C_{\phi,1}^{(t)} \|g_{sum}^{(t-1)}(\eta) - g_{sum}^{(t-1)}(\eta')\|_2 + C_{\phi,2}^{(t)} \left\| \int g_{sum}^{(t-1)} \, d\mu - \int g_{sum}^{(t-1)} \, d\mu' \right\|_2.$$

**Bounding the integral term** Let $G \colon H_{t-1} \to \mathbb{R}^{p_{t-1}}$ denote $g_{sum}^{(t-1)}$. For any unit vector $a \in \mathbb{R}^{p_{t-1}}$ with $\|a\|_2 \leq 1$, define the scalar function $f_a \colon H_{t-1} \to \mathbb{R}$ as $f_a(z) := \langle a, G(z) \rangle$. Then

$$\|f_a\|_\infty \leq B_{t-1}, \qquad \mathrm{Lip}(f_a) \leq C_{t-1}.$$

where $\mathrm{Lip}(f_a)$ denotes the smallest Lipschitz constant of $f_a$.

Set $\lambda_{t-1} := B_{t-1} + C_{t-1}$ and define $\widetilde{f}_a := f_a/\lambda_{t-1}$. Then $\|\widetilde{f}_a\|_\infty \leq 1$ and $\mathrm{Lip}(\widetilde{f}_a) \leq 1$, so by definition of $\mathbf{K}$,

$$\left| \int f_a \, d\mu - \int f_a \, d\mu' \right| = \lambda_{t-1} \left| \int \widetilde{f}_a \, d\mu - \int \widetilde{f}_a \, d\mu' \right| \leq \lambda_{t-1} \, \mathbf{K}(\mu, \mu').$$

Taking the supremum over $\|a\|_2 \leq 1$ and using Lemma 100 yields

$$\left\|\int G \, d\mu - \int G \, d\mu'\right\|_2 \leq \lambda_{t-1} \, \mathbf{K}(\mu, \mu').$$

**Lipschitzess of $g^{(t)}$** By the induction hypothesis,

$$\|g_{\text{sum}}^{(t-1)}(\eta) - g_{\text{sum}}^{(t-1)}(\eta')\|_2 \leq C_{t-1} \, d_{\mathbf{K},t-1}(\eta, \eta').$$

Combining with the previous gives

$$\|g_{\text{sum}}^{(t)}(x) - g_{\text{sum}}^{(t)}(x')\|_2 \leq C_{\phi,1}^{(t)} C_{t-1} \, d_{\mathbf{K},t-1}(\eta, \eta') + C_{\phi,2}^{(t)} \lambda_{t-1} \, \mathbf{K}(\mu, \mu').$$

Since $d_{\mathbf{K},t}(x, x') = d_{\mathbf{K},t-1}(\eta, \eta') + \mathbf{K}(\mu, \mu')$, we obtain

$$\|g_{\text{sum}}^{(t)}(x) - g_{\text{sum}}^{(t)}(x')\|_2 \leq C_t \, d_{\mathbf{K},t}(x, x'), \qquad C_t := \max\left\{C_{\phi,1}^{(t)} C_{t-1}, \; C_{\phi,2}^{(t)}(B_{t-1} + C_{t-1})\right\}.$$

**Boundedness of $g^{(t)}$** Let $x = (\eta, \mu) \in H_t$. Using the Lipschitz bound for $\phi_t$ and the offset assumption,

$$\|g^{(t)}(x)\|_2 \leq \|\phi_t(0,0)\|_2 + C_{\phi,1}^{(t)} \|g^{(t-1)}(\eta)\|_2 + C_{\phi,2}^{(t)} \left\|\int g^{(t-1)}(z) \, d\mu(z)\right\|_2.$$

Since $\mu$ has total mass at most 1 in the normalized-sum case, we have

$$\left\|\int g^{(t-1)}(z) \, d\mu(z)\right\|_2 \leq \int \|g^{(t-1)}(z)\|_2 \, d\mu(z) \leq B_{t-1} \, \mu(H_{t-1}) \leq B_{t-1}.$$

Therefore,

$$\|g^{(t)}(x)\|_2 \leq B_t + (C_{\phi,1}^{(t)} + C_{\phi,2}^{(t)}) B_{t-1},$$

so it suffices to take

$$B_t := B_t^{(\phi)} + (C_{\phi,1}^{(t)} + C_{\phi,2}^{(t)}) B_{t-1}.$$

This completes the induction.

Overall, we have that for $(G, u), (G', u') \in V_1(\mathcal{G}_{r,p_0})$, we have

$$\|h_u^{(L)} - h_{u'}^{(L)}\|_2 = \|g^{(L)}(\eta_u^{(L)}) - g^{(L)}(\eta_{u'}^{(L)})\|_2 \leq C_L \, d_{\mathbf{K},L}(\eta_u^{(L)}, \eta_{u'}^{(L)}) = C_L \, d_{\text{sum},L}((G, u), (G', u')).$$

Thus $f_{\text{sum}}$ is Lipschitz with Lipschitz constant $C_{\text{sum},L} := C_L$. $\qquad\square$

**Proposition 102** (Lipschitzness of mean-aggregation MPNNs). *Fix $L \in \mathbb{N}$ and consider the mean-aggregation MPNN in Equation (22). For each $t \in \{0, \ldots, L\}$, let*

$$H_t^{\mathcal{G}} := \left\{\overline{\eta}_u^{(t)} \;\middle|\; (G, u) \in \mathcal{G}_{r,p_0} \otimes \mathcal{V}\right\} \subseteq H_t$$

*denote the set of mean-induced IDMs of order $t$. Let $\overline{g}^{(t)} \colon H_t^{\mathcal{G}} \to \mathbb{R}^{p_t}$ be defined as in Lemma 99, i.e., $\overline{h}_u^{(t)} = \overline{g}^{(t)}(\overline{\eta}_u^{(t)})$ for all $(G, u) \in V_1(\mathcal{G}_{r,p_0})$. Then, for each $t \in [L]$, the map $\overline{g}^{(t)}$ is Lipschitz with respect to $d_{\mathbf{K},t}$ restricted to $H_t^{\mathcal{G}}$. In particular, the function $f_{\text{mean}} \colon V_1(\mathcal{G}_{r,p_0}) \to \mathbb{R}^{p_L}$ defined by $f_{\text{mean}}(G, u) := \overline{h}_u^{(L)}$ is Lipschitz with respect to $d_{\text{mean},L}$, i.e., there exists $C_{\text{mean},L} > 0$ such that*

$$\|f_{\text{mean}}(G, u) - f_{\text{mean}}(G', u')\|_2 \leq C_{\text{mean},L} \, d_{\text{mean},L}((G, u), (G', u')), \quad \forall (G, u), (G', u') \in V_1(\mathcal{G}_{r,p_0}).$$

*Proof.* The proof follows the same argument as in Proposition 101, with the only changes that we work on the restricted domain $H_t^{\mathcal{G}}$, so that mean-aggregation is well defined, and we replace the induced measures $\mu_u^{(t)}$ by $\pi_u^{(t)}$. $\qquad\square$

### K.3. Conclusion: finite Lipschitz classes for sum/mean aggregation

We now state the main consequence for normalized-sum and mean aggregation, corresponding to Theorem 5 in the main paper, and explain how it follows directly from the Lipschitz continuity established above together with compactness (hence total boundedness) of the representation space $(H_L, d_{\mathbf{K},L})$.

**Theorem 103** (Theorem 3 (sum/mean aggregation) in the main text). *Fix $L \in \mathbb{N}$. Consider the hypothesis class induced by the normalized-sum MPNN Equation* (21) *(respectively, the mean-aggregation MPNN Equation* (22)*) with Lipschitz update maps $\{\phi_t\}_{t \in [L]}$ as assumed in the main text. Endow $V_1(\mathcal{G}_{r,p_0})$ with the pseudo-metric $d_{\mathrm{sum},L}$ (respectively, $d_{\mathrm{mean},L}$). Then the resulting hypothesis class is a finite Lipschitz class in the sense of Definition* 1.

*Proof.* By Proposition 101 (respectively, Proposition 102), the hypothesis map $f_{\mathrm{sum}}(G, u) = h_u^{(L)}$ (respectively $f_{\mathrm{mean}}(G, u) = \overline{h}_u^{(L)}$) is Lipschitz with respect to the induced pseudo-metric on $V_1(\mathcal{G}_{r,p_0})$. By Theorem 96, the metric space $(H_L, d_{\mathbf{K},L})$ is compact, and therefore has a finite covering number for every radius $\varepsilon > 0$. Since $d_{\mathrm{sum},L}$ and $d_{\mathrm{mean},L}$ are defined as pullbacks of $d_{\mathbf{K},L}$ through the induced IDM maps, $\big(V_1(\mathcal{G}_{r,p_0}), d_{\mathrm{sum},L}\big)$ and $\big(V_1(\mathcal{G}_{r,p_0}), d_{\mathrm{mean},L}\big)$ admit finite $\varepsilon$-covers for every $\varepsilon > 0$ as well. $\qquad\square$

**Remark** In the above result, the Lipschitz constant $M_{\boldsymbol{\theta}}$ appearing in Definition 1 depends only on the Lipschitz constants $\{C_{\phi,1}^{(t)}, C_{\phi,2}^{(t)}\}_{t \in [L]}$ of the update maps $\{\phi_t\}_{t \in [L]}$ and on the number of layers $L$, and is independent of the size or structure of the input graph. In particular, in the special case

$$\mathcal{F}_{\Theta} := \Big\{ f: V_1(\mathcal{G}_{r,p_0}) \to \mathbb{R}^{p_L} \ \Big| \ f(G, u) = h_u^{(L)}, \ \phi_t(x, y) = \sigma\Big(\boldsymbol{W}_1^{(t)} x + \boldsymbol{W}_2^{(t)} y\Big),$$
$$\boldsymbol{W}_1^{(t)}, \boldsymbol{W}_2^{(t)} \in \mathbb{R}^{p_t \times p_{t-1}}, \ t \in [L] \Big\}.$$

where $\sigma$ is the ReLu function applied elementwise, the Lipschitz constant $M_{\boldsymbol{\theta}}$ depends only on the operator 2-norms $\|\boldsymbol{W}_1^{(t)}\|_2$ and $\|\boldsymbol{W}_2^{(t)}\|_2$ of the weight matrices and on $L$.

## L. Proof of Theorem 3 for max/min-aggregation

In this section, we prove Theorem 3 for max-aggregation MPNNs. We work in the slightly more general setting where edge weights are vectors in a compact set $E \subset \mathbb{R}^{d_e}$, for $d_e \in \mathbb{N}$; the result In the main paper, it corresponds to the special case $d_e = 1$. As in the sum/mean case, the proof proceeds by constructing a canonical representation space endowed with a compact metric and showing that the MPNN maps are Lipschitz with respect to the induced pseudo-metric on $V_1(\mathcal{G}_{r,p_0})$.

**Roadmap**

The argument follows the same high-level structure as for sum/mean aggregation:

1. We introduce induced *iterated neighborhood-set* objects that encode rooted graph neighborhoods recursively.

2. We define an induced pseudo-metric on $V_1(\mathcal{G}_{r,p_0})$ via a recursive Hausdorff metric.

3. We show that max-aggregation MPNNs are Lipschitz with respect to this pseudo-metric.

4. We conclude by combining Lipschitzness with compactness to verify Definition 1.

**Max-aggregation MPNNs** We recall the max-aggregation scheme defined in Equation (23). Fix $L \in \mathbb{N}$. For $(G, u) \in V_1(\mathcal{G}_{r,p_0})$, define $\hat{h}_u^{(0)} := x_u$ and, for $t \in [L]$,

$$\hat{h}_u^{(t)} = \phi_t\Big(\hat{h}_u^{(t-1)}, \max_{v \in N(u)} M_t\Big(\hat{h}_v^{(t-1)}, w_{uv}\Big)\Big), \tag{24}$$

where $M_t : \mathbb{R}^{p_{t-1}} \times \mathbb{R}^{d_e} \to \mathbb{R}^{p_{t-1}}$ is a message map and $\phi_t : \mathbb{R}^{p_{t-1}} \times \mathbb{R}^{p_{t-1}} \to \mathbb{R}^{p_t}$ is an update map. The maximum is taken coordinatewise in $\mathbb{R}^{p_{t-1}}$. For each $t \in [L]$, assume that $\phi_t$ and $M_t$ are Lipschitz, i.e., there exist constants $C_{\phi,1}^{(t)}, C_{\phi,2}^{(t)}, C_{M,1}^{(t)}, C_{M,2}^{(t)} > 0$ such that

$$\|\phi_t(x, y) - \phi_t(x', y')\|_2 \le C_{\phi,1}^{(t)} \|x - x'\|_2 + C_{\phi,2}^{(t)} \|y - y'\|_2,$$

and

$$\|M_t(x, y) - M_t(x', y')\|_2 \le C_{M,1}^{(t)}\|x - x'\|_2 + C_{M,2}^{(t)}\|y - y'\|_2.$$

### L.1. Induced iterated neighborhood-set objects

Here we represent neighborhoods as sets of feature–edge pairs. This leads to a recursive representation in terms of iterated neighborhood-set objects equipped with Hausdorff metrics. Let $(G, u) \in V_1(\mathcal{G}_{r,p_0})$. Define $\eta_u^{\max,(t)} \in H_t^{\max}$ recursively by $\eta_u^{\max,(0)} := x_u \in H_0^{\max}$ and, for $t \ge 0$,

$$\eta_u^{\max,(t+1)} := \left(\eta_u^{\max,(t)}, A_u^{(t+1)}\right) \in H_{t+1}^{\max}, \qquad A_u^{(t+1)} := \left\{(\eta_v^{\max,(t)}, w_{uv}) : v \in N(u)\right\}.$$

Nonemptiness of $A_u^{(t+1)}$ follows since graphs in $\mathcal{G}_{r,p_0}$ have no isolated nodes.

Let

$$H_t^{\mathcal{G},\max} := \left\{\eta_u^{\max,(t)} \mid (G, u) \in V_1(\mathcal{G}_{r,p_0})\right\} \subseteq H_t^{\max}$$

denote the collection of neighborhood-set objects induced by the graph-node pairs. We define maps $g_{\max}^{(t)} : H_t^{\mathcal{G},\max} \to \mathbb{R}^{p_t}$ recursively by $g_{\max}^{(0)}(x) = x$ and, for $t \in [L]$,

$$g_{\max}^{(t)}(\eta, A) = \phi_t\left(g_{\max}^{(t-1)}(\eta), \max_{(z,w)\in A} \Gamma_t(z, w)\right), \qquad \Gamma_t(z, w) := M_t(g_{\max}^{(t-1)}(z), w),$$

where the maximum is taken coordinatewise in $\mathbb{R}^{p_{t-1}}$.

**Lemma 104** (Equivalent formulation for max-aggregation). *For all* $(G, u) \in V_1(\mathcal{G}_{r,p_0})$ *and all* $t$,

$$\hat{h}_u^{(t)} = g_{max}^{(t)}\left(\eta_u^{max,(t)}\right).$$

*Proof.* We prove by induction on $t$.

For $t = 0$, $\eta_u^{\max,(0)} = x_u$ and $\hat{h}_u^{(0)} = x_u$, so define $g_{\max}^{(0)}(\eta) := \eta$. Assume the claim holds at depth $t - 1$. Fix $(G, u)$. By construction,

$$\eta_u^{\max,(t)} = \left(\eta_u^{\max,(t-1)}, A_u^{(t)}\right), \qquad A_u^{(t)} = \{(\eta_v^{\max,(t-1)}, w_{uv}) : v \in N(u)\}.$$

Using (24) and the induction hypothesis $h_v^{(t-1)} = g_{\max}^{(t-1)}(\eta_v^{\max,(t-1)})$,

$$\hat{h}_u^{(t)} = \phi_t\left(g_{\max}^{(t-1)}(\eta_u^{\max,(t-1)}), \max_{v\in N(u)} M_t\left(g_{\max}^{(t-1)}(\eta_v^{\max,(t-1)}), w_{uv}\right)\right).$$

Since $A_u^{(t)}$ is precisely the set of pairs $(\eta_v^{\max,(t-1)}, w_{uv})$, the maximum equals $\max_{(z,e)\in A_u^{(t)}} \Gamma_t(z, e)$ with $\Gamma_t(z, e) := M_t(g_{\max}^{(t-1)}(z), e)$. Define $g_{\max}^{(t)}$ on induced elements by the stated recursion; then $h_u^{(t)} = g_{\max}^{(t)}(\eta_u^{\max,(t)})$, completing the induction. $\square$

### L.2. Induced pseudo-metric

Since $(H_L^{\max}, d_{\mathbf{H},L})$ is a compact metric space (see Theorem 98), we may pull back its metric to obtain a pseudo-metric on rooted graphs. Define

$$d_{\max,L}\left((G, u), (G', u')\right) := d_{\mathbf{H},L}\left(\eta_u^{\max,(L)}, \eta_{u'}^{\max,(L)}\right), \qquad (G, u), (G', u') \in V_1(\mathcal{G}_{r,p_0}). \tag{25}$$

### L.3. Lipschitzness of max-aggregation MPNNs

The next lemma shows that the coordinatewise maximum over a compact set is stable under perturbations measured by the Hausdorff distance.

**Lemma 105.** *Let* $(\mathcal{X}, d)$ *be a metric space and let* $f : \mathcal{X} \to \mathbb{R}$ *be Lipschitz with constant* $L_f$. *Define*

$$F(A) := \max_{x\in A} f(x), \qquad A \in \text{Haus}(\mathcal{X}).$$

*Then* $F$ *is* $L_f$*-Lipschitz with respect to* $\mathbf{H}_d$.

*Proof.* Fix $A, B \in \text{Haus}(\mathcal{X})$. Let $x \in A$ be such that $f(x) = \max_{a \in A} f(a)$. By definition of the Hausdorff distance $\mathbf{H}_d(A, B)$, there exists $y \in B$ such that

$$d(x, y) \leq \mathbf{H}_d(A, B).$$

Then

$$F(A) - F(B) = \max_{a \in A} f(a) - \max_{b \in B} f(b) \leq f(x) - f(y) \leq L_f \, d(x, y) \leq L_f \, \mathbf{H}_d(A, B).$$

Similarly, for the difference $F(B) - F(A)$. $\qquad\qquad\square$

We are now ready to establish the Lipschitz continuity of max-aggregation MPNNs with respect to the induced pseudo-metric.

**Proposition 106** (Lipschitzness of max-aggregation MPNNs)**.** *Fix $L \in \mathbb{N}$ and consider the max-aggregation MPNN defined in Equation* (24)*. Then for each $t \in [L]$, the function $f_{max} : V_1(\mathcal{G}_{r,p_0}) \to \mathbb{R}^{p_L}$, $f_{max}(G, u) := \hat{h}_u^{(L)}$, is Lipschitz with respect to the induced pseudo-metric $d_{max,L}$ (Equation* (25)*).*

*Proof.* We prove by induction on $t$ that there exist constants $C_t, B_t > 0$ such that for all $x, x' \in H_t^{\mathcal{G},\max}$,

$$\|g_{\max}^{(t)}(x) - g_{\max}^{(t)}(x')\|_2 \leq C_t \, d_{\mathbf{H},t}(x, x'),$$

then the result follows directly by Lemma 104.

For $t = 0$, $g_{\max}^{(0)}(\eta) = \eta$, hence $C_0 = 1$. Assume the claim holds for $t - 1$. Let $x = (\eta, A)$ and $x' = (\eta', A')$ in $H_t^{\mathcal{G},\max}$. By Lemma 104,

$$g_{\max}^{(t)}(x) = \phi_t\Big(g_{\max}^{(t-1)}(\eta), \ \max_{(z,e) \in A} \Gamma_t(z, e)\Big), \qquad \Gamma_t(z, e) = M_t(g_{\max}^{(t-1)}(z), e),$$

and similarly for $x'$.

Using the Lipschitz property of $\phi_t$,

$$\|g_{\max}^{(t)}(x) - g_{\max}^{(t)}(x')\|_2 \leq C_{\phi,1}^{(t)} \|g_{\max}^{(t-1)}(\eta) - g_{\max}^{(t-1)}(\eta')\|_2 + C_{\phi,2}^{(t)} \|\max_A \Gamma_t - \max_{A'} \Gamma_t\|_2,$$

where $\max_A \Gamma_t$ abbreviates $\max_{(z,e) \in A} \Gamma_t(z, e)$.

The first term is bounded by $C_{\phi,1}^{(t)} C_{t-1} \, d_{\max,t-1}(\eta, \eta')$ by the induction hypothesis.

For the max term, fix a unit direction $a \in \mathbb{R}^{p_{t-1}}$ with $\|a\|_2 \leq 1$ and consider the scalar functional $f_a(z, e) := \langle a, \Gamma_t(z, e) \rangle$ on $Y_{t-1}^{\max} = H_{t-1}^{\max} \times E$. By Lipschitzness of $M_t$ and the induction hypothesis, $f_a$ is Lipschitz on $(Y_{t-1}^{\max}, d_{Y,t-1})$ with constant at most $C_M^{(t)} C_{t-1}$, uniformly in $\|a\|_2 \leq 1$, $C_M^{(t)} := \max\{C_{M,1}^{(t)}, C_{M,2}^{(t)}\}$ Applying Lemma 105 on $(\text{Haus}(Y_{t-1}^{\max}), \mathbf{H}_{t-1})$ yields

$$\Big|\max_{(z,e) \in A} f_a(z, e) - \max_{(z,e) \in A'} f_a(z, e)\Big| \leq C_M^{(t)} C_{t-1} \, \mathbf{H}_{t-1}(A, A').$$

Taking the supremum over $\|a\|_2 \leq 1$ and using an argument identical to Lemma 100, gives

$$\|\max_A \Gamma_t - \max_{A'} \Gamma_t\|_2 \leq C_M^{(t)} C_{t-1} \, \mathbf{H}_{t-1}(A, A').$$

Combining the bounds and recalling $d_{\mathbf{H},t}(x, x') = d_{\mathbf{H},t-1}(\eta, \eta') + \mathbf{H}_{t-1}(A, A')$ yields a recursion of the form

$$\|g_{\max}^{(t)}(x) - g_{\max}^{(t)}(x')\|_2 \leq C_t \, d_{\mathbf{H},t}(x, x'), \quad C_t := \max\Big\{ C_{\phi,1}^{(t)} C_{t-1}, \ C_{\phi,2}^{(t)} C_M^{(t)} C_{t-1} \Big\}.$$

$\qquad\qquad\square$

### L.4. Conclusion: finite Lipschitz class for max aggregation

We now state the final consequence for max aggregation, completing the proof of Theorem 3.

**Theorem 107** (Theorem 3 (max/min aggregation) in the main text)**.** *Fix $L \in \mathbb{N}$. Consider the hypothesis class induced by the max-aggregation MPNN Equation* (24)*. Endow $V_1(\mathcal{G}_{r,p_0})$ with the pseudo-metric $d_{max,L}$. Then the resulting hypothesis class is a finite Lipschitz class in the sense of Definition 1.*

*Proof.* By Proposition 106, the hypothesis map $f_{\max}(G, u) = \hat{h}_u^{(L)}$ is Lipschitz with respect to $d_{\max,L}$. By Theorem 98, $(H_L^{\max}, d_{\mathbf{H},L})$ is compact which implies finite covering numbers for every radius $\varepsilon$. Since $d_{\max,L}$ is obtained by pulling back $d_{\mathbf{H},L}$ through the induced neighborhood-set map, the space $V_1(\mathcal{G}_{r,p_0})$ endowed with $d_{\max,L}$ also admits finite $\varepsilon$-covers for all $\varepsilon > 0$. This verifies Definition 1. $\qquad\square$

**Remark.** The Lipschitz constant $M_{\boldsymbol{\theta}}$ in Definition 1 depends only on the Lipschitz constants of the message maps $\{M_t\}_{t\in[L]}$, the update maps $\{\phi_t\}_{t\in[L]}$, and on the number of layers $L$, and is independent of the size or structure of the input graph. In particular, in the special case

$$
\begin{aligned}
\mathcal{F}_{\Theta} := \Big\{ f\colon V_1(\mathcal{G}_{r,p_0}) \to \mathbb{R}^{p_L} \; \Big| \; & f(G, u) = h_u^{(L)}, \; \phi_t(x, y) = \sigma\Big(\boldsymbol{W}_1^{(t)} x + \boldsymbol{W}_2^{(t)} y\Big), \\
& M_t(z, w) = \sigma\Big(\boldsymbol{U}_1^{(t)} z + \boldsymbol{U}_2^{(t)} w\Big), \\
& \boldsymbol{W}_1^{(t)}, \boldsymbol{W}_2^{(t)} \in \mathbb{R}^{p_t \times p_{t-1}}, \; \boldsymbol{U}_1^{(t)} \in \mathbb{R}^{q_t \times p_{t-1}}, \; \boldsymbol{U}_2^{(t)} \in \mathbb{R}^{q_t \times e}, \; t \in [L] \Big\},
\end{aligned}
$$

where $\sigma$ is the ReLu function applied elementwise, the Lipschitz constant $M_{\boldsymbol{\theta}}$ depends only on the 2-norms $\|\boldsymbol{W}_1^{(t)}\|_2$, $\|\boldsymbol{W}_2^{(t)}\|_2$, $\|\boldsymbol{U}_1^{(t)}\|_2$, and $\|\boldsymbol{U}_2^{(t)}\|_2$ of the weight matrices and on the depth $L$.

The same conclusions hold for min-aggregation, since $\min(-\cdot) = -\max(-\cdot)$ and negation preserves all Lipschitz properties.

## M. Iterated degree measures as computation trees

Computation trees or unrolling trees (see Appendix H.2) provide a convenient way to understand how message passing neural networks (MPNNs) propagate and aggregate information across a graph. Starting from a root node, a computation tree records its neighbors, the neighbors of those neighbors, and so on, up to a prescribed depth. Since MPNNs update node representations by repeatedly aggregating information from local neighborhoods, such trees encode precisely the information required to compute node features layer by layer.

Iterated degree measures (IDMs) can be viewed as a measure-theoretic analog of these computation trees. Instead of explicitly storing finite sets of neighbors, IDMs represent neighborhoods as probability measures and neighborhoods of neighborhoods as measures over measures, recursively. This abstraction is particularly natural in the setting of graphons, which are continuous objects extending finite graphs. Formally, a graphon is a measurable function $W\colon [0,1]^2 \to [0,1]$, which can be interpreted as a weighted graph with an uncountable node set $[0,1]$, where the edge weight between two nodes $x, y \in [0,1]$ is given by $W(x, y)$.

In a graphon, each node has infinitely many neighbors, and its neighborhood cannot be described by a finite multiset. Instead, the local structure around a node is naturally captured by a measure encoding the distribution of its neighbors and their attributes. Since edge weights are integrable, these neighborhood measures have total mass at most one. Iterating this construction—taking measures of neighborhood measures—yields exactly the hierarchy of spaces $\{M_\ell\}_{\ell=0}^L$ and $\{H_\ell\}_{\ell=0}^L$ introduced above. In this sense, IDMs serve as computation trees for graphons, providing a compact, recursive representation of increasingly deep neighborhood information. This perspective aligns with recent work connecting graph neural networks applied on graphons, and measure-based representations of local structure (see, e.g., (Grebik & Rocha, 2022)). See also (Böker et al., 2023), for a definition of the 1-WL algorithm applied on graphons based on the above analysis.

## N. Experimental study

In the following, we outline details related to the experimental evaluation of **Q1** to **Q3** in Section 5. In addition, we provide further results on size generalization and training set construction. The source code for all experiments is available in the supplementary material.

**Dataset creation**    To investigate **Q1**, we construct multiple training and test datasets. Throughout the experiments, we consider two training sets of graphs. Building on theoretical results on minimal training samples for the SSSP problem, we construct a training set of the minimum number of path graphs, as outlined in Section 3. This results in a set of three graphs for a Bellman–Ford problem with two steps. In addition, we construct an extended training set to help with training in our

*Table 3.* Parameters for generation of Erdős–Reyni and stochastic block model (SBM) graphs in **Q1** to **Q3**.

| Dataset/Graphs | n | p | weight |
|---|---|---|---|
| ER-constdeg | 64-1024 | 6.4/n | Uniform(1,100) |
| ER | 64-1024 | 0.1 | Uniform(1,100) |
| Star graphs | 64-1024 | - | Uniform(1,100) |
| Complete graphs | 64-1024 | - | Uniform(1,100) |
| SBM | 64-1024 | $\begin{bmatrix} 0.7 & 0.05 & 0.02 \\ 0.05 & 0.6 & 0.03 \\ 0.02 & 0.03 & 0.4 \end{bmatrix}$ | Uniform(1,100) |

MPNN setting. For this, we consider additional path graphs constructed in the same way as for the minimal training set with edge weight $x$, but with edges scaled by $0.5$ and $2$. Moreover, we add a special version of the path graph, which includes multiple paths to the furthest reachable node. In all training datasets, the initial Bellman–Ford state and edge weights are given. Moreover, the initial starting node is marked with $0$ unless otherwise specified. For all experiments, we set the edge weight $x = 50$ and the value indicating unvisited nodes to $1000$. In addition, all graphs in both the training and test datasets have a self-loop edge for each node.

As the simplest case of size generalization, we provide a test set, which we call ER-constdeg, of 200 randomly generated Erdős–Rényi graphs with an average node degree of $6.4$. We fix the average node degree and increase the number of nodes, generating one dataset for each from 64 to 1024 nodes. Furthermore, we provide a second dataset generated from Erdös–Reyni graphs, but with an unbounded average degree. For this, we set $p = 0.1$ as the edge probability in the graph generator provided by NetworkX (Hagberg et al., 2008). We call this dataset ER. Again, we provide a test dataset for 64 to 1024 nodes each. As the most general case of the test set, we provide a set of 50 graphs, each consisting of Erdös–Reyni graphs with unbounded degree, stochastic block model graphs with probability matrices outlined in Table 3, complete graphs, as well as star graphs and path graphs as shown in Figure 4. This dataset is called General. Across all test datasets, the weight distribution is uniform on $[1,100]$ for all graphs.

For **Q2** and **Q3**, we consider the same training and test sets as in **Q1**. However, we restrict our evaluation to a subset of the test datasets, since **Q1** showed that the General and ER test sets are sufficient for evaluating generalization capabilities.

**Neural architectures**  The MPNN architecture discussed in Section 3 consists of update and aggregation functions that map node and edge features to an intermediate representation. The ReLU activation function is used, except for the last layer, as this resulted in unstable training behavior. Moreover, all functions in each layer are implemented using a two-layer FNN with the configurations outlined in the following paragraph.

**Hyperparameters and hardware**  Across all experiments, we trained the MPNN architectures for 160000 steps at a learning rate of $0.001$. Table 4 showcases tuned parameters with selected parameters highlighted. A constant learning rate was used without a specific scheduler. Furthermore, the Adam optimizer (Kingma & Ba, 2015) was used across all experiments.

For size generalization and regularization experiments in **Q1** and **Q3**, we used a two-layer MPNN as outlined in Section 3. For all experiments, we used the design outlined in the theory of Section 3. In addition, we set the update and aggregation functions to two-layer MLPs with a hidden dimension of 64. The first aggregation FNN uses minimum aggregation with an output dimension of 16, while the second layer reduces it to 1. All layers are randomly initialized via the uniform initialization provided by PyTorch (Paszke et al., 2019).

Furthermore, we report the runtime and memory usage of our experiments in Table 5. We provide a PyTorch Geometric implementation for each model. All our experiments were executed on a system with 12 CPU cores, an Nvidia L40 GPU, and 120GB of memory.

**Experimental protocol**  In all experiments, we use the Bellman–Ford state of the nodes and the edge weights from the graph construction as input to our model. Furthermore, the target is given by the result obtained from Bellman-Ford after $K$ additional steps from the starting iteration. To calculate the training loss, a combined loss consisting of an $\ell_1$-loss $L_{\text{emp}}$ and a regularization term $L_{\text{reg}}$ is given:

$$\mathcal{L} = \mathcal{L}_{\text{emp}} + \eta \mathcal{L}_{\text{reg}}.$$

*Table 4.* Hyperparameter selection for each experiment in **Q1** to **Q3**. Selected hyperparameters are highlighted, with the tuned ones shown in brackets.

| Hyperparameter | Q1 | Q2 | Q3 |
|---|---|---|---|
| Learning rate | $\{0.0001, \mathbf{0.001}, 0.01\}$ | $\{0.0001, \mathbf{0.001}, 0.01\}$ | $\{0.0001, \mathbf{0.001}, 0.01\}$ |
| Weight decay | 0 | 0 | 0 |
| Optimzier | Adam | Adam | Adam |
| Number of steps | 160000 | 160000 | 160000 |
| Batch size | 1 | 1 | 1 |
| Edge weight (x) | 50 | 50 | 50 |
| Initial node value | 0 | 0 | 0 |
| Hidden dim. | $\{32, \mathbf{64}, 128\}$ | $\{32, \mathbf{64}, 128\}$ | $\{32, \mathbf{64}, 128\}$ |
| Number of MPNN layers | 2 | 2 | 2 |
| Number of MLP layers | 2 | 2 | 2 |
| 1st layer output dim. | $\{8, \mathbf{16}, 32\}$ | $\{8, \mathbf{16}, 32\}$ | $\{8, \mathbf{16}, 32\}$ |
| 2nd layer output dim. | 1 | 1 | 1 |

*Table 5.* Runtime and Memory Usage for each experiment in Section 5. The first value denotes the runtime in minutes (m) and seconds (s) of each experiment, and the second value denotes the used VRAM in MB. All results were obtained on a single computing node with an Nvidia L40 GPU and 120GB of RAM. For each experiment, the longest runtime was considered, obtained from test datasets with 1024 nodes.

| | Dataset | | |
|---|---|---|---|
| Task | ER-CONSTDEG | ER | GENERAL |
| Q1 | 18m48s/606.33 | 27m20s/881.06 | 25m3s/891.55 |
| Q2 | 12m3s/573.33 | 14m17s/571.28 | 24m44s/887.95 |
| Q3 $L_1$ | -/- | -/- | 23m37s/887.94 |
| Q3 $L_2$ | -/- | -/- | 24m16s/887.95 |

Throughout **Q1**, this loss is used to train the model with the regularization outlined in Section 3. For $\eta$, a value of $0\,1$ is used across experiments. The test score, however, is computed slightly differently. Given the precision $h_v$ of a node value after $K$ additional Bellman-Ford steps and the underlying ground truth $x_v$ the test score is computed as follows:

$$L_{\text{test}} = \frac{1}{|G_{\text{test}}|} \sum_{v \in G_{\text{test}}} \frac{|h_v - x_v|}{(x_v + 1)}.$$

Therefore, a lower test score implies better generalization by the model to the unseen test dataset.

For **Q2**, we use slightly modified training and test sets compared to **Q1** and **Q3**. Since the more expressive MPNN architecture from Section 3 requires a marked starting node for Bellman-Ford, we remove this special label from the training and test data and only provide the Bellman–Ford initialization from the execution of the algorithm. Otherwise, the parameters and training/test data remain the same as in **Q1**.

Finally, in **Q3**, we consider the regularization term introduced in Section 3 and whether it improves performance over standard $\ell_1$ or $\ell_2$ regularization. To conduct the experiment, we select the ER and General datasets, as outlined in Appendix N, to evaluate both regularization terms. To provide a fair comparison, both architectures were kept the same as in **Q1** and executed with the same seeds across experiments.

**Additional results** To supplement the results for **Q1**, we present detailed results in Figure 7, highlighting similar test behavior across test graph sizes. Furthermore, we provide an additional training dataset containing both random graphs and the minimum-path graphs for the training outlined in Section 3. In addition to synthetic graphs used throughout **Q1** to **Q3** we apply a two-layer MPNN to real world graphs given by Cora and Citeseer. Results for these experiments are outlined in Table 8, indicating similar performance to results on synthetic graphs.

Moreover, we provide distributions of the weight matrices for **Q1**. We omit showing bias value distributions, as they converge to 0 across all FNN layers. In addition, the weight matrices for update MLP layers contain singular non-zero values, with most entries remaining zero during training. Similar results can be observed for the aggregation FNN layers. Furthermore, the

*Table 6.* Additional results for the ER-constdeg and ER dataset used for size generalization in **Q1**. Results are outlined for the given MPNN from **Q1**, with the test dataset changed accordingly. A list of hyperparameters can be found under **Q1** in Table 4.

| | Nodes | | | | |
|---|---|---|---|---|---|
| **Test Set** (Test score ↓) | 64 | 128 | 256 | 512 | 1024 |
| ER-constdeg | $0.0035 \pm 0.0002$ | $0.0034 \pm 0.0002$ | $0.0033 \pm 0.0002$ | $0.0031 \pm 0.0002$ | $0.0030 \pm 0.0001$ |
| ER | $0.0034 \pm 0.0004$ | $0.0033 \pm 0.0006$ | $0.0037 \pm 0.0001$ | $0.0038 \pm 0.0002$ | $0.0038 \pm 0.0002$ |
| General | $0.0032 \pm 0.0002$ | $0.0033 \pm 0.0002$ | $0.0033 \pm 0.0002$ | $0.0033 \pm 0.0003$ | $0.0033 \pm 0.0002$ |

*Table 7.* Additional results indicating test scores for the application of a three-layer MPNN as an extension of **Q1** and the repeated application of a two-layer MPNN to learn four steps of Bellman-Ford. The MPNNs used are the same as in the experiments for **Q1**.

| | Nodes | |
|---|---|---|
| **Test Set** (Test score ↓) | General (64) | ER (64) |
| Three steps BF | $0.0075 \pm 0.0023$ | $0.0155 \pm 0.0014$ |
| Repeated application | $0.0040 \pm 0.0011$ | $0.0051 \pm 0.0013$ |

singular non-zero weights converge to positive values across all layers, with a maximum of 1.5 observed in the aggregation FNN of the first MPNN layer. We observe similar behavior for $\ell_1$ and $\ell_2$ regularization, but not as pronounced as in Figure 6.

Since size generalization experiments in **Q1** were conducted only for two Bellman–Ford steps, we aim to provide additional insights into predicting future Bellman–Ford steps. Using the setup from **Q1**, we predict three steps of Bellman-Ford instead of two. For this, we use the same MPNN as in **Q1** and highlight the results obtained in Table 9. We note that a two-layer MPNN is insufficient to learn to predict three steps of Bellman-Ford from scratch using the given training set from **Q1**. However, this aligns with theoretical results, indicating that at least one layer is needed for each step of Bellman-Ford to be sufficiently predicted.

Furthermore, we provide results for a three-layer MPNN trained on three steps of Bellman-Ford. Similar to results for **Q1**, application of a three-layer MPNN to three steps of Bellman-Ford obtains test scores in a similar range as results for two Bellman-Ford steps. Additionally, results from Table 9 indicate the difference between a two-layer and three-layer MPNN for predicting three steps of Bellman-Ford. Results are outlined in Table 7.

Moreover, we consider the case of repeated application of the trained two-layer MPNN from **Q1** to predict more than two Bellman–Ford steps. As a proof of concept, we provide results for predicting the four steps of Bellman-Ford using the two-time application of the MPNN from **Q1**. As indicated in Table 7 repeated application yields favorable results in the outlined case of four Bellman-Ford steps for both tasks. Moreover, results are also improved over Table 9, similar to the application of a three-layer MPNN to learn three steps of Bellman-Ford.

Finally, Table 10 shows additional results for **Q2**, highlighting differences between training and test results due to the increased expressivity of the 1-iWL-MPNN. As shown, the training loss and test score are significantly higher for **Q2** than for the MPNN in **Q1**. This aligns empirical results with theoretical observations on the required expressivity for the SSSP problem, as seen in Proposition 5.

*Table 8.* Results indicating the application of the two-layer MPNN to real-world graph datasets Cora and Citeseer.

| | Nodes | |
|---|---|---|
| **Test Set** (Test score ↓) | Cora | Citeseer |
| Two steps BF | $0.0083 \pm 0.0035$ | $0.0080 \pm 0.0040$ |

*Figure 6.* Weights associated with the two MPNN layers and the corresponding MLPs for **Q1** with 1024 nodes in the test set. Bias values are not shown as they converge towards 0. 1st or 2nd denotes the MPNN layer, whereas layer 1 or 2 denotes the FNN layer.

*Table 9.* Results for the application of the 2-layer MPNN from **Q1** to the prediction task of three Bellman–Ford steps. Following the same protocol as for **Q1**, the node features are given by the node values at the initial Bellman–Ford state. The target is given by the Bellman-Ford state after three steps.

| | **Nodes** | | | | |
|---|---|---|---|---|---|
| **Test Set** (Test score ↓) | 64 | 128 | 256 | 512 | 1024 |
| General | $1.5381 \pm 0.0057$ | $1.5030 \pm 0.0043$ | $1.0831 \pm 0.0015$ | $0.6433 \pm 0.0015$ | $0.5721 \pm 0.0039$ |

*Table 10.* Additional results for **Q2** with size generalization properties. The same setup as in **Q1** was used in **Q2**, without special node labeling, unlike **Q1**.

| | **Nodes - General** | | | | |
|---|---|---|---|---|---|
| **Task** (Score ↓) | 64 | 128 | 256 | 512 | 1024 |
| Q2 - Training | $28.4869 \pm 0.0015$ | $28.4869 \pm 0.0015$ | $28.4869 \pm 0.0015$ | $28.4869 \pm 0.0015$ | $28.4869 \pm 0.0015$ |
| Q2 - Test | $0.8393 \pm 0.0001$ | $0.8414 \pm 0.0008$ | $0.8368 \pm 0.0010$ | $0.8306 \pm 0.0006$ | $0.8217 \pm 0.0010$ |
| Q1 - Training | $2.0544 \pm 0.0151$ | $2.0544 \pm 0.0151$ | $2.0544 \pm 0.0151$ | $2.0544 \pm 0.0151$ | $2.0544 \pm 0.0151$ |
| Q1 - Test | $0.0032 \pm 0.0002$ | $0.0033 \pm 0.0002$ | $0.0033 \pm 0.0002$ | $0.0033 \pm 0.0003$ | $0.0033 \pm 0.0002$ |

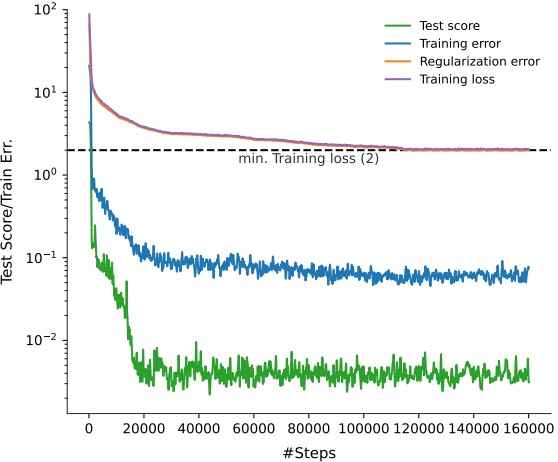

*(a)* Results with General test set for 64 nodes

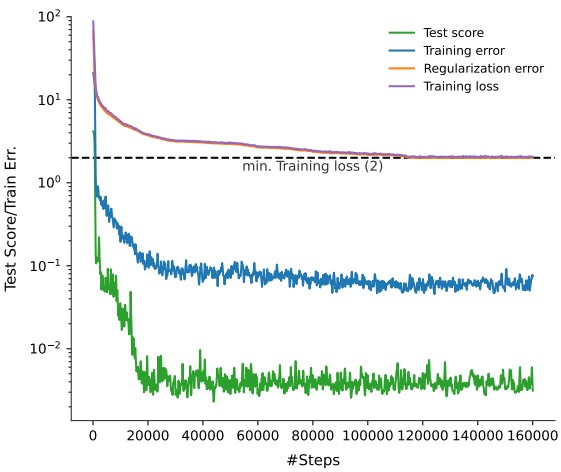

*(b)* Results with General test set for 128 nodes

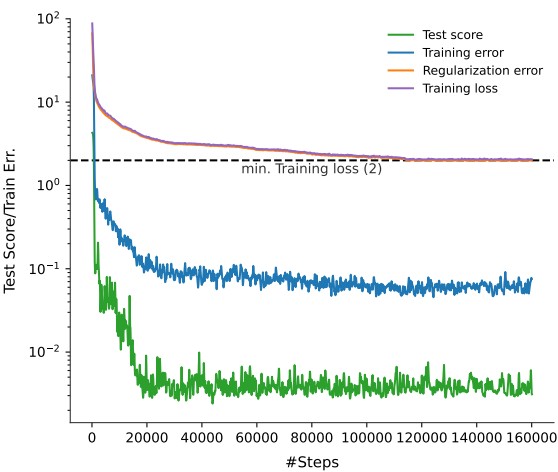

*(c)* Results with General test set for 256 nodes

*Figure 7.* Extended results on size generalization obtained in **Q1** for the General dataset. Each plot shows training error $\mathcal{L}_{\text{emp}}$, training loss $\mathcal{L}$, regularization loss $\eta\mathcal{L}_{\text{reg}}$, and test score for each of the experiments. All plots are generated from the same seed and smoothed using Gaussian smoothing with $\sigma = 1$ (continued on next page).

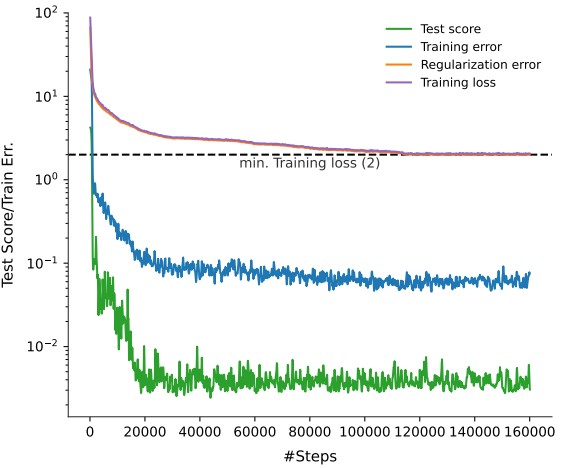

*(d)* Results with General test set for 512 nodes

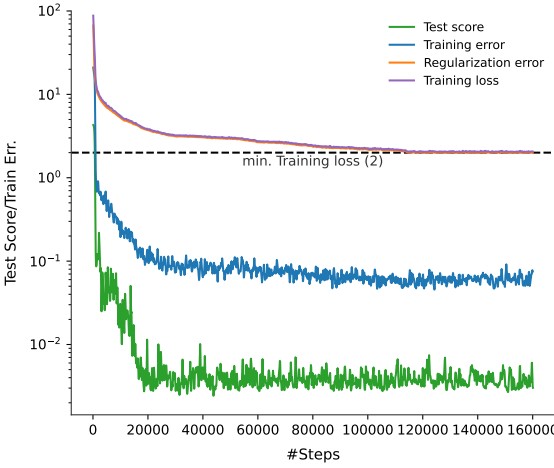

*(e)* Results with General test set for 1024 nodes

