# OpenReview forum: "Which Algorithms Can Graph Neural Networks Learn?"
_ICML.cc/2026/Conference — ICML 2026 spotlight_

### Official Review · Reviewer_VAGH · 2026-02-24

**Soundness:** 4
**Presentation:** 3
**Significance:** 4
**Originality:** 4
**Overall Recommendation:** 6
**Confidence:** 2

**Summary:**

This paper asks when message-passing GNNs (MPNNs) can genuinely learn discrete graph algorithms from finite data and then generalize to larger graphs. The authors develop a learning-theoretic framework built on graph-space pseudo-metrics, finite covering numbers, and Lipschitz certificates, and show that for carefully constructed finite training sets, sufficiently small regularized training loss can imply uniform approximation over graphs of arbitrary size when the target algorithm is Lipschitz and approximable with a bounded certificate.
Using this lens, the paper gives both positive and negative results. On the positive side, it identifies learnable tasks that fall within the finite-Lipschitz / certificate-controlled framework for mean/max/min/normalized-sum MPNNs, including truncated PageRank, $K$-step Bellman-Ford-style shortest paths, and dynamic-programming reductions through shortest paths. On the negative side, it shows that standard MPNNs cannot approximate some classical invariants such as SSSP/MST costs, and that even expressible invariants may still fail to be learnable because of covering-number issues. The Bellman-Ford case is refined further through an explicit small training set and a differentiable $\ell_1$-style regularizer, improving on prior analyses that relied on non-differentiable penalties. Synthetic experiments are included and appear broadly consistent with the theory.

**Compliance With Llm Reviewing Policy:**

Affirmed.

**Final Justification:**

I do think the paper is solid and interesting, and I believe that this paper should be accepted. However, after reviewing the comments and discussions from the other reviewers, I realize I may have been a bit too generous because I am sharing some of the concerns raised by the other reviewers. Therefore, I intend to maintain my current rating but slightly lower my confidence.

**Key Questions For Authors:**

1. Your main guarantees rely on datasets that approximate minimum-cardinality covers of the graph space. What practical sampling-based surrogates would you recommend, and how would the bounds change if one only had such approximate cover constructions with high probability?
2. How should practitioners estimate or upper-bound the relevant certificate budget, for example the target's Lipschitz certificate $B_{f^*}$ or an analogous quantity, when the target is not exactly in-class? Is there a concrete diagnostic that could be used in a standard training workflow?
3. The refined Bellman-Ford result imposes specific architectural and scaling conditions, such as depth tied to $K$ and regularization/training weights with particular behavior. How robust are the empirical results when these conditions are relaxed, e.g., larger aggregation dimension, deeper-than-$K$ models, or noisier supervision?

**Limitations:**

yes

**Strengths And Weaknesses:**

**Strengths**

I think the paper is addressing a genuinely important question: not merely what MPNNs can represent, but when they can be trained on finite instances and still extrapolate algorithmically to larger graphs. The overall story is fairly well organized, and I found the mix of positive and negative results useful for locating the real boundary of the framework.
- The move from pure expressivity to finite-sample learnability and size generalization is valuable, and the metric/cover/certificate perspective feels general enough to matter beyond the specific examples studied here.
- I appreciated that the paper does not oversell only the positive side. By combining learnability results with impossibility and non-learnability statements, it gives a clearer picture of where the approach should and should not work.
- The Bellman-Ford refinement is a meaningful technical step. Replacing the earlier non-differentiable regularizer with a differentiable alternative, while also giving an explicit small training-set construction, makes the result feel closer to something trainable in practice.
- Although the experiments are synthetic, they do serve as a reasonable sanity check: the observed failures of standard MPNNs and the gains from the proposed regularization/architectural choices line up with the qualitative theory.

**(Minor) Weaknesses**

I’m somewhat worried that the main guarantees lean on assumptions that may be difficult to operationalize in realistic training settings. The framework is elegant, but it sometimes feels like it requires a “carefully curated” dataset and certificate budget that practitioners don’t naturally have access to. I also think the empirical section could do more to justify the breadth of the claims.
- A central requirement is access to datasets that approximate minimum-cardinality covers (or near-covers) of the graph input space. In practice, training data are usually sampled from some distribution, so it’s not obvious to me how one would obtain or even approximate such cover-like datasets without exponential blow-up.
- The bounded certificate / certificate budget assumptions are theoretically clean but practically opaque. I’m not sure how a practitioner would diagnose whether a target algorithm is “within budget,” or how to tune those certificate-related quantities in a robust way when the target is only approximately in-class.
- The empirical validation feels narrow relative to the range of algorithmic examples mentioned in the theory (mostly a Bellman–Ford-style shortest-path setting on synthetic graphs). Additional tasks (e.g., DP reductions or other classical invariants) and broader distribution shifts would make the evidence match the ambition of the framework more convincingly.

---

> ### Author Rebuttal · Authors · 2026-03-30
>
> We thank the reviewer for their fair and constructive review.
>
> On the requirement of cover-like training datasets:
>
> We agree that this is a drawback of our framework, which also enables the generality of our results. In practice, one can approximate such datasets via sampling-based surrogates. For example, given a suitable pseudometric on graphs, one can compute pairwise distances and construct an approximate cover via a standard set-cover procedure (e.g., greedy selection of representative samples). This provides a finite-budget approximation to the ideal construction in our theory. More broadly, we believe that future work should study high-probability sampling-based approaches to compute such covers.
>
> On certificate / budget assumptions:
>
> See our response to Q2
>
> On the scope of empirical validation:
>
> We agree that the empirical validation is somewhat narrow. We mainly view our work as a theoretical contribution. To address distributional shifts, we extend our evaluation to the real-world Cora and Citeseer datasets. For further details and additional experiments, we refer to our response to Reviewer SRWb, W3.
> A valuable direction for further empirical evaluation is described below in our response to Q1.
>
> [Q1]
> In practice, one can compute pairwise distances between graphs under the relevant pseudometric (Appendices K.1, L.2). The computational complexity of computing such distances are similar to the ones described in the papers "Tree Mover’s Distance: Bridging Graph Metrics and Stability of Graph Neural Networks",  and "Generalization, and Expressivity, and Universality of Graph Neural Networks on Attributed Graphs" by solving a corresponding optimal transport problem, which can be solved efficiently. Given these distances, constructing an approximate minimum cover reduces to a set cover problem: each graph covers those within a fixed radius, and a dataset can be obtained using a standard set cover algorithm (e.g., greedy). This provides a practical surrogate for the ideal minimum-cover construction in our theory. If we only have a cover with high probability, then the guarantees also hold with high probability. However, achieving such guarantees typically requires larger sample sizes, and analyzing such sampling procedures is an important direction for future work. We will describe this experimental process in more detail in the revised version.
>
> [Q2]
> In practice, when the target is not exactly in-class, we do not currently have a principled, general method for estimating the certificate budget. The most reasonable diagnostic is to tune the regularization strength and trace the tradeoff with training loss: ideally, one uses as much regularization as possible while still fitting the training signal well. This is consistent with our theory and with the discussion in the supplement (see App. D: Learning non-realizable functions and the remark therein). There, the key tension is that excessive regularization can shift the optimum away from a good approximation of the target, while insufficient regularization yields weaker Lipschitz control and higher sample complexity. More broadly, making these quantities practically estimable is, in our view, an important next step.
>
> [Q3]
> We agree these are interesting questions deserving further investigation. We have not yet explored learning $K$ iterations of Bellman-Ford with more than $K$ layers or under noisier supervision — we leave this for future work. Regarding noisy supervision, we note that this falls outside our theoretical setting. Regarding aggregation dimensions, we have observed that training works best with hidden dimensions between 2 and 32, with larger dimensions appearing harder to train.
>
> We are happy to answer any remaining questions.

---

> > ### Author Rebuttal · Reviewer_VAGH · 2026-04-03
> >
> > I appreciate the authors' efforts in providing a thorough rebuttal.

---

### Official Review · Reviewer_SGGx · 2026-03-10

**Soundness:** 4
**Presentation:** 3
**Significance:** 2
**Originality:** 4
**Overall Recommendation:** 4
**Confidence:** 4

**Summary:**

This paper presents an analysis of the generalization capabilities of message passing neural networks (MPNNs) in the context of neural algorithmic reasoning (NAR), i.e., in their ability to learn to simulate combinatorial algorithms across graph sizes. The main result of the paper is that for an hypothesis class of Lipschitz functions and a target Lipschitz function, there exists a regularization term that guarantees good generalization.
The paper establishes that MPNNS are Lipschitz continuous with respect to some pseudometric on graphs' nodes (by using WL unfolding trees), and that common algorithms like PageRank and Bellman-Ford are Lipschitz continuous to the same pseudometric, therefore allowing the applicability of the theorem above.
The paper also establishes some impossibility results, which are mainly inherited from the WL expressivity limitations of MPNNs. Some additional impossibility results on the learnability (rather than expressivity) are derived based on the lack of structure of the psudometric (e.g., for graphs with unbounded degree).
The theoretical analysis is complemented by proof-of-concept experimental results, which show that (individualized) MPNNs are able to generalize to larger graph sizes for the task of simulating two steps of Bellman-Ford.

**Compliance With Llm Reviewing Policy:**

Affirmed.

**Final Justification:**

Despite my concerns on the relevance of the problem, I retain my generally positive assessment of the paper, which is technically solid and interesting.

**Key Questions For Authors:**

See weaknesses.

**Limitations:**

Yes (in fact, very well).

**Strengths And Weaknesses:**

Strengths:
- the paper is really well-written, it is clear, with intuitive explanations and illustrations, and at the same time very formal. The related work is also analyzed (in the appendix) in great depth, giving a precise context for the paper.
- the technical contributions are very significant, and the ideas are original. I could not check all the details, but at a high-level the theoretical claims seem correct.
- understanding the generalization abilities of GNNs, including in the size extrapolation scenario is an important direction for the field (which has been stuck on merely expressivity for too long)
- the experimental evaluation, albeit concise, provides some empirical evidence supporting the theoretical claims. For example, they show that the weighted $\ell_1$ regularization they derive is better than a naive $\ell_1$ regularization.

Weaknesses:
- while I appreciate the beauty of these theoretical results, I wonder about their broader significance. The main question is: why would we want a GNN to simulate Bellman-Ford? This gets even worse for general DPs (e.g., knapsack), where we would have to first construct an ad-hoc graph, and then run a GNN on top, while running the DP would ensure us to get the correct result and would likely be faster. I understand that for some problems, like subgraph isomorphism and PageRank, one could link this to the ability of the GNN to extract some features that can be used in a downstream task (such as molecular property prediction). But for SSSP or DPs, I’m not convinced.
- My understanding (actually, this could be made cleared in the paper) is that the main result (Theorem 2) is only an existence result, and therefore does not yield any guidance on how big the dataset should be or how to construct it (as it would entail computing the covering number of $\mathcal{X}$, which is not immediate). This is improved only for the case of Bellman-Ford. I understand that it might be really hard or impossible to give a constructive proof, so this is a minor limitation.
- The paper over-states a bit the claim that they are among the firsts to deal with the sample efficiency of models in algorithmic reasoning, stating “Such analyses typically […] provide little insight into sample efficiency or generalization beyond the training distribution”. This is not completely true. Among the papers cited there, Xu et al. (Theorem 3.6.) provides sample complexity bounds via algorithmic alignment, albeit in a peculiar learning scenario. Pellizzoni et al. (2025, Section 4) derives sample complexity bounds in a uniform learnability setting. de Luca et al. (2025, Section 6) derives sample complexity bounds for a transformer architecture that aligns with MPC algorithms. Perhaps these (and possibly others) could be discussed.

Minor remarks:
- “assumes familiarity with the classical graph learning problems” should probably be “graph theory” or similar, not learning.
- while reading, I found it confusing that Theorem 4 says that we can approximate the execution of Bellman-Ford and then the next result (Proposition 5) states that MPNNs cannot solve SSSP. My understanding is that the issue lies in the very first step, i.e., assigning distance 0 to the root and +inf to the other nodes, which cannot be done by a MPNN. This should be clarified in the text, I think.

---

> ### Author Rebuttal · Authors · 2026-03-30
>
> We thank the reviewer for their fair and constructive review.
>
> [W1]
> We agree that, when Bellman-Ford or a standard dynamic program is already known and the goal is simply to solve that problem, the classical algorithm is the better tool. Our goal is not to argue that one should replace such algorithms with GNNs in these standalone settings. Rather, we use Bellman-Ford, PageRank, and dynamic programs as canonical test cases for a different question: which algorithmic target functions can MPNNs learn from finite data and then generalize with worst-case guarantees? These examples are useful because they isolate different computational primitives. This matters beyond direct simulation of known algorithms, since differentiable learned modules can be incorporated into larger end-to-end pipelines, whereas the classical discrete procedures generally cannot. We will revise the introduction to make the motivation clearer and to state more explicitly that our contribution primarily concerns understanding learnability and size generalization, rather than replacing classical solvers where they are already available.
>
> [W2]
> Yes, this is correct. Theorem 2 is primarily an existence result, and in full generality it does not by itself give a simple constructive recipe for choosing the dataset. We will clarify this more explicitly in the paper. That said, the theorem proof yields an explicit bound on the required dataset size as a function of the allowable training error, and therefore identifies the relevant tradeoff. Moreover, if one can estimate the certificate, then one can in principle compute the corresponding cover radius and obtain a practical surrogate by computing pairwise distances under the pseudometric and then solving the induced set-cover problem approximately, for example with a greedy algorithm. We agree, however, that this is still much more constructive in the Bellman-Ford case than in the full generality of Theorem 2, and we will present this limitation more clearly.
>
> [W3]
> Thank you for pointing this out, and we agree that our current phrasing is too broad, and will revise it. Our intended claim is not that prior work provides no insight into sample efficiency, but rather that our focus is on the relatively unstudied setting of learning target functions with bounded error uniformly over all test instances in the MPNN setting. In the revision we will explicitly position our contribution relative to Xu et al., Pellizzoni et al., and de Luca et al., and clarify that our novelty is in providing a framework for uniform bounded-error learnability of target functions by MPNNs, rather than in being the first work to study sample complexity in neural algorithmic reasoning.
>
> [R1]
> Thank you, good point. We will replace “classical graph learning problems” with “classical graph problems”.
>
> [R2]
> Yes, this is exactly the right interpretation, and we will clarify it in the paper.
> Proposition 5 states that standard MPNNs, without any special marking of an initial node, are not expressive enough to represent the target function induced by the Bellman–Ford algorithm. However, if we increase their expressivity by allowing a special marking before message passing, the resulting hypothesis class can capture this function.
> We will make this distinction much more explicit in the text so that the relation between the two results is clear.
>
>
> Please consider updating your score if we have addressed your concerns. We are happy to answer any remaining questions.

---

> > ### Author Rebuttal · Reviewer_SGGx · 2026-04-02
> >
> > [W2, W3, R1, R2] Thanks for your clarification.
> >
> > [W1] I see that my concern on the relevance of the problem is shared with most reviewers. You even wrote an extremely vague "...into larger end-to-end pipelines", which only seems to consolidate our concerns.
> >
> > Despite W1, I retain my generally **positive** assessment of the paper, which is technically solid and interesting, also in light of the responses to the other reviews.

---

> > > ### Author Response · Authors · 2026-04-03
> > >
> > > We thank re Reviewer for the positive feedback. We try to further clarify our response to W1 since we acknowledge that our previous response may have placed too much emphasis on the general motivation behind NAR models, and might therefore have seemed a somewhat vague answer to your specific question.
> > >
> > > We clarify that our main contribution is not specifically about learning the Bellman–Ford algorithm with GNNs from a finite dataset. Rather, our theoretical results apply to a broader class of target functions that satisfy certain structural properties (Theorem 2). The Bellman–Ford algorithm is used primarily as a simple and illustrative example of such a target function. In particular, it provides a setting where the required finite dataset can be constructed exactly and in a non-expensive way (Theorem 4), thanks to its simple structure.  More generally, as follow by Theorems 2,3, without additional assumptions on the target function, one would need to construct a cover using the pseudometrics we define in order to obtain such a finite dataset. In this fully general setting, this procedure may be computationally expensive, which further motivates the use of structured examples such as BF for a better understanding of GNNs learnability properties.
> > >
> > > Thank you again for your feedback, and please let us know if you have any further questions.

---

### Official Review · Reviewer_SRWb · 2026-03-12

**Soundness:** 3
**Presentation:** 4
**Significance:** 2
**Originality:** 4
**Overall Recommendation:** 5
**Confidence:** 4

**Summary:**

This paper addresses the theoretical gap in neural algorithmic reasoning to show how message-passing neural networks (MPNNs) can learn discrete graph algorithms from finite training sets. Based on the learnability of Lipschitz functions, they establish finite Lipschitzness for MPNNs with normalized-sum, mean, and max/min aggregations, and proposed a differentiable regularization term to control the computable Lipschitz certificate during training. Their show that under specific conditions, regularized MPNNs can learn standard algorithms like PageRank and Bellman-Ford and generalize to arbitrary sizes. They also identify which invariant algorithms MPNNs cannot learn.

**Compliance With Llm Reviewing Policy:**

Affirmed.

**Final Justification:**

The authors have addressed the comments adequately, and thus, I have increased the score.

**Key Questions For Authors:**

1. Do you have any thoughts on investigating the effect of regularizing the Lipschitz certificate on different common objective functions used for neural algorithmic reasoning on graph combinatorial problems?

2. Have you thought about investigating this method on predictive tasks like node classification? How are the setting and findings expected to change in such scenarios?

**Limitations:**

Yes

**Strengths And Weaknesses:**

### **Strengths**
* The paper tackles generalizaition of neural algorithmic reasoning to out-of-distribution graph sizes. Beyond expressivity they provide formal sample efficiency and generalization guarantees.
* The paper establishes explicit Lipschitzness guarantees for max- and min-aggregation MPNNs.
* Showing computable certificate as a replacement for Lipschitz constant, the authors derive a practical and differentiable regularization strategy that empirically reduces the required training data.

### **Weaknesses**
* Not only the function class must be approximated, but also, as the authors mention in their limitations, the assumptions rarely happened in real-world. Like the bounded graph-diameter for the Bellman-Ford algorithm as graph diameter usually scales with the graph size. This makes the MPNN iterations $K$ be dependent of graph size that breaks the generalization guarantees. Another example is that the target algorithm often does not lie strictly within the hypothesis class. Or the assumption of fixed number of nodes for the normalized-sum aggregation to achieve 1-WL distinguishing power that the goal of arbitrary size generalization. These all put the work distant from practical scenarios.
* Suggesting practical regularizer relies on bounding the computable Lipschitz certificate using parameter norms. But this term might directly misalign with or constrain the actual objectives used for training the MPNN models, resulting in their capacity and expressivity to go on more complex algorithms.
* The experiments only evaluate two steps of a single algorithm (Bellman-Ford) on synthetic toy datasets, which fails to prove the model can learn full algorithmic execution until convergence. Standard benchmarks, state-of-the-art baseline comparisons, and scalability to large graphs are absent in the experiments section. These weaken the applicability or practicality of the paper.

---

> ### Author Rebuttal · Authors · 2026-03-30
>
> We thank the reviewer for their fair and constructive review.
>
> W1 - Practical limitations of the results
>
> Bringing our results closer to practice is, indeed, an important next step. However, because of the dependence on the fixed number of iterations, it is not possible to derive a size-generalization guarantee independent of that number. That said, recurrent GNNs, which share weights across layers, might enable this, and we view it as an interesting direction for future work.
> We note that the other two important practical limitations raised by the reviewer have largely been addressed in our work. With respect to the target algorithm not belonging exactly to the hypothesis class, we note that our framework does not require exact representability; approximate solutions within the hypothesis class are explicitly accounted for (see App. D). For the order-normalized sum, our results do imply size generalization, i.e., the learned parameter transfers to larger graphs even though the normalization factor depends on the graph size.  We will edit the manuscript to highlight these features.
>
> W2 - Restrictions in expressivity caused by the regulariser
>
> We acknowledge this trade-off, which is also present in supervised machine learning more broadly. Note that, by design, the loss in Theorem 2 has a minimum value of zero, meaning the regularization does not harm the ability to fit the training data in this setting. We will make this more apparent in our revision. We prove that several non-trivial algorithms — including Bellman-Ford and Truncated PageRank — are learnable within this framework without being restricted by the regularizer, demonstrating that the trade-off is not prohibitive for practically relevant algorithms. For the case of learning functions outside of the hypothesis class, the effect of regularization can be more detrimental, particularly to the sample complexity (see remark in App. D).
>
> W3 - Experiments
>
> Good point.  To address this, we have extended our evaluation to additional Bellman-Ford iterations and larger graphs.
> We include the following experiments in our rebuttal:
>
> 1.) We train our model on three steps of Bellman-Ford applied to our synthetic datasets from section 5. Furthermore, we extend our evaluation to the Cora and Citeseer datasets. For this real-world data, we remove node and edge labels already present in the datasets and compute Bellman-Ford steps as in our previous experiments. Edge weights and starting nodes are randomly selected, as in the synthetic experiments.
>
> 2.) We also investigate recurrently applying our two-layer MPNN that was trained on two Bellman-Ford steps. In particular, we evaluate how well two iterative applications of this MPNN are able to approximate four Bellman-Ford steps.
>
> From our results below, an extension of our previous results to these new tasks can be seen, with slightly worse MAE scores for the repeated application task compared to two steps of Bellman-Ford.
>
> | | **General (64)** | **ER (64)** |
> |---|---|---|
> | Three steps BF | 0.0075 ±0.0023 | 0.0155 ±0.0014 |
> | Repeated application | 0.0040 ±0.0011 | 0.0051 ±0.0013 |
>
> | | **Cora** | **Citeseer** |
> |---|---|---|
> | Two steps BF | 0.0083 ±0.0035 | 0.0080 ±0.0040 |
>
>
> Q1
>
> We are not certain we fully understood this comment - we suspect it asks how our regularisation is expected to impact results across different graph combinatorial problems, and respond accordingly.
>
> Investigating the effect of our proposed Lipschitz certificate regularisation across different graph problems is an interesting direction, which we leave for future work. We would nevertheless expect the regularisation to be beneficial in settings where enforcing a bounded Lipschitz constant is desirable - for instance, when stability under small perturbations of the input graph is important, or as in our case, when the target function is known to be Lipschitz.
>
> Q2 - Applying method to other tasks such as node prediction
>
> Our results are not tied to specific tasks: we prove that any target function representable by our MPNN function classes is learnable. For tasks beyond this function class, our framework still applies, but additional challenges arise — most notably, the construction of a suitable training set. Unlike Bellman–Ford, where we can exploit the target function's known structure to generate data, this may not be available for general predictive tasks. We consider this an interesting direction for future work.

---

> > ### Author Rebuttal · Reviewer_SRWb · 2026-04-02
> >
> > My concerns are adequately addressed. Thanks for the new experiments. The comments that are left for future work make sense. This is a good paper and I have enjoyed reading it. I will increase my score.

---

### Official Review · Reviewer_hh4R · 2026-03-17

**Soundness:** 3
**Presentation:** 2
**Significance:** 3
**Originality:** 3
**Overall Recommendation:** 4
**Confidence:** 3

**Summary:**

This paper investigates the expressive power and theoretical limits of Message Passing Neural Networks (MPNNs) in solving classical algorithmic problems on graphs. Inspired by foundational results such as universal approximation theorems for neural networks, the authors aim to characterize which classes of algorithms (e.g., combinatorial optimization problems like 0-1 knapsack and minimum spanning tree) can or cannot be represented by MPNNs. The paper provides theoretical analysis of MPNN capabilities and discusses their generalization behavior, attempting to bridge the gap between graph algorithms and neural network-based computation.

**Compliance With Llm Reviewing Policy:**

Affirmed.

**Final Justification:**

The authors' replies clearly resolved some of my concerns.

**Key Questions For Authors:**

Q1. What is the primary motivation for studying classical combinatorial problems (e.g., knapsack, MST) in the context of MPNNs? Are these problems meant as theoretical probes of expressiveness, or do the authors envision practical scenarios where MPNNs would be preferable to classical algorithms?

Q2. How does the ability (or inability) of MPNNs to solve these problems relate to generalization in real-world graph learning tasks?

Q3. Can the authors more precisely present the limitations of existing theoretical frameworks that their work addresses?

Q4. Would it be possible to restructure the paper such that core theoretical results are highlighted in the main text, while extended discussions (e.g., interpretation, implications, limitations) are consolidated into a dedicated section?

**Strengths And Weaknesses:**

**Strengths**

S1. The paper tackles a fundamentally important and long-standing question: understanding the expressive limits of MPNNs from a theoretical perspective. This aligns with classical works (e.g., universal approximation theorems) and contributes to a deeper understanding of graph neural networks at a foundational level.

S2. The work attempts to connect neural network expressiveness with classical algorithmic problems, which is intellectually appealing and could potentially inspire future research at the intersection of learning and algorithms.

**Concerns**

(1) Soundness

C1. The connection between solving classical combinatorial problems (e.g., 0-1 knapsack, minimum spanning tree) and the capabilities of MPNNs is not sufficiently justified. These problems already have well-established exact or efficient algorithms, and the paper does not clearly explain why studying them through MPNNs is meaningful from either a theoretical or practical standpoint. For example, early research on GNNs to solve graph isomorphism problems was primarily driven by the enormous complexity of graph isomorphism problems.

C2. The relationship between the ability to solve these problems and the notion of generalization is unclear. It is not well explained how success (or failure) on such algorithmic tasks translates into broader insights about generalization in graph learning settings.

(2) Presentation

C3. The discussion of limitations in existing theory is somewhat vague, which weakens the motivation for the proposed analysis. As a result, it is difficult to clearly identify what gap in the literature the paper is filling.

C4. It would be better if authors can include only the most important theoretical conclusions in the main text. The focus should be on the discussion, which should both demonstrate the practical significance of the theory through experiments and explore its limitations. (may merge sections 4 and 5).

(3) Significance

Please check C1.

(4) Originality

C5. The work is conceptually aligned with existing studies on the expressive power of GNNs. While it extends this line of research, the novelty is somewhat limited by the lack of strong new insights or clearly differentiated contributions.

---

> ### Author Rebuttal · Authors · 2026-03-30
>
> We thank the reviewer for their fair and constructive review.
>
> [C1]
> Our goal is not to solve well-known combinatorial problems with GNNs, but to characterize which target functions can be learned by MPNNs through training, with worst-case bounds. We use classical dynamic programming problems (e.g., Bellman–Ford, Knapsack) as canonical instances of such functions, since their structure allows us to precisely identify properties that make them learnable. Theorem 2 formalizes these conditions at the level of the hypothesis class and target function, while later results provide explicit examples that satisfy them. Importantly, our results go beyond expressivity: we show that these functions are not only representable but also learnable, with worst-case error bounds, given appropriate training data and a loss function. A key motivation is neural combinatorial optimization, where learning algorithmic primitives (e.g., shortest paths) helps understand how MPNNs can exploit structure in more complex tasks without known algorithms. We will clarify this in the introduction.
>
> [C2]
> Our goal is to understand when a model can learn a target function, thereby supporting insights into (size) generalization. In our setting, this means that once the function is learned, it can be applied to all unseen graphs, including those outside the training distribution and of arbitrary size. The key point is that generalization depends on structural properties of the target function. The algorithmic tasks we consider (e.g., Bellman–Ford) serve as canonical examples, but our main result (Theorem 2) applies to any function satisfying the stated assumptions. We study a strong notion of generalization by providing worst-case guarantees over all inputs, in contrast to standard PAC-style guarantees that hold with high probability under a distribution. Our goal is to understand when neural models can achieve worst-case guarantees comparable to those of classical algorithms.
>
>
> [C3]
> Classical learning theory typically provides guarantees in distribution, allowing large errors on low-probability inputs, and thus does not ensure worst-case performance over all inputs. In contrast, classical algorithms often come with such guarantees. Our work addresses this gap by studying when MPNNs can learn target functions with uniform (worst-case) guarantees. Moreover, most existing theoretical work focuses on whether a target function belongs to the hypothesis class, whereas we study whether such functions can be learned by minimizing empirical risk, the central objective in machine learning. Our framework thus connects learnability, training, and generalization.
>
> [C4]
> Thank you for the suggestion. In the current version, we already include informal explanations of several theoretical results to improve accessibility. In the revision, we will streamline the presentation by highlighting the main results while strengthening the discussion of practical implications.
>
> [Q1]
> We are primarily interested in the size-generalization abilities of MPNNs, rather than expressivity alone. While the considered problems are computationally well understood, it remains largely unclear which algorithmic procedures MPNNs can learn (and generalize to all instances). We therefore use classical problems as controlled test cases to probe which algorithmic components GNNs can learn. Starting from well-understood algorithms provides a principled way to build this understanding before moving to more complex or less structured tasks (see also our responses to C1, C3).
>
> [Q2]
> Our results show that finite training datasets can enable size generalization, clarifying which types of algorithms can be learned and generalized by MPNNs. This provides guidance for practitioners on what to expect from training on limited data. For example, our construction of finite training sets may inspire extensions or modifications of real-world datasets to improve generalization. More broadly, our framework helps identify when a learned model is likely to generalize beyond the training distribution.
>
> [Q3]
> Most existing theoretical work on MPNNs focuses on expressivity, i.e., what functions can be represented. In contrast, we study when such functions can be learned from data with worst-case guarantees.
> To the best of our knowledge, very few works address this question (e.g., Nerem et al., 2025, for shortest paths). Our contribution is to provide a general framework that characterizes when training recovers such algorithmic behavior, going beyond expressivity analysis. We will revise the manuscript to make this distinction clearer.
>
> [Q4]
> As noted above, we will use the additional page to better balance technical results and discussion by highlighting the main contributions and expanding their implications.
>
> Please consider updating your score if we answered your questions. We are happy to answer any remaining questions.

---

> > ### Author Rebuttal · Reviewer_hh4R · 2026-04-04
> >
> > Thank you for the author's reply. It clearly resolved some of my concerns. I will increase my score.

---

### Decision · Program_Chairs · 2026-04-30

**Decision:**

Accept (spotlight)

**Comment:**

In this paper, the authors tackle the problem of MPGNNs' expressiveness by studying their ability to implement various discrete graph algorithms. The reviewers globally praised the importance and scope of the theoretical results, on a well-studied but globally still open significant question in GNNs' theory. Some concerns were raised about the applicability of these results in realistic scenarii, but without diminishing the positive view on the theory developed by the authors.